# On the Power of Multitask Representation Learning with Gradient Descent

## Abstract

Representation learning, particularly multi-task representation learning, has gained widespread popularity in various deep learning applications, ranging from computer vision to natural language processing, due to its remarkable generalization performance. Despite its growing use, our understanding of the underlying mechanisms remains limited. In this paper, we provide a theoretical analysis elucidating why multi-task representation learning outperforms its single-task counterpart in scenarios involving over-parameterized two-layer convolutional neural networks trained by gradient descent. Our analysis is based on a data model that encompasses both task-shared and task-specific features, a setting commonly encountered in real-world applications. We also present experiments on synthetic and real-world data to illustrate and validate our theoretical findings.

## 1 Introduction

Multi-task representation learning (Caruana, 1997) is an important machine learning paradigm that simultaneously learns multiple tasks within a single model. The goal is to extract the shared representations from the input data so that they can benefit all tasks. By exploiting the relationships between tasks, multi-task learning aims to enhance generalization, optimize resource utilization, and promote transferability. Coupled with the advancements in deep learning models, it has gained substantial popularity and made notable progress (He et al., 2017; Finn et al., 2017; Liu et al., 2019; Yao et al., 2022). Take BERT (Devlin et al., 2018) as a well-known example. By training on both the pre-training task and fine-tuning task, BERT can capture general representations more effectively and combine them with task-specific features to achieve outstanding performance. However, while learning across tasks produces representations that generalize remarkably well, the formal understanding of the underlying mechanism remains less explored.

Drawing upon foundational research in generalization and transferability (Baxter, 2000; Yosinski et al., 2014), a line of theoretical investigations (Evgeniou et al., 2005; Du et al., 2017; Frei et al., 2022; Shen et al., 2022) has sought to explain the exceptional performance achieved by multi-task representation learning. However, even for simplified mathematical models, a gap persists between theoretical understanding and the many observed improvements of multi-task learning in practice. Specifically, when selecting different tasks to be learned at the same time, there may exist similarities in problem structures or task types among them. For example, the tasks can share the same goal such as classification or regression. However, they could have totally different or even orthogonal features. Compared to learning on a single task, jointly learning these tasks can still yield improvement in generalization. Consequently, a crucial question emerges: how do the intrinsic relationships between tasks help consistently improve model's performances across different tasks with different features? The fundamental conditions behind this phenomenon and their important role in obtaining the exceptional generalization results remain undisclosed.

In this paper, we aim to rigorously investigate the underlying mechanism behind the intriguing enhancement in generalization observed for of multi-task learning. To achieve this, we perform a formal examination and comparison of the learning procedures of single-task learning and multi-task learning, considering a two-layer convolutional neural network with smoothed ReLU activation. Within our theoretical framework, we consider an image-like data model, where each data consists of a feature patch and several noise patches. Given the joint training of multiple related tasks, we consider two possibilities for the feature patch: it could be either a task-shared feature, common to all tasks, or task-specific feature, unique to each task. With careful design of the setting, we present a comprehensive theoretical analysis summarized as follows:

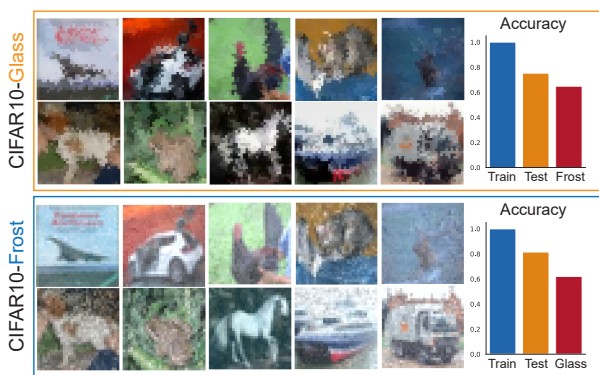

Figure 1: We demonstrate data examples for two tasks, CIFAR10-Glass and CIFAR10-Frost, to illustrate the following: (1) shared features exist across the tasks as they share the same original images in CIFAR10; (2) each task contains unique features as the corruption significantly changes the image style. The bar plots demonstrate the training and test accuracy for models trained on each task, where last column show the test accuracy of a model on its unseen task. The drop in test accuracy also indicates the existence of unique features.

1. Through careful examination and analysis of the learning processes, we separate these procedures into distinct stages. We clarify the learning behaviors exhibited at each stage and elucidate their influence on learning features of tasks within both multi-task learning and single-task learning settings.
2. By reviewing learning results of training samples, we identify the types of data that exhibit significantly different performances between multi-task learning and single-task learning. Additionally, we delve into theoretical explanations for this observed phenomenon.
3. Combining the above aspects, we establish bounds on optimization and generalization for single-task learning and multi-task learning. Furthermore, we offer explanations on intrinsic relationships underlying different results, with a particular focus on generalization.

## 1.1 OUR CONTRIBUTION

The contributions of this paper are summarized as follows.

- We prove the global convergence of gradient descent with weight decay for both multi-task learning and single-task learning. In a polynomial number of iterations, the two-layer convolutional neural network can be trained to attain zero training error on both multiple tasks and a single task.
- We further prove that the global solutions attained by multi-task learning and single-task learning are different and they have very different generalization abilities. While the model trained on multiple tasks can generalize well with a nearly zero test error, the model trained on a single task has much worse performance on generalization with a test error no less than a constant.
- We depict the learning process of neural networks under these two different circumstances. For multi-task learning, the model fits the training data on the feature patches including both task-shared features and task-specific features, so that the solution contracts all the information of features. But for single-task learning, the model fits the noise patches instead for the part of the data with task-shared feature, which ends in losing some information on features.
- We conduct extensive experiments on both synthetic data generated from our data model and real-world benchmark data including MNIST-C (Mu & Gilmer, 2019) and CIFAR10-C (Hendrycks & Dietterich, 2018). The empirical results confirm and support our theoretical findings.

**Notation.** We use lower case letters, lower case bold face letters, and upper case bold face letters to denote scalars, vectors, and matrices respectively. For a vector $\mathbf{v} = (v_1, \cdots, v_d)^\top$, we denote by $\|\mathbf{v}\|_2 := \left( \sum_{j=1}^d v_j^2 \right)^{1/2}$ its 2-norm. For two sequence $\{a_k\}$ and $\{b_k\}$, we denote $a_k = O(b_k)$ if $|a_k| \leq C|b_k|$ for some absolute constant $C$, denote $a_k = \Omega(b_k)$ if $b_k = O(a_k)$, and denote $a_k = \Theta(b_k)$ if $a_k = O(b_k)$ and $a_k = \Omega(b_k)$. We also denote $a_k = o(b_k)$ if $\lim |a_k/b_k| = 0$. Finally, we use $\widetilde{O}(\cdot)$ and $\widetilde{\Omega}(\cdot)$ to omit logarithmic terms in the notation. We denote the set $\{1, \cdots, n\}$ by $[n]$. The carnality of a set $S$ is denoted by $|S|$. Additionally, we denote $x_n = \text{poly}(y_n)$ if $x_n = O(y_n^D)$ for some positive constant $D$, and $x_n = \text{polylog}(y_n)$ if $x_n = \text{poly}(\log(y_n))$.

## 2 RELATED WORK

**Multi-task Representation Learning.** The ideas of representation learning and multi-task representation learning have been studied since 90's (Caruana, 1997; Thrun & Pratt, 1998; Baxter,

2000). Since its excellent performance has been discovered on many applications, a growing number of multi-task learning methods on different specific problems have been studied (Ben-David & Schuller, 2003; Ando et al., 2005; Argyriou et al., 2006; Cavallanti et al., 2010; Kuzborskij & Orabona, 2013; Maurer et al., 2013; Pontil & Maurer, 2013; Pentina & Lampert, 2014; Widmer et al., 2014). Notably, it yields some of the best results in computer vision (He et al., 2017) and natural language processing (Devlin et al., 2018). On the theoretical analyses, a series of works (Baxter, 2000; Maurer et al., 2016; Maurer, 2006; Cavallanti et al., 2010; Lounici et al., 2011; Tripuraneni et al., 2020) focus on analyzing multi-task representation learning, including its generalization capabilities and sample complexities. While these works study the linear setting, Tripuraneni et al. (2021) generalizes to non-linear settings. Our work focuses on the more complex non-linear setting. However, few of them consider a model as close to real-world as ours, which is to train neural networks on data that consists of different sources of features, and multiple sparse noises with feature noises. Also, a thorough analysis of feature learning and noise memorization during the process is also relatively rare.

**Optimization and generalization in deep learning.** There have been many theoretical explorations on the optimization and generalization mechanisms of neural networks (Du et al., 2018; 2019; Allen-Zhu et al., 2019b; Zou et al., 2020; Allen-Zhu et al., 2019a; Arora et al., 2019b;a; Ji & Telgarsky, 2019; Chen et al., 2021). Most of them are in the neural tangent kernel (NTK) regime (Jacot et al., 2018) or lazy training regime (Chizat et al., 2019). More recently, more works start to study the learning properties of neural networks outside of NTK regime (Allen-Zhu & Li, 2019; Bai & Lee, 2019; Shen et al., 2022). We note that our work also makes analysis beyond the NTK regime, following the line of works that applies tensor power method for analysis (Allen-Zhu & Li, 2020; Zou et al., 2021; Kou et al., 2022; Chen et al., 2022b).

## 3 PRELIMINARIES

In this section, we outline our problem setup that serves as the foundation for our theoretical development.

**Problem Setup.** We consider a scenario involving $K(K \geq 2)$ distinct classification tasks. Our approach involves training a Convolutional Neural Network (CNN) using gradient descent on training examples $(\mathbf{x}, y)$ generated from a data model $\mathcal{D}$. For each of the $K$ tasks, we possess a collection of $n$ training samples, denoted as $\{(\mathbf{x}_{i,k}, y_{i,k})\}_{i=1}^{n}$.

Within this data model, the input data is composed of both feature and noise patches. Specifically, we represent $\mathbf{x}$ as a vector consisting of $H$ patches, i.e., $\mathbf{x} = (\mathbf{x}_1, \mathbf{x}_2, ..., \mathbf{x}_H) \in \mathbb{R}^{Hd}$. Among these patches, one represents the feature patch generated according to Definition 3.1, while the others represent noise patches generated as per Definition 3.2.

**Definition 3.1** (Feature Patch Generation). For data $(\mathbf{x}, y)$ which belongs to the $k$-th task, the feature patch is generated as one of the following case randomly:

- **Case 1: task-shared feature.** This patch is given by $\alpha y \cdot \mathbf{v}$, where $0 < \alpha < 1$, $\mathbf{v} = (1, 0, ..., 0)$ denotes the task-shared feature and it is all the same for different tasks.
- **Case 2: task-specific feature.** This patch is given by $y \cdot \mathbf{v}_k$, where $\mathbf{v}_k$ denotes the task-specific feature for the $k$-th task and $\mathbf{v}_k$, which is 1 on its $k$-th coordinate and 0 on others, is orthogonal to $\mathbf{v}$.

With probability $p$, the feature patch is taken as the task-shared feature $\alpha y \cdot \mathbf{v}$, and with probability $q = 1 - p$, the feature patch is taken as the task-specific feature $y \cdot \mathbf{v}_k$.

**Definition 3.2** (Noise Patch Generation). For data $(\mathbf{x}, y)$ which belongs to the $k$-th task, after the generation of feature patch $\mathbf{v}$, the noise patch $\mathbf{x}_h$ is generated according to the following process:

- Randomly select $s$ coordinates from $[d] \backslash [K + 1]$ uniformly, denoted as a vector $\mathbf{s}_h \in \{0, 1\}^d$.
- Generate $\boldsymbol{\zeta}_h$ from distribution $\mathcal{N}\left(0, \sigma_p^2 \mathbf{I}_d\right)$, and then mask off the first $K + 1$ coordinates and other $d - s - K - 1$ coordinates, i.e., $\boldsymbol{\zeta}_h = \boldsymbol{\zeta}_h \odot \mathbf{s}_h$.
- Add feature noise to $\boldsymbol{\zeta}_h$ to form the final noise patch $\boldsymbol{\xi}_h$, i.e., $\boldsymbol{\xi}_h = \boldsymbol{\zeta}_h - \beta_h y \mathbf{v}$, where $0 < \beta_h < 1$ is the strength of the feature noise.

This data model is motivated by multitask representation learning, where data samples originate from various learning tasks. It accounts for the presence of task-shared features common across all tasks and task-specific features unique to individual tasks, often perturbed by noise. Such data

models are prevalent in various applications, especially in classification problems. For instance, when dealing with multiple similar image classification tasks, the labels may depend on general information shared among all tasks (e.g., animals/vehicles) or more specific information limited to one task (e.g., dogs/cats), while being influenced by noise.

Our data model can be thought of as representing the output of an intermediate layer of a CNN, assuming entry-wise independent noise and sparsity. These assumptions draw inspiration from prior works such as Yang (2019), which discussed scenarios where hidden nodes in an intermediate layer are independently sampled, and Papyan et al. (2017), which demonstrated the sparsity of outputs in such layers. Importantly, our proof techniques can be extended to settings where features and noise exhibit higher density, as long as a sparsity gap between features and noise is maintained.

Given our data model, our objective is to optimize the CNN model defined as follows:

$$F_j(\mathbf{W}, \mathbf{x}) = \sum_{r=1}^{m} \sum_{h=1}^{H} \sigma\left(\langle \mathbf{w}_{j,r}, \mathbf{x}_h \rangle\right), j \in \{\pm 1\}. \tag{1}$$

Specifically, we employ the same smoothed ReLU activation function as described in Allen-Zhu & Li (2020), and its definition can be found in Definition 3.3.

**Definition 3.3.** For an integer $q \geq 3$ and a threshold $\rho = \frac{1}{\text{polylog}(n)}$ , we define the following smoothed ReLU function:

$$\widetilde{\text{ReLU}}(z) = \begin{cases} 0 & \text{if } z \leq 0; \\ \frac{z^q}{q\rho^{q-1}} & \text{if } z \in [0, \rho]; \\ z - (1 - \frac{1}{q})\rho & \text{if } z \geq \rho \end{cases}$$

Our goal is to minimize the regularized logistic loss function. Therefore, for single-task learning trained on $k$-th task, the target function is

$$L_k(\mathbf{W}) = \frac{1}{n} \sum_{i=1}^{n} L_{i,k}(\mathbf{W}) + \frac{\lambda}{2} \|\mathbf{W}\|_{\text{F}}^2. \tag{2}$$

and for multi-task learning, the target function is

$$L(\mathbf{W}) = \frac{1}{nK} \sum_{k=1}^{K} \sum_{i=1}^{n} L_{i,k}(\mathbf{W}) + \frac{\lambda}{2} \|\mathbf{W}\|_{\text{F}}^2. \tag{3}$$

Here, $L_{i,k}(\mathbf{W}) = -\log \frac{e^{F_{y_{i,k}}(\mathbf{W}, \mathbf{x}_{i,k})}}{\sum_{j \in \{-1,1\}} e^{F_j(\mathbf{W}, \mathbf{x}_{i,k})}}$ represents the individual loss for data $(\mathbf{x}_{i,k}, y_{i,k})$, which is the $i$-th data point from the $k$-th task. We use gradient descent to optimize the loss:

$$\mathbf{w}_{j,r}^{(0)} \sim \mathcal{N}(0, \sigma_0^2 \mathbf{I}_d),$$
$$\mathbf{w}_{j,r}^{(t+1)} = \mathbf{w}_{j,r}^{(t)} - \eta \cdot \nabla_{\mathbf{w}_{j,r}} L(\mathbf{W}^{(t)}), \qquad\qquad j \in \{\pm 1\}, r \in [m], \tag{4}$$

where $\eta$ is the learning rate.

**Other requirements.** We also have certain requirements for parameter magnitude, which are listed here. Based on $n$, we have $s = \Theta(n^c)$, $K = \Omega(n^{\frac{c}{8q}})$, $d = \Omega(n^{2c+4}K)$, $\sigma_0 = \Theta(n^{-\frac{c}{6q}})$, $\sigma_p = \Theta(n^{-\frac{1}{2}+\frac{1}{4q}})$, $\alpha = \Theta(n^{-\frac{c}{10q^2}})$, $\beta_h = \Theta(n^{-\frac{c}{18q}})$, $\forall h \in [2, H]$, $\lambda = O(n^{-\frac{c}{6}+\frac{3c}{10q}})$ and $m = \text{polylog}(n)$. Other parameters not listed are absolute constants with some logarithmic terms.

## 4 MAIN RESULTS

In this section we will introduce our main results. We will give optimization and generalization guarantees of gradient descent for both training a two-layer CNN model on multiple tasks and a single task in the following theorems. There are two parts in our main theorems: single-task learning and multi-task learning.

### 4.1 SINGLE-TASK LEARNING

In this part, we consider the analysis for the learning problem of the $k$-th task.

**Theorem 4.1.** *(Single-task Setting) Consider a two-layer CNN defined in* (1) *and regularized training objective* (2) *with a regularization parameter $\lambda > 0$, with the data distribution following Definition 3.1 and 3.2, then we have the following guarantees on the training and test errors for the models trained by Gradient descent. Suppose we run gradient descent for $T = \frac{\text{poly}(n, \lambda^{-1})}{\eta}$ iterations with learning rate $\eta$, then with probability at least $1 - O(n^{-1})$, we can find a NN model $\mathbf{W}_{\text{single}}^*$ such that $\|\nabla L(\mathbf{W}_{\text{single}}^*)\|_F^2 \leq \frac{1}{T\eta}$. Moreover, the model $\mathbf{W}_{\text{single}}^*$ also satisfies:*

- *Training error is zero: $\frac{1}{n} \sum_{i=1}^n \mathbf{1}[F_{y_{i,k}}(\mathbf{W}_{\text{single}}^*, \mathbf{x}_{i,k}) \leq F_{-y_{i,k}}(\mathbf{W}_{\text{single}}^*, \mathbf{x}_{i,k})] = 0$;*
- *Test error is high: $\mathbb{P}_{(\mathbf{x}, y) \sim \mathcal{D}}[F_y(\mathbf{W}_{\text{single}}^*, \mathbf{x}) \leq F_{-y}(\mathbf{W}_{\text{single}}^*, \mathbf{x})] \geq \frac{p}{4} - \text{poly}(n^{-1})$.*

From Theorem 4.1, we can see that the optimization on training data of one single task via gradient descent with a two-layer convolution neural network will converge to a global solution which has both a zero classification error and a small gradient. However, this solution does not generalize well, as it has a constant lower bound for test error.

## 4.2 MULTI-TASK LEARNING

In this part, we consider the analysis for the learning problem of all $K$ tasks.

**Theorem 4.2.** *(Multi-task Setting) Consider a two-layer CNN defined in* (1) *with and regularized training objective* (3) *with a regularization parameter $\lambda > 0$, with the data distribution following Definition 3.1 and 3.2, then we have the following guarantees on the training and test errors for the models trained by Gradient descent. Suppose we run gradient descent for $T = \frac{\text{poly}(n, \lambda^{-1})}{\eta}$ iterations with learning rate $\eta$, then with probability at least $1 - O(n^{-1})$, we can find a NN model $\mathbf{W}_{\text{multi}}^*$ such that $\|\nabla L(\mathbf{W}_{\text{multi}}^*)\|_F^2 \leq \frac{1}{T\eta}$. Moreover, the model $\mathbf{W}_{\text{multi}}^*$ also satisfies:*

- *Training error is zero: $\frac{1}{nK} \sum_{k=1}^K \sum_{i=1}^n \mathbf{1}[F_{y_{i,k}}(\mathbf{W}_{\text{multi}}^*, \mathbf{x}_{i,k}) \leq F_{-y_{i,k}}(\mathbf{W}_{\text{multi}}^*, \mathbf{x}_{i,k})] = 0$;*
- *Test error is nearly zero: $\mathbb{P}_{(\mathbf{x}, y) \sim \mathcal{D}}[F_y(\mathbf{W}_{\text{multi}}^*, \mathbf{x}) \leq F_{-y}(\mathbf{W}_{\text{multi}}^*, \mathbf{x})] = \text{poly}(n^{-1})$.*

From Theorem 4.2, we observe that when using gradient descent with the same neural network but trained on samples from all tasks, it achieves optimal training optimization performance characterized by a small gradient and perfect accuracy. Importantly, this solution also exhibits strong generalization capabilities on unseen test data.

## 4.3 DISCUSSION

Comparing Theorem 4.1 and Theorem 4.2, we observe similar optimization performance in both cases. However, there is a significant disparity in terms of generalization to test data, even when utilizing weight decay regularization. Models trained on multiple tasks exhibit remarkable generalization capabilities and achieve nearly zero test error. In contrast, models trained on a single task cannot attain a test error lower than a constant. This gap is formed during the training process, as outlined in the following section.

# 5 PROOF SKETCH OF MAIN RESULTS

In this section, we provide an outline of our proof and introduce critical techniques employed in proving Theorem 4.1 and Theorem 4.2. Subsequently, we present a fundamental stage separation in the learning processes of single-task learning and multi-task learning. Both learning scenarios are structured similarly, with two main stages: the pattern learning stage and the regularization stage. Therefore, we present an overview of each stage for both scenarios concurrently. Our focus centers on several key quantities: $\langle \mathbf{w}_{j,r}^{(t)}, j \cdot \mathbf{v} \rangle$, $\langle \mathbf{w}_{j,r}^{(t)}, j \cdot \mathbf{v}_k \rangle$, $\langle \mathbf{w}_{j,r}^{(t)}, \boldsymbol{\zeta}_{i,k,h} \rangle$. A larger value of $\langle \mathbf{w}_{j,r}^{(t)}, j \cdot \mathbf{v} \rangle$ signifies improved task-shared feature learning, while a larger $\langle \mathbf{w}_{j,r}^{(t)}, j \cdot \mathbf{v}_k \rangle$ indicates better task-specific feature learning. Furthermore, a greater $\langle \mathbf{w}_{j,r}^{(t)}, \boldsymbol{\zeta}_{i,k,h} \rangle$ represents enhanced noise memorization.

## 5.1 IMPORTANT LEMMA: TENSOR-POWER METHOD

In this section, we present an important result regarding the depiction and comparison of the growth rates of two non-linear sequences, which have different parameter magnitudes. This can be viewed as a weaker version of Lemma D.19 in Allen-Zhu & Li (2020).

**Lemma 5.1.** *Let $\{x_t, y_t\}_{t=1,\dots}$ be two positive sequences that satisfy the following conditions:*

$$x_{t+1} \geq x_t + \eta \cdot A x_t^{q-1},$$
$$y_{t+1} \leq y_t + \eta \cdot B y_t^{q-1},$$

*for some $A$ and $B$ such that $A x_0^{q-2} \geq B y_0^{q-2} \log\left(1/x_0\right)$ and $\eta = o(1)$, $A x_0^{q-2} = o(1)$ and $B y_0^{q-2} = o(1)$. Then, for any $q \geq 3$, for some $C \in (x_0, O(1))$, let $T_x$ be the first iteration such that $x_t \geq C$, then we have*

$$T_x \eta \leq \widetilde{\Theta}(A^{-1} x_0^{2-q}) \quad and \quad y_{T_x} \leq O(y_0).$$

Owing to the characteristics of the pattern learning stage and our activation function, we will observe that some of the updating rules for the target inner products in this stage exhibit similarities with this type of non-linear growth. Consequently, we can apply the tensor-power method to establish bounds on the time and magnitude of each inner product.

**Assumptions and notations.** Without loss of generality, we consider the first task for single-task learning and assume the first patch of data is the feature path and the other patches are noise patches. Denote $\mathcal{D}_1$ as the index set of training data whose feature patch is taken as the task-shared feature, and $\mathcal{D}_2$ as the index of the rest of the training data whose feature patch is taken as the task-specific feature. For the simplicity of proof, we define $\Lambda_j^{(t)} = \max_{r \in [m]}[\langle \mathbf{w}_{j,r}^{(t)}, j \cdot \mathbf{v} \rangle]_+$, $\Psi_j^{(t)} = \max_{r \in [m]}[\langle \mathbf{w}_{j,r}^{(t)}, j \cdot \mathbf{v}_1 \rangle]_+$, and $\Phi_{j,i}^{(t)} = \max_{r \in [m], h \in [2,H]}[\langle \mathbf{w}_{j,r}^{(t)}, \boldsymbol{\zeta}_{i,k,h} \rangle]_+$ for single-task learning, $\Phi_{j,i}^{(t)} = \max_{k \in [K]} \max_{r \in [m], h \in [2,H]}[\langle \mathbf{w}_{j,r}^{(t)}, \boldsymbol{\zeta}_{i,k,h} \rangle]_+$ for multi-task learning to quantify task-shared feature learning, task-specific feature learning and noise memorization respectively. Let $P_j$ be the iteration number that $\Psi_j^{(t)}$ reaches $\Theta(1/m)$ for $j \in \{\pm 1\}$, $T_i$ be the iteration number that $\Phi_{y_{i,1},i}^{(t)}$ reaches $\Theta(1/m)$ for $i \in \mathcal{D}_1$, and $Q_j$ be the iteration number that $\Lambda_j^{(t)}$ reaches $\Theta(1/m)$ for $j \in \{\pm 1\}$.

## 5.2 PATTERN LEARNING STAGE

In this initial learning phase, weight decay has minimal impact, and gradient descent guides the model to learn from training samples. Notably, single-task and multi-task learning exhibit different behaviors in this stage. Based on our data distribution definition, the task-shared feature is "weakened" by a factor $\alpha$. Consequently, in single-task learning, for data with task-shared feature (in $\mathcal{D}_2$), the noise patches will have a greater "impact" than the task-shared feature patch. As a result, the model focuses on learning the noise patches. Conversely, in multi-task learning, the "impact" of the task-shared feature from data across all tasks is combined, surpassing that of the noise patches. Therefore, the model prioritizes learning the task-shared patch. For data with task-specific feature patches (in $\mathcal{D}_1$), in both single-task and multi-task learning, the task-specific feature consistently carries a greater "impact" than the noises. Therefore, the model concentrates on learning the task-specific feature.

For single-task learning during the pattern learning stage, we observe the following inequalities regarding the increasing patterns of various metrics:

$$\Psi_j^{(t)} \geq \Psi_j^{(t-1)} + \frac{\eta}{2\rho^{q-1}} \Theta((\Psi_j^{(t-1)})^{q-1}), \qquad \forall j \in \{\pm 1\}, t \leq P_j;$$

$$\Phi_{j,i}^{(t)} \leq \Phi_{j,i}^{(t-1)} + \frac{\eta}{\rho^{q-1}} \Theta(s\sigma_p^2/n) \Theta((\Phi_{j,i}^{(t-1)} + \widetilde{\Theta}(\sigma_0^{\frac{2}{3}}))^{q-1}), \quad \forall j \in \{\pm 1\}, i \in \mathcal{D}_2, t \leq \max_{j \in \{\pm 1\}} P_j;$$

$$\Phi_{y_{i,1},i}^{(t)} \geq \Phi_{y_{i,1},i}^{(t-1)} + \frac{\eta}{2\rho^{q-1}} \Theta(s\sigma_p^2/n) \Theta((\Phi_{y_{i,1},i}^{(t-1)} - \widetilde{\Theta}(\sigma_0))^{q-1}), \quad \forall i \in \mathcal{D}_1, t \leq T_i;$$

$$\Lambda_j^{(t)} \leq \max\{\Lambda_j^{(t-1)} + \frac{\eta\alpha^q}{\rho^{q-1}} \Theta((\Lambda_j^{(t-1)})^{q-1}), \widetilde{\Theta}(\alpha^q \sigma_0^{\frac{2}{3}})\}, \qquad \forall j \in \{\pm 1\}, t \leq \max_{i \in \mathcal{D}_1} T_i.$$

Considering $\{\Psi_j^{(t)}\}, \{\Phi_{y_{i,1},i}^{(t)}\}(j \in \{\pm 1\}, i \in \mathcal{D}_1)$ as $\{x_t\}$, and $\{\Lambda_j^{(t)}\}, \{\Phi_{j,i}^{(t)}\}(j \in \{\pm 1\}, i \in \mathcal{D}_2)$ as $\{y_t\}$, we can leverage Lemma 5.1 in conjunction with specific techniques to yield the subsequent results:

**Lemma 5.2.** *For single-task learning, the iteration numbers $P_j$ for task-specific feature learning metric $\Psi_j^{(t)}$ to reach $\Theta(1/m)$ and $T_i$ for noise memorization metric $\Phi_{y_{i,1},i}^{(t)}$ to reach $\Theta(1/m)$ follow these orders:*

$$P_j = \widetilde{O}(\sigma_0^{2-q}/\eta), \qquad \forall j \in \{\pm 1\}, \qquad T_i = \widetilde{O}(n(\sqrt{s}\sigma_p\sigma_0)^{2-q}/\eta s\sigma_p^2), \qquad \forall i \in \mathcal{D}_1.$$

*The orders of $\Phi_{j,i}^{(t)}$ and $\Lambda_j^{(t)}$ are as follows:*

$$\Phi_{j,i}^{(t)} = \widetilde{O}(\sqrt{s}\sigma_p\sigma_0), \quad \forall j \in \{\pm 1\}, i \in \mathcal{D}_2, t \leq \max_{j \in \{\pm 1\}} P_j, \quad \Lambda_j^{(t)} = \widetilde{O}(\sigma_0), \quad \forall j \in \{\pm 1\}, t \leq \max_{i \in \mathcal{D}_1} T_i.$$

This implies that at the end of the feature learning stage in single-task learning: For data with task-shared feature patches (in $\mathcal{D}_1$), the inner products of neurons with noises will reach a constant level, while the inner products with task-shared features remain small. For data with task-specific feature patches (in $\mathcal{D}_2$), the inner products of neurons with task-specific features will reach a constant level, while the inner products with noises remain small.

For multi-task learning during the pattern learning stage, we have these inequalities:

$$\Psi_{j,k}^{(t)} \geq \Psi_{j,k}^{(t-1)} + \frac{\eta}{2K\rho^{q-1}}\Theta((\Psi_{j,k}^{(t-1)})^{q-1}), \forall j \in \{\pm 1\}, k \in [K], t \leq P_j;$$

$$\Phi_{j,i}^{(t)} \leq \Phi_j^{(t-1)} + \frac{\eta}{K\rho^{q-1}}\Theta(s\sigma_p^2/n)\Theta((\Phi_j^{(t-1)} + \widetilde{\Theta}(\sigma_0^{\frac{2}{3}}))^{q-1}), \forall j \in \{\pm 1\}, i \in \mathcal{D}_2, t \leq \max_{j \in \{-1,1\}} P_j;$$

$$\Phi_{j,i}^{(t)} \leq \Phi_j^{(t-1)} + \frac{\eta}{K\rho^{q-1}}\Theta(s\sigma_p^2/n)\Theta((\Phi_j^{(t-1)} + \widetilde{\Theta}(\sigma_0^{\frac{2}{3}}))^{q-1}), \forall j \in \{\pm 1\}, i \in \mathcal{D}_1, t \leq \max_{j \in \{-1,1\}} Q_j;$$

$$\Lambda_j^{(t)} \geq \Lambda_j^{(t-1)} + \frac{\eta\alpha^q}{\rho^{q-1}}\Theta((\Lambda_j^{(t-1)})^{q-1}) - \widetilde{\Theta}((\sqrt{s}\sigma_p\sigma_0)^{q-1}), \forall j \in \{\pm 1\}, \forall t \leq Q_j;$$

Taking $\{\Psi_{j,k}^{(t)}\}, \{\Lambda_j^{(t)}\}(j \in \{\pm 1\}, k \in [K])$ as $\{x_t\}$, and $\{\Phi_{j,i}^{(t)}\}(j \in \{\pm 1\})$ as $\{y_t\}$, we can once more apply Lemma 5.1 with specific techniques to derive the following results:

**Lemma 5.3.** *For multi-task learning, the following orders of iteration numbers $P_j$ for task-specific feature learning metric $\Psi_j^{(t)}$ reaching $\Theta(1/m)$ and of iteration number $Q_j$ for task-shared feature learning metric $\Lambda_j^{(t)}$ reaching $\Theta(1/m)$ hold:*

$$\textcolor{blue}{P_j = \widetilde{O}(K\sigma_0^{2-q}/\eta), \quad Q_j = \widetilde{O}(\alpha^{-q}\sigma_0^{2-q}/\eta), \qquad \forall j \in \{\pm 1\}.}$$

*And following order of $\Phi_{j,i}^{(t)}$ holds:*

$$\Phi_{j,i}^{(t)} = \widetilde{O}(\sqrt{s}\sigma_p\sigma_0), \qquad \forall j \in \{\pm 1\}, \forall (i \in \mathcal{D}_2, t \leq \max_{j \in \{\pm 1\}} P_j), \forall (i \in \mathcal{D}_1, t \leq \max_{j \in \{\pm 1\}} Q_j).$$

This implies that at the end of the feature learning stage in multi-task learning: For data with task-shared feature patches (in $\mathcal{D}_1$), the inner products of neurons with task-shared features will reach a constant level, while the inner products with noises are still small. For data with task-specific feature patches (in $\mathcal{D}_2$), the inner products of neurons with task-specific features will reach a constant level, while the inner products with noises stay small.

### 5.3 REGULARIZATION STAGE

In the regularization stage, which follows the acquisition of basic data directions, weight decay plays a crucial role in retaining learned knowledge and driving the model towards convergence. Consequently, a single-task learning model converges while retaining some noise and loses information related to shared features. This ultimately results in a constant lower bound for generalization loss. In contrast, a multi-task learning model accommodates all features and achieves nearly perfect test error.

**Lemma 5.4** (Maintain the pattern). *In the regularization stage, for single-task learning, it holds that*

$$\Phi_{y_{i,1},i}^{(t)} = \widetilde{\Theta}(1), \qquad \forall i \in \mathcal{D}_1, \qquad\qquad \Psi_j^{(t)} = \widetilde{\Theta}(1), \qquad \forall j \in \{\pm 1\},$$

$$\Phi_{j,i}^{(t)} \leq \widetilde{\Theta}(\sqrt{s}\sigma_p\sigma_0), \qquad \forall i \in \mathcal{D}_2, j \in \{\pm 1\}, \qquad \Lambda_j^{(t)} \leq \widetilde{\Theta}(\sigma_0), \qquad \forall j \in \{\pm 1\}.$$

*For multi-task learning, it holds that*

$$\Psi_j^{(t)} = \widetilde{\Theta}(1), \Lambda_j^{(t)} = \widetilde{\Theta}(1), \quad \forall j \in \{\pm 1\}, \quad \Phi_{j,i}^{(t)} \leq \widetilde{\Theta}(\sqrt{s}\sigma_p\sigma_0), \quad \forall j \in \{\pm 1\}, i \in [n].$$

The first part of this lemma implies that, during the regularization stage in single-task learning, for data with task-shared feature patches, the inner products of neurons with noise will remain constant, while the inner products with task-shared features will remain at a small magnitude. For data with task-specific feature patches, the inner products of neurons with task-specific features will continue at a constant level, while the inner products with noise remain small.

The second part of this lemma implies that, during the regularization stage in multi-task learning, for data with task-shared feature patches, the inner products of neurons with task-shared features will remain constant, while the inner products with noises will remain at a small magnitude. For data with task-specific feature patches, the inner products of neurons with task-specific features will continue at a constant level, while the inner products with noise remain small.

## 6 EXPERIMENTS

In this section, we present experiment results on both synthetic and real data in verification of our theory.

### 6.1 SYNTHETIC EXPERIMENTS

We consider a two-layer CNN model as outlined in (1) with $m = 10$ and $q = 3$. We present the results of the experiment under the following two data settings that strictly follows Definition 3.2. In Appendix A, we additionally present a generalized setting to demonstrate that our theoretical results can be extended to more general cases.

- Setting 1: we let the number of tasks $K = 3$, number of data $N = 45$, patch dimension $d = 100$, and number of patches $H = 2$. Moreover, we set the parameters $\alpha = 0.8$, $p = 0.2$ and $s = 1$.
- Setting 2: keeping all other parameters the same as setting 1, we let $d = 200$ and $s = 2$.

Adhering to Definition 3.2, all three tasks have a shared feature $\mathbf{v}$ and respectively has a specific feature $\mathbf{v}_k$. Both test and training data are generated under identical settings and share the same size. The models are trained to loss convergence as per (2) for single-task learning and (3) for multi-task learning, using gradient descent as in (4), with parameters set at $\lambda = 0.01$ and $\eta = 0.1$. For single-task training, the model is trained on the same data of task $k$ in our generated multi-task training data. As we show in Figure 2, we train the models until the loss converged for both single-task learning and multi-task learning. The comparison results in setting 1 and 2 for models under multi-task and single-task training are presented in Table 1, using identical test data from task $k$. In alignment with our theoretical results of Theorems 4.1 and 4.2, we can observe a distinct gap in test accuracy between the two learning schemes across the all tasks in both settings.

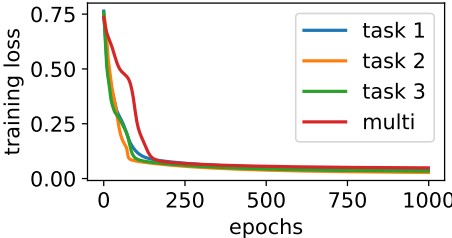

Figure 2: Change of training loss in setting 1 for single-task and multi-task learning with regard to training epochs. As shown, we train until all losses converged.

Table 1: Synthetic experiment results on setting 1 and 2. We report the test accuracy (%) of task $k$.

| Setting 1 | Single-task | Multi-task |
|---|---|---|
| Task 1 | 92.31 | 100 |
| Task 2 | 81.25 | 100 |
| Task 3 | 87.50 | 100 |

| Setting 2 | Single-task | Multi-task |
|---|---|---|
| Task 1 | 88.24 | 100 |
| Task 2 | 64.29 | 100 |
| Task 3 | 78.57 | 100 |

### 6.2 REAL DATA EXPERIMENTS

We further support our theoretical findings with the following real-data experiments. Recall our theory finds that (1) multi-task learning reaches better generalization result, due to (2) its better learning of the shared feature. We use real data to confirm both points.

**Dataset.** We consider CIFAR10-C (Hendrycks & Dietterich, 2018) for our real data experiments. In specifics, CIFAR10-C is a corrupted CIFAR10 benchmark dataset, with 19 different corruptions applied to the original testset of CIFAR10. In appendix, we also present experiments on MNIST-C (Mu & Gilmer, 2019), similar corruption datasets used in previous literature (Chen et al., 2022a) investigating multi-task learning for different theoretical perspectives. Due to the nature of having

the same original dataset but differing significantly in the applied corruptions, the classification problem within each corruption well fits our data model containing both task-specific feature and shared feature. Therefore, we consider learning on each specific corruption as the single-task setting and learning on a union of corruptions as the multi-task setting. We make the **same** split on each corruption data, letting the first $8,000$ images as the training data and the last $2,000$ as the test data. Lastly, we let the severity value of corruptions to be $5$.

**Model.** For the model architecture, we consider ResNet18 as our network. Models for single-task and multi-task learning are all trained to convergence on training loss using SGD optimizer with a learning rate of $0.1$, momentum of $0.9$ and weight decay of 5e-4.

**Multi-task learning outperforms single-task learning.** Each corruption type in the classification problem is treated as a distinct task. We begin with considering single-task training on the 9 tasks specified in Table 5. Moreover, we consider multi-task learning on the union of the training data of the 9 tasks. Again, we note that the train-test splits are the same for all tasks. Lastly, multi-task learning are evaluated on the test data for each single task to compare with single-task learning. Table 5 shows that multi-task learning indeed outperforms single-task learning on each of the specific task.

**Multi-task learning better learns the shared feature.** We further present the following experiment to confirm our theoretical reasoning on why multi-task learning outperforms single-task learning. That is, we demonstrate that multi-task learning can better learn the shared feature. Table 3 presents the test accuracy of both single-task and multi-task training across various test data on unseen tasks. We make the following two observations. Firstly, while single-task learning manages to maintain a decent accuracy on unseen tasks, there remains a distinguished gap in performance compared to that on the test data of its own task. This indicates the existence of both shared feature and distinct specific features. Secondly, multi-task learning consistently outperforms single-task learning across different new tasks. Such an observation provides evidence that multi-task learning can better learn the shared feature that enhances its performance on the other tasks.

Table 2: We report the model's accuracy ($\%$) on the test data with regard to each task (corruption). For single-task learning, we report the model trained on the specific task. For multi-task learning, we report the model trained on the union of the 9 considered tasks.

|  | Bright | Contrast | Defocus | Elastic | Fog | Frost | Gauss blur | Gauss noise | Glass |
|---|---|---|---|---|---|---|---|---|---|
| Single | 83.95 | 78.55 | 82.45 | 78.70 | 82.95 | 81.50 | 84.20 | 76.80 | 75.25 |
| Multi | **84.75** | **85.90** | **84.70** | **81.15** | **84.05** | **84.15** | **84.70** | **82.15** | **80.35** |

Table 3: We report the model's accuracy ($\%$) on the test data with regard to each unseen task (corruption). For single-task learning, we report the model trained on the listed task. For multi-task learning, we report the model trained on the union of the 9 considered tasks.

|  | Impulse | Jpeg | Motion | Pixelate | Saturate | Shot | Snow | Spatter | Speckle | Zoom |
|---|---|---|---|---|---|---|---|---|---|---|
| Bright | 17.00 | 61.10 | 34.15 | 51.20 | 65.15 | 24.95 | 67.05 | 58.50 | 24.75 | 38.85 |
| Contrast | 13.35 | 34.95 | 39.20 | 37.45 | 23.85 | 18.45 | 27.85 | 28.45 | 18.20 | 41.90 |
| Defocus | 23.40 | 59.30 | 67.95 | 62.05 | 56.80 | 35.15 | 51.70 | 45.00 | 36.05 | 73.40 |
| Elastic | 43.30 | 74.80 | 64.80 | 77.85 | 66.35 | 56.00 | 67.25 | 71.30 | 55.45 | 71.20 |
| Fog | 26.35 | 62.20 | 62.80 | 46.70 | 50.40 | 24.25 | 60.45 | 60.70 | 26.65 | 62.35 |
| Frost | 38.40 | 69.25 | 54.85 | 69.50 | 61.60 | 58.20 | 74.65 | 71.95 | 57.15 | 57.80 |
| Gauss blur | 14.45 | 44.90 | 60.15 | 48.45 | 49.55 | 15.55 | 34.35 | 36.05 | 15.55 | 67.95 |
| Gauss noise | 67.55 | 71.65 | 43.00 | 67.05 | 66.65 | 77.05 | 65.70 | 67.55 | 75.25 | 48.50 |
| Glass | 43.85 | 73.20 | 66.35 | 77.10 | 62.30 | 57.30 | 66.30 | 69.35 | 56.30 | 68.25 |
| Multi | **77.50** | **78.80** | **78.00** | **82.20** | **78.75** | **82.15** | **79.55** | **77.40** | **82.20** | **82.55** |

## 7 CONCLUSION AND FUTURE WORK

In this paper, we theoretically studied how multi-task learning reaches better generalization results than single-task learning. Our theoretical results showed that, when both single-task learning and multi-task learning reach zero training error, multi-task learning has a much better generalization performance than single-task learning on the test data. We explain the mechanism of the improvement by separating the learning process into different stages appropriately and analyzing the different situations of feature learning and noise memorization in each stage. Our proof reveals how

learning on multiple tasks can help raise the impact of features in data, resulting in stronger generalization capabilities. Experiments on both synthetic data and real data verify our theory.

Our findings suggest several interesting future research directions. An important direction is to extend our analysis to a broader range of conditions. This may include extending two-layer neural networks to deeper neural networks, exploring different data types, and extending related learning schemes such as meta-learning. A comprehensive exploration of generalization capabilities across diverse scenarios promises to furnish deeper insights into the underlying landscape.

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

# A    ADDITIONAL EXPERIMENTS

## A.1    SYNTHETIC DATA

**Inner product.**   To corroborate our theoretical analysis, which delineates the two-stage training process described in Subsection 5.2 and 5.3, we have plotted the inner products between the model weight vectors and both the key feature vectors and noises identified in our study, i.e., $\Lambda_j^{(t)}$, $\Psi_j^{(t)}$ and $\Phi_{j,i}^{(t)}$ defined in 5.1.

We consider setting 1 for these experiments, and took the average of all noise vectors in training data to show the overall trend of noise learning. In the case of single-task learning, our observations confirm that the model primarily learns the task-specific feature, while the learning of noise surpassed the learning of the shared feature. Conversely, multi-task learning demonstrates a successful learning of the shared feature, surpassing the learning of the noise feature.

Furthermore, the learning trends observed in our synthetic experiments are consistent with our two-stage training analysis. In single-task learning, the learning processes of the task-specific feature and noise are going through the pattern learning stage before approximate 50th and 350th epoch respectively, in which the inner products are growing rapidly; and they both keep steady and enter the regularization stage after the turning points. And the inner product with the task-shared feature remains near initialization. In multi-task learning, the inner products with all features are increasing fast in the pattern learning stage before around 250th epoch, until each of them reach a certain level and keep the same magnitude after that in the regularization stage. And the learning level of noises stays at a low level.

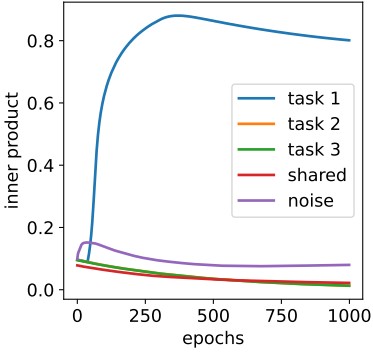

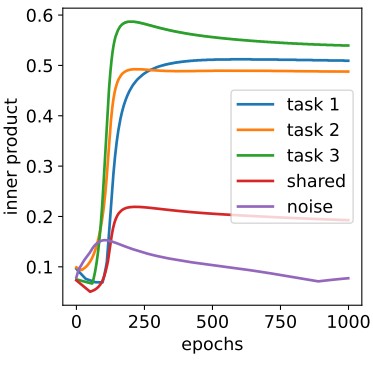

(a) Single-task training (task 1)          (b) Multi-task training

Figure 3: We demonstrate the maximum inner product over the model weight vectors with the feature vector. We compare the growth of these quantities when training with weight decay or training without weight decay. It can be observed that training without weight decay will result in the inner product continuously growing, while training with weight decay will not.

**Weight decay.**   We also investigate the training process with or without weight decay on the synthetic data. The results are illustrated in Figure 4. From both figures, we can see that although there are turning points in the dynamics of inner products between weight node vectors and features both with and without weight decay, the curves after the turning points are obviously different between these two settings.

With weight decay, which is the setting considered in our original model, the learning processes of features enter the regularization stage after the turning points, and the inner products with features basically stay still, and even decrease a little. Without weight decay, however, the inner products with features keep increasing at lower speeds, so there are not very significant regularization stages in these situations.

**Setting 3.**   We consider a more realistic setting without restricting orthogonality on the noise patches across data points. Keeping the same parameters as in Setting 1, we let $K = 5$, $N = 75$, and $s = 94$. In other words, the noise patches are random noise vectors only orthogonal to the feature vectors. The results are shown in Table 4. Although the noise patches in this setting do not enforce sparsity

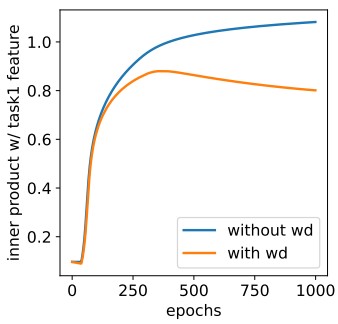 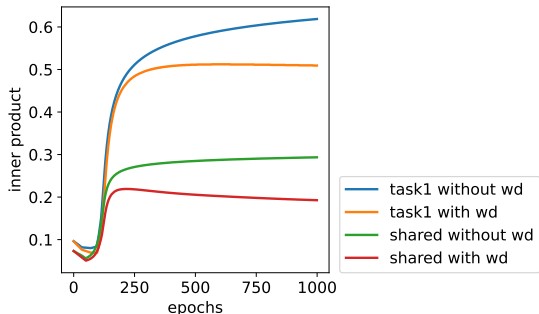

(a) Single-task training (task 1)       (b) Multi-task training

Figure 4: We demonstrate the maximum inner product over the model weight vectors with the feature vector. We compare the growth of these quantities when training with weight decay or training without weight decay. It can be observed that training without weight decay will result in the inner product continuously growing, while training with weight decay will not.

and orthogonality, a clear difference in test accuracy is evident across all tasks. Multi-task learning consistently yields better performance compared to single-task learning.

Table 4: Synthetic experiment results on setting 3. We report the test accuracy (%) of task $k$.

|  | Single-task | Multi-task |
|---|---|---|
| Task 1 | 81.25 | 100 |
| Task 2 | 31.25 | 100 |
| Task 3 | 75.00 | 100 |
| Task 4 | 75.00 | 100 |
| Task 5 | 73.33 | 100 |

## A.2 CIFAR10-C

We conduct additional experiments to confirm the analysis of our theory, especially the different capabilities of learning the task-shared feature between single-task learning and multi-task learning and different performances between two stages during the learning process on both task-shared feature and task-specific feature. Since in the CIFAR10-C dataset, one can not explicitly separate task-shared feature and task-specific feature, so we try to use the test errors under various training and testing situations to represent the learning process and the learning results of task-shared feature and task-specific features.

In Figure 5, we present a comparison of test accuracy for models trained on a single task (brightness) and a multi-task framework (encompassing the nine datasets discussed previously). The test data includes impulse noise, an element not present in the training data for either the single-task or multi-task models. The performance on impulse noise serves as an indicator of the model's proficiency in learning shared features. Additionally, we examine test data related to brightness, a corruption encountered in both single-task and multi-task training. This assessment is used to indicate the model's effectiveness in learning task-specific data.

As shown in Figure 5, multi-task learning successfully learned the shared feature while single-task learning failed to do so. Moreover, the trend in the figure aligns with our two-stage analysis. For single-task learning, before around 80th step, the task-specific feature is learned relatively rapidly, which corresponds to our pattern learning stage; after around 80th step, it basically remains at the same level, which corresponds to our regularization stage. And the learning level of the task-shared feature stays low. For multi-task learning, before around 50th step, both the task-specific feature and the task-shared feature are gained rapidly by the neural network, which matches our pattern learning stage; after around 50th stage, they both keep at a consistent level, which matches our regularization stage.

**Less Training Data.** Previously we considered a train-test split of $80\%$ and $20\%$. Here, we additionally consider the train-test split of $60\%$ and $40\%$. The other setting follow our previous settings.

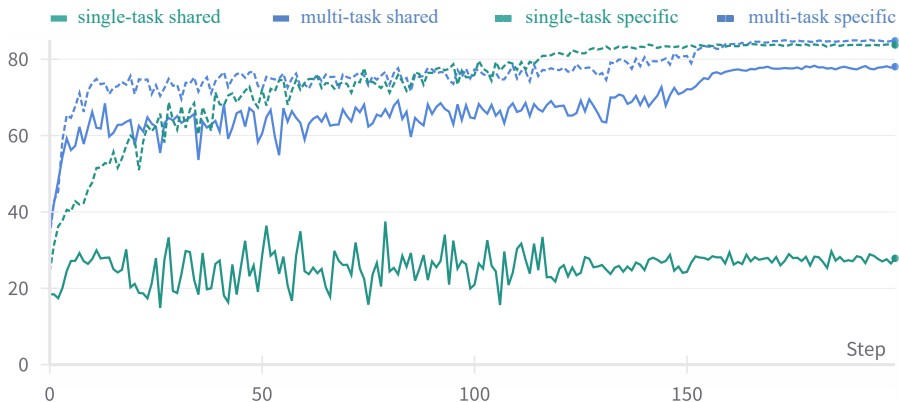

Figure 5: Change of test accuracy with regard to training epochs. We consider single-task training on brightness and multi-task training on the 9 tasks we discussed previously. For the test data, we consider the test data from impulse noise (as an indicator for shared feature learning) and brightness (as an indicator for specific feature learning). We note that both models are not trained on impulse noise.

When training data are less, we can observe that the improvement gain of multi-task learning is more significant.

Table 5: We report the model's accuracy (%) on the test data with regard to each task (corruption). For single-task learning, we report the model trained on the specific task. For multi-task learning, we report the model trained on the union of the 9 considered tasks.

|        | Bright | Contrast | Defocus | Elastic | Fog   | Frost | Gauss blur | Gauss noise | Glass |
|--------|--------|----------|---------|---------|-------|-------|------------|-------------|-------|
| Single | 71.66  | 61.42    | 70.34   | 65.16   | 49.68 | 66.16 | 55.80      | 43.10       | 44.76 |
| Multi  | **80.60** | **80.75** | **80.24** | **77.28** | **79.30** | **79.60** | **79.82** | **77.78** | **75.74** |

### A.3 MNIST-C

**Datasets.** We additionally present the real-data experiments on MNIST-C. Similarly, MNIST-C is a corrupted MNIST benchmark dataset with 15 different corruptions applied to MNIST. We similarly consider learning on each corruption as the single-task setting and learning on a union of corruptions as the multi-task setting. As MNIST-C has train and test split, we do not make our own splits.

**Models.** For experiments on MNIST-C (Mu & Gilmer, 2019), we consider a CNN architecture with two layers of convolution and two layers of fully connected layers. Specifically, both convolutional layers have kernel size of 3 and stride of 1, each with 32 and 64 output channels. The two linear layers have the dimension of $(9216, 128)$ and $(128, 10)$.

| Training task | Test  | Motion blur | Rotate | Scale | Shear | Stripe | Translate |
|---------------|-------|-------------|--------|-------|-------|--------|-----------|
| Brightness    | 98.98 | 84.50       | 89.53  | 91.83 | 96.34 | 94.17  | 42.46     |
| Canny edges   | 98.69 | 33.64       | 45.61  | 51.56 | 49.68 | 82.63  | 30.02     |
| Dotted line   | 98.94 | 92.79       | 91.24  | 93.65 | 97.29 | 96.05  | 46.03     |
| Fog           | 99.03 | 93.35       | 87.77  | 92.44 | 96.13 | 95.26  | 41.35     |
| Glass blur    | 98.22 | 76.48       | 88.30  | 72.47 | 93.77 | 83.02  | 36.19     |
| Impulse noise | 98.53 | 92.11       | 89.22  | 93.17 | 96.85 | 95.26  | 42.37     |
| Shot noise    | 98.79 | 93.40       | 90.89  | 92.92 | 96.47 | 95.61  | 44.27     |
| Spatter       | 98.74 | 94.47       | 90.98  | 92.63 | 97.22 | 96.95  | 45.68     |
| Zigzag        | 98.66 | 92.18       | 90.99  | 94.13 | 96.69 | 95.95  | 43.02     |
| Multi-task    | 98.65 | **95.72**   | **92.40** | **94.72** | **97.46** | **97.19** | **48.02** |

Table 6: Test accuracy (%) of single-task training and multi-task training on different unseen tasks in MNIST-C. Test denotes the test data that corresponds to each training task. For multi-task learning, we report its performance on the combined test data of its tasks.

# B  PREPARATION FOR THE PROOFS

In this section, we will get some preparations for our proof of main results.

## B.1  GRADIENT CALCULATIONS

By following data assumptions, we can calculate the gradient update rules on single-task learning and multi-task learning respectively.

We consider a two-layer CNN model $F$ using the smoothed ReLU activation function $\sigma(z) = \widetilde{\text{ReLU}}(z)$, where $q \geq 2$. From 1, given the data $(\mathbf{x}, y)$ which belongs to the $k$-th task, the $j$-th output of the neural network can be formulated as

$$F_j(\mathbf{W}, \mathbf{x}) = \sum_{r=1}^{m} \sum_{h=1}^{H} \sigma(\langle \mathbf{w}_{j,r}, \mathbf{x}_h \rangle)$$

So if $(\mathbf{x}, y)$ is the first part of the data, then we have

$$F_j(\mathbf{W}, \mathbf{x}) = \sum_{r=1}^{m} \left[ \sigma(\langle \mathbf{w}_{j,r}, \alpha y \cdot \mathbf{v} \rangle) + \sum_{h=2}^{H} \sigma(\langle \mathbf{w}_{j,r}, \boldsymbol{\xi}_{\mathbf{x},h} \rangle) \right]$$

$$= \sum_{r=1}^{m} \left[ \sigma(\langle \mathbf{w}_{j,r}, \alpha y \cdot \mathbf{v} \rangle) + \sum_{h=2}^{H} \sigma(\langle \mathbf{w}_{j,r}, \boldsymbol{\zeta}_{\mathbf{x},h} - \beta_h y \mathbf{v} \rangle) \right]$$

where $m$ is the width of the network, $\mathbf{w}_{j,r} \in \mathbb{R}^d$ denotes the weight at the $r$-th neuron, and $\mathbf{W}$ is the collection of model weights.

If $(\mathbf{x}, y)$ is the second part of the data which belongs to the $k$-th task, then we have

$$F_j(\mathbf{W}, \mathbf{x}) = \sum_{r=1}^{m} \left[ \sigma(\langle \mathbf{w}_{j,r}, \alpha y \cdot \mathbf{v}_k \rangle) + \sum_{h=2}^{H} \sigma(\langle \mathbf{w}_{j,r}, \boldsymbol{\xi}_{\mathbf{x},h} \rangle) \right]$$

$$= \sum_{r=1}^{m} \left[ \sigma(\langle \mathbf{w}_{j,r}, y \cdot \mathbf{v}_k \rangle) + \sum_{h=2}^{H} \sigma(\langle \mathbf{w}_{j,r}, \boldsymbol{\zeta}_{\mathbf{x},h} - \beta_h y \mathbf{v} \rangle) \right]$$

Combining with optimization objective function 2 for single-task learning, 3 for multi-task learning, and the gradient descent algorithm 4, we can obtain the following formulas.

**Lemma B.1.** *According to the update rule of gradient descent, the feature learning and noise memorization of gradient descent for $(\mathbf{x}_{i,k}, y_{i,k})$ which belongs to the $k$-th task for single-task learning can be formulated by*

$$\left\langle \mathbf{w}_{j,r}^{(t+1)}, j \cdot \mathbf{v} \right\rangle = (1 - \eta\lambda) \left\langle \mathbf{w}_{j,r}^{(t)}, j \cdot \mathbf{v} \right\rangle + \frac{\eta}{n} \cdot j \cdot \left( \sum_{i \in \mathcal{D}_1} \alpha y_{i,k} l_{j,i,k}^{(t)} \sigma' \left( \left\langle \mathbf{w}_{j,r}^{(t)}, \alpha y_{i,k} \cdot \mathbf{v} \right\rangle \right) \right.$$

$$\left. - \sum_{h=2}^{H} \sum_{i=1}^{n} \beta_h y_{i,k} l_{j,i,k}^{(t)} \sigma' \left( \left\langle \mathbf{w}_{j,r}^{(t)}, \boldsymbol{\zeta}_{i,k,h} - \beta_h y_{i,k} \mathbf{v} \right\rangle \right) \right)$$

$$= (1 - \eta\lambda) \left\langle \mathbf{w}_{j,r}^{(t)}, j \cdot \mathbf{v} \right\rangle + \frac{\eta}{n} \cdot j \cdot \left( \sum_{i \in \mathcal{D}_1} \alpha y_{i,k} l_{j,i,k}^{(t)} \sigma' \left( \left\langle \mathbf{w}_{j,r}^{(t)}, \alpha y_{i,k} \cdot \mathbf{v} \right\rangle \right) \right.$$

$$\left. - \sum_{h=2}^{H} \sum_{i=1}^{n} \beta_h y_{i,k} l_{j,i,k}^{(t)} \sigma' \left( \left\langle \mathbf{w}_{j,r}^{(t)}, \boldsymbol{\xi}_{i,k,h} \right\rangle \right) \right)$$

$$\left\langle \mathbf{w}_{j,r}^{(t+1)}, j \cdot \mathbf{v}_k \right\rangle = (1 - \eta\lambda) \left\langle \mathbf{w}_{j,r}^{(t)}, j \cdot \mathbf{v}_k \right\rangle + \frac{\eta}{n} \cdot j \cdot \left( \sum_{i \in \mathcal{D}_2} y_{i,k} l_{j,i,k}^{(t)} \sigma' \left( \left\langle \mathbf{w}_{j,r}^{(t)}, y_{i,k} \cdot \mathbf{v}_k \right\rangle \right) \right)$$

$$\left\langle \mathbf{w}_{y_{i,k},r}^{(t+1)}, \boldsymbol{\zeta}_{i,k,h} \right\rangle = (1 - \eta\lambda) \left\langle \mathbf{w}_{y_{i,k},r}^{(t)}, \boldsymbol{\zeta}_{i,k,h} \right\rangle$$

$$+ \frac{\eta}{n} \sum_{g=2}^{H} \sum_{s=1}^{n} l_{y_{i,k},s,k}^{(t)} \sigma' \left( \left\langle \mathbf{w}_{y_{i,k},r}^{(t)}, \boldsymbol{\xi}_{s,k,g} \right\rangle \right) \langle \boldsymbol{\zeta}_{s,k,g}, \boldsymbol{\zeta}_{i,k,h} \rangle$$

$$\left\langle \mathbf{w}_{y_{i,k},r}^{(t+1)}, \boldsymbol{\xi}_{i,k,h} \right\rangle = (1-\eta\lambda)\left\langle \mathbf{w}_{y_{i,k},r}^{(t)}, \boldsymbol{\xi}_{i,k,h} \right\rangle - \frac{\eta}{n}\sum_{s\in\mathcal{D}_1}\alpha\beta_h y_{i,k}y_{s,k}l_{y_{i,k},s,k}^{(t)}\sigma'\left(\left\langle \mathbf{w}_{y_{i,k},r}^{(t)}, \alpha y_{s,k}\mathbf{v} \right\rangle\right)$$

$$+ \frac{\eta}{n}\sum_{g=2}^{H}\sum_{s=1}^{n}l_{y_{i,k},s,k}^{(t)}\sigma'\left(\left\langle \mathbf{w}_{y_{i,k},r}, \boldsymbol{\xi}_{s,k,g} \right\rangle\right)\left\langle \boldsymbol{\xi}_{s,k,g}, \boldsymbol{\xi}_{i,k,h} \right\rangle$$

*Proof.* According to our model, the training objective for single task learning given training data $\{(\mathbf{x}_{i,k}, y_{i,k})\}_{i=1}^{n}$ is the empirical loss function with weight decay,

$$L_k(\mathbf{W}) = \frac{1}{n}\sum_{i=1}^{n}L_{i,k}(\mathbf{W}) + \frac{\lambda}{2}\|\mathbf{W}\|_{\mathrm{F}}^2$$

where $L_{i,k}(\mathbf{W}) = -\log\frac{e^{F_{i,k}(\mathbf{W},\mathbf{x}_{i,k})}}{\sum_{j\in\{-1,1\}}e^{F_j(\mathbf{W},\mathbf{x}_{i,k})}}$ denotes the individual loss for the data $(\mathbf{w}_{i,k}, y_{i,k})$.
Since we are using gradient descent, the update rule of parameters is

$$\mathbf{w}_{j,r}^{(t+1)} = \mathbf{w}_{j,r}^{(t)} - \eta\cdot\nabla_{\mathbf{w}_{j,r}^{(t)}}L_k\left(\mathbf{W}^{(t)}\right)$$

Combining these two parts, we can obtain that

$$\mathbf{w}_{j,r}^{(t+1)} = \mathbf{w}_{j,r}^{(t)} - \eta\nabla_{\mathbf{w}_{j,r}^{(t)}}L_k\left(\mathbf{W}^{(t)}\right)$$

$$= \mathbf{w}_{j,r}^{(t)} - \eta\nabla_{\mathbf{w}_{j,r}^{(t)}}\left(\frac{1}{n}\sum_{i=1}^{n}L_{i,k}\left(\mathbf{W}^{(t)}\right) + \frac{\lambda}{2}\|\mathbf{W}\|_{\mathrm{F}}^2\right)$$

$$= \mathbf{w}_{j,r}^{(t)} - \frac{\eta}{n}\sum_{i=1}^{n}\nabla_{\mathbf{w}_{j,r}}L_{i,k}\left(\mathbf{W}^{(t)}\right) - \frac{\eta\lambda}{2}\nabla_{\mathbf{w}_{j,r}^{(t)}}\left\|\mathbf{W}^{(t)}\right\|_{\mathrm{F}}^2$$

$$= \mathbf{w}_{j,r}^{(t)} + \frac{\eta}{n}\sum_{i=1}^{n}\nabla_{\mathbf{w}_{j,r}^{(t)}}\log\frac{e^{F_{y_{i,k}}(\mathbf{W}^{(t)},\mathbf{x}_{i,k})}}{\sum_{l\in\{-1,1\}}e^{F_l(\mathbf{W}^{(t)},\mathbf{x}_{i,k})}} - \frac{\eta\lambda}{2}\nabla_{\mathbf{w}_{j,r}^{(t)}}\left\|\mathbf{W}^{(t)}\right\|_{\mathrm{F}}^2$$

$$= \mathbf{w}_{j,r}^{(t)} + \frac{\eta}{n}\sum_{i:y_{i,k}=j}\frac{e^{F_{-j}(\mathbf{W}^{(t)},\mathbf{x}_{i,k})}}{\sum_{l\in\{-1,1\}}e^{F_l(\mathbf{W}^{(t)},\mathbf{x}_{i,k})}}\nabla_{\mathbf{w}_{j,r}^{(t)}}F_j\left(\mathbf{W}^{(t)},\mathbf{x}_{i,k}\right)$$

$$+ \frac{\eta}{n}\sum_{i:y_{i,k}=-j}\frac{-e^{F_j(\mathbf{W}^{(t)},\mathbf{x}_{i,k})}}{\sum_{l\in\{-1,1\}}e^{F_l(\mathbf{W}^{(t)},\mathbf{x}_{i,k})}}\nabla_{\mathbf{w}_{j,r}^{(t)}}F_j\left(\mathbf{W}^{(t)},\mathbf{x}_{i,k}\right) - \eta\lambda\mathbf{w}_{j,r}^{(t)}$$

$$= (1-\eta\lambda)\mathbf{w}_{j,r}^{(t)} + \frac{\eta}{n}\sum_{i=1}^{n}l_{j,i,k}^{(t)}\nabla_{\mathbf{w}_{j,r}^{(t)}}F_j^{(\mathbf{W}^{(t)},\mathbf{x}_{i,k})}$$

$$= (1-\eta\lambda)\mathbf{w}_{j,r}^{(t)} + \frac{\eta}{n}\sum_{i\in\mathcal{D}_1}l_{j,i,k}^{(t)}\nabla_{\mathbf{w}_{j,r}^{(t)}}\left(\sigma\left(\left\langle \mathbf{w}_{j,r}^{(t)}, \alpha y_{i,k}\mathbf{v} \right\rangle\right) + \sum_{h=2}^{H}\sigma\left(\left\langle \mathbf{w}_{j,r}^{(t)}, \boldsymbol{\xi}_{i,k,h} \right\rangle\right)\right)$$

$$+ \frac{\eta}{n}\sum_{i\in\mathcal{D}_2}l_{j,i,k}^{(t)}\nabla_{\mathbf{w}_{j,r}^{(t)}}\left(\sigma\left(\left\langle \mathbf{w}_{j,r}^{(t)}, y_{i,k}\mathbf{v}_k \right\rangle\right) + \sum_{h=2}^{H}\sigma\left(\left\langle \mathbf{w}_{j,r}^{(t)}, \boldsymbol{\xi}_{i,k,h} \right\rangle\right)\right)$$

$$= (1-\eta\lambda)\mathbf{w}_{j,r}^{(t)} + \frac{\eta}{n}\sum_{i\in\mathcal{D}_1}\alpha y_{i,k}\left(\sigma'\left(\left\langle \mathbf{w}_{j,r}^{(t)}, \alpha y_{i,k}\mathbf{v} \right\rangle\right)\mathbf{v} + \sum_{h=2}^{H}\sigma'\left(\left\langle \mathbf{w}_{j,r}^{(t)}, \boldsymbol{\xi}_{i,k,h} \right\rangle\right)\boldsymbol{\xi}_{i,k,h}\right)$$

$$+ \frac{\eta}{n}\sum_{i\in\mathcal{D}_2}y_{i,k}\left(\sigma'\left(\left\langle \mathbf{w}_{j,r}^{(t)}, y_{i,k}\mathbf{v}_k \right\rangle\right)\mathbf{v}_k + \sum_{h=2}^{H}\sigma'\left(\left\langle \mathbf{w}_{j,r}^{(t)}, \boldsymbol{\xi}_{i,k,h} \right\rangle\right)\boldsymbol{\xi}_{i,k,h}\right)$$

Since $\langle\mathbf{v},\mathbf{v}\rangle = 1$, $\langle\mathbf{v},\mathbf{v}_k\rangle = 0$, $\langle\mathbf{v},\boldsymbol{\xi}_{i,k,h}\rangle = -\beta_h y_{i,k}$, $\langle\mathbf{v}_k,\mathbf{v}_k\rangle = 1$, $\langle\mathbf{v}_k,\boldsymbol{\xi}_{i,k,h}\rangle = 0$, $\langle\boldsymbol{\zeta}_{i,k,h},\mathbf{v}\rangle = 0$, $\langle\boldsymbol{\zeta}_{i,k,h},\boldsymbol{\xi}_{s,k,g}\rangle = \langle\boldsymbol{\zeta}_{i,k,h},\boldsymbol{\zeta}_{s,l,g}\rangle$ according to our data assumption, so by multiplying with task-shared feature, task-specific feature and random noise respectively, we can get the above update rules of each inner product. $\qquad\square$

**Lemma B.2.** *According to the update rule of gradient descent, the feature learning and noise memorization of gradient descent for $(\mathbf{x}_{i,k}, y_{i,k})$ which belongs to the $k$-th task for multi-task learning can be formulated by*

$$
\left\langle \mathbf{w}_{j,r}^{(t+1)}, j \cdot \mathbf{v} \right\rangle = (1 - \eta\lambda) \left\langle \mathbf{w}_{j,r}^{(t)}, j \cdot \mathbf{v} \right\rangle + \frac{\eta}{nK} \cdot j \cdot \left( \sum_{k=1}^{K} \sum_{i \in \mathcal{D}_1} \alpha y_{i,k} l_{j,i,k}^{(t)} \sigma' \left( \left\langle \mathbf{w}_{j,r}^{(t)}, \alpha y_{i,k} \cdot \mathbf{v} \right\rangle \right) \right.
$$
$$
\left. - \sum_{k=1}^{K} \sum_{h=2}^{H} \sum_{i=1}^{n} \beta_h y_{i,k} l_{j,i,k}^{(t)} \sigma' \left( \left\langle \mathbf{w}_{j,r}^{(t)}, \boldsymbol{\zeta}_{i,k,h} - \beta_h y_{i,k} \mathbf{v} \right\rangle \right) \right)
$$
$$
= (1 - \eta\lambda) \left\langle \mathbf{w}_{j,r}^{(t)}, j \cdot \mathbf{v} \right\rangle + \frac{\eta}{nK} \cdot j \cdot \left( \sum_{k=1}^{K} \sum_{i \in \mathcal{D}_1} \alpha y_{i,k} l_{j,i,k}^{(t)} \sigma' \left( \left\langle \mathbf{w}_{j,r}^{(t)}, \alpha y_{i,k} \cdot \mathbf{v} \right\rangle \right) \right.
$$
$$
\left. - \sum_{k=1}^{K} \sum_{h=2}^{H} \sum_{i=1}^{n} \beta_h y_{i,k} l_{j,i,k}^{(t)} \sigma' \left( \left\langle \mathbf{w}_{j,r}^{(t)}, \boldsymbol{\xi}_{i,k,h} \right\rangle \right) \right)
$$
$$
\left\langle \mathbf{w}_{j,r}^{(t+1)}, j \cdot \mathbf{v}_k \right\rangle = (1 - \eta\lambda) \left\langle \mathbf{w}_{j,r}^{(t)}, j \cdot \mathbf{v}_k \right\rangle + \frac{\eta}{nK} \cdot j \cdot \left( \sum_{i \in \mathcal{D}_2} y_{i,k} l_{j,i,k}^{(t)} \sigma' \left( \left\langle \mathbf{w}_{j,r}^{(t)}, y_{i,k} \cdot \mathbf{v}_k \right\rangle \right) \right)
$$
$$
\left\langle \mathbf{w}_{j,r}^{(t+1)}, \boldsymbol{\zeta}_{i,k,h} \right\rangle = (1 - \eta\lambda) \left\langle \mathbf{w}_{j,r}^{(t)}, \boldsymbol{\zeta}_{i,k,h} \right\rangle
$$
$$
+ \frac{\eta}{nK} \sum_{l=1}^{K} \sum_{g=2}^{H} \sum_{s=1}^{n} l_{y_{i,k},s,l}^{(t)} \sigma' \left( \left\langle \mathbf{w}_{y_{i,k},r}, \boldsymbol{\xi}_{s,l,g} \right\rangle \right) \left\langle \boldsymbol{\zeta}_{s,l,g}, \boldsymbol{\zeta}_{i,k,h} \right\rangle
$$
$$
\left\langle \mathbf{w}_{y_{i,k},r}^{(t+1)}, \boldsymbol{\xi}_{i,k,h} \right\rangle = (1 - \eta\lambda) \left\langle \mathbf{w}_{y_{i,k},r}^{(t)}, \boldsymbol{\xi}_{i,k,h} \right\rangle
$$
$$
- \frac{\eta}{nK} \sum_{l=1}^{K} \sum_{s \in \mathcal{D}_1} \alpha \beta_h y_{s,l} y_{i,k} l_{y_{i,k},s,l}^{(t)} \sigma' \left( \left\langle \mathbf{w}_{y_{i,k},r}^{(t)}, \alpha y_{s,l} \cdot \mathbf{v} \right\rangle \right)
$$
$$
+ \frac{\eta}{nK} \sum_{l=1}^{K} \sum_{g=2}^{H} \sum_{s=1}^{n} l_{y_{i,k},s,l}^{(t)} \sigma' \left( \left\langle \mathbf{w}_{y_{i,k},r}, \boldsymbol{\xi}_{s,l,g} \right\rangle \right) \left\langle \boldsymbol{\xi}_{s,l,g}, \boldsymbol{\xi}_{i,k,h} \right\rangle
$$

*Proof.* According to our model, the training objective for single task learning given training data $\{(\mathbf{x}_{i,k}, y_{i,k})\}_{i=1}^{n}$ is the empirical loss function with weight decay,

$$
L(\mathbf{W}) = \frac{1}{nK} \sum_{k=1}^{K} \sum_{i=1}^{n} L_{i,k}(\mathbf{W}) + \frac{\lambda}{2} \|\mathbf{W}\|_{\mathrm{F}}^2
$$

where $L_{i,k}(\mathbf{W}) == -\log \frac{e^{F_{i,k}(\mathbf{W}, \mathbf{x}_{i,k})}}{\sum_{j \in \{-1,1\}} e^{F_j(\mathbf{W}, \mathbf{x}_{i,k})}}$ denotes the individual loss for the data $(\mathbf{w}_{i,k}, y_{i,k})$. Since we are using gradient descent, the update rule of parameters is

$$
\mathbf{w}_{j,r}^{(t+1)} = \mathbf{w}_{j,r}^{(t)} - \eta \cdot \nabla_{\mathbf{w}_{j,r}^{(t)}} L\left(\mathbf{W}^{(t)}\right)
$$

Combining these two parts, we can obtain that

$$
\mathbf{w}_{j,r}^{(t+1)} = \mathbf{w}_{j,r}^{(t)} - \eta \nabla_{\mathbf{w}_{j,r}^{(t)}} L\left(\mathbf{W}^{(t)}\right)
$$
$$
= \mathbf{w}_{j,r}^{(t)} - \eta \nabla_{\mathbf{w}_{j,r}^{(t)}} \left( \frac{1}{nK} \sum_{k=1}^{K} \sum_{i=1}^{n} L_{i,k}\left(\mathbf{W}^{(t)}\right) + \frac{\lambda}{2} \|\mathbf{W}\|_{\mathrm{F}}^2 \right)
$$
$$
= \mathbf{w}_{j,r}^{(t)} - \frac{\eta}{nK} \sum_{k=1}^{K} \sum_{i=1}^{n} \nabla_{\mathbf{w}_{j,r}} L_{i,k}\left(\mathbf{W}^{(t)}\right) - \frac{\eta\lambda}{2} \nabla_{\mathbf{w}_{j,r}^{(t)}} \left\|\mathbf{W}^{(t)}\right\|_{\mathrm{F}}^2
$$
$$
= \mathbf{w}_{j,r}^{(t)} + \frac{\eta}{nK} \sum_{k=1}^{K} \sum_{i=1}^{n} \nabla_{\mathbf{w}_{j,r}^{(t)}} \log \frac{e^{F_{y_{i,k}}(\mathbf{W}^{(t)}, \mathbf{x}_{i,k})}}{\sum_{l \in \{-1,1\}} e^{F_l(\mathbf{W}^{(t)}, \mathbf{x}_{i,k})}} - \frac{\eta\lambda}{2} \nabla_{\mathbf{w}_{j,r}^{(t)}} \left\|\mathbf{W}^{(t)}\right\|_{\mathrm{F}}^2
$$

$$= \mathbf{w}_{j,r}^{(t)} + \frac{\eta}{nK} \sum_{k=1}^{K} \sum_{i:y_{i,k}=j} \frac{e^{F_{-j}(\mathbf{W}^{(t)}, \mathbf{x}_{i,k})}}{\sum_{l \in \{-1,1\}} e^{F_l(\mathbf{W}^{(t)}, \mathbf{x}_{i,k})}} \nabla_{\mathbf{w}_{j,r}^{(t)}} F_j\left(\mathbf{W}^{(t)}, \mathbf{x}_{i,k}\right)$$

$$+ \frac{\eta}{nK} \sum_{k=1}^{K} \sum_{i:y_{i,k}=-j} \frac{-e^{F_j(\mathbf{W}^{(t)}, \mathbf{x}_{i,k})}}{\sum_{l \in \{-1,1\}} e^{F_l(\mathbf{W}^{(t)}, \mathbf{x}_{i,k})}} \nabla_{\mathbf{w}_{j,r}^{(t)}} F_j\left(\mathbf{W}^{(t)}, \mathbf{x}_{i,k}\right) - \eta\lambda \mathbf{w}_{j,r}^{(t)}$$

$$= (1-\eta\lambda)\mathbf{w}_{j,r}^{(t)} + \frac{\eta}{nK} \sum_{k=1}^{K} \sum_{i=1}^{n} l_{j,i,k}^{(t)} \nabla_{\mathbf{w}_{j,r}}^{(t)} F_j^{(\mathbf{W}^{(t)}, \mathbf{x}_{i,k})}$$

$$= (1-\eta\lambda)\mathbf{w}_{j,r}^{(t)} + \frac{\eta}{nK} \sum_{k=1}^{K} \sum_{i \in \mathcal{D}_1} l_{j,i,k}^{(t)} \nabla_{\mathbf{w}_{j,r}^{(t)}} \left( \sigma\left(\left\langle \mathbf{w}_{j,r}^{(t)}, \alpha y_{i,k}\mathbf{v}\right\rangle\right) + \sum_{h=2}^{H} \sigma\left(\left\langle \mathbf{w}_{j,r}^{(t)}, \boldsymbol{\xi}_{i,k,h}\right\rangle\right) \right)$$

$$+ \frac{\eta}{nK} \sum_{k=1}^{K} \sum_{i \in \mathcal{D}_2} l_{j,i,k}^{(t)} \nabla_{\mathbf{w}_{j,r}^{(t)}} \left( \sigma\left(\left\langle \mathbf{w}_{j,r}^{(t)}, y_{i,k}\mathbf{v}_k\right\rangle\right) + \sum_{h=2}^{H} \sigma\left(\left\langle \mathbf{w}_{j,r}^{(t)}, \boldsymbol{\xi}_{i,k,h}\right\rangle\right) \right)$$

$$= (1-\eta\lambda)\mathbf{w}_{j,r}^{(t)} + \frac{\eta}{nK} \sum_{k=1}^{K} \sum_{i \in \mathcal{D}_1} \alpha y_{i,k} \left( \sigma'\left(\left\langle \mathbf{w}_{j,r}^{(t)}, \alpha y_{i,k}\mathbf{v}\right\rangle\right)\mathbf{v} + \sum_{h=2}^{H} \sigma'\left(\left\langle \mathbf{w}_{j,r}^{(t)}, \boldsymbol{\xi}_{i,k,h}\right\rangle\right)\boldsymbol{\xi}_{i,k,h} \right)$$

$$+ \frac{\eta}{nK} \sum_{k=1}^{K} \sum_{i \in \mathcal{D}_2} y_{i,k} \left( \sigma'\left(\left\langle \mathbf{w}_{j,r}^{(t)}, y_{i,k}\mathbf{v}_k\right\rangle\right)\mathbf{v}_k + \sum_{h=2}^{H} \sigma'\left(\left\langle \mathbf{w}_{j,r}^{(t)}, \boldsymbol{\xi}_{i,k,h}\right\rangle\right)\boldsymbol{\xi}_{i,k,h} \right)$$

Since $\langle \mathbf{v}, \mathbf{v}\rangle = 1$, $\langle \mathbf{v}, \mathbf{v}_k\rangle = 0$, $\langle \mathbf{v}, \boldsymbol{\xi}_{i,k,h}\rangle = -\beta_h y_{i,k}$, $\langle \mathbf{v}_k, \mathbf{v}_l\rangle = 1_{k=l}$, $\langle \mathbf{v}_k, \boldsymbol{\xi}_{i,l,h}\rangle = 0$, $\langle \boldsymbol{\zeta}_{i,l,h}, \mathbf{v}\rangle = 0$, $\langle \boldsymbol{\zeta}_{i,k,h}, \boldsymbol{\xi}_{s,l,g}\rangle = \langle \boldsymbol{\zeta}_{i,k,h}, \boldsymbol{\zeta}_{s,l,g}\rangle$ according to our data assumption, so by multiplying with task-shared feature, task-specific features, and random noises respectively, we can get the above update rules of each inner product. $\qquad\square$

## B.2  DATA STRUCTURE

Now we highlight an essential property of our data distribution $\mathcal{D}$. As previously mentioned, sparsity is a distinctive characteristic of our model. The following lemma from Zou et al. (2021) demonstrates that according to Definition 3.2, all random noise vectors in training samples will have disjoint support sets with high probability.

**Lemma B.3.** *Let $\{(\mathbf{x}_{i,k}, y_{i,k})\}_{(i,k) \in [n] \times [K]}$ be the training dataset generated by Definition 1.1, and $\{\boldsymbol{\zeta}_{i,k,h}\}_{h=2}^{H}$ be the corresponding random noises of $\mathbf{x}_{i,k}$. Then with probability at least $1 - n^{-2}$, $\langle \boldsymbol{\zeta}_{i,k,h}, \boldsymbol{\zeta}_{s,l,g}\rangle = 0$ for all $(i, k, h) \neq (s, l, g)$.*

*Proof.* For any vector $a$, denote $a^u$ its $u$-th coordinate, and let $\mathcal{B}_{i,k,h} = \text{supp}(\boldsymbol{\zeta}_{i,k,h})$ be the support of $\boldsymbol{\zeta}_{i,k,h}$. Then according to the definition of inner product, we have that for any $(i, k, h) \neq (s, l, g)$,

$$\langle \boldsymbol{\zeta}_{i,k,h}, \boldsymbol{\zeta}_{s,l,g}\rangle = \sum_{u=1}^{d} \boldsymbol{\zeta}_{i,k,h}^u \boldsymbol{\zeta}_{s,l,g}^u$$

$$= \sum_{u \in [d] \backslash (\mathcal{B}_{i,k,h} \cup \mathcal{B}_{s,l,g})} \boldsymbol{\zeta}_{i,k,h}^u \boldsymbol{\zeta}_{s,l,g}^u + \sum_{u \in \mathcal{B}_{i,k,h} \backslash \mathcal{B}_{s,l,g}} \boldsymbol{\zeta}_{i,k,h}^u \boldsymbol{\zeta}_{s,l,g}^u$$

$$+ \sum_{u \in \mathcal{B}_{s,l,g} \backslash \mathcal{B}_{i,k,h}} \boldsymbol{\zeta}_{i,k,h}^u \boldsymbol{\zeta}_{s,l,g}^u + \sum_{u \in \mathcal{B}_{i,k,h} \cap \backslash \mathcal{B}_{s,l,g}} \boldsymbol{\zeta}_{i,k,h}^u \boldsymbol{\zeta}_{s,l,g}^u$$

$$= \sum_{u \in \mathcal{B}_{i,k,h} \cap \mathcal{B}_{s,l,g}} \boldsymbol{\zeta}_{i,k,h}^u \boldsymbol{\zeta}_{s,l,g}^u$$

so if $\mathcal{B}_{i,k,h} \cap \mathcal{B}_{s,l,g} = \emptyset$, then $\langle \boldsymbol{\zeta}_{i,k,h}, \boldsymbol{\zeta}_{s,l,g}\rangle = \sum_{u=1}^{d} \boldsymbol{\zeta}_{i,k,h}^u \boldsymbol{\zeta}_{s,l,g}^u = \sum_{u \in \mathcal{B}_{i,k,h} \cap \mathcal{B}_{s,l,g}} \boldsymbol{\zeta}_{i,k,h}^u \boldsymbol{\zeta}_{s,l,g}^u = 0$. Therefore,

$$\mathbb{P}\{\forall (i, k, h), (s, l, g) \in [n] \times [K] \times [2, H], (i, k, h) \neq (s, l, g) : \langle \boldsymbol{\zeta}_{i,k,h}, \boldsymbol{\zeta}_{s,l,g}\rangle = 0\}$$
$$\geq \mathbb{P}\{\forall (i, k, h), (s, l, g) \in [n] \times [K] \times [2, H], (i, k, h) \neq (s, l, g) : \mathcal{B}_{s,l,g} \cap \mathcal{B}_{i,k,h} = \emptyset\}$$
$$= 1 - \mathbb{P}\{\exists (i, k, h), (s, l, g) \in [n] \times [K] \times [2, H], (i, k, h) \neq (s, l, g) : \mathcal{B}_{s,l,g} \cap \mathcal{B}_{i,k,h} \neq \emptyset\}$$

From definition, we can get that

$$\mathbb{P}\left\{\exists\left(i,k,h\right),\left(s,l,g\right)\in[n]\times[K]\times[2,H],,\left(i,k,h\right)\neq\left(s,l,g\right):\mathcal{B}_{s,l,g}\cap\mathcal{B}_{i,k,h}\neq\emptyset\right\}$$
$$=\mathbb{P}\left\{\exists\left(i,k,h\right)\in[n]\times[K]\times[2,H],\left(s,l,g\right)\in[n]\times[K]\times[2,H]\backslash\left\{\left(i,k,h\right)\right\}:\mathcal{B}_{s,l,g}\cap\mathcal{B}_{i,k,h}\neq\emptyset\right\}$$
$$=\mathbb{P}\left\{\exists\left(i,k,h\right)\in[n]\times[K]\times[2,H],j\in\mathcal{B}_{i,k,h},\left(s,l,g\right)\in[n]\times[K]\times[2,H]\backslash\left\{\left(i,k,h\right)\right\}:\boldsymbol{\xi}_{s,l,g}^{j}\neq0\right\}$$

For any fixed $\left(i,k,h\right)\in[n]\times[K]\times[2,H]$ and $j\in\mathcal{B}_{i,k,h}$, then by the model assumption we have

$$\mathbb{P}\left\{\boldsymbol{\xi}_{s,l,g}^{j}\neq0\right\}=\frac{s}{d-K-1}$$

for all $\left(s,l,g\right)\in[n]\times[K]\times[2,H]$, since the first $K+1$ coordinates of the random noises are equal to $0$ according to the definition. Therefore, by the fact that all the noises are independent with each other, we have

$$\mathbb{P}\left\{\exists\left(s,l,g\right)\in[n]\times[K]\times[2,H]\backslash\left\{\left(i,k,h\right)\right\}:\boldsymbol{\xi}_{s,l,g}^{j}\neq0\right\}=1-\left[1-\frac{s}{d-K-1}\right]^{(H-1)nK-1}$$

Applying a union bound over all $\left(i,k,h\right)\in[n]\times[K]\times[2,H]$ and $j\in\mathcal{B}_{i,k,h}$, we obtain

$$\mathbb{P}\left\{\exists\left(i,k,h\right)\in[n]\times[K]\times[2,H],j\in\mathcal{B}_{i,k,h},\left(s,l,g\right)\in[n]\times[K]\times[2,H]\backslash\left\{\left(i,k,h\right)\right\}:\boldsymbol{\xi}_{s,l,g}^{j}\neq0\right\}$$
$$\leq(H-1)nKs\cdot\left\{1-\left[1-\frac{s}{d-K-1}\right]^{(H-1)nK-1}\right\}$$

By the data distribution assumption we have $s\leq\frac{\sqrt{d}}{2HKn^{2}}$, which clearly implies that $\frac{s}{d-K-1}\leq\frac{1}{2}$. Therefore, we have

$$(H-1)nKs\cdot\left\{1-\left[1-\frac{s}{d-K-1}\right]^{(H-1)nK-1}\right\}$$
$$=(H-1)nKs\cdot\left\{1-\exp\left\{[(H-1)nK-1]\log\left(1-\frac{s}{d-K-1}\right)\right\}\right\}$$
$$\leq(H-1)nKs\cdot\left\{1-\exp\left\{[(H-1)nK-1]\cdot\frac{2s}{d-K-1}\right\}\right\}$$
$$\leq(H-1)nKs\cdot\left\{1-\exp\left\{[(H-1)nK-1]\cdot\frac{4s}{d}\right\}\right\}$$
$$\leq(H-1)nKs\cdot\frac{[(H-1)nK-1]\cdot4s}{d}$$
$$\leq HnKs\cdot\frac{HnK\cdot4s}{d}=\frac{4H^{2}n^{2}K^{2}s^{2}}{d}\leq n^{-2}$$

where the first inequality follows by the inequality $\log\left(1-z\right)\geq-2z$ for $z\in\left[0,\frac{1}{2}\right]$, the second inequality follows by $\frac{s}{d-1}\geq\frac{2s}{d}$, the third inequality follows by the inequality $1-\exp\left(-z\right)\leq z$ for $z\in\mathbb{R}$, and the last inequality follows by our model assumption. Combining all the previous results, we have

$$\mathbb{P}\left\{\forall\left(i,k,h\right),\left(s,l,g\right)\in[n]\times[K]\times[2,H],\left(i,k,h\right)\neq\left(s,l,g\right):\langle\boldsymbol{\zeta}_{i,k,h},\boldsymbol{\zeta}_{s,l,g}\rangle=0\right\}$$
$$\geq1-\mathbb{P}\left\{\exists\left(i,k,h\right)\in[n]\times[K]\times[2,H],\left(i,k,h\right)\neq\left(s,l,g\right):\mathcal{B}_{s,l,g}\cap\mathcal{B}_{i,k,h}\neq\emptyset\right\}$$
$$=1-\mathbb{P}\left\{\exists\left(i,k,h\right),\left(s,l,g\right)\in[n]\times[K]\times[2,H],\left(s,l,g\right)\in[n]\times[K]\times[2,H]\backslash\left\{\left(i,k,h\right)\right\},j\in\mathcal{B}_{i,k,h}:\boldsymbol{\xi}_{s,l,g}^{j}\neq0\right\}$$
$$\geq1-(H-1)nK\cdot\left\{1-\left[1-\frac{s}{d-K-1}\right]^{(H-1)nK-1}\right\}$$
$$\geq1-n^{-2}$$

Then we finish the proof. $\qquad\square$

This lemma provides insight into the optimization of the model parameter $\mathbf{W}$ for coordinates without features. Since the support sets do not overlap, each coordinate will be influenced by only one sample. This implies that the update for each coordinate will be primarily determined by the corresponding data. Consequently, the optimization process simplifies to a coordinate-wise landscape, facilitating the analysis of the learning and generalization behavior of gradient descent in both single-task and multi-task learning settings.

### B.3    Proof of Lemma 5.1

*Proof of Lemma 5.1.* Since $x_t$ is positive for each $t \geq 0$, obviously $x_t$ is monotonically increasing, so $x_t \geq x_0$. Then we have

$$x_{t+1} \geq x_t + \eta A x_t^{q-1} \geq \left(1 + \eta A x_0^{q-2}\right) x_t$$
$$\geq \left(1 + \eta A x_0^{q-2}\right)^t x_0$$

Since $\log(1+x) \geq x \log 2$ for all $x \in (0,1)$, then we have

$$T_x \leq \frac{\log(1/x_0)}{\log\left(1 + \eta A x_0^{q-2}\right)} \leq \frac{\log(1/x_0)}{\eta A x_0^{q-2} \log 2}$$

Therefore, we can get that for those $t$ satisfies $y_t \leq 2y_0$, we have that

$$y_{t+1} \leq y_t + \eta B y_t^{q-1} \leq \left(1 + \eta 2^{q-2} B y_0^{q-2}\right)^t y_0$$
$$\leq \exp\left(2^{q-2} \eta B y_0^{q-2} t\right) y_0$$
$$\leq \exp\left(\frac{2^{q-2} \eta B y_0^{q-2} \log(1/x_0)}{\eta A x_0^{q-2}}\right) y_0 \leq 2y_0$$

due to our assumptions. So we finish the proof. $\qquad\square$

## C    Single-Task Learning

For single-task learning, our analysis primarily focuses on the $k$-th task, where we assume without loss of generality that $k = 1$. To prove Theorem 4.1, we rely on the following crucial technical lemmas:

**Lemma C.1** (Convergence Guarantee). *If the step size satisfies $\eta \leq O(\sigma_0)$, then for any $t \geq 0$, it holds that*
$$L_1(\mathbf{W}^{(t+1)}) - L_1(\mathbf{W}^{(t)}) \leq -\frac{\eta}{2}\|\nabla L_1(\mathbf{W}^{(t)})\|_{\mathrm{F}}^2$$

This lemma demonstrates that optimization on the training data using gradient descent with a two-layer convolutional neural network will converge to a solution with a small gradient.

**Lemma C.2** (Generalization Performance of GD). *Let*
$$\mathbf{W}^* = \arg\min_{\{\mathbf{W}^{(1)},...,\mathbf{W}^{(T)}\}} \|\nabla L_1(\mathbf{W}^{(t)})\|_{\mathrm{F}}.$$

*Then by selecting $T = \frac{\mathrm{poly}\left(n, \lambda^{-1}\right)}{\eta}$, for all training data, we have*

$$\frac{1}{n}\sum_{i=1}^{n} \mathbf{1}\left[F_{-y_{i,1}}\left(\mathbf{W}^*, \mathbf{x}_{i,1}\right) \leq F_{y_{i,1}}\left(\mathbf{W}^*, \mathbf{x}_{i,1}\right)\right] = 0$$

*Moreover, in terms of the test data $(\mathbf{x}, y) \sim \mathcal{D}$, we have*

$$\mathbb{P}_{(\mathbf{x},y)\sim\mathcal{D}}\left[F_y\left(\mathbf{W}^*, \mathbf{x}\right) < F_{-y}\left(\mathbf{W}^*, \mathbf{x}\right)\right] \geq \frac{p}{4} - \mathrm{poly}\left(n^{-1}\right).$$

This lemma reveals that even though the neural network can achieve zero training error, it cannot perform well on generalization and has a constant lower bound for test error.

# D PROOF OF LEMMAS IN APPENDIX C

In order to prove Lemma C.1 and Lemma C.2, we need the following technical lemmas.

**Lemma D.1** (Off-diagonal Correlations for Task-specific Feature). *For any $j \in \{-1, 1\}$ and any $t$, it holds that $[\langle \mathbf{w}_{-j,r}^{(t)}, j \cdot \mathbf{v}_1 \rangle]_+ \leq \widetilde{\Theta}(\sigma_0)$.*

**Lemma D.2** (Off-diagonal correlations for Random Noises). *For any data $(\mathbf{x}_{i,1}, y_{i,1})$, any $h \in [2, H]$ and any $t$, it holds that $[\langle \mathbf{w}_{-y_{i,1},r}^{(t)}, \boldsymbol{\zeta}_{i,1,h} \rangle]_+ \leq \widetilde{\Theta}(\sqrt{s}\sigma_p\sigma_0)$.*

**Lemma D.3.** *Suppose the training data is generated according to Definition 3.1 and Definition 3.2. Let $\Lambda_j^{(t)} = \max_{r \in [m]}[\langle \mathbf{w}_{j,r}^{(t)}, j \cdot \mathbf{v} \rangle]_+$, $\Psi_j^{(t)} = \max_{r \in [m]}[\langle \mathbf{w}_{j,r}^{(t)}, j \cdot \mathbf{v}_1 \rangle]_+$, $\Phi_{j,i,h}^{(t)} = \max_{r \in [m]}[\langle \mathbf{w}_{j,r}^{(t)}, \boldsymbol{\zeta}_{i,1,h} \rangle]_+$, $\Phi_{j,i}^{(t)} = \max_{h \in [2,H]} \Phi_{j,i,h}^{(t)}$, and $\Phi_j^{(t)} = \max_{i \in [n]} \Phi_{j,i}^{(t)}$. Then let $P_j$ be the iteration number that $\Psi_j^{(t)}$ reaches $\Theta(1/m)$ for $j \in \{-1, 1\}$, $T_i$ be the iteration number that $\Phi_{y_{i,1},i}^{(t)}$ reaches $\Theta(1/m)$ for $i \in \mathcal{D}_1$, we have $P_j \leq \widetilde{\Theta}(\sigma_0^{2-q}/\eta)$ for all $j \in \{-1, 1\}$ and $T_i \leq \widetilde{\Theta}(n(\sqrt{s}\sigma_p\sigma_0)^{2-q}/\eta d\sigma_p^2)$ for all $i \in \mathcal{D}_1$. Moreover, let $P_0 = \max_{j\{-1,1\}} P_j$ and $T_0 = \max_{i \in \mathcal{D}_1} T_i$. For all $t \geq 0$ and $r \in [m]$ it holds that $\Lambda_j^{(t)} = \widetilde{O}(\sigma_0)$ for all $j \in \{-1, 1\}$, $\Phi_{j,i}^{(t)} = \widetilde{O}(\sqrt{s}\sigma_p\sigma_0)$ for all $j \in \{-1, 1\}$ and $i \in \mathcal{D}_2$, and $[\langle \mathbf{w}_{-j,r}^{(t)}, j \cdot \mathbf{v} \rangle]_+ \leq \widetilde{\Theta}(\sigma_0^{\frac{1}{3}})$ for all $j \in \{-1, 1\}$.*

Now we are ready to prove Lemma C.1 and Lemma C.2.

## D.1 PROOF OF LEMMA C.1

*Proof of Lemma C.1.* The proof is basically relying the smoothness property of the loss function $L_1(\mathbf{W})$ given certain constraints on the inner products with each patch.

Let $\Delta F_{j,i} = F_j\left(\mathbf{W}^{(t+1)}, \mathbf{x}_{i,1}\right) - F_j\left(\mathbf{W}^{(t)}, \mathbf{x}_{i,1}\right)$, we can get that following Taylor expansion on the loss function $L_{i,1}\left(\mathbf{W}^{(t+1)}\right)$,

$$L_{i,1}\left(\mathbf{W}^{(t+1)}\right) - L_{i,1}\left(\mathbf{W}^{(t)}\right) \leq \sum_j \frac{\partial L_{i,1}\left(\mathbf{W}^{(t)}\right)}{\partial F_j\left(\mathbf{W}^{(t),\mathbf{x}_{i,1}}\right)} \cdot \Delta F_{j,i} + \sum_j (\Delta F_{j,i})^2$$

In particular, by Lemma D.1 to Lemma D.3, we know that $\left[\left\langle \mathbf{w}_{j,r}^{(t)}, y_{i,1} \cdot \mathbf{v} \right\rangle\right]_+ \leq \widetilde{\Theta}(\sigma_0)$, $\left[\left\langle \mathbf{w}_{j,r}^{(t)}, y_{i,1} \cdot \mathbf{v}_1 \right\rangle\right]_+ \leq \widetilde{\Theta}(1)$ and $\left[\left\langle \mathbf{w}_{j,r}^{(t)}, \boldsymbol{\xi}_{i,1,h} \right\rangle\right]_+ \leq \widetilde{\Theta}(1)$. Then we can apply first order Taylor expansion to $F_j\left(\mathbf{W}^{(t+1)}, \mathbf{x}_{i,1}\right)$, which requires to characterize the second-order error of the Taylor expansion on $\sigma\left(\left\langle \mathbf{w}_{j,r}^{(t+1)}, y_{i,1} \cdot \mathbf{v} \right\rangle\right)$, $\sigma\left(\left\langle \mathbf{w}_{j,r}^{(t+1)}, y_{i,1} \cdot \mathbf{v}_1 \right\rangle\right)$ and $\sigma\left(\left\langle \mathbf{w}_{j,r}^{(t+1)}, \boldsymbol{\xi}_{i,1,h} \right\rangle\right)$ as the following:

$$\left| \sigma\left(\left\langle \mathbf{w}_{j,r}^{(t+1)}, y_{i,1} \cdot \mathbf{v} \right\rangle\right) - \sigma\left(\left\langle \mathbf{w}_{j,r}^{(t)}, y_{i,1} \cdot \mathbf{v} \right\rangle\right) - \left\langle \nabla_{\mathbf{w}_{j,r}} \sigma\left(\left\langle \mathbf{w}_{j,r}^{(t)}, y_{i,1} \cdot \mathbf{v} \right\rangle\right), \mathbf{w}_{j,r}^{(t+1)} - \mathbf{w}_{j,r}^{(t)} \right\rangle \right|$$
$$\leq \widetilde{\Theta}\left(\left\| \mathbf{w}_{j,r}^{(t+1)} - \mathbf{w}_{j,r}^{(t)} \right\|_2^2\right) = \widetilde{\Theta}\left(\eta^2 \left\| \nabla_{\mathbf{w}_{j,r}} L_1\left(\mathbf{W}^{(t)}\right) \right\|_2^2\right);$$

$$\left| \sigma\left(\left\langle \mathbf{w}_{j,r}^{(t+1)}, y_{i,1} \cdot \mathbf{v}_1 \right\rangle\right) - \sigma\left(\left\langle \mathbf{w}_{j,r}^{(t)}, y_{i,1} \cdot \mathbf{v}_1 \right\rangle\right) - \left\langle \nabla_{\mathbf{w}_{j,r}} \sigma\left(\left\langle \mathbf{w}_{j,r}^{(t)}, y_{i,1} \cdot \mathbf{v}_1 \right\rangle\right), \mathbf{w}_{j,r}^{(t+1)} - \mathbf{w}_{j,r}^{(t)} \right\rangle \right|$$
$$\leq \widetilde{\Theta}\left(\left\| \mathbf{w}_{j,r}^{(t+1)} - \mathbf{w}_{j,r}^{(t)} \right\|_2^2\right) = \widetilde{\Theta}\left(\eta^2 \left\| \nabla_{\mathbf{w}_{j,r}} L\left(\mathbf{W}^{(t)}\right) \right\|_2^2\right);$$

$$\left| \sigma\left(\left\langle \mathbf{w}_{j,r}^{(t+1)}, \boldsymbol{\xi}_{i,1} \right\rangle\right) - \sigma\left(\left\langle \mathbf{w}_{j,r}^{(t)}, \boldsymbol{\xi}_{i,1,h} \right\rangle\right) - \left\langle \nabla_{\mathbf{w}_{j,r}} \sigma\left(\left\langle \mathbf{w}_{j,r}^{(t)}, \boldsymbol{\xi}_{i,1,h} \right\rangle\right), \mathbf{w}_{j,r}^{(t+1)} - \mathbf{w}_{j,r}^{(t)} \right\rangle \right|$$
$$\leq \widetilde{\Theta}\left(\left\| \mathbf{w}_{j,r}^{(t+1)} - \mathbf{w}_{j,r}^{(t)} \right\|_2^2\right) = \widetilde{\Theta}\left(\eta^2 \left\| \nabla_{\mathbf{w}_{j,r}} L\left(\mathbf{W}^{(t)}\right) \right\|_2^2\right);$$

Then combining the above bounds for every $r \in [m]$, we can get the following bound for $\Delta F_{j,i}$

$$\left| \Delta F_{j,i} - \left\langle \nabla_{\mathbf{W}} F_j\left(\mathbf{W}^{(t)}, \mathbf{x}_{i,1}\right), \mathbf{W}^{(t+1)} - \mathbf{W}^{(t)} \right\rangle \right| \leq \widetilde{\Theta}\left(\eta^2 \sum_{r \in [m]} \left\| \nabla_{\mathbf{w}_{j,r}} L_1\left(\mathbf{W}^{(t)}\right) \right\|_2^2\right)$$

$$\leq \widetilde{\Theta} \left( \eta^2 \left\| \nabla L_1 \left( \mathbf{W}^{(t)} \right) \right\|_{\mathrm{F}}^2 \right)$$

Moreover, since $\left\langle \mathbf{w}_{j,r}^{(t)}, y_{i,1} \cdot \mathbf{v} \right\rangle \leq \widetilde{\Theta}(1)$, $\left\langle \mathbf{w}_{j,r}^{(t)}, y_{i,1} \cdot \mathbf{v}_1 \right\rangle \leq \widetilde{\Theta}(1)$, $\left\langle \mathbf{w}_{j,r}^{(t)}, \boldsymbol{\xi}_{i,1,h} \right\rangle \leq \widetilde{\Theta}(1)$ and $\sigma(\cdot)$ is convex, then we have

$$\left| \sigma \left( \left\langle \mathbf{w}_{j,r}^{(t+1)}, \alpha y_{i,1} \mathbf{v} \right\rangle \right) - \sigma \left( \left\langle \mathbf{w}_{j,r}^{(t+1)}, \alpha y_{i,1} \mathbf{v} \right\rangle \right) \right|$$
$$\leq \max \left\{ \left| \sigma' \left( \left\langle \mathbf{w}_{j,r}^{(t+1)}, \alpha y_{i,1} \mathbf{v} \right\rangle \right) \right|, \left| \sigma' \left( \left\langle \mathbf{w}_{j,r}^{(t)}, \alpha y_{i,1} \mathbf{v} \right\rangle \right) \right| \right\} \cdot \left| \left\langle \mathbf{v}, \mathbf{w}_{j,r}^{(t+1)} - \mathbf{w}_{j,r}^{(t)} \right\rangle \right|$$
$$\leq \widetilde{\Theta} \left( \left\| \mathbf{w}_{j,r}^{(t+1)} - \mathbf{w}_{j,r}^{(t)} \right\|_2 \right).$$

Similarly we also have

$$\left| \sigma \left( \left\langle \mathbf{w}_{j,r}^{(t+1)}, y_{i,1} \mathbf{v}_1 \right\rangle \right) - \sigma \left( \left\langle \mathbf{w}_{j,r}^{(t+1)}, y_{i,1} \mathbf{v}_1 \right\rangle \right) \right| \leq \widetilde{\Theta} \left( \left\| \mathbf{w}_{j,r}^{(t+1)} - \mathbf{w}_{j,r}^{(t)} \right\|_2 \right).$$

and

$$\left| \sigma \left( \left\langle \mathbf{w}_{j,r}^{(t+1)}, \boldsymbol{\xi}_{i,1,h} \right\rangle \right) - \sigma \left( \left\langle \mathbf{w}_{j,r}^{(t+1)}, \boldsymbol{\xi}_{i,1,h} \right\rangle \right) \right| \leq \widetilde{\Theta} \left( \left\| \mathbf{w}_{j,r}^{(t+1)} - \mathbf{w}_{j,r}^{(t)} \right\|_2 \right).$$

Combining the above inequalities for every $r \in [m]$, we have

$$|\Delta F_{j,i}|^2 \leq \widetilde{\Theta} \left( \left[ \sum_{r \in [m]} \left\| \mathbf{w}_{j,r}^{(t+1)} - \mathbf{w}_{j,r}^{(t)} \right\|_2 \right]^2 \right) \leq \widetilde{\Theta} \left( m \eta^2 \left\| \nabla L_1 \left( \mathbf{W}^{(t)} \right) \right\|_{\mathrm{F}}^2 \right)$$
$$= \widetilde{\Theta} \left( \eta^2 \left\| \nabla L_1 \left( \mathbf{W}^{(t)} \right) \right\|_{\mathrm{F}}^2 \right)$$

Now we can combine all the above inequalities, which gives

$$L_{i,1} \left( \mathbf{W}^{(t+1)} \right) - L_{i,1} \left( \mathbf{W}^{(t)} \right) \leq \sum_j \frac{\partial L_{i,1} \left( \mathbf{W}^{(t)} \right)}{\partial F_j \left( \mathbf{W}^{(t)}, \mathbf{x}_{i,1} \right)} \cdot \Delta F_{j,i} + \sum_j \left( \Delta F_{j,i} \right)^2$$
$$= \left\langle \nabla L_{i,1} \left( \mathbf{W}^{(t)} \right), \mathbf{W}^{(t+1)} - \mathbf{W}^{(t)} \right\rangle$$
$$+ \widetilde{\Theta} \left( \eta^2 \left\| \nabla L_1 \left( \mathbf{W}^{(t)} \right) \right\|_{\mathrm{F}}^2 \right)$$

Taking sum over $i \in [n]$ and applying the smoothness property of the regularization function $\lambda \|\mathbf{W}\|_{\mathrm{F}}^2$, we can get

$$L_1 \left( \mathbf{W}^{(t+1)} \right) - L_1 \left( \mathbf{W}^{(t)} \right)$$
$$= \frac{1}{n} \sum_{i=1}^n \left\{ \left[ L_{i,1} \left( \mathbf{W}^{(t+1)} \right) - L_{i,1} \left( \mathbf{W}^{(t)} \right) \right] + \lambda \left( \left\| \mathbf{W}^{(t+1)} \right\|_{\mathrm{F}}^2 - \left\| \mathbf{W}^{(t)} \right\|_{\mathrm{F}}^2 \right) \right\}$$
$$\leq \left\langle \nabla L_1 \left( \mathbf{W}^{(t)} \right), \mathbf{W}^{(t+1)} - \mathbf{W}^{(t)} \right\rangle + \widetilde{\Theta} \left( \eta^2 \left\| \nabla L_1 \left( \mathbf{W}^{(t)} \right) \right\|_{\mathrm{F}}^2 \right)$$
$$= - \left( \eta - \widetilde{\Theta} \left( \eta^2 \right) \right) \cdot \left\| \nabla L_1 \left( \mathbf{W}^{(t)} \right) \right\|_{\mathrm{F}}^2$$
$$\leq - \frac{\eta}{2} \left\| \nabla L_1 \left( \mathbf{W}^{(t)} \right) \right\|_{\mathrm{F}}^2$$

where the last inequality is due to our choice of step size $\eta = o(1)$. This completes the proof. $\qquad \square$

## D.2 PROOF OF LEMMA C.2

*Proof of Lemma C.2.* From Lemma D.2 and Lemma D.3, we can get that

$$F_{y_{i,1}} \left( \mathbf{W}^*, \mathbf{x}_{i,1} \right) = \sum_{r=1}^m \left[ \sigma \left( \left\langle \mathbf{w}_{y_{i,1},r}^*, \alpha y_{i,1} \mathbf{v} \right\rangle \right) + \sum_{h=2}^H \sigma \left( \left\langle \mathbf{w}_{y_{i,1},r}^*, \boldsymbol{\xi}_{i,1,h} \right\rangle \right) \right]$$

$$\geq \max_{h\in[2,H]} \max_{r\in[m]} \sigma\left(\left\langle \mathbf{w}^*_{y_{i,1},r}, \boldsymbol{\xi}_{i,1,h}\right\rangle\right) = \widetilde{\Theta}\left(1\right)$$

$$F_{-y_{i,1}}\left(\mathbf{W}^*, \mathbf{x}_{i,1}\right) = \sum_{r=1}^{m}\left[\sigma\left(\left\langle \mathbf{w}^*_{-y_{i,1},r}, \alpha y_{i,1}\mathbf{v}\right\rangle\right) + \sum_{h=2}^{H}\sigma\left(\left\langle \mathbf{w}^*_{-y_{i,1},r}, \boldsymbol{\xi}_{i,1,h}\right\rangle\right)\right]$$

$$\leq m\max_{r\in[m]}\sigma\left(\mathbf{w}^*_{-y_{i,1},r}, \alpha y_{i,1}\mathbf{v}\right)$$

$$+ m\left(H-1\right)\sigma\left(\left(\max_{r\in[m]}\max_{h\in[2,H]}\left\langle \mathbf{w}^*_{-y_{i,1},r}, \boldsymbol{\zeta}_{i,1,h}\right\rangle\right) + \max_{h\in[2,H]}\beta_h \max_{r\in[m]}\left\langle \mathbf{w}^*_{-y_{i,1},r}, -y_{i,1}\mathbf{v}\right\rangle\right)$$

$$\leq m\widetilde{\Theta}\left(\alpha^q \sigma_0^{\frac{q}{3}}\right) + m\left(H-1\right)\widetilde{\Theta}\left(\left(\sqrt{s}\sigma_p\sigma_0\right)^q\right) = o\left(1\right).$$

so $F_{y_{i,1}}\left(\mathbf{W}^*, \mathbf{x}_{i,1}\right) \geq F_{-y_{i,1}}\left(\mathbf{W}^*, \mathbf{x}_{i,1}\right)$ holds for $i\in\mathcal{D}_1$. Similarly, from Lemma D.1 to Lemma D.3, we also have

$$F_{y_{i,1}}\left(\mathbf{W}^*, \mathbf{x}_{i,1}\right) = \sum_{r=1}^{m}\left[\sigma\left(\left\langle \mathbf{w}^*_{y_{i,1},r}, y_{i,1}\mathbf{v}_1\right\rangle\right) + \sum_{h=2}^{H}\sigma\left(\left\langle \mathbf{w}^*_{y_{i,1},r}, \boldsymbol{\xi}_{i,1,h}\right\rangle\right)\right]$$

$$\geq \max_{r\in[m]}\sigma\left(\left\langle \mathbf{w}^*_{y_{i,1},r}, y_{i,1}\mathbf{v}_1\right\rangle\right) = \widetilde{\Theta}\left(1\right)$$

$$F_{-y_{i,1}}\left(\mathbf{W}^*, \mathbf{x}_{i,1}\right) = \sum_{r=1}^{m}\left[\sigma\left(\left\langle \mathbf{w}^*_{-y_{i,1},r}, y_{i,1}\mathbf{v}_1\right\rangle\right) + \sum_{h=2}^{H}\sigma\left(\left\langle \mathbf{w}^*_{-y_{i,1},r}, \boldsymbol{\xi}_{i,1,h}\right\rangle\right)\right]$$

$$\leq m\max_{r\in[m]}\sigma\left(\left\langle \mathbf{w}^*_{-y_{i,1},r}, y_{i,1}\mathbf{v}_1\right\rangle\right)$$

$$+ m\left(H-1\right)\sigma\left(\max_{r\in[m]}\max_{h\in[2,H]}\left\langle \mathbf{w}^*_{-y_{i,1},r}, \boldsymbol{\zeta}_{i,1,h}\right\rangle + \max_{h\in[2,H]}\beta_h \max_{r\in[m]}\left\langle \mathbf{w}^*_{-y_{i,1},r}, -y_{i,1}\mathbf{v}\right\rangle\right)$$

$$\leq m\widetilde{\Theta}\left(\sigma_0^q\right) + m\left(H-1\right)\widetilde{\Theta}\left(\left(\sqrt{s}\sigma_p\sigma_0\right)^q\right) = o\left(1\right).$$

so $F_{y_{i,1}}\left(\mathbf{W}^*, \mathbf{x}_{i,1}\right) \geq F_{-y_{i,1}}\left(\mathbf{W}^*, \mathbf{x}_{i,1}\right)$ holds for $i\in\mathcal{D}_2$. Combining these two parts, we have that for $i\in[n]$, $F_{y_{i,1}}\left(\mathbf{W}^*, \mathbf{x}_{i,1}\right) \geq F_{-y_{i,1}}\left(\mathbf{W}^*, \mathbf{x}_{i,1}\right)$ holds, which directly implies that

$$\frac{1}{n}\sum_{i=1}^{n}\mathbf{1}\left[F_{-y_{i,1}}\left(\mathbf{W}^*, \mathbf{x}_{i,1}\right) \leq F_{y_{i,1}}\left(\mathbf{W}^*, \mathbf{x}_{i,1}\right)\right] = 0$$

Therefore, $\mathbf{W}^*$ can correctly classify all training data and thus achieve zero training error.
In terms of the test data $(\mathbf{x}, y)$ which is generated according to our assumptions, then with probability $p$, it will have the patch of task-shared feature and the patches of noise, like the training data for $i\in\mathcal{D}_1$, then $\mathbf{x} = [\alpha y\mathbf{v}, \boldsymbol{\xi}_2, ..., \boldsymbol{\xi}_H]$. For each $i\in[n]$, denote $(r_i^*, h_i^*) = \arg\max_{(r,h)\in[2,H]\times[m]}\left\langle \mathbf{w}^*_{y_{i,1},r}, \boldsymbol{\xi}_{i,1,h}\right\rangle$, then we have

$$\left\langle \mathbf{w}^*_{y_{i,1},r_i^*}, \boldsymbol{\zeta}_{i,1,h_i^*}\right\rangle = \left\langle \mathbf{w}^*_{y_{i,1},r_i^*}, \boldsymbol{\xi}_{i,1,h_i^*}\right\rangle - \left\langle \mathbf{w}^*_{y_{i,1},r_i^*}, \beta_{h_i^*}\cdot(-y_{i,1})\mathbf{v}\right\rangle$$

$$= \left\langle \mathbf{w}^*_{y_{i,1},r_i^*}, \boldsymbol{\xi}_{i,1,h_i^*}\right\rangle - \beta_{h^*}\left\langle \mathbf{w}^*_{y_{i,1},r_i^*}, (-y_{i,1})\mathbf{v}\right\rangle \geq \widetilde{\Theta}\left(1\right) - \beta_{h_i^*}\cdot\frac{\rho}{\alpha} \geq \widetilde{\Theta}\left(1\right)$$

Then according to the gradient calculations, for $i\in[n]$, we have

$$\nabla_{\mathbf{w}_{y_{i,1},r_i^*}}L_1\left(\mathbf{W}^{(t)}\right)$$

$$= \lambda\mathbf{w}^{(t)}_{y_{i,1},r_i^*}$$

$$- \frac{1}{n}\sum_{s\in\mathcal{D}_1}l^{(t)}_{y_{i,1},s,1}\left[\sigma'\left(\left\langle \mathbf{w}^{(t)}_{y_{i,1},r_i^*}, \alpha y_{s,1}\mathbf{v}\right\rangle\right)\cdot \alpha y_{s,1}\mathbf{v} + \sum_{g=2}^{H}\sigma'\left(\left\langle \mathbf{w}^{(t)}_{y_{i,1},r_i^*}, \boldsymbol{\xi}_{s,1,g}\right\rangle\right)\cdot \boldsymbol{\xi}_{s,1,h}\right]$$

$$- \frac{1}{n}\sum_{s\in\mathcal{D}_2}l^{(t)}_{y_{i,1},s,1}\left[\sigma'\left(\left\langle \mathbf{w}^{(t)}_{y_{i,1},r_i^*}, y_{s,1}\mathbf{v}_1\right\rangle\right)\cdot y_{s,1}\mathbf{v}_1 + \sum_{g=2}^{H}\sigma'\left(\left\langle \mathbf{w}^{(t)}_{y_{i,1},r_i^*}, \boldsymbol{\xi}_{s,1,g}\right\rangle\right)\cdot \boldsymbol{\xi}_{s,1,g}\right]$$

By taking inner product with $\boldsymbol{\zeta}_{i,1,h_i^*}$, and notice that $\mathbf{v}$ and $\mathbf{v}_1$ are orthogonal to $\boldsymbol{\zeta}_{i,1,h_i^*}$, we can get that

$$\left\langle \nabla_{\mathbf{w}_{y_{i,1},r_i^*}} L_1\left(\mathbf{W}^{(t)}\right), \boldsymbol{\zeta}_{i,1,h_i^*} \right\rangle$$

$$= \lambda \left\langle \mathbf{w}_{y_{i,1},r_i^*}^{(t)}, \boldsymbol{\zeta}_{i,1,h_i^*} \right\rangle - \frac{1}{n} \sum_{s \in \mathcal{D}_1} l_{y_{i,1},s,1}^{(t)} \sum_{g=2}^{H} \sigma'\left(\left\langle \mathbf{w}_{y_{i,1},r_i^*}^{(t)}, \boldsymbol{\xi}_{s,1,g} \right\rangle\right) \cdot \left\langle \boldsymbol{\xi}_{s,1,g}, \boldsymbol{\zeta}_{i,1,h_i^*} \right\rangle$$

$$\qquad - \frac{1}{n} \sum_{s \in \mathcal{D}_2} l_{y_{i,1},s,1}^{(t)} \sum_{g=2}^{H} \sigma'\left(\left\langle \mathbf{w}_{y_{i,1},r_i^*}^{(t)}, \boldsymbol{\xi}_{s,1,g} \right\rangle\right) \cdot \left\langle \boldsymbol{\xi}_{s,1,g}, \boldsymbol{\zeta}_{i,1,h_i^*} \right\rangle$$

$$= \lambda \left\langle \mathbf{w}_{y_{i,1},r_i^*}^{(t)}, \boldsymbol{\zeta}_{i,1,h_i^*} \right\rangle - \frac{1}{n} \sum_{s=1}^{n} l_{y_{i,1},s,1}^{(t)} \sum_{g=2}^{H} \sigma'\left(\left\langle \mathbf{w}_{y_{i,1},r_i^*}^{(t)}, \boldsymbol{\xi}_{s,1,g} \right\rangle\right) \cdot \left\langle \boldsymbol{\zeta}_{s,1,g} - \beta_g y_{s,1} \mathbf{v}, \boldsymbol{\zeta}_{i,1,h_i^*} \right\rangle$$

$$= \lambda \left\langle \mathbf{w}_{y_{i,1},r_i^*}^{(t)}, \boldsymbol{\zeta}_{i,1,h_i^*} \right\rangle - \frac{1}{n} \sum_{s=1}^{n} l_{y_{i,1},s,1}^{(t)} \sum_{g=2}^{H} \sigma'\left(\left\langle \mathbf{w}_{y_{i,1},r_i^*}^{(t)}, \boldsymbol{\xi}_{s,1,g} \right\rangle\right) \cdot \left\langle \boldsymbol{\zeta}_{s,1,g}, \boldsymbol{\zeta}_{i,1,h_i^*} \right\rangle$$

$$= \lambda \left\langle \mathbf{w}_{y_{i,1},r_i^*}^{(t)}, \boldsymbol{\zeta}_{i,1,h_i^*} \right\rangle - \frac{1}{n} l_{y_{i,1},i,1}^{(t)} \sigma'\left(\left\langle \mathbf{w}_{y_{i,1},r_i^*}^{(t)}, \boldsymbol{\xi}_{i,1,h_i^*} \right\rangle\right) \left\langle \boldsymbol{\zeta}_{i,1,h_i^*}, \boldsymbol{\zeta}_{i,1,h_i^*} \right\rangle$$

$$\qquad - \frac{1}{n} \sum_{(s,g) \neq (i,h_i^*)} l_{y_{i,1},s,1}^{(t)} \sigma'\left(\left\langle \mathbf{w}_{y_{i,1},r_i^*}^{(t)}, \boldsymbol{\xi}_{s,1,g} \right\rangle\right) \cdot \left\langle \boldsymbol{\zeta}_{s,1,g}, \boldsymbol{\zeta}_{i,1,h_i^*} \right\rangle$$

$$\overset{(i)}{\geq} \lambda \left\langle \mathbf{w}_{y_{i,1},r_i^*}^{(t)}, \boldsymbol{\zeta}_{i,1,h_i^*} \right\rangle - \frac{1}{n} \left| l_{y_{i,1},i,1}^{(t)} \right| \sigma'\left(\left\langle \mathbf{w}_{y_{i,1},r_i^*}^{(t)}, \boldsymbol{\xi}_{i,1,h_i^*} \right\rangle\right) \left\langle \boldsymbol{\zeta}_{i,1,h_i^*}, \boldsymbol{\zeta}_{i,1,h_i^*} \right\rangle$$

$$\qquad - \frac{1}{n} \sum_{(s,g) \neq (i,h_i^*)} \left| l_{y_{i,1},s,1}^{(t)} \right| \sigma'\left(\left\langle \mathbf{w}_{y_{i,1},r_i^*}^{(t)}, \boldsymbol{\xi}_{s,1,g} \right\rangle\right) \cdot \left| \left\langle \boldsymbol{\zeta}_{s,1,g}, \boldsymbol{\zeta}_{i,1,h_i^*} \right\rangle \right|$$

$$\overset{(ii)}{\geq} \lambda \left\langle \mathbf{w}_{y_{i,1},r_i^*}^{(t)}, \boldsymbol{\zeta}_{i,1,h_i^*} \right\rangle - \frac{1}{n} \left| l_{y_{i,1},i,1}^{(t)} \right| \sigma'\left(\left\langle \mathbf{w}_{y_{i,1},r_i^*}^{(t)}, \boldsymbol{\xi}_{i,1,h_i^*} \right\rangle\right) \left\langle \boldsymbol{\zeta}_{i,1,h_i^*}, \boldsymbol{\zeta}_{i,1,h_i^*} \right\rangle$$

$$\qquad - \frac{1}{n} \sum_{(s,g) \neq (i,h_i^*)} \sigma'\left(\left\langle \mathbf{w}_{y_{i,1},r_i^*}^{(t)}, \boldsymbol{\xi}_{s,1,g} \right\rangle\right) \cdot \left| \left\langle \boldsymbol{\zeta}_{s,1,g}, \boldsymbol{\zeta}_{i,1,h_i^*} \right\rangle \right|$$

in which (i) holds since $l_{y_{i,1},i,1}^{(t)} \geq 0$ according to the definition, and (ii) holds since $\left| l_{y_{i,1},s,1}^{(t)} \right| \leq 1$. For $y_{s,1} = y_{i,1}$, from Lemma D.3, $\left\langle \mathbf{w}_{y_{s,1},r_i^*}^{(t)}, \boldsymbol{\xi}_{s,1,g} \right\rangle \leq \widetilde{\Theta}(1)$, so $\left\langle \mathbf{w}_{y_{i,1},r_i^*}^{(t)}, \boldsymbol{\xi}_{s,1,g} \right\rangle = \left\langle \mathbf{w}_{y_{s,1},r_i^*}^{(t)}, \boldsymbol{\xi}_{s,1,g} \right\rangle \leq \widetilde{\Theta}(1)$. For $y_{s,1} \neq y_{i,1}$, from Lemma D.2 and Lemma D.3, $\left\langle \mathbf{w}_{-y_{s,1},r_i^*}^{(t)}, \boldsymbol{\xi}_{s,1,g} \right\rangle \leq \widetilde{\Theta}(\sqrt{\sigma_0}) \leq \widetilde{\Theta}(1)$, so $\left\langle \mathbf{w}_{y_{i,1},r_i^*}^{(t)}, \boldsymbol{\xi}_{s,1,g} \right\rangle = \left\langle \mathbf{w}_{-y_{s,1},r}^{(t)}, \boldsymbol{\xi}_{s,1,g} \right\rangle \leq \widetilde{\Theta}(1)$. Then $\left\langle \mathbf{w}_{y_{i,1},r_i^*}^{(t)}, \boldsymbol{\xi}_{s,1,g} \right\rangle \leq \widetilde{\Theta}(1)$ holds for $(s,g) \neq (i,h_i^*)$, which implies that $\sigma'\left(\left\langle \mathbf{w}_{y_{i,1},r_i^*}^{(t)}, \boldsymbol{\xi}_{s,1,g} \right\rangle\right) \leq \sigma'\left(\widetilde{\Theta}(1)\right) = \widetilde{\Theta}(1)$. Since $\left\langle \mathbf{w}_{y_{i,1},r_i^*}^{(t)}, \boldsymbol{\xi}_{i,1,h_i^*} \right\rangle = \widetilde{\Theta}(1)$, then $\sigma'\left(\left\langle \mathbf{w}_{y_{i,1},r_i^*}^{(t)}, \boldsymbol{\xi}_{i,1,h_i^*} \right\rangle\right) = \sigma'\left(\widetilde{\Theta}(1)\right) = \widetilde{\Theta}(1)$. Besides, using the same calculations as in Lemma D.3, with probability exceeding $1 - 2n^{-1}$,

$$\left\langle \boldsymbol{\zeta}_{i,1,h_i^*}, \boldsymbol{\zeta}_{i,1,h_i^*} \right\rangle = \sum_{u=1}^{d} {\zeta_{i,1,h_i^*}^u}^2 = \widetilde{\Theta}\left(s\sigma_p^2\right)$$

and for $(s,g) \neq (i,h_i^*)$, $\left| \left\langle \boldsymbol{\zeta}_{s,1,g}, \boldsymbol{\zeta}_{i,1,h_i^*} \right\rangle \right| = 0$ according to Lemma B.3. Combining all these results, we can get that

$$\left\langle \nabla_{\mathbf{w}_{y_{i,1},r_i^*}} L_1\left(\mathbf{W}^{(t)}\right), \boldsymbol{\zeta}_{i,1,h_i^*} \right\rangle$$

$$\geq \lambda \left\langle \mathbf{w}_{y_{i,1},r_i^*}^{(t)}, \boldsymbol{\zeta}_{i,1,h_i^*} \right\rangle - \frac{1}{n} \left| l_{y_{i,1},i,1}^{(t)} \right| \sigma'\left(\left\langle \mathbf{w}_{y_{i,1},r_i^*}^{(t)}, \boldsymbol{\xi}_{i,1,h_i^*} \right\rangle\right) \left\langle \boldsymbol{\zeta}_{i,1,h_i^*}, \boldsymbol{\zeta}_{i,1,h_i^*} \right\rangle$$

$$\qquad - \frac{1}{n} \sum_{(s,g) \neq (i,h_i^*)} \sigma'\left(\left\langle \mathbf{w}_{y_{i,1},r_i^*}^{(t)}, \boldsymbol{\xi}_{s,1,g} \right\rangle\right) \cdot \left| \left\langle \boldsymbol{\zeta}_{s,1,g}, \boldsymbol{\zeta}_{i,1,h_i^*} \right\rangle \right|$$

$$\geq \lambda \cdot \widetilde{\Theta}\left(1\right) - \frac{1}{n}\left|l_{y_{i,1},i,1}^{(t)}\right| \cdot \widetilde{\Theta}\left(1\right) \cdot \widetilde{\Theta}\left(s\sigma_p^2\right)$$

$$= \widetilde{\Theta}\left(\lambda\right) - \widetilde{\Theta}\left(\frac{s\sigma_p^2}{n}\right)\left|l_{y_{i,1},i,1}^{(t)}\right|$$

Since

$$\left\langle \nabla_{\mathbf{w}_{y_{i,1},r_i^*}} L_1\left(\mathbf{W}^{(t)}\right), \boldsymbol{\zeta}_{i,1,h_i^*}\right\rangle \leq \left\|\nabla_{\mathbf{w}_{y_{i,1},r_i^*}} L_1\left(\mathbf{W}^{(t)}\right)\right\|_2 \left\|\boldsymbol{\zeta}_{i,1,h_i^*}\right\|_2$$

$$\leq \left\|\nabla L_1\left(\mathbf{W}^{(t)}\right)\right\|_{\mathrm{F}} \left\|\boldsymbol{\zeta}_{i,1,h_i^*}\right\|_2$$

$$\leq \widetilde{\Theta}\left(\frac{\lambda}{n}\right),$$

so we can get that

$$\left|l_{y_{i,1},i,1}^{(t)}\right| \geq \widetilde{\Theta}\left(\frac{n}{s\sigma_p^2}\right)\left(\widetilde{\Theta}\left(\lambda\right) - \left\langle \nabla_{\mathbf{w}_{y_{i,1},r_i^*}} L_1\left(\mathbf{W}^{(t)}\right), \boldsymbol{\zeta}_{i,1,h_i^*}\right\rangle\right)$$

$$\geq \widetilde{\Theta}\left(\frac{s\sigma_p^2}{n}\right)\left(\widetilde{\Theta}\left(\lambda\right) - \widetilde{\Theta}\left(\frac{\lambda}{n}\right)\right) \geq \widetilde{\Theta}\left(\frac{n\lambda}{s\sigma_p^2}\right)$$

According to the definition of $l_{j,i,1}^{(t)}$, we can get that $\left|l_{-y_{i,1},i,1}^{(t)}\right| = \left|l_{y_{i,1},i,1}^{(t)}\right| = \widetilde{\Theta}\left(\frac{n\lambda}{s\sigma_p^2}\right)$, so for $j \in \{-1,1\}$ and $i \in [n]$, $\left|l_{j,i,1}^{(t)}\right| \geq \widetilde{\Theta}\left(\frac{n\lambda}{s\sigma_p^2}\right)$. By taking inner product of $\nabla_{\mathbf{w}_{y_{i,1},r_i^*}} L_1\left(\mathbf{W}^{(t)}\right)$ with $-y_{i,1}\mathbf{v}$, and notice that $\mathbf{v}_1$ and $\boldsymbol{\zeta}_{s,1,g}$ are orthogonal to $\mathbf{v}$, we can get that

$$\left\langle \nabla_{\mathbf{w}_{y_{i,1},r_i^*}^{(t)}} L_1\left(\mathbf{W}^{(t)}\right), -y_{i,1}\mathbf{v}\right\rangle$$

$$= \lambda\left\langle \mathbf{w}_{y_{i,1},r_i^*}^{(t)}, -y_{i,1}\mathbf{v}\right\rangle - \frac{1}{n}\sum_{s\in\mathcal{D}_1} l_{y_{i,1},s,1}^{(t)}\sigma'\left(\left\langle \mathbf{w}_{y_{i,1},r_i^*}, \alpha y_{s,1}\mathbf{v}\right\rangle\right)\cdot\left\langle \alpha y_{s,1}\mathbf{v}, -y_{i,1}\mathbf{v}\right\rangle$$

$$- \frac{1}{n}\sum_{s\in\mathcal{D}_1} l_{y_{i,1},s,1}^{(t)}\sum_{g=2}^{H}\sigma'\left(\left\langle \mathbf{w}_{y_{i,1},r_i^*}^{(t)}, \boldsymbol{\xi}_{s,1,g}\right\rangle\right)\left\langle \boldsymbol{\xi}_{s,1,g}, -y_{i,1}\mathbf{v}\right\rangle$$

$$- \frac{1}{n}\sum_{s\in\mathcal{D}_2} l_{y_{i,1},s,1}^{(t)}\sum_{g=2}^{H}\sigma'\left(\left\langle \mathbf{w}_{y_{i,1},r_i^*}^{(t)}, \boldsymbol{\xi}_{s,1,g}\right\rangle\right)\left\langle \boldsymbol{\xi}_{s,1,g}, -y_{i,1}\mathbf{v}\right\rangle$$

$$= \lambda\left\langle \mathbf{w}_{y_{i,1},r_i^*}, -y_{i,1}\mathbf{v}\right\rangle - \frac{1}{n}\sum_{s\in\mathcal{D}_1}\left|l_{y_{i,1},s,1}^{(t)}\right|\sigma'\left(\left\langle \mathbf{w}_{y_{i,1},r_i^*}^{(t)}, \alpha y_{s,1}\mathbf{v}\right\rangle\right)\cdot\left\langle \alpha y_{s,1}\mathbf{v}, -y_{i,1}\mathbf{v}\right\rangle$$

$$- \frac{1}{n}\sum_{s=1}^{n} l_{y_{i,1},s,1}^{(t)}\sum_{g=2}^{H}\sigma'\left(\left\langle \mathbf{w}_{y_{i,1},r_i^*}^{(t)}, \boldsymbol{\xi}_{s,1,g}\right\rangle\right)\left\langle \boldsymbol{\zeta}_{s,1,g} - \beta_g y_{s,1}\mathbf{v}, -y_{i,1}\mathbf{v}\right\rangle$$

$$= \lambda\left\langle \mathbf{w}_{y_{i,1},r_i^*}^{(t)}, -y_{i,1}\mathbf{v}\right\rangle - \frac{1}{n}\sum_{s\in\mathcal{D}_1} l_{y_{i,1},s,1}^{(t)}\sigma'\left(\left\langle \mathbf{w}_{y_{i,1},r_i^*}^{(t)}, \alpha y_{s,1}\mathbf{v}\right\rangle\right)\cdot\left\langle \alpha y_{s,1}\mathbf{v}, -y_{i,1}\mathbf{v}\right\rangle$$

$$- \frac{1}{n}\sum_{s=1}^{n} l_{y_{i,1},s,1}^{(t)}\sum_{g=2}^{H}\sigma'\left(\left\langle \mathbf{w}_{y_{i,1},r_i^*}^{(t)}, \boldsymbol{\xi}_{s,1,g}\right\rangle\right)\left\langle -\beta_g y_{s,1}\mathbf{v}, -y_{i,1}\mathbf{v}\right\rangle$$

$$\overset{(\mathrm{i})}{=} \lambda\left\langle \mathbf{w}_{y_{i,1},r_i^*}^{(t)}, -y_{i,1}\mathbf{v}\right\rangle + \frac{\alpha}{n}\sum_{s\in\mathcal{D}_1}\left|l_{y_{i,1},s,1}^{(t)}\right|\sigma'\left(\left\langle \mathbf{w}_{y_{i,1},r_i^*}^{(t)}, \alpha y_{s,1}\mathbf{v}\right\rangle\right)$$

$$- \frac{1}{n}\sum_{s=1}^{n}\left|l_{y_{i,1},s,1}^{(t)}\right|\sum_{g=2}^{H}\beta_g\sigma'\left(\left\langle \mathbf{w}_{y_{i,1},r_i^*}^{(t)}, \boldsymbol{\xi}_{s,1,g}\right\rangle\right)$$

$$= \lambda\left\langle \mathbf{w}_{y_{i,1},r_i^*}^{(t)}, -y_{i,1}\mathbf{v}\right\rangle + \frac{\alpha}{n}\sum_{s:s\in\mathcal{D}_1, y_{s,1}=y_{i,1}}\left|l_{y_{i,1},s,1}^{(t)}\right|\sigma'\left(\left\langle \mathbf{w}_{y_{i,1},r_i^*}^{(t)}, \alpha y_{s,1}\mathbf{v}\right\rangle\right)$$

$$+ \frac{\alpha}{n} \sum_{s:s\in\mathcal{D}_1,y_{s,1}\neq y_{i,1}} \left| l^{(t)}_{y_{i,1},s,1} \right| \sigma'\left( \left\langle \mathbf{w}^{(t)}_{y_{i,1},r_i^*}, \alpha y_{s,1}\mathbf{v} \right\rangle \right)$$

$$- \frac{1}{n} \sum_{s=1}^{n} \left| l^{(t)}_{y_{i,1},s,1} \right| \sum_{g=2}^{H} \beta_g \sigma'\left( \left\langle \mathbf{w}^{(t)}_{y_{i,1},r_i^*}, \boldsymbol{\xi}_{s,1,g} \right\rangle \right)$$

in which (i) holds since $\mathrm{sgn}\left( y_{s,1}y_{i,1}l^{(t)}_{y_{i,1},s,1} \right) = 1$. For $y_{s,1} = y_{i,1}$, from Lemma D.3, $\left\langle \mathbf{w}^{(t)}_{y_{s,1},r_i^*}, y_{s,1}\mathbf{v} \right\rangle \leq \widetilde{O}\left( \sigma_0 \right)$, so $\left\langle \mathbf{w}_{y_{i,1},r_i^*}, \alpha y_{s,1}\mathbf{v} \right\rangle = \alpha \left\langle \mathbf{w}_{y_{s,1},r_i^*}, y_{s,1}\mathbf{v} \right\rangle \leq \left\langle \mathbf{w}_{y_{s,1},r_i^*}, y_{s,1}\mathbf{v} \right\rangle \leq \widetilde{O}\left( \sigma_0 \right) \leq \rho$, which implies that the activation function for $\left\langle \mathbf{w}_{y_{i,1},r_i^*}, \alpha y_{s,1}\mathbf{v} \right\rangle$ is $\frac{z^q}{q\rho^{q-1}}$, then $\sigma'\left( \left\langle \mathbf{w}_{y_{i,1},r_i^*}, \alpha y_{s,1}\mathbf{v} \right\rangle \right) = \frac{\left( \left\langle \mathbf{w}_{y_{i,1},r_i^*}, \alpha y_{s,1}\mathbf{v} \right\rangle \right)^{q-1}}{\rho^{q-1}} \leq \widetilde{\Theta}\left( \widetilde{O}\left( \sigma_0 \right) \right)^{q-1} = \widetilde{O}\left( \sigma_0^{q-1} \right)$. For $y_{s,1} \neq y_{i,1}$, $\left\langle \mathbf{w}^{(t)}_{y_{i,1},r_i^*}, \alpha y_{s,1}\mathbf{v} \right\rangle = \alpha \left\langle \mathbf{w}^{(t)}_{-y_{s,1},r_i^*}, y_{s,1}\mathbf{v} \right\rangle \leq \alpha \max_{j\in\{-1,1\}} \max_{r\in[m]} \left\langle \mathbf{w}^{(t)}_{-j,r}, j\cdot\mathbf{v} \right\rangle \leq \alpha \cdot \frac{\rho}{\alpha} = \rho$, which implies that the activation function for $\left\langle \mathbf{w}^{(t)}_{y_{i,1},r_i^*}, \alpha y_{s,1}\mathbf{v} \right\rangle$ is $\frac{z^q}{q\rho^{q-1}}$, so $\sigma'\left( \left\langle \mathbf{w}^{(t)}_{y_{i,1},r_i^*}, \alpha y_{s,1}\mathbf{v} \right\rangle \right) = \frac{\left( \alpha \left\langle \mathbf{w}^{(t)}_{y_{i,1},r_i^*}, y_{s,1}\mathbf{v} \right\rangle \right)^{q-1}}{\rho^{q-1}} \leq \widetilde{\Theta}\left( \alpha \right) \left\langle \mathbf{w}^{(t)}_{y_{i,1},r_i^*}, y_{s,1}\mathbf{v} \right\rangle \leq \widetilde{\Theta}\left( \alpha \right) \max_{j\in\{-1,1\}} \max_{r\in[m]} \left\langle \mathbf{w}^{(t)}_{-j,r}, j\cdot\mathbf{v} \right\rangle$. From Lemma D.3 and the definition of $(r_i^*, h_i^*)$, $\left\langle \mathbf{w}^{(t)}_{y_{i,1},r_i^*}, \boldsymbol{\xi}_{i,1,h_i^*} \right\rangle = \widetilde{\Theta}(1)$, so $\sigma'\left( \left\langle \mathbf{w}^{(t)}_{y_{i,1},r_i^*}, \boldsymbol{\xi}_{i,1,h_i^*} \right\rangle \right) = \sigma'\left( \widetilde{\Theta}(1) \right) = \widetilde{\Theta}(1)$. Combining all these results, we can get that

$$\left\langle \nabla_{\mathbf{w}^{(t)}_{y_{i,1},r_i^*}} L_1\left( \mathbf{W}^{(t)} \right), -y_{i,1}\mathbf{v} \right\rangle$$

$$= \lambda \left\langle \mathbf{w}^{(t)}_{y_{i,1},r_i^*}, -y_{i,1}\mathbf{v} \right\rangle + \frac{\alpha}{n} \sum_{s:s\in\mathcal{D}_1,y_{s,1}=y_{i,1}} \left| l^{(t)}_{y_{i,1},s,1} \right| \sigma'\left( \left\langle \mathbf{w}_{y_{i,1},r_i^*}, \alpha y_{s,1}\mathbf{v} \right\rangle \right)$$

$$+ \frac{\alpha}{n} \sum_{s:s\in\mathcal{D}_1,y_{s,1}\neq y_{i,1}} \left| l^{(t)}_{y_{i,1},s,1} \right| \sigma'\left( \left\langle \mathbf{w}_{y_{i,1},r_i^*}, \alpha y_{s,1}\mathbf{v} \right\rangle \right)$$

$$- \frac{1}{n} \sum_{s=1}^{n} \left| l^{(t)}_{y_{i,1},s,1} \right| \sum_{g=2}^{H} \beta_g \sigma'\left( \left\langle \mathbf{w}^{(t)}_{y_{i,1},r_i^*}, \boldsymbol{\xi}_{s,1,g} \right\rangle \right)$$

$$\leq \lambda \max_{j\in\{-1,1\}} \max_{r\in[m]} \left\langle \mathbf{w}^{(t)}_{-j,r}, j\cdot\mathbf{v} \right\rangle + \frac{\alpha}{n} \cdot \max_{s\in\mathcal{D}_1} \left| l^{(t)}_{y_{i,1},s,1} \right| \cdot \sum_{s:s\in\mathcal{D}_1,y_{i,1}=y_{s,1}} \sigma'\left( \left\langle \mathbf{w}_{y_{i,1},r_i^*}, \alpha y_{s,1}\mathbf{v} \right\rangle \right)$$

$$+ \frac{\alpha}{n} \cdot \max_{s\in\mathcal{D}_1} \left| l^{(t)}_{y_{i,1},s,1} \right| \cdot \sum_{s:s\in\mathcal{D}_1,y_{s,1}\neq y_{i,1}} \sigma'\left( \left\langle \mathbf{w}^{(t)}_{y_{i,1},r_i^*}, \alpha y_{s,1}\mathbf{v} \right\rangle \right)$$

$$- \frac{1}{n} \left| l^{(t)}_{y_{i,1},i,1} \right| \beta_{h_i^*} \sigma'\left( \left\langle \mathbf{w}^{(t)}_{y_{i,1},r_i^*}, \boldsymbol{\xi}_{i,1,h_i^*} \right\rangle \right)$$

$$\leq \lambda \max_{j\in\{-1,1\}} \max_{r\in[m]} \left\langle \mathbf{w}^{(t)}_{-j,r}, j\cdot\mathbf{v} \right\rangle + \alpha \cdot \max_{s\in\mathcal{D}_1} \left| l^{(t)}_{y_{i,1},s,1} \right| \cdot \widetilde{\Theta}\left( \max_{h\in[2,H]} \beta_h^4 \right)$$

$$+ \alpha \cdot \max_{s\in\mathcal{D}_1} \left| l^{(t)}_{y_{i,1},s,1} \right| \cdot \widetilde{\Theta}\left( \alpha \right) \max_{j\in\{-1,1\}} \max_{r\in[m]} \left\langle \mathbf{w}^{(t)}_{-j,r}, j\cdot\mathbf{v} \right\rangle - \frac{1}{n} \left| l^{(t)}_{y_{i,1},i,1} \right| \beta_{h_i^*} \cdot \widetilde{\Theta}(1)$$

Since

$$\left\langle \nabla_{\mathbf{w}^{(t)}_{y_{i,1},r_i^*}} L_1\left( \mathbf{W}^{(t)} \right), -y_{i,1}\mathbf{v} \right\rangle \geq - \left\| \nabla_{\mathbf{w}^{(t)}_{y_{i,1},r_i^*}} L_1\left( \mathbf{W}^{(t)} \right) \right\|_2 \left\| -y_{i,1}\mathbf{v} \right\|_2$$

$$\geq - \left\| \nabla L\left( \mathbf{W}^{(t)} \right) \right\|_{\mathrm{F}} \left\| -y_{i,1}\mathbf{v} \right\|_2$$

$$\geq -\widetilde{\Theta}\left( \frac{\lambda}{n} \right)$$

then we can solve that $\max_{j\in\{-1,1\}} \max_{r\in[m]} \left\langle \mathbf{w}^{(t)}_{-j,r}, j\cdot\mathbf{v} \right\rangle \geq \widetilde{\Theta}\left( \frac{\beta_{h_{i^*}}}{n} \right)$. Without loss of generality, we can assume that $\max_{r\in[m]} \left\langle \mathbf{w}^{(t)}_{-1,r}, \mathbf{v} \right\rangle \geq \widetilde{\Theta}\left( \frac{\beta_{h_{i^*}}}{n} \right)$. Then according to our data model,

with probability $\frac{p}{4}$, we will get data with $y = 1$ and its feature patch is task-shared feature. Similar to Lemma B.3, the support set of the random noise of this data will have no interpolation with the support sets of random noises in training samples with probability larger than $1 - n^{-2}$, which implies that $\left\langle \mathbf{w}_{1,r}^*, \boldsymbol{\zeta}_{\mathbf{x},h} \right\rangle = 0$. For this kind of data, we have

$$
\begin{aligned}
F_1\left(\mathbf{W}^*, \mathbf{x}\right) &= \sum_{r=1}^{m}\left[\sigma\left(\left\langle \mathbf{w}_{1,r}^*, \alpha\mathbf{v}\right\rangle\right) + \sum_{h=2}^{H} \sigma\left(\left\langle \mathbf{w}_{1,r}^*, \boldsymbol{\zeta}_{\mathbf{x},h} - \beta_h\mathbf{v}\right\rangle\right)\right] \\
&= \sum_{r=1}^{m}\left[\sigma\left(\left\langle \mathbf{w}_{1,r}^*, \alpha\mathbf{v}\right\rangle\right) + \sum_{h=2}^{H} \sigma\left(\left\langle \mathbf{w}_{1,r}^*, -\beta_h\mathbf{v}\right\rangle\right)\right] \\
&\leq m\left[\sigma\left(\max_{r\in[m]}\left\langle \mathbf{w}_{1,r}^*, \alpha\mathbf{v}\right\rangle\right) + (H-1)\sigma\left(\beta_h \max_{r\in[m]}\left\langle \mathbf{w}_{1,r}^*, -\mathbf{v}\right\rangle\right)\right] \\
&\leq mH\sigma\left(\max_{h\in[2,H]}\beta_h\sigma_0^{\frac{1}{3}}\right) = \widetilde{\Theta}\left(mH\left(\max_{h\in[2,H]}\beta_h\sigma_0^{\frac{1}{3}}\right)^q\right)
\end{aligned}
$$

and

$$
\begin{aligned}
F_{-1}\left(\mathbf{W}^*, \mathbf{x}\right) &= \sum_{r=1}^{m}\left[\sigma\left(\left\langle \mathbf{w}_{-1,r}^*, \alpha\mathbf{v}\right\rangle\right) + \sum_{h=2}^{H} \sigma\left(\left\langle \mathbf{w}_{-1,r}^*, \boldsymbol{\zeta}_{\mathbf{x},h} - \beta_h\mathbf{v}\right\rangle\right)\right] \\
&\geq \sigma\left(\max_{r\in[m]}\left\langle \mathbf{w}_{-1,r}^*, \alpha\mathbf{v}\right\rangle\right) \geq \sigma\left(\widetilde{\Theta}\left(\frac{\alpha\beta_{h^*}}{n}\right)\right) = \widetilde{\Theta}\left(\left(\frac{\alpha\beta_{h_{i^*}}}{n}\right)^q\right)
\end{aligned}
$$

then $F_{-1}\left(\mathbf{W}^*, \mathbf{x}\right) > F_1\left(\mathbf{W}^*, \mathbf{x}\right)$ holds, so for this kind of data, it will fail in classification. Therefore, the test error will be at least $\frac{p}{4} - \text{poly}\left(n^{-1}\right)$. □

# E    PROOF OF LEMMAS IN APPENDIX D

## E.1    PROOF OF LEMMA D.1

*Proof of Lemma D.1.* Note that at the initialization, since $\mathbf{w}_{j,r}^{(0)} \sim \mathcal{N}\left(0, \sigma_0^2\mathbf{I}_d\right)$, then denote $\mathbf{w}_{j,r}^{(0)\,u}$ the $u$-th coordinate of $\mathbf{w}_{j,r}^{(0)}$, denote $\mathbf{v}_1^u$ the $u$-th coordinate of $\mathbf{v}_1$,
then using Hoeffding's inequality, we can get that

$$
\mathbb{P}\left\{\left|\left\langle \mathbf{w}_{j,r}^{(0)}, \mathbf{v}_1\right\rangle\right| \geq a\right\} = \mathbb{P}\left\{\left|\sum_{u=1}^{d}\mathbf{w}_{j,r}^{(0)\,u}\mathbf{v}_1^u\right| \geq t\right\} \leq 2\exp\left(-\frac{ca^2}{\sigma_0^2\left\|\mathbf{v}_1\right\|^2}\right) = 2\exp\left(-\frac{ca^2}{\sigma_0^2}\right)
$$

so one can conclude that with probability exceeding $1 - 2n^{-2}$,

$$
\left|\left\langle \mathbf{w}_{j,r}^{(0)}, \mathbf{v}_1\right\rangle\right| = \left|\sum_{u=1}^{d}\mathbf{w}_{j,r}^{(0)\,u}\mathbf{v}_1^u\right| \leq \widetilde{\Theta}\left(\sigma_0\right)
$$

then with probability exceeding $1 - 4mn^{-2} \geq 1 - 4n^{-1}$, $\left|\left\langle \mathbf{w}_{j,r}^{(0)}, \mathbf{v}_1\right\rangle\right| \leq \widetilde{\Theta}\left(\sigma_0\right)$ holds for all $r \in [m]$ and all $j \in \{-1, 1\}$. Similarly, we can get that with probability exceeding $1 - 4n^{-1}$, $\left|\left\langle \mathbf{w}_{j,r}^{(0)}, \mathbf{v}\right\rangle\right| \leq \widetilde{\Theta}\left(\sigma_0\right)$ holds for all $r \in [m]$ and $j \in \{-1, 1\}$. Then we have

$$
\left[\left\langle \mathbf{w}_{-j,r}^{(0)}, j \cdot \mathbf{v}_1\right\rangle\right]_+ \leq \left|\left\langle \mathbf{w}_{-j,r}^{(0)}, \mathbf{v}_1\right\rangle\right| \leq \widetilde{\Theta}\left(\sigma_0\right)
$$

According to our calculations of update, we can get that

$$
\begin{aligned}
&\left[\left\langle \mathbf{w}_{-j,r}^{(t+1)}, j \cdot \mathbf{v}_1\right\rangle\right]_+ \\
&= \left[(1 - \eta\lambda) \cdot \left\langle \mathbf{w}_{-j,r}^{(t)}, j \cdot \mathbf{v}_1\right\rangle + \frac{\eta}{n} \cdot j \cdot \left(\sum_{s\in\mathcal{D}_2} y_{s,1}l_{-j,s,1}^{(t)}\sigma'\left(\left\langle \mathbf{w}_{-j,r}^{(t)}, y_{s,1} \cdot \mathbf{v}_1\right\rangle\right)\right)\right]_+
\end{aligned}
$$

$$\stackrel{(i)}{=} \left[(1-\eta\lambda) \cdot \left\langle \mathbf{w}_{-j,r}^{(t)}, j \cdot \mathbf{v}_1 \right\rangle - \frac{\eta}{n} \sum_{s \in \mathcal{D}_2} \left| l_{-j,s,1}^{(t)} \right| \sigma' \left( \left\langle \mathbf{w}_{-j,r}^{(t)}, y_{s,1} \cdot \mathbf{v}_1 \right\rangle \right) \right]_+$$

$$\stackrel{(ii)}{\leq} (1-\eta\lambda) \left[ \left\langle \mathbf{w}_{-j,r}^{(t)}, j \cdot \mathbf{v}_1 \right\rangle \right]_+ \leq \left[ \left\langle \mathbf{w}_{-j,r}^{(t)}, j \cdot \mathbf{v}_1 \right\rangle \right]_+$$

where (i) holds since $\text{sgn}\left(j y_{s,1} l_{-j,s,1}^{(t)}\right) = -1$, (ii) holds since $[\cdot]_+$ is a monotone function. By applying the above inequality recursively, we can get that

$$\left[ \left\langle \mathbf{w}_{-j,r}^{(t+1)}, j \cdot \mathbf{v}_1 \right\rangle \right]_+ \leq \left[ \left\langle \mathbf{w}_{-j,r}^{(t)}, j \cdot \mathbf{v}_1 \right\rangle \right]_+ \leq \left[ \left\langle \mathbf{w}_{-j,r}^{(0)}, j \cdot \mathbf{v}_1 \right\rangle \right]_+ \leq \widetilde{\Theta}(\sigma_0)$$

$\square$

### E.2 PROOF OF LEMMA D.2

*Proof of Lemma D.2.* Note that at the initialization, according to the definition, $\boldsymbol{\zeta}_{i,1,h}$ is s random vector which selects $s$ coordinates from a random vector which follows $\mathcal{N}\left(0, \sigma_p^2 \mathbf{I}_d\right)$, then denote $\mathcal{B}_{i,1,h} = \text{supp}\left(\boldsymbol{\zeta}_{i,1,h}\right)$ be the support of $\boldsymbol{\zeta}_{i,1,h}$, then $|\boldsymbol{\zeta}_{i,1,h}| = s$. Denote $\zeta_{i,1,h}^u$ the $u$-th coordinate of $\boldsymbol{\zeta}_{i,1,h}$. Since $\mathbf{w}_{j,r}^{(0)} \sim \mathcal{N}\left(0, \sigma_0^2 \mathbf{I}_s\right)$, then denote $\mathbf{w}_{j,r}^{(0)u}$ the $u$-th coordinate of $\mathbf{w}_{j,r}^{(0)}$, and using Bernstein's inequality, we can get that

$$\mathbb{P}\left\{\left|\left\langle \mathbf{w}_{j,r}^{(0)}, \boldsymbol{\zeta}_{i,1,h} \right\rangle\right| \geq a\right\} = \mathbb{P}\left\{\left|\sum_{u=1}^d \mathbf{w}_{j,r}^{(0)u} \zeta_{i,1,h}^u\right| \geq a\right\}$$

$$= \mathbb{P}\left\{\left|\sum_{u \in \mathcal{B}_{i,1,h}} \mathbf{w}_{j,r}^{(0)u} \zeta_{i,1,h}^u\right| \geq a\right\} \leq 2\exp\left(-c\min\left(\frac{a^2}{s\sigma_p^2\sigma_0^2}, \frac{t}{\sigma_p\sigma_0}\right)\right)$$

so one can conclude that with probability exceeding $1 - 2n^{-2}$,

$$\left|\left\langle \mathbf{w}_{j,r}^{(0)}, \boldsymbol{\zeta}_{i,1,h} \right\rangle\right| = \left|\sum_{u=1}^d \mathbf{w}_{j,r}^{(0)u} \zeta_{i,1,h}^u\right| \leq \widetilde{\Theta}\left(\sqrt{s}\sigma_p\sigma_0\right)$$

then with probability exceeding $1 - 4mHn^{-2} \geq 1 - 4n^{-1}$, $\left|\left\langle \mathbf{w}_{j,r}^{(0)}, \boldsymbol{\zeta}_{i,1,h} \right\rangle\right| \leq \widetilde{\Theta}\left(\sqrt{s}\sigma_p\sigma_0\right)$ holds for all $r \in [m]$, $h \in [2, H]$ and $j \in \{-1, 1\}$ Then we have

$$\left[\left\langle \mathbf{w}_{-y_{i,1},r}^{(0)}, \boldsymbol{\zeta}_{i,1,h} \right\rangle\right]_+ \leq \left|\left\langle \mathbf{w}_{-y_{i,1},r}^{(0)}, \boldsymbol{\zeta}_{i,1,h} \right\rangle\right| \leq \widetilde{\Theta}\left(\sqrt{s}\sigma_p\sigma_0\right)$$

According to our calculations of update, we can get that

$$\left[\left\langle \mathbf{w}_{-y_{i,1},r}^{(t+1)}, \boldsymbol{\zeta}_{i,1,h} \right\rangle\right]_+$$

$$= \left[(1-\eta\lambda) \cdot \left\langle \mathbf{w}_{y_{i,1},r}^{(t)}, \boldsymbol{\zeta}_{i,1,h} \right\rangle + \frac{\eta}{n} \sum_{g=2}^H \sum_{s=1}^n l_{-y_{i,1},s,1}^{(t)} \sigma'\left(\left\langle \mathbf{w}_{-y_{i,1},r}^{(t)}, \boldsymbol{\xi}_{s,1,g} \right\rangle\right) \left\langle \boldsymbol{\zeta}_{s,1,g}, \boldsymbol{\zeta}_{i,1,h} \right\rangle\right]_+$$

According to Lemma B.3, with probability exceeding $1 - n^{-2}$, for all $(s,g) \neq (i,h)$, $\left\langle \boldsymbol{\zeta}_{s,1,g}, \boldsymbol{\zeta}_{i,1,h} \right\rangle = 0$, then we have

$$\left[\left\langle \mathbf{w}_{-y_{i,1},r}^{(t+1)}, \boldsymbol{\zeta}_{i,1,h} \right\rangle\right]_+$$

$$= \left[(1-\eta\lambda) \cdot \left\langle \mathbf{w}_{y_{i,1},r}^{(t)}, \boldsymbol{\zeta}_{i,1,h} \right\rangle + \frac{\eta}{n} l_{-y_{i,1},i,1}^{(t)} \sigma'\left(\left\langle \mathbf{w}_{-y_{i,1},r}^{(t)}, \boldsymbol{\xi}_{i,1,h} \right\rangle\right) \left\langle \boldsymbol{\zeta}_{i,1,h}, \boldsymbol{\zeta}_{i,1,h} \right\rangle\right]_+$$

$$\stackrel{(i)}{=} \left[(1-\eta\lambda) \cdot \left\langle \mathbf{w}_{-y_{i,1},r}^{(t)}, \boldsymbol{\zeta}_{i,1,h} \right\rangle - \frac{\eta}{n} \left| l_{-y_{i,1},i,1}^{(t)} \right| \sigma'\left(\left\langle \mathbf{w}_{-y_{i,1},r}^{(t)}, \boldsymbol{\xi}_{i,1,h} \right\rangle\right) \|\boldsymbol{\zeta}_{i,1,h}\|^2\right]_+$$

$$\stackrel{(ii)}{\leq} (1-\eta\lambda) \left[\left\langle \mathbf{w}_{-y_{i,1},r}^{(t)}, \boldsymbol{\zeta}_{i,1,h} \right\rangle\right]_+ \leq \left[\left\langle \mathbf{w}_{-y_{i,1},r}^{(t)}, \boldsymbol{\zeta}_{i,1,h} \right\rangle\right]_+$$

where (i) holds since $\text{sgn}\left(l_{-y_{i,1},i,1}^{(t)}\right) = -1$, (ii) holds since $[\cdot]_+$ is a monotone functions. By applying the above inequality recursively, we can get that

$$\left[\left\langle \mathbf{w}_{-y_{i,1},r}^{(t+1)}, \boldsymbol{\zeta}_{i,1,h} \right\rangle\right]_+ \leq \left[\left\langle \mathbf{w}_{-y_{i,1},r}^{(t)}, \boldsymbol{\zeta}_{i,1,h} \right\rangle\right]_+ \leq \left[\left\langle \mathbf{w}_{-y_{i,1},r}^{(0)}, \boldsymbol{\zeta}_{i,1,h} \right\rangle\right]_+ \leq \widetilde{\Theta}\left(\sqrt{s}\sigma_p\sigma_0\right)$$

$\square$

### E.3 PROOF OF LEMMA D.3

*Proof of Lemma D.3.* From the calculations of initialization in Lemma D.1 and Lemma D.2, we have that with probability exceeding $1 - 12n^{-1}$, $\left|\left\langle \mathbf{w}_{j,r}^{(0)}, \mathbf{v}_1 \right\rangle\right| \leq \widetilde{\Theta}\left(\sigma_0\right)$, $\left|\left\langle \mathbf{w}_{j,r}^{(0)}, \mathbf{v} \right\rangle\right| \leq \widetilde{\Theta}\left(\sigma_0\right)$, and $\left|\left\langle \mathbf{w}_{j,r}^{(0)}, \boldsymbol{\zeta}_{i,1,h} \right\rangle\right| \leq \widetilde{\Theta}\left(\sqrt{s}\sigma_p\sigma_0\right)$ holds simultaneously for all $j \in \{-1, 1\}$, $r \in [m]$ and $h \in [2, H]$. Therefore,

$$\Psi_j^{(0)} = \max_{r \in [m]}\left[\left\langle \mathbf{w}_{j,r}^{(0)}, j \cdot \mathbf{v}_1 \right\rangle\right]_+ \leq \max_{r \in [m]}\left|\left\langle \mathbf{w}_{j,r}^{(0)}, j \cdot \mathbf{v}_1 \right\rangle\right| = \max_{r \in [m]}\left|\left\langle \mathbf{w}_{j,r}^{(0)}, \mathbf{v}_1 \right\rangle\right| \leq \widetilde{\Theta}\left(\sigma_0\right) \leq \rho$$

$$\Lambda_j^{(0)} = \max_{r \in [m]}\left[\left\langle \mathbf{w}_{j,r}^{(0)}, j \cdot \mathbf{v} \right\rangle\right]_+ \leq \max_{r \in [m]}\left|\left\langle \mathbf{w}_{j,r}^{(0)}, j \cdot \mathbf{v} \right\rangle\right| = \max_{r \in [m]}\left|\left\langle \mathbf{w}_{j,r}^{(0)}, \mathbf{v} \right\rangle\right| \leq \widetilde{\Theta}\left(\sigma_0\right) \leq \rho$$

$$\Phi_{j,i,h}^{(0)} \leq \Phi_{j,i}^{(0)} \leq \Phi_j^{(0)} = \max_{i \in [n]}\max_{h \in [2,H]}\max_{r \in [m]}\left[\left\langle \mathbf{w}_{j,r}^{(0)}, \boldsymbol{\zeta}_{i,1,h} \right\rangle\right]_+$$

$$\leq \max_{i \in [n]}\max_{h \in [2,H]}\max_{r \in [m]}\left|\left\langle \mathbf{w}_{j,r}^{(0)}, \boldsymbol{\zeta}_{i,1,h} \right\rangle\right| \leq \widetilde{\Theta}\left(\sqrt{s}\sigma_p\sigma_0\right) \leq \rho$$

holds simultaneously, which also implies that

$$\max_{i \in [n]}\max_{h \in [2,H]}\max_{r \in [m]}\left[\left\langle \mathbf{w}_{j,r}^{(0)}, \boldsymbol{\xi}_{i,1,h} \right\rangle\right]_+ = \max_{i \in [n]}\max_{h \in [2,H]}\max_{r \in [m]}\left[\left\langle \mathbf{w}_{j,r}^{(0)}, \boldsymbol{\zeta}_{i,1,h} - \beta_h y_{i,1}\mathbf{v} \right\rangle\right]_+$$

$$\leq \max_{i \in [n]}\max_{r \in [2,H]}\max_{r \in [m]}\left[\left\langle \mathbf{w}_{j,r}^{(0)}, \boldsymbol{\zeta}_{i,1,h} \right\rangle\right]_+ + \max_{r \in [m]}\left|\left\langle \mathbf{w}_{j,r}^{(0)}, \mathbf{v} \right\rangle\right|$$

$$\leq \widetilde{\Theta}\left(\sqrt{s}\sigma_p\sigma_0\right) + \widetilde{\Theta}\left(\sigma_0\right) \leq \rho$$

so that the activation function for all of them are $\frac{z^q}{q\rho^{q-1}}$.

Besides, for any $r \in [m]$, since $\mathbf{w}_{j,r}^{(0)} \sim \mathcal{N}\left(0, \sigma_0^2\mathbf{I}_d\right)$, we can get that $\left\langle \mathbf{w}_{j,r}^{(0)}, j \cdot \mathbf{v}_1 \right\rangle \sim \mathcal{N}\left(0, \sigma_0^2 \|\mathbf{v}_1\|^2\right) = \mathcal{N}\left(0, \sigma_0^2\right)$, so $\mathbb{P}\left(\left\langle \mathbf{w}_{j,r}^{(0)}, j \cdot \mathbf{v}_1 \right\rangle < \frac{\sigma_0}{2}\right)$ is an absolute constant, then we can get that

$$\mathbb{P}\left(\max_{r \in [m]}\left\langle \mathbf{w}_{j,r}^{(0)}, j \cdot \mathbf{v}_1 \right\rangle \geq \frac{\sigma_0}{2}\right) = 1 - \mathbb{P}\left(\max_{r \in [m]}\left\langle \mathbf{w}_{j,r}^{(0)}, j \cdot \mathbf{v}_1 \right\rangle < \frac{\sigma_0}{2}\right)$$

$$= 1 - \mathbb{P}\left(\left\langle \mathbf{w}_{j,r}^{(0)}, j \cdot \mathbf{v}_1 \right\rangle < \frac{\sigma_0}{2}, \forall r \in [m]\right)$$

$$= 1 - \mathbb{P}\left(\left\langle \mathbf{w}_{j,r}^{(0)}, j \cdot \mathbf{v}_1 \right\rangle < \frac{\sigma_0}{2}\right)^m$$

$$\geq 1 - n^{-1}$$

so with probability exceeding $1 - n^{-1}$, $\Psi_j^{(0)} = \max_{r \in [m]}\left[\left\langle \mathbf{w}_{j,r}^{(0)}, j \cdot \mathbf{v}_1 \right\rangle\right]_+ = \max_{r \in [m]}\left\langle \mathbf{w}_{j,r}^{(0)}, j \cdot \mathbf{v}_1 \right\rangle \geq \frac{\sigma_0}{2}$. Similarly, we can get that with probability exceeding $1 - n^{-1}$, $\Lambda_j^{(0)} = \max_{r \in [m]}\left[\left\langle \mathbf{w}_{j,r}^{(0)}, j \cdot \mathbf{v} \right\rangle\right]_+ = \max_{r \in [m]}\left\langle \mathbf{w}_{j,r}^{(0)}, j \cdot \mathbf{v} \right\rangle \geq \frac{\sigma_0}{2}$. And conditioning on $\boldsymbol{\zeta}_{i,1,h}$, we can get that $\left\langle \mathbf{w}_{j,r}^{(0)}, \boldsymbol{\zeta}_{i,1,h} \right\rangle \sim \mathcal{N}\left(0, \sigma_0^2 \|\boldsymbol{\zeta}_{i,1,h}\|^2\right)$, so $\mathbb{P}\left(\left\langle \mathbf{w}_{j,r}^{(0)}, \boldsymbol{\zeta}_{i,1,h} \right\rangle < \frac{\sigma_0\|\boldsymbol{\zeta}_{i,1,h}\|}{2}\right)$ is an absolute constant, then we can get that

$$\mathbb{P}\left(\max_{r \in [m]}\left\langle \mathbf{w}_{j,r}^{(0)}, \boldsymbol{\zeta}_{i,1,h} \right\rangle \geq \frac{\sigma_0 \|\boldsymbol{\zeta}_{i,1,h}\|}{2}\right) = 1 - \mathbb{P}\left(\max_{r \in [m]}\left\langle \mathbf{w}_{j,r}^{(0)}, \boldsymbol{\zeta}_{i,1,h} \right\rangle < \frac{\sigma_0 \|\boldsymbol{\zeta}_{i,1,h}\|}{2}\right)$$

$$= 1 - \mathbb{P}\left(\left\langle \mathbf{w}_{j,r}^{(0)}, \boldsymbol{\zeta}_{i,1,h} \right\rangle < \frac{\sigma_0 \|\boldsymbol{\zeta}_{i,1,h}\|}{2}, \forall r \in [m]\right)$$

$$= 1 - \mathbb{P}\left(\left\langle \mathbf{w}_{j,r}^{(0)}, \boldsymbol{\zeta}_{i,1,h} \right\rangle < \frac{\sigma_0 \|\boldsymbol{\zeta}_{i,1,h}\|}{2}\right)^m$$

$$\geq 1 - n^{-1}$$

so conditioning on $\boldsymbol{\zeta}_{i,1,h}$, with probability exceeding $1 - n^{-1}$, $\max_{r \in [m]}\left\langle \mathbf{w}_{j,r}^{(0)}, \boldsymbol{\zeta}_{i,1,h} \right\rangle \geq \frac{\sigma_0\|\boldsymbol{\zeta}_{i,1,h}\|}{2}$. According to the definition, $\boldsymbol{\zeta}_{i,1,h}$ is s random vector which selects $s$ coordinates from a random

vector which follows $\mathcal{N}\left(0, \sigma_p^2 \mathbf{I}_d\right)$, then denote $\mathcal{B}_{i,1,h} = \text{supp}\left(\zeta_{i,1,h}\right)$ be the support of $\zeta_{i,1,h}$, then $|\zeta_{i,1,h}| = s$, $\|\zeta_{i,1,h}\|^2 = \sum_{u \in \mathcal{B}_{i,1,h}} {\zeta_{i,1,h}^u}^2$ and for each $u \in \mathcal{B}_{i,1,h}$, $\zeta_{i,1,h}^u \sim \mathcal{N}\left(0, \sigma_p^2\right)$. Then we have that

$$\mathbb{E}\left\|\zeta_{i,1,h}\right\|^2 = \mathbb{E}\sum_{u \in \mathcal{B}_{i,1,h}} {\zeta_{i,1,h}^u}^2 = \sum_{u \in \mathcal{B}_{i,1,h}} \mathbb{E}{\zeta_{i,1,h}^u}^2 = s\sigma_p^2$$

By using Bernstein's inequality, we can get that

$$\mathbb{P}\left\{\left|\left\|\zeta_{i,1,h}\right\|^2 - \mathbb{E}\left\|\zeta_{i,1,h}\right\|^2\right| \geq a\right\} = \mathbb{P}\left\{\left|\sum_{u \in \mathcal{B}_{i,1,h}} {\zeta_{i,1,h}^u}^2 - \mathbb{E}\sum_{u \in \mathcal{B}_{i,1,h}} {\zeta_{i,1,h}^u}^2\right| \geq a\right\}$$

$$= \mathbb{P}\left\{\left|\sum_{u \in \mathcal{B}_{i,1,h}} \zeta_{i,1,h}^2 - s\sigma_p^2\right| \geq a\right\}$$

$$\leq 2\exp\left(-c\min\left(\frac{a^2}{s\sigma_p^2}, \frac{t}{\sigma_p}\right)\right)$$

so one conclude that with probability exceeding $1 - 2n^{-2}$,

$$\left|\left\|\zeta_{i,1,h}\right\|^2 - s\sigma_p^2\right| \leq \widetilde{\Theta}\left(\sqrt{s}\sigma_p^2\right)$$

which implies that which probability exceeding $1 - 2n^{-2}$,

$$\left\|\zeta_{i,1,h}\right\| \geq O\left(\sqrt{s}\sigma_p\right)$$

Combining these two parts, we have that with probability exceeding $1 - n^{-1} - 2n^{-2} \geq 1 - 2n^{-1}$,

$$\max_{r \in [m]}\left\langle\mathbf{w}_{j,r}^{(0)}, \zeta_{i,1,h}\right\rangle \geq \frac{\sigma_0\left\|\zeta_{i,1,h}\right\|}{2} \geq O\left(\sqrt{s}\sigma_p\sigma_0\right)$$

So we have that with probability exceeding $1 - 2n^{-1}$,

$$\Phi_j^{(0)} \geq \Phi_{j,i}^{(0)} \geq \Phi_{j,i,h}^{(0)} = \max_{r \in [m]}\left[\left\langle\mathbf{w}_{j,r}^{(0)}, \zeta_{i,1,h}\right\rangle\right]_+ = \max_{r \in [m]}\left\langle\mathbf{w}_{j,r}^{(0)}, \zeta_{i,1,h}\right\rangle \geq O\left(\sqrt{s}\sigma_p\sigma_0\right)$$

Therefore, with probability exceeding $1 - 8n^{-1}$,

$$\Psi_j^{(0)} = \max_{r \in [m]}\left[\left\langle\mathbf{w}_{j,r}^{(0)}, j \cdot \mathbf{v}_1\right\rangle\right]_+ \geq \frac{\sigma_0}{2}$$

$$\Lambda_j^{(0)} = \max_{r \in [m]}\left[\left\langle\mathbf{w}_{j,r}^{(0)}, j \cdot \mathbf{v}\right\rangle\right]_+ \geq \frac{\sigma_0}{2}$$

$$\Phi_j^{(0)} \geq \Phi_{j,i}^{(0)} \geq \Phi_{j,i,h}^{(0)} = \max_{r \in [m]}\left[\left\langle\mathbf{w}_{j,r}^{(0)}, \zeta_{i,1,h}\right\rangle\right]_+ \geq O\left(\sqrt{s}\sigma_p\sigma_0\right)$$

holds simultaneously for all $j \in \{-1, 1\}$. Combining these two parts, we have that with probability exceeding $1 - 20n^{-1}$,

$$\Psi_j^{(0)} = \widetilde{\Theta}\left(\sigma_0\right), \Lambda_j^{(0)} = \widetilde{\Theta}\left(\sigma_0\right), \Phi_j^{(0)} = \widetilde{\Theta}\left(\sqrt{s}\sigma_p\sigma_0\right), \Phi_{j,i}^{(0)} = \widetilde{\Theta}\left(\sqrt{s}\sigma_p\sigma_0\right), \Phi_{j,i,h}^{(0)} = \widetilde{\Theta}\left(\sqrt{s}\sigma_p\sigma_0\right)$$

Then according to our definition, it can be shown that for $i \in \mathcal{D}_1$,

$$F_j\left(\mathbf{W}^{(0)}, \mathbf{x}_{i,1}\right) = \sum_{r=1}^m\left[\sigma\left(\left\langle\mathbf{w}_{j,r}^{(0)}, \alpha y_{i,1} \cdot \mathbf{v}\right\rangle\right) + \sum_{h=2}^H \sigma\left(\left\langle\mathbf{w}_{j,r}^{(0)}, \xi_{i,1,h}\right\rangle\right)\right]$$

$$= \sum_{r=1}^m\left[\frac{\left[\left\langle\mathbf{w}_{j,r}^{(0)}, \alpha y_{i,1} \cdot \mathbf{v}\right\rangle\right]_+^q}{q\rho^{q-1}} + \sum_{h=2}^H \frac{\left[\left\langle\mathbf{w}_{j,r}^{(0)}, \xi_{i,1,h}\right\rangle\right]_+^q}{q\rho^{q-1}}\right]$$

$$\leq m\left(\frac{\rho^q}{q\rho^{q-1}} + \sum_{h=2}^H \frac{\rho^q}{q\rho^{q-1}}\right) = \frac{mH\rho}{q} = o(1)$$

for all $j \in \{-1, 1\}$, and similarly for $i \in \mathcal{D}_2$,

$$
\begin{aligned}
F_j\left(\mathbf{W}^{(0)}, \mathbf{x}_{i,1}\right) &= \sum_{r=1}^{m}\left[\sigma\left(\left\langle \mathbf{w}_{j,r}^{(0)}, y_{i,1} \cdot \mathbf{v}_1\right\rangle\right) + \sum_{h=2}^{H}\sigma\left(\left\langle \mathbf{w}_{j,r}^{(0)}, \boldsymbol{\xi}_{i,1,h}\right\rangle\right)\right] \\
&= \sum_{r=1}^{m}\left[\frac{\left[\left\langle \mathbf{w}_{j,r}^{(0)}, y_{i,1} \cdot \mathbf{v}_1\right\rangle\right]_+^q}{q\rho^{q-1}} + \sum_{h=2}^{H}\frac{\left[\left\langle \mathbf{w}_{j,r}^{(0)}, \boldsymbol{\xi}_{i,1,h}\right\rangle\right]_+^q}{q\rho^{q-1}}\right] \\
&\leq m\left(\frac{\rho^q}{q\rho^{q-1}} + \sum_{h=2}^{H}\frac{\rho^q}{q\rho^{q-1}}\right) = \frac{mH\rho}{q} = o(1)
\end{aligned}
$$

for all $j \in \{-1, 1\}$. Then we have

$$
\frac{e^{F_j\left(\mathbf{W}^{(0)}, \mathbf{x}_{i,1}\right)}}{\sum_s e^{F_s\left(\mathbf{W}^{(0)}, \mathbf{x}_{i,1}\right)}} = \Theta(1).
$$

so that

$$
\left|l_{j,i,1}^{(0)}\right| = \Theta(1).
$$

According to the definition of $l_{j,i,1}^{(t)}$, we have $\operatorname{sgn}\left(l_{j,i,1}^{(t)}\right) = jy_{i,1}$, so $\operatorname{sgn}\left(jy_{i,1}l_{j,i,1}^{(t)}\right) = 1$. We will prove the desired argument based on the following induction hypothesis:

$$
\Psi_j^{(t)} \geq \Psi_j^{(t-1)} + \frac{\eta}{2\rho^{q-1}}\Theta\left(\left(\Psi_j^{(t-1)}\right)^{q-1}\right),
$$

$$
\Psi_j^{(t)} = \max_{r \in [m]}\left\langle \mathbf{w}_{j,r}^{(t)}, j \cdot \mathbf{v}_1\right\rangle,
$$

$$
\Psi_j^{(t)} \text{ is monotonically non-decreasing}, \forall j \in \{-1,1\}, t \leq P_j \tag{1}
$$

$$
\Phi_{y_{i,1},i}^{(t)} \geq \Phi_{y_{i,1},i}^{(t-1)} + \frac{\eta}{2\rho^{q-1}}\Theta\left(\frac{s\sigma_p^2}{n}\right)\Theta\left(\left(\Phi_{y_{i,1},i}^{(t-1)} - \widetilde{\Theta}\left(\sigma_0\right)\right)^{q-1}\right),
$$

$$
\Phi_{y_{i,1},i}^{(t)} = \max_{h \in [2,H]}\max_{r \in [m]}\left\langle \mathbf{w}_{y_{i,1},r}^{(t)}, \boldsymbol{\zeta}_{i,1,h}\right\rangle,
$$

$$
\Phi_{y_{i,1},i}^{(t)} \text{ is monotonically non-decreasing}, \forall i \in \mathcal{D}_1, t \leq T_i \tag{2}
$$

$$
\Phi_{y_{i,1},i}^{(t)} \leq \Phi_{y_{i,1},i}^{(t-1)} + \frac{\eta}{\rho^{q-1}}\Theta\left(\frac{s\sigma_p^2}{n}\right)\Theta\left(\left(\Phi_{y_{i,1},i}^{(t-1)} + \widetilde{\Theta}\left(\sigma_0^{\frac{2}{3}}\right)\right)^{q-1}\right), \forall i \in \mathcal{D}_2, t \leq P_0 \tag{3}
$$

$$
\Lambda_j^{(t)} \leq \max\left\{\Lambda_j^{(t-1)} + \frac{\eta\alpha^q}{\rho^{q-1}}\Theta\left(\left(\Lambda_j^{(t-1)}\right)^{q-1}\right), \widetilde{\Theta}\left(\alpha^q\sigma_0^{\frac{2}{3}}\right)\right\}, \forall j \in \{-1,1\}, t \leq T_0 \tag{4}
$$

$$
\Psi_j^{(t)} \leq \widetilde{\Theta}(1), \forall j \in \{-1,1\}, t \leq P_j,
$$

$$
\Psi_j^{(t)} = \widetilde{\Theta}(1), \Psi_j^{(t)} = \max_{r \in [m]}\left\langle \mathbf{w}_{j,r}^{(t)}, j \cdot \mathbf{v}_1\right\rangle, \forall j \in \{-1,1\}, P_j \leq t \leq T \tag{5}
$$

$$
\Phi_{y_{i,1},i}^{(t)} \leq \widetilde{\Theta}(1), \forall i \in \mathcal{D}_1, t \leq T_i,
$$

$$
\Phi_{y_{i,1},i}^{(t)} = \widetilde{\Theta}(1), \Phi_{y_{i,1},i}^{(t)} = \max_{h \in [2,H]}\max_{r \in [m]}\left\langle \mathbf{w}_{y_{i,1},r}^{(t)}, \boldsymbol{\zeta}_{i,1,h}\right\rangle, \forall i \in \mathcal{D}_1, T_i \leq t \leq T; \tag{6}
$$

$$
\Phi_{j,i}^{(t)} \leq \widetilde{\Theta}\left(\sqrt{s}\sigma_p\sigma_0\right), \forall j \in \{-1,1\}, \forall i \in \mathcal{D}_2, t \leq T \tag{7}
$$

$$
\Lambda_j^{(t)} \leq \widetilde{\Theta}\left(\sigma_0\right), \forall j \in \{-1,1\}, t \leq T \tag{8}
$$

$$
\left\langle \mathbf{w}_{-j,r}^{(t)}, j \cdot \mathbf{v}\right\rangle \leq \widetilde{\Theta}\left(\sigma_0^{\frac{1}{3}}\right), \forall j \in \{-1,1\}, t \leq T \tag{9}
$$

and for $t = -1$, we set $\Psi_j^{(-1)} = 0$ for all $j \in \{-1, 1\}$, $\Phi_{j,i}^{(-1)} = 0$ for all $j \in \{-1, 1\}$ and $i \in \mathcal{D}_1$, $\Phi_{j,i}^{(-1)} = 1$ for all $j \in \{-1, 1\}$ and $i \in \mathcal{D}_2$, and $\Lambda_j^{(-1)} = 1$ for all $j \in \{-1, 1\}$. Now we consider

the situation at $t = 0$.

(i) In terms of Hypothesis 1 and 5, since the distribution of $\mathbf{w}_{j,r}^{(0)}$ is symmetric, then for each $r \in [m]$ and $j \in \{-1, 1\}$, with probability $\frac{1}{2}$, $\left\langle \mathbf{w}_{j,r}^{(0)}, j \cdot \mathbf{v}_1 \right\rangle < 0$. So with probability $\frac{1}{2^m}$, $\left\langle \mathbf{w}_{j,r}^{(0)}, j \cdot \mathbf{v}_1 \right\rangle < 0$ holds for all $r \in [m]$. Denote $r^* = \arg\max_{r \in [m]} \left\langle \mathbf{w}_{j,r}^{(0)}, j \cdot \mathbf{v}_1 \right\rangle$, then

$$
\begin{aligned}
&\mathbb{P}\left\{ \left\langle \mathbf{w}_{j,r^*}^{(0)}, j \cdot \mathbf{v}_1 \right\rangle \geq 0, \forall j \in \{-1, 1\} \right\} \\
&\geq 1 - \mathbb{P}\left\{ \left\langle \mathbf{w}_{1,r^*}^{(0)}, 1 \cdot \mathbf{v}_1 \right\rangle < 0 \right\} - \mathbb{P}\left\{ \left\langle \mathbf{w}_{-1,r^*}^{(0)}, -1 \cdot \mathbf{v}_1 \right\rangle < 0 \right\} \\
&= 1 - \mathbb{P}\left\{ \left\langle \mathbf{w}_{1,r}^{(0)}, 1 \cdot \mathbf{v}_1 \right\rangle < 0, \forall r \in [m] \right\} - \mathbb{P}\left\{ \left\langle \mathbf{w}_{-1,r}^{(0)}, -1 \cdot \mathbf{v}_1 \right\rangle < 0, \forall r \in [m] \right\} \\
&= 1 - \frac{1}{2^{m-1}} \geq 1 - n^{-1}
\end{aligned}
$$

so with probability exceeding $1 - n^{-1}$, $\Psi_j^{(0)} = \left\langle \mathbf{w}_{j,r^*}^{(0)}, j \cdot \mathbf{v}_1 \right\rangle \geq 0$ holds for both $j \in \{-1, 1\}$. Since $\Psi_j^{(-1)} = 0$, then the inequality

$$
\Psi_j^{(0)} \geq \Psi_j^{(-1)} + \frac{\eta}{2\rho^{q-1}} \Theta\left( \left( \Psi_j^{(-1)} \right)^{q-1} \right)
$$

holds, so we verify Hypothesis 1 at $t = 0$. According to previous calculations, we have that $\Psi_j^{(0)} = \widetilde{\Theta}(\sigma_0) \leq \widetilde{\Theta}(1)$ for all $j \in \{-1, 1\}$, so we verify Hypothesis 5 at $t = 0$.

(ii) In terms of Hypothesis 2 and 6, since both the distribution of $\mathbf{w}_{j,r}^{(0)}$ and the distribution of $\boldsymbol{\zeta}_{i,1,h}$ are symmetric, then for each $i \in [n]$, $h \in [2, H]$ and $r \in [m]$, with probability $\frac{1}{2}$, $\left\langle \mathbf{w}_{j,r}^{(0)}, \boldsymbol{\zeta}_{i,1,h} \right\rangle < 0$. So with probability $\frac{1}{2^m}$, $\left\langle \mathbf{w}_{j,r}^{(0)}, \boldsymbol{\zeta}_{i,1,h} \right\rangle < 0$ holds for all $r \in [m]$. Denote $r_{i,h}^* = \arg\max_{r \in [m]} \left\langle \mathbf{w}_{j,r}^{(0)}, \boldsymbol{\zeta}_{i,1,h} \right\rangle$, then

$$
\begin{aligned}
&\mathbb{P}\left\{ \left\langle \mathbf{w}_{y_{i,1},r_{i,h}^*}^{(0)}, \boldsymbol{\zeta}_{i,1,h} \right\rangle \geq 0, \forall i \in [n], h \in [2, H] \right\} \\
&\geq 1 - \sum_{i=1}^{n} \sum_{h=2}^{H} \mathbb{P}\left\{ \left\langle \mathbf{w}_{y_{i,1},r_{i,h}^*}^{(0)}, \boldsymbol{\zeta}_{i,1,h} \right\rangle < 0 \right\} \\
&= 1 - \sum_{i=1}^{n} \sum_{h=2}^{H} \mathbb{P}\left\{ \left\langle \mathbf{w}_{y_{i,1},r}^{(0)}, \boldsymbol{\zeta}_{i,1,h} \right\rangle < 0, \forall r \in [m] \right\} \\
&= 1 - \frac{n(H-1)}{2^m} \geq 1 - n^{-1}
\end{aligned}
$$

so with probability exceeding $1 - n^{-1}$,

$$
\begin{aligned}
\Phi_{y_{i,1},i}^{(0)} &= \max_{h \in [2,H]} \max_{r \in [m]} \left[ \left\langle \mathbf{w}_{y_{i,1},r}^{(0)}, \boldsymbol{\zeta}_{i,1,h} \right\rangle \right]_+ \\
&= \max_{h \in [2,H]} \left\langle \mathbf{w}_{y_{i,1},r_{i,h}^*}^{(0)}, \boldsymbol{\zeta}_{i,1,h} \right\rangle \\
&= \max_{h \in [2,H]} \max_{r \in [m]} \left\langle \mathbf{w}_{y_{i,1},r}^{(0)}, \boldsymbol{\zeta}_{i,1,h} \right\rangle \geq 0
\end{aligned}
$$

holds for all $i \in \mathcal{D}_1$. Since $\Phi_{y_{i,1},i}^{(-1)} = 0$ for all $i \in \mathcal{D}_1$, then the inequality

$$
\Phi_{y_{i,1},i}^{(0)} \geq \Phi_{y_{i,1},i}^{(-1)} + \frac{\eta}{2\rho^{q-1}} \widetilde{\Theta}\left( \frac{s\sigma_p^2}{n} \right) \Theta\left( \left( \Phi_{y_{i,1},i}^{(-1)} - \widetilde{\Theta}(\sigma_0) \right)^{q-1} \right)
$$

holds, so we verify Hypothesis 2 at $t = 0$. According to previous calculations, we have that $\Phi_{y_{i,1},i}^{(0)} = \widetilde{\Theta}(\sqrt{s}\sigma_p\sigma_0) \leq \widetilde{\Theta}(1)$ for all $i \in \mathcal{D}_1$, so we verify Hypothesis 6 at $t = 0$.

(iii) In terms of Hypothesis 3 and 7, from the calculations of initialization in Lemma D.2, we have that for any $i \in \mathcal{D}_2$, $\Phi_{y_{i,1},i}^{(0)} \leq \Phi_{y_{i,1}}^{(0)} \leq \widetilde{\Theta}\left(\sqrt{s}\sigma_p\sigma_0\right)$. Since $\Phi_{y_{i,1},i}^{(-1)} = 1$, then

$$\Phi_{y_{i,1},i}^{(0)} \leq \widetilde{\Theta}\left(\sqrt{s}\sigma_p\sigma_0\right) \leq 1 \leq \Phi_{y_{i,1},i}^{(-1)} + \frac{\eta}{\rho^{q-1}}\widetilde{\Theta}\left(\frac{d\sigma_p^2}{n}\right)\Theta\left(\left(\Phi_{y_{i,1},i}^{(-1)} + \widetilde{\Theta}\left(\sigma_0^{\frac{2}{3}}\right)\right)^{q-1}\right)$$

so we verify the hypothesis at $t = 0$.

(iv) In terms of Hypothesis 4 and 8, from the calculations of initialization in Lemma D.1, we have that for any $j \in \{-1, 1\}$, $\Lambda_j^{(0)} \leq \widetilde{\Theta}\left(\sigma_0\right) \leq 1 \leq \max\left\{\Lambda_j^{(-1)} + \frac{\eta\alpha^q}{\rho^{q-1}}\Theta\left(\left(\Lambda_j^{(-1)}\right)^{q-1}\right), \widetilde{\Theta}\left(\alpha^q\sigma_0^{\frac{2}{3}}\right)\right\}$, so we verify the hypothesis at $t = 0$.

(v) In terms of Hypothesis 9, from the calculations of initialization in Lemma D.1, we have that for any $j \in \{-1, 1\}$, $\left\langle\mathbf{w}_{-j,r}^{(0)}, j \cdot \mathbf{v}\right\rangle \leq \left|\left\langle\mathbf{w}_{-j,r}^{(0)}, \mathbf{v}\right\rangle\right| \leq \widetilde{\Theta}\left(\sigma_0\right) \leq \widetilde{\Theta}\left(\sigma_0^{\frac{1}{3}}\right)$, so we verify the hypothesis at $t = 0$.

By induction, we assume that all these hypotheses holds for all $\tau \in [0, t-1]$. Then we consider the case at $t$.

(i) In terms of Hypothesis 1, denote $r^* = \arg\max_{r \in [m]}\left\langle\mathbf{w}_{j,r}^{(t-1)}, j \cdot \mathbf{v}_1\right\rangle$, then we can apply Hypothesis 1 at time $t-1$ and get that $\Psi_j^{(t-1)} = \left\langle\mathbf{w}_{j,r^*}^{(t-1)}, j \cdot \mathbf{v}_1\right\rangle \geq 0$, and apparently $\Psi_j^{(t)} \geq \left\langle\mathbf{w}_{j,r^*}^{(t)}, j \cdot \mathbf{v}_1\right\rangle$, then the gradient calculation implies that

$$\left\langle\mathbf{w}_{j,r^*}^{(t)}, j \cdot \mathbf{v}_1\right\rangle = (1 - \eta\lambda) \cdot \left\langle\mathbf{w}_{j,r^*}^{(t-1)}, j \cdot \mathbf{v}_1\right\rangle + \frac{\eta}{n} \cdot j \cdot \left(\sum_{i \in \mathcal{D}_2} y_{i,1}l_{j,i,1}^{(t-1)}\sigma'\left(\left\langle\mathbf{w}_{j,r^*}^{(t-1)}, y_{i,1} \cdot \mathbf{v}_1\right\rangle\right)\right)$$

$$\overset{(i)}{=} (1 - \eta\lambda) \cdot \left\langle\mathbf{w}_{j,r^*}^{(t-1)}, j \cdot \mathbf{v}_1\right\rangle + \frac{\eta}{n} \cdot \left(\sum_{i \in \mathcal{D}_2}\left|l_{j,i,1}^{(t-1)}\right|\sigma'\left(\left\langle\mathbf{w}_{j,r^*}^{(t-1)}, y_{i,1} \cdot \mathbf{v}_1\right\rangle\right)\right)$$

$$\overset{(ii)}{\geq} (1 - \eta\lambda) \cdot \left\langle\mathbf{w}_{j,r^*}^{(t-1)}, j \cdot \mathbf{v}_1\right\rangle + \Theta(\eta)(1-p) \cdot \sigma'\left(\left\langle\mathbf{w}_{j,r^*}^{(t-1)}, j \cdot \mathbf{v}_1\right\rangle\right)$$

$$\overset{(iii)}{=} (1 - \eta\lambda) \cdot \Psi_j^{(t-1)} + \frac{\eta(1-p)}{\rho^{q-1}}\Theta\left(\left(\Psi_j^{(t-1)}\right)^{q-1}\right)$$

$$\overset{(iv)}{\geq} (1 - \eta\lambda) \cdot \Psi_j^{(t-1)} + \frac{\eta}{\rho^{q-1}}\Theta\left(\left(\Psi_j^{(t-1)}\right)^{q-1}\right) \geq 0$$

where (i) holds since $\mathrm{sgn}\left(jy_{i,1}l_{j,i,1}^{(t-1)}\right) = 1$, (ii) holds since $y_{i,1} = j$ with probability $\frac{1}{2}$ and $i \in \mathcal{D}_2$ with probability $1-p$ according to our data generation method, as well as $\left|l_{j,i,1}^{(t-1)}\right| = \Theta(1)$ according to our assumption, (iii) holds since $\Psi_j^{(t-1)} = \left\langle\mathbf{w}_{j,r^*}^{(t-1)}, j \cdot \mathbf{v}_1\right\rangle \geq 0$ according to Hypothesis 1 at $t-1$, and the activation function for $\left\langle\mathbf{w}_{j,r^*}^{(t-1)}, j \cdot \mathbf{v}_1\right\rangle$ is $\frac{z^q}{q\rho^{q-1}}$ according to our assumption, which implies that $\sigma'\left(\left\langle\mathbf{w}_{j,r^*}^{(t-1)}, j \cdot \mathbf{v}_1\right\rangle\right) = \frac{\left\langle\mathbf{w}_{j,r^*}^{(t-1)}, j \cdot \mathbf{v}_1\right\rangle^{q-1}}{\rho^{q-1}} = \frac{\left(\Psi_j^{(t-1)}\right)^{q-1}}{\rho^{q-1}}$, and (iv) holds since $p = O(1)$. So we have

$$\max_r\left\langle\mathbf{w}_{j,r}^{(t)}, j \cdot \mathbf{v}_1\right\rangle \geq \left\langle\mathbf{w}_{j,r^*}^{(t)}, j \cdot \mathbf{v}_1\right\rangle \geq 0$$

which implies that $\Psi_j^{(t)} = \max_r\left\langle\mathbf{w}_{j,r}^{(t)}, j \cdot \mathbf{v}_1\right\rangle \geq 0$. Then

$$\Psi_j^{(t)} \geq (1 - \eta\lambda)\Psi_j^{(t-1)} + \frac{\eta}{\rho^{q-1}}\Theta\left(\left(\Psi_j^{(t-1)}\right)^{q-1}\right)$$

Furthermore, according to Hypothesis 1 at $[0, t-1]$, we can get that $\Psi_j^{(\tau)}$ is monotonically non-decreasing for $\tau \in [0, t-1]$, which implies that $\Psi_j^{(t-1)} \geq \Psi_j^{(0)}$. Then $\lambda < \widetilde{\Theta}\left(\frac{\sigma_0^{q-2}}{\rho^{q-1}}\right) \leq$

$$\Theta\left(\frac{\left(\Psi_j^{(0)}\right)^{q-2}}{\rho^{q-1}}\right) \le \Theta\left(\frac{\left(\Psi_j^{(t)}\right)^{q-2}}{\rho^{q-1}}\right), \text{ which implies that}$$

$$\Psi_j^{(t)} \ge (1-\eta\lambda)\cdot\Psi_j^{(t-1)} + \frac{\eta}{\rho^{q-1}}\Theta\left(\left(\Psi_j^{(t-1)}\right)^{q-1}\right)$$

$$= \Psi_j^{(t-1)} + \frac{\eta}{2\rho^{q-1}}\Theta\left(\left(\Theta_j^{(t-1)}\right)^{q-1}\right) + \frac{\eta}{2\rho^{q-1}}\Theta\left(\left(\Psi_j^{(t-1)}\right)^{q-1}\right) - \eta\lambda\Psi_j^{(t-1)}$$

$$= \Psi_j^{(t-1)} + \frac{\eta}{2\rho^{q-1}}\Theta\left(\left(\Psi_j^{(t-1)}\right)^{q-1}\right) + \eta\Psi_j^{(t-1)}\left(\frac{\Theta\left(\left(\Psi_j^{(t-1)}\right)^{q-2}\right)}{2\rho^{q-1}} - \lambda\right)$$

$$\ge \Psi_j^{(t-1)} + \frac{\eta}{2\rho^{q-1}}\Theta\left(\left(\Psi_j^{(t-1)}\right)^{q-1}\right).$$

so the hypothesis holds at $t$. Using Lemma 5.1 with our initialization, we can conclude that for each $j \in \{-1,1\}$, $P_j = \widetilde{\Theta}\left(\left(\Psi_j^{(0)}\right)^{2-q}\left(\frac{\eta}{2\rho^{q-1}}\right)^{-1}\right) = \widetilde{\Theta}\left(\sigma_0^{2-q}/\eta\right)$.

(ii) In terms of Hypothesis 2, denote $r_h^* = \arg\max_{r\in[m]}\left\langle \mathbf{w}_{y_{i,1},r}^{(t-1)}, \boldsymbol{\zeta}_{i,1,h}\right\rangle$, then we can apply Hypothesis 2 at time $t-1$ and get that $\Phi_{y_{i,1},i}^{(t-1)} = \max_{h\in[2,H]}\max_{r\in[m]}\left\langle \mathbf{w}_{y_{i,1},r}^{(t-1)}, \boldsymbol{\zeta}_{i,1,h}\right\rangle = \max_{h\in[2,H]}\left\langle \mathbf{w}_{y_{i,1},r_h^*}^{(t-1)}, \boldsymbol{\zeta}_{i,1,h}\right\rangle \ge 0$, and apparently $\Phi_{y_{i,1},i,h}^{(t)} \ge \left\langle \mathbf{w}_{y_{i,1},r_h^*}^{(t)}, \boldsymbol{\zeta}_{i,1,h}\right\rangle$, then the gradient calculation implies that

$$\left\langle \mathbf{w}_{y_{i,1},r_h^*}^{(t)}, \boldsymbol{\zeta}_{i,1,h}\right\rangle = (1-\eta\lambda)\cdot\left\langle \mathbf{w}_{y_{i,1},r_h^*}^{(t-1)}, \boldsymbol{\zeta}_{i,1,h}\right\rangle$$
$$+ \frac{\eta}{n}\sum_{g=2}^{H}\sum_{s=1}^{n} l_{y_{i,1},s,1}^{(t-1)}\sigma'\left(\left\langle \mathbf{w}_{y_{i,1},r_h^*}^{(t-1)}, \boldsymbol{\xi}_{s,1,g}\right\rangle\right)\left\langle \boldsymbol{\zeta}_{s,1,g}, \boldsymbol{\zeta}_{i,1,h}\right\rangle$$

According to Lemma B.3, with probability exceeding $1 - n^{-2}$, for all $(s,g) \ne (i,h)$, $\left\langle \boldsymbol{\zeta}_{s,1,g}, \boldsymbol{\zeta}_{i,1,h}\right\rangle = 0$. By using the calculations above, we can get that with probability exceeding $1 - 2n^{-2}$, $\left|\|\boldsymbol{\zeta}_{i,1,h}\|^2 - s\sigma_p^2\right| \le \widetilde{\Theta}\left(\sqrt{s}\sigma_p^2\right)$, which implies that $\|\boldsymbol{\zeta}_{i,1,h}\| \ge \Theta\left(\sqrt{s}\sigma_p\right)$, then we have

$$\left\langle \mathbf{w}_{y_{i,1},r_h^*}^{(t)}, \boldsymbol{\zeta}_{i,1,h}\right\rangle \ge (1-\eta\lambda)\cdot\left\langle \mathbf{w}_{y_{i,1},r_h^*}^{(t-1)}, \boldsymbol{\zeta}_{i,1,h}\right\rangle + \frac{\eta}{n}\Theta\left(s\sigma_p^2\right) l_{y_{i,1},i,1}^{(t-1)}\sigma'\left(\left\langle \mathbf{w}_{y_{i,1},r_h^*}^{(t-1)}, \boldsymbol{\xi}_{i,1,h}\right\rangle\right)$$

According to Hypothesis 2 at $[0,t-1]$, we have $\Phi_{y_{i,1},i}^{(t-1)} = \max_{h\in[2,H]}\left\langle \mathbf{w}_{y_{i,1},r_h^*}^{(t-1)}, \boldsymbol{\zeta}_{i,1,h}\right\rangle \ge 0$ and $\Phi_{y_{i,1},i}^{(\tau)}$ is monotonically non-decreasing for $\tau \in [0,t-1]$, which implies that $\Phi_{y_{i,1},i}^{(t-1)} \ge \Phi_{y_{i,1},i}^{(0)} \ge \widetilde{\Theta}\left(\sqrt{s}\sigma_p\sigma_0\right) \ge \widetilde{\Theta}\left(\sigma_0\right)$. According to Hypothesis 8 at time $t-1$, we have $\max_{r\in[m]}\left\langle \mathbf{w}_{y_{i,1},r}^{(t-1)}, y_{i,1}\mathbf{v}\right\rangle \le \Lambda_{y_{i,1}}^{(t-1)} \le \widetilde{\Theta}\left(\sigma_0\right)$. Then taking maximum according to $h \in [2,H]$ on the both side of the inequality, we further get that

$$\max_{h\in[2,H]}\left\langle \mathbf{w}_{y_{i,1},r_h^*}^{(t)}, \boldsymbol{\zeta}_{i,1,h}\right\rangle$$

$$\overset{(i)}{\ge} (1-\eta\lambda)\cdot\max_{h\in[2,H]}\left\langle \mathbf{w}_{y_{i,1},r_h^*}^{(t-1)}, \boldsymbol{\zeta}_{i,1,h}\right\rangle + \frac{\eta}{n}\Theta\left(s\sigma_p^2\right)\left|l_{y_{i,1},i,1}^{(t-1)}\right|\max_{h\in[2,H]}\sigma'\left(\left\langle \mathbf{w}_{y_{i,1},r_h^*}^{(t-1)}, \boldsymbol{\xi}_{i,1,h}\right\rangle\right)$$

$$\overset{(ii)}{\ge} (1-\eta\lambda)\cdot\max_{h\in[2,H]}\left\langle \mathbf{w}_{y_{i,1},r_h^*}^{(t-1)}, \boldsymbol{\zeta}_{i,1,h}\right\rangle + \frac{\eta}{n}\Theta\left(s\sigma_p^2\right)\sigma'\left(\max_{h\in[2,H]}\left\langle \mathbf{w}_{y_{i,1},r_h^*}^{(t-1)}, \boldsymbol{\xi}_{i,1,h}\right\rangle\right)$$

$$= (1-\eta\lambda)\cdot\max_{h\in[2,H]}\left\langle \mathbf{w}_{y_{i,1},r_h^*}^{(t-1)}, \boldsymbol{\zeta}_{i,1,h}\right\rangle + \frac{\eta}{n}\Theta\left(s\sigma_p^2\right)\sigma'\left(\max_{h\in[2,H]}\left\langle \mathbf{w}_{y_{i,1},r_h^*}^{(t-1)}, \boldsymbol{\zeta}_{i,1,h} - \beta_h y_{i,1}\mathbf{v}\right\rangle\right)$$

$$\ge (1-\eta\lambda)\cdot\max_{h\in[2,H]}\left\langle \mathbf{w}_{y_{i,1},r_h^*}^{(t-1)}, \boldsymbol{\zeta}_{i,1,h}\right\rangle$$
$$+ \frac{\eta}{n}\Theta\left(s\sigma_p^2\right)\sigma'\left(\max_{h\in[2,H]}\left\langle \mathbf{w}_{y_{i,1},r_h^*}^{(t-1)}, \boldsymbol{\zeta}_{i,1,h}\right\rangle - \beta_h\max_{h\in[2,H]}\left\langle \mathbf{w}_{y_{i,1},r_h^*}^{(t-1)}, y_{i,1}\mathbf{v}\right\rangle\right)$$

$$\geq (1 - \eta\lambda) \cdot \max_{h \in [2,H]} \left\langle \mathbf{w}_{y_{i,1},r_h^*}^{(t-1)}, \boldsymbol{\zeta}_{i,1,h} \right\rangle$$

$$+ \frac{\eta}{n} \Theta\left(s\sigma_p^2\right) \sigma'\left( \max_{h \in [2,H]} \left\langle \mathbf{w}_{y_{i,1},r_h^*}^{(t-1)}, \boldsymbol{\zeta}_{i,1,h} \right\rangle - \max_{r \in [m]} \left\langle \mathbf{w}_{y_{i,1},r_h^*}^{(t-1)}, y_{i,1}\mathbf{v} \right\rangle \right)$$

$$\geq (1 - \eta\lambda) \cdot \max_{h \in [2,H]} \left\langle \mathbf{w}_{y_{i,1},r_h^*}^{(t-1)}, \boldsymbol{\zeta}_{i,1,h} \right\rangle + \frac{\eta}{n} \Theta\left(s\sigma_p^2\right) \sigma'\left( \max_{h \in [2,H]} \left\langle \mathbf{w}_{y_{i,1},r_h^*}^{(t-1)}, \boldsymbol{\zeta}_{i,1,h} \right\rangle - \widetilde{\Theta}\left(\sigma_0\right) \right)$$

$$= (1 - \eta\lambda) \Phi_{y_{i,1},i}^{(t-1)} + \frac{\eta}{n} \Theta\left(s\sigma_p^2\right) \sigma'\left( \Phi_{y_{i,1},i}^{(t-1)} - \widetilde{\Theta}\left(\sigma_0\right) \right)$$

$$\overset{\text{(iii)}}{=} (1 - \eta\lambda) \Phi_{y_{i,1},i}^{(t-1)} + \frac{\eta}{\rho^{q-1}} \Theta\left(\frac{s\sigma_p^2}{n}\right) \Theta\left( \left(\Phi_{y_{i,1},i}^{(t-1)} - \widetilde{\Theta}\left(\sigma_0\right)\right)^{q-1} \right) \geq 0$$

in which (i) holds since $\text{sgn}\left(l_{y_{i,1},i,1}^{(t-1)}\right) = 1$, (ii) holds since $\sigma'\left(\cdot\right)$ is a monotonic function and $\left|l_{y_{i,1},i,1}^{(t-1)}\right| = \Theta\left(1\right)$ according to our assumption, and (iii) holds since $0 \leq \Phi_{y_{i,1},i}^{(t-1)} - \widetilde{\Theta}\left(\sigma_0\right) \leq \Phi_{y_{i,1},i}^{(t-1)} \leq \rho$ according to our assumption, then the activation function for $\Phi_{y_{i,1},i}^{(t-1)}$ is $\frac{z^q}{q\rho^{q-1}}$, which implies that $\sigma'\left( \Phi_{y_{i,1},i}^{(t-1)} - \widetilde{\Theta}\left(\sigma_0\right) \right) = \frac{\left(\Phi_{y_{i,1},i}^{(t-1)} - \widetilde{\Theta}(\sigma_0)\right)^{q-1}}{\rho^{q-1}}$. So we have

$$\max_{h \in [2,H]} \left\langle \mathbf{w}_{y_{i,1},r_h^*}^{(t)}, \boldsymbol{\zeta}_{i,1,h} \right\rangle = \max_{h \in [2,H]} \max_{r \in [m]} \left\langle \mathbf{w}_{y_{i,1},r}^{(t)}, \boldsymbol{\zeta}_{i,1,h} \right\rangle \geq 0$$

which implies that $\Phi_{y_{i,1},i}^{(t)} = \max_{h \in [2,H]} \max_{r \in [m]} \left\langle \mathbf{w}_{y_{i,1},r}^{(t)}, \boldsymbol{\zeta}_{i,1,h} \right\rangle \geq 0$. Then

$$\Phi_{y_{i,1},i}^{(t)} \geq (1 - \eta\lambda) \Phi_{y_{i,1},i}^{(t-1)} + \frac{\eta}{\rho^{q-1}} \Theta\left(\frac{s\sigma_p^2}{n}\right) \Theta\left( \left(\Phi_{y_{i,1},i}^{(t-1)} - \widetilde{\Theta}\left(\sigma_0\right)\right)^{q-1} \right)$$

Furthermore, according Hypothesis 2 at $[0, t-1]$, we can get that $\Phi_{y_{i,1},i}^{(\tau)}$ is monotonically non-decreasing for $\tau \in [0, t-1]$, which implies that $\Phi_{y_{i,1},i}^{(t-1)} \geq \Phi_{y_{i,1},i}^{(0)} \geq \widetilde{\Theta}\left(\sqrt{s}\sigma_p\sigma_0\right)$, so $\Phi_{y_{i,1},i}^{(t-1)} - \widetilde{\Theta}\left(\sigma_0\right) \geq \widetilde{\Theta}\left(\sqrt{s}\sigma_p\sigma_0\right) - \widetilde{\Theta}\left(\sigma_0\right) \geq \widetilde{\Theta}\left(\sigma_0\right)$ and $\Phi_{y_{i,1},i}^{(t-1)} - \widetilde{\Theta}\left(\sigma_0\right) = \frac{\Phi_{y_{i,1},i}^{(t-1)}}{2} + \frac{\Phi_{y_{i,1},i}^{(t-1)}}{2} - \widetilde{\Theta}\left(\sigma_0\right) \geq \frac{\Phi_{y_{i,1},i}^{(t-1)}}{2} + \frac{\widetilde{\Theta}\left(\sqrt{s}\sigma_p\sigma_0\right)}{2} - \widetilde{\Theta}\left(\sigma_0\right) \geq \frac{\Phi_{y_{i,1},i}^{(t-1)}}{2}$. Then $\lambda < \Theta\left(\frac{s\sigma_p^2\sigma_0^{q-2}}{2n\rho^{q-1}}\right) \leq \Theta\left(\frac{s\sigma_p^2\left(\Phi_{y_{i,1},i}^{(0)} - \widetilde{\Theta}(\sigma_0)\right)^{q-2}}{n\rho^{q-1}}\right) \leq \Theta\left(\frac{s\sigma_p^2\left(\Phi_{y_{i,1},i}^{(t-1)} - \widetilde{\Theta}(\sigma_0)\right)^{q-2}}{n\rho^{q-1}}\right)$, which implies that

$$\Phi_{y_{i,1},i}^{(t)} \geq (1 - \eta\lambda) \Phi_{y_{i,1},i}^{(t-1)} + \frac{\eta}{\rho^{q-1}} \Theta\left(\frac{s\sigma_p^2}{n}\right) \Theta\left( \left(\Phi_{y_{i,1},i}^{(t-1)} - \widetilde{\Theta}\left(\sigma_0\right)\right)^{q-1} \right)$$

$$= \Phi_{y_{i,1},i}^{(t-1)} + \frac{\eta}{2\rho^{q-1}} \Theta\left(\frac{s\sigma_p^2}{n}\right) \Theta\left( \left(\Phi_{y_{i,1},i}^{(t-1)} - \widetilde{\Theta}\left(\sigma_0\right)\right)^{q-1} \right)$$

$$+ \frac{\eta}{2\rho^{q-1}} \Theta\left(\frac{s\sigma_p^2}{n}\right) \Theta\left( \left(\Phi_{y_{i,1},i}^{(t-1)} - \widetilde{\Theta}\left(\sigma_0\right)\right)^{q-1} \right) - \eta\lambda\Phi_{y_{i,1},i}^{(t-1)}$$

$$\geq \Phi_{y_{i,1},i}^{(t-1)} + \frac{\eta}{2\rho^{q-1}} \Theta\left(\frac{s\sigma_p^2}{n}\right) \Theta\left( \left(\Phi_{y_{i,1},i}^{(t-1)} - \widetilde{\Theta}\left(\sigma_0\right)\right)^{q-1} \right)$$

$$+ \frac{\eta}{2\rho^{q-1}} \Theta\left(\frac{s\sigma_p^2}{n}\right) \Theta\left( \left(\Phi_{y_{i,1},i}^{(t-1)} - \widetilde{\Theta}\left(\sigma_0\right)\right)^{q-1} \right) - 2\eta\lambda\left(\Phi_{y_{i,1},i}^{(t-1)} - \widetilde{\Theta}\left(\sigma_0\right)\right)$$

$$= \Phi_{y_{i,1},i}^{(t-1)} + \frac{\eta}{2\rho^{q-1}} \Theta\left(\frac{s\sigma_p^2}{n}\right) \Theta\left( \left(\Phi_{y_{i,1},i}^{(t-1)} - \widetilde{\Theta}\left(\sigma_0\right)\right)^{q-1} \right)$$

$$+ \eta \left( \Phi_{y_{i,1},i}^{(t-1)} - \widetilde{\Theta} \left( \sigma_0 \right) \right) \left( \Theta \left( \frac{s \sigma_p^2 \left( \Phi_{y_{i,1},i}^{(t-1)} - \widetilde{\Theta} \left( \sigma_0 \right) \right)^{q-2}}{2 n \rho^{q-1}} \right) - \lambda \right)$$

$$\geq \Phi_{y_{i,1},i}^{(t-1)} + \frac{\eta}{2 \rho^{q-1}} \Theta \left( \frac{s \sigma_p^2}{n} \right) \Theta \left( \left( \Phi_{y_{i,1},i}^{(t-1)} - \widetilde{\Theta} \left( \sigma_0 \right) \right)^{q-1} \right)$$

so the hypothesis holds at $t$.

(iii) In terms of Hypothesis 3, denote $r_h^* = \arg\max_{r \in [m]} \left\langle \mathbf{w}_{y_{i,1},r}^{(t)}, \boldsymbol{\zeta}_{i,1,h} \right\rangle$, then apparently $\Phi_{y_{i,1},i,h}^{(t)} = \left[ \left\langle \mathbf{w}_{y_{i,1},r_h^*}^{(t)}, \boldsymbol{\zeta}_{i,1,h} \right\rangle \right]_+$ and $\Phi_{y_{i,1},i,h}^{(t-1)} \geq \left[ \left\langle \mathbf{w}_{y_{i,1},r_h^*}^{(t-1)}, \boldsymbol{\zeta}_{i,1,h} \right\rangle \right]_+$. Then the gradient calculation implies that

$$\Phi_{y_{i,1},i,h}^{(t)} = \left[ \left\langle \mathbf{w}_{y_{i,1},r_h^*}^{(t)}, \boldsymbol{\zeta}_{i,1,h} \right\rangle \right]_+$$

$$= \left[ (1 - \eta\lambda) \cdot \left\langle \mathbf{w}_{y_{i,1},r_h^*}^{(t-1)}, \boldsymbol{\zeta}_{i,1,h} \right\rangle + \frac{\eta}{n} \sum_{g=2}^{H} \sum_{s=1}^{n} l_{y_{i,1},s,1}^{(t-1)} \sigma' \left( \left\langle \mathbf{w}_{y_{i,1},r_h^*}^{(t-1)}, \boldsymbol{\xi}_{s,1,g} \right\rangle \right) \left\langle \boldsymbol{\zeta}_{s,1,g}, \boldsymbol{\zeta}_{i,1,h} \right\rangle \right]_+$$

According to Lemma B.3, with probability exceeding $1 - n^{-2}$, for all $(s,g) \neq (i,h)$, $\left\langle \boldsymbol{\zeta}_{s,1,g}, \boldsymbol{\zeta}_{i,1,h} \right\rangle = 0$. By using the calculations above, we can get that with probability exceeding $1 - 2n^{-2}$, $\left| \|\boldsymbol{\zeta}_{i,1,h}\|^2 - s\sigma_p^2 \right| \leq \widetilde{\Theta} \left( \sqrt{s}\sigma_p^2 \right)$, which implies that $\|\boldsymbol{\zeta}_{i,1,h}\| \leq \Theta \left( \sqrt{s}\sigma_p \right)$, then we have

$$\Phi_{y_{i,1},i,h}^{(t)} = \left[ (1 - \eta\lambda) \cdot \left\langle \mathbf{w}_{y_{i,1},r_h^*}^{(t-1)}, \boldsymbol{\zeta}_{i,1,h} \right\rangle + \frac{\eta}{n} l_{y_{i,1},i,1}^{(t-1)} \sigma' \left( \left\langle \mathbf{w}_{y_{i,1},r_h^*}^{(t-1)}, \boldsymbol{\xi}_{i,1,h} \right\rangle \right) \|\boldsymbol{\zeta}_{i,1,h}\|^2 \right]_+$$

$$\overset{(i)}{\leq} (1 - \eta\lambda) \cdot \left[ \left\langle \mathbf{w}_{y_{i,1},r_h^*}^{(t-1)}, \boldsymbol{\zeta}_{i,1,h} \right\rangle \right]_+ + \frac{\eta}{n} \left| l_{y_{i,1},i,1}^{(t-1)} \right| \sigma' \left( \left\langle \mathbf{w}_{y_{i,1},r_h^*}^{(t-1)}, \boldsymbol{\xi}_{i,1,h} \right\rangle \right) \|\boldsymbol{\zeta}_{i,1,h}\|^2$$

$$\overset{(ii)}{\leq} \left[ \left\langle \mathbf{w}_{y_{i,1},r_h^*}^{(t-1)}, \boldsymbol{\zeta}_{i,1,h} \right\rangle \right]_+ + \frac{\eta}{n} \Theta \left( s\sigma_p^2 \right) \sigma' \left( \left\langle \mathbf{w}_{y_{i,1},r_h^*}^{(t-1)}, \boldsymbol{\xi}_{i,1,h} \right\rangle \right)$$

$$\overset{(iii)}{=} \left[ \left\langle \mathbf{w}_{y_{i,1},r_h^*}^{(t-1)}, \boldsymbol{\zeta}_{i,1,h} \right\rangle \right]_+ + \frac{\eta}{\rho^{q-1}} \Theta \left( \frac{s\sigma_p^2}{n} \right) \Theta \left( \left[ \left\langle \mathbf{w}_{y_{i,1},r_h^*}^{(t-1)}, \boldsymbol{\xi}_{i,1,h} \right\rangle \right]_+ \right)^{q-1}$$

$$\overset{(iv)}{\leq} \left[ \left\langle \mathbf{w}_{y_{i,1},r_h^*}^{(t-1)}, \boldsymbol{\zeta}_{i,1,h} \right\rangle \right]_+$$
$$+ \frac{\eta}{\rho^{q-1}} \Theta \left( \frac{s\sigma_p^2}{n} \right) \Theta \left( \left[ \left\langle \mathbf{w}_{y_{i,1},r_h^*}^{(t-1)}, \boldsymbol{\zeta}_{i,1,h} \right\rangle \right]_+ + \beta_h \left[ \left\langle \mathbf{w}_{y_{i,1},r_h^*}^{(t-1)}, y_{i,1}\mathbf{v} \right\rangle \right]_+ \right)^{q-1}$$

$$\leq \Phi_{y_{i,1},i}^{(t-1)} + \frac{\eta}{\rho^{q-1}} \Theta \left( \frac{s\sigma_p^2}{n} \right) \Theta \left( \left( \Phi_{y_{i,1},i}^{(t-1)} + \beta_h \left\langle \mathbf{w}_{y_{i,1},r_h^*}^{(t-1)}, -y_{i,1}\mathbf{v} \right\rangle \right)^{q-1} \right)$$

$$\overset{(v)}{\leq} \Phi_{y_{i,1},i}^{(t-1)} + \frac{\eta}{\rho^{q-1}} \Theta \left( \frac{s\sigma_p^2}{n} \right) \Theta \left( \left( \Phi_{y_{i,1},i}^{(t-1)} + \widetilde{\Theta} \left( \sigma_0^{\frac{2}{3}} \right) \right)^{q-1} \right)$$

where (i) holds according to the definition of $[\cdot]_+$ and the triangle inequality, (ii) holds since $\left| l_{y_{i,1},i,1}^{(t-1)} \right| \leq 1$, (iii) holds since the activation function for $\left\langle \mathbf{w}_{y_{i,1},r_h^*}^{(t-1)}, \boldsymbol{\xi}_{i,1,h} \right\rangle$ is $\frac{z^q}{q\rho^{q-1}}$ according to our assumption, which implies that $\sigma' \left( \left\langle \mathbf{w}_{y_{i,1},r_h^*}^{(t-1)}, \boldsymbol{\xi}_{i,1,h} \right\rangle \right) = \frac{\left( \left[ \left\langle \mathbf{w}_{y_{i,1},r_h^*}^{(t-1)}, \boldsymbol{\xi}_{i,1,h} \right\rangle \right]_+ \right)^{q-1}}{\rho^{q-1}}$, (iv) holds according to the triangle inequality, and (v) holds since $\left\langle \mathbf{w}_{y_{i,1},r_h^*}^{(t-1)}, -y_{i,1}\mathbf{v} \right\rangle \leq \widetilde{\Theta} \left( \sigma_0^{\frac{1}{3}} \right)$ according to Hypothesis 9 at $t - 1$. Taking maximum according to $h \in [2, H]$ on the left hand side of the inequality, we further get that

$$\Phi_{y_{i,1},i}^{(t)} \leq \Phi_{y_{i,1},i}^{(t-1)} + \frac{\eta}{\rho^{q-1}} \Theta \left( \frac{s\sigma_p^2}{n} \right) \Theta \left( \left( \Phi_{y_{i,1},i}^{(t-1)} + \widetilde{\Theta} \left( \sigma_0^{\frac{2}{3}} \right) \right)^{q-1} \right)$$

so the hypothesis holds at $t$.

(iv) In terms of Hypothesis 4, denote $r^* = \arg\max_{r \in [m]} \left\langle \mathbf{w}_{j,r}^{(t)}, j \cdot \mathbf{v} \right\rangle$, then apparently $\Lambda_j^{(t)} = \left[ \left\langle \mathbf{w}_{j,r^*}^{(t)}, j \cdot \mathbf{v} \right\rangle \right]_+$ and $\Lambda_j^{(t-1)} \geq \left[ \left\langle \mathbf{w}_{j,r^*}^{(t-1)}, j \cdot \mathbf{v} \right\rangle \right]_+$. So we can get from the gradient calculation that

$$
\begin{aligned}
\Lambda_j^{(t)} &= \left[ \left\langle \mathbf{w}_{j,r^*}^{(t)}, j \cdot \mathbf{v} \right\rangle \right]_+ \\
&= \left[ (1 - \eta\lambda) \cdot \left\langle \mathbf{w}_{j,r^*}^{(t-1)}, j \cdot \mathbf{v} \right\rangle + \frac{\eta}{n} \cdot j' \cdot \left( \sum_{i \in \mathcal{D}_1} \alpha y_{i,1} l_{j,i,1}^{(t-1)} \sigma' \left( \left\langle \mathbf{w}_{j,r^*}^{(t-1)}, \alpha y_{i,1} \cdot \mathbf{v} \right\rangle \right) \right. \right. \\
&\quad \left. \left. - \sum_{h=2}^{H} \sum_{i=1}^{n} \beta_h y_{i,1} l_{j',i,1}^{(t-1)} \sigma' \left( \left\langle \mathbf{w}_{j,r^*}^{(t-1)}, \boldsymbol{\xi}_{i,1,h} \right\rangle \right) \right) \right]_+ \\
&\overset{(i)}{=} \left[ (1 - \eta\lambda) \cdot \left\langle \mathbf{w}_{j,r^*}^{(t-1)}, j \cdot \mathbf{v} \right\rangle + \frac{\eta}{n} \cdot \sum_{i \in \mathcal{D}_1} \alpha \left| l_{j,i,1}^{(t-1)} \right| \sigma' \left( \left\langle \mathbf{w}_{j,r^*}^{(t-1)}, \alpha y_{i,1} \cdot \mathbf{v} \right\rangle \right) \right. \\
&\quad \left. - \frac{\eta}{n} \sum_{h=2}^{H} \sum_{i=1}^{n} \beta_h \left| l_{j,i,1}^{(t-1)} \right| \sigma' \left( \left\langle \mathbf{w}_{j,r^*}^{(t-1)}, \boldsymbol{\xi}_{i,1,h} \right\rangle \right) \right]_+ \\
&\overset{(ii)}{\leq} \left[ (1 - \eta\lambda) \left\langle \mathbf{w}_{j,r^*}^{(t-1)}, j \cdot \mathbf{v} \right\rangle + \frac{\eta}{n} \cdot \sum_{i \in \mathcal{D}_1} \alpha \left| l_{j,i,1}^{(t-1)} \right| \sigma' \left( \left\langle \mathbf{w}_{j,r^*}^{(t-1)}, \alpha y_{i,1} \cdot \mathbf{v} \right\rangle \right) \right]_+ \\
&\overset{(iii)}{\leq} (1 - \eta\lambda) \left[ \left\langle \mathbf{w}_{j,r^*}^{(t-1)}, j \cdot \mathbf{v} \right\rangle \right]_+ + \left[ \frac{\eta}{n} \cdot \sum_{i \in \mathcal{D}_1} \alpha \left| l_{j,i,1}^{(t-1)} \right| \sigma' \left( \left\langle \mathbf{w}_{j,r^*}^{(t-1)}, \alpha y_{i,1} \cdot \mathbf{v} \right\rangle \right) \right]_+ \\
&= (1 - \eta\lambda) \cdot \left[ \left\langle \mathbf{w}_{j,r^*}^{(t-1)}, j \cdot \mathbf{v} \right\rangle \right]_+ + \frac{\eta}{n} \cdot \left( \sum_{i \in \mathcal{D}_1} \alpha \left| l_{j,i,1}^{(t-1)} \right| \sigma' \left( \left\langle \mathbf{w}_{j,r^*}^{(t-1)}, \alpha y_{i,1} \cdot \mathbf{v} \right\rangle \right) \right) \\
&\leq \left[ \left\langle \mathbf{w}_{j,r^*}^{(t-1)}, j \cdot \mathbf{v} \right\rangle \right]_+ + \frac{\eta}{n} \cdot \left( \sum_{i=1}^{n} \alpha \left| l_{j,i,1}^{(t-1)} \right| \sigma' \left( \left\langle \mathbf{w}_{j,r^*}^{(t-1)}, \alpha y_{i,1} \cdot \mathbf{v} \right\rangle \right) \right) \\
&= \left[ \left\langle \mathbf{w}_{j,r^*}^{(t-1)}, j \cdot \mathbf{v} \right\rangle \right]_+ + \frac{\eta}{n} \cdot \left( \sum_{i:y_{i,1}=j} \alpha \left| l_{j,i,1}^{(t-1)} \right| \sigma' \left( \left\langle \mathbf{w}_{j,r^*}^{(t-1)}, \alpha y_{i,1} \cdot \mathbf{v} \right\rangle \right) \right. \\
&\quad \left. + \sum_{i:y_{i,1}=-j} \alpha \left| l_{j,i,1}^{(t-1)} \right| \sigma' \left( \left\langle \mathbf{w}_{j,r^*}^{(t-1)}, \alpha y_{i,1} \cdot \mathbf{v} \right\rangle \right) \right) \\
&= \left[ \left\langle \mathbf{w}_{j,r^*}^{(t-1)}, j \cdot \mathbf{v} \right\rangle \right]_+ + \frac{\eta}{n} \cdot \left( \sum_{i:y_{i,1}=j} \alpha \left| l_{j,i,1}^{(t-1)} \right| \sigma' \left( \left\langle \mathbf{w}_{j,r^*}^{(t-1)}, \alpha j \cdot \mathbf{v} \right\rangle \right) \right. \\
&\quad \left. + \sum_{i:y_{i,1}=-j} \alpha \left| l_{j,i,1}^{(t-1)} \right| \sigma' \left( \left\langle \mathbf{w}_{j,r^*}^{(t-1)}, \alpha \cdot (-j) \cdot \mathbf{v} \right\rangle \right) \right) \\
&\overset{(iv)}{\leq} \left[ \left\langle \mathbf{w}_{j,r^*}^{(t-1)}, j \cdot \mathbf{v} \right\rangle \right]_+ + \Theta(\eta) \cdot \alpha \sigma' \left( \left\langle \mathbf{w}_{j,r^*}^{(t-1)}, \alpha j \cdot \mathbf{v} \right\rangle \right) \\
&\quad + \Theta(\eta) \cdot \alpha \sigma' \left( \left\langle \mathbf{w}_{j,r^*}^{(t-1)}, \alpha \cdot (-j) \mathbf{v} \right\rangle \right)
\end{aligned}
$$

where (i) holds since $\text{sgn}\left( j y_{i,1} l_{j,i,1}^{(t-1)} \right) = 1$, (ii) holds since $[\cdot]_+$ is a monotone function, (iii) holds because of the triangle inequality, and (iv) holds since $\left| l_{j,i,1}^{(t-1)} \right| \leq 1$ and $y_{i,1} = j$ with probability $\frac{1}{2}$ according to our data generation methodThen there are two situations:

1. If $\left\langle \mathbf{w}_{j,r*}^{(t-1)}, j \cdot \mathbf{v} \right\rangle \geq 0$, then $\left\langle \mathbf{w}_{j,r}^{(t-1)}, \alpha \cdot (-j) \cdot \mathbf{v} \right\rangle \leq 0$, which implies that

$\sigma'\left(\left\langle \mathbf{w}_{j,r*}^{(t-1)}, \alpha \cdot (-j) \cdot \mathbf{v} \right\rangle\right) = 0$, then we have

$$\Lambda_j^{(t)} \le \Lambda_j^{(t-1)} + \Theta(\eta) \cdot \alpha \sigma'\left(\alpha \Lambda_j^{(t-1)}\right)$$

$$\overset{(i)}{\le} \Lambda_j^{(t-1)} + \frac{\eta \alpha^q}{\rho^{q-1}} \Theta\left(\left(\Lambda_j^{(t-1)}\right)^{q-1}\right)$$

$$\le \max\left\{\Lambda_j^{(t-1)} + \frac{\eta \alpha^q}{\rho^{q-1}} \Theta\left(\left(\Lambda_j^{(t-1)}\right)^{q-1}\right), \widetilde{\Theta}(\sigma_0)\right\}$$

where (i) holds since the activation function for $\alpha \Lambda_j^{(t-1)}$ is $\frac{z^q}{q\rho^{q-1}}$ according to our assumption, which implies that $\sigma'\left(\alpha \Lambda_j^{(t-1)}\right) = \frac{\alpha^{q-1}\left(\Lambda_j^{(t-1)}\right)^{q-1}}{\rho^{q-1}}$.

2. If $\left\langle \mathbf{w}_{j,r*}^{(t-1)}, j \cdot \mathbf{v} \right\rangle < 0$, then $\left\langle \mathbf{w}_{j,r*}^{(t-1)}, \alpha j' \cdot \mathbf{v} \right\rangle < 0$, which implies that $\left[\left\langle \mathbf{w}_{j,r*}^{(t-1)}, j \cdot \mathbf{v} \right\rangle\right]_+ = \sigma'\left(\left\langle \mathbf{w}_{j,r*}^{(t-1)}, \alpha j \cdot \mathbf{v} \right\rangle\right) = 0$, then we have

$$\Lambda_j^{(t)} \le \Theta(\eta) \cdot \alpha \sigma'\left(\left\langle \mathbf{w}_{j,r*}^{(t-1)}, \alpha \cdot (-j) \cdot \mathbf{v} \right\rangle\right)$$

$$\overset{(i)}{\le} \Theta(\eta) \cdot \alpha \sigma'\left(\alpha \widetilde{\Theta}\left(\sigma_0^{\frac{1}{3}}\right)\right)$$

$$\overset{(ii)}{\le} \frac{\eta}{\rho^{q-1}} \alpha^q \widetilde{\Theta}\left(\sigma_0^{\frac{q-1}{3}}\right) \le \frac{\eta}{\rho^{q-1}} \alpha^q \widetilde{\Theta}\left(\sigma_0^{\frac{2}{3}}\right)$$

$$\le \max\left\{\Lambda_j^{(t-1)} + \frac{\eta \alpha^q}{\rho^{q-1}} \Theta\left(\left(\Lambda_j^{(t-1)}\right)^{q-1}\right), \widetilde{\Theta}\left(\alpha^q \sigma_0^{\frac{2}{3}}\right)\right\}$$

where (i) holds since $\left[\left\langle \mathbf{w}_{j,r*}^{(t-1)}, j \cdot \mathbf{v} \right\rangle\right]_+ \le \widetilde{\Theta}\left(\sigma_0^{\frac{1}{3}}\right)$ according to Hypothesis 9 at $t-1$, and (ii) holds since the activation function for $\alpha \widetilde{\Theta}\left(\sigma_0^{\frac{1}{3}}\right)$ is $\frac{z^q}{q\rho^{q-1}}$, which implies that $\sigma'\left(\alpha \widetilde{\Theta}\left(\sigma_0^{\frac{1}{3}}\right)\right) = \frac{\left(\alpha \widetilde{\Theta}\left(\sigma_0^{\frac{1}{3}}\right)\right)^{q-1}}{\rho^{q-1}} = \frac{\alpha^{q-1}\widetilde{\Theta}\left(\sigma_0^{\frac{q-1}{3}}\right)}{\rho^{q-1}}$.

Combining these two situations, the hypothesis holds at $t$.

(v) In terms of Hypothesis 5, there are two different stages:

1. If $t \le P_j$, then according to our definition of $P_j$, $\Psi_j^{(P_j)}$ is the first time in the sequence that reaches $\Theta(1/m)$, which implies that $\Psi_j^{(t)} \le \Theta(1/m) = \widetilde{\Theta}(1)$;

2. If $t \ge P_j$, without loss of generality, we will prove the case when $j = 1$. Now we want to prove the following inequality by induction. For $\tau \in [0, t]$,

$$\Psi_1^{(\tau)} \ge (1 - \eta\lambda) \Psi_1^{(\tau-1)} + \frac{\eta}{n}\left(\sum_{i:y_{i,1}=1,i\in\mathcal{D}_2} \left|l_{1,i,1}^{(\tau)}\right| \sigma'\left(\Psi_1^{(\tau-1)}\right)\right), \quad \text{and} \quad \Psi_1^{(\tau)} = \max_{r\in[m]} \left\langle \mathbf{w}_{1,r}^{(\tau)}, \mathbf{v}_1 \right\rangle$$

and we set $\Psi_1^{(-1)} = 0$.

For $\tau = 0$, according to previous calculations, we have $\Psi_1^{(0)} = \widetilde{\Theta}(\sigma_0)$. Since $\Psi_1^{(-1)} = 0$, then the inequality holds at $\tau = 0$, so we verify the hypothesis at $\tau = 0$. By induction, we assume that for any $\tau_0 \in [0, \tau-1]$, $\Psi_1^{(\tau_0)} = \max_{r\in[m]} \left\langle \mathbf{w}_{1,r}^{(\tau_0)}, \mathbf{v}_1 \right\rangle \ge 0$, then for $\tau$, denote $r^* = \arg\max_{r\in[m]} \left\langle \mathbf{w}_{1,r}^{(\tau-1)}, j \cdot \mathbf{v}_1 \right\rangle$. Then we can apply the hypothesis at time $\tau-1$ and get $\Psi_1^{(\tau-1)} = \left\langle \mathbf{w}_{1,r*}^{(\tau-1)}, \mathbf{v}_1 \right\rangle \ge 0$ and apparently $\Psi_1^{(\tau)} \ge \left\langle \mathbf{w}_{1,r*}^{(\tau)}, j \cdot \mathbf{v}_1 \right\rangle$. From the gradient calculations on $\left\langle \mathbf{w}_{1,r*}^{(t+1)}, \mathbf{v}_1 \right\rangle$, we can get that

$$\left\langle \mathbf{w}_{1,r*}^{(\tau)}, \mathbf{v}_1 \right\rangle = (1 - \eta\lambda) \cdot \left\langle \mathbf{w}_{1,r*}^{(\tau-1)}, \mathbf{v}_1 \right\rangle + \frac{\eta}{n} \cdot \left(\sum_{i\in\mathcal{D}_2} y_{i,1} l_{1,i,1}^{(\tau)} \sigma'\left(\left\langle \mathbf{w}_{1,r*}^{(\tau-1)}, y_{i,1} \cdot \mathbf{v}_1 \right\rangle\right)\right)$$

$$\overset{(i)}{=} (1 - \eta\lambda) \cdot \left\langle \mathbf{w}_{1,r*}^{(\tau-1)}, \mathbf{v}_1 \right\rangle + \frac{\eta}{n} \cdot \left(\sum_{i\in\mathcal{D}_2} \left|l_{1,i,1}^{(\tau-1)}\right| \sigma'\left(\left\langle \mathbf{w}_{1,r*}^{(\tau-1)}, y_{i,1} \cdot \mathbf{v}_1 \right\rangle\right)\right)$$

$$\geq (1 - \eta\lambda)\,\Psi_1^{(\tau-1)} + \frac{\eta}{n}\left(\sum_{i:y_{i,1}=1, i\in\mathcal{D}_2}\left|l_{1,i,1}^{(\tau-1)}\right|\sigma'\left(\left\langle \mathbf{w}_{1,r^*}^{(\tau-1)}, y_{i,1}\cdot\mathbf{v}_1\right\rangle\right)\right)$$

$$= (1 - \eta\lambda)\,\Psi_1^{(\tau-1)} + \frac{\eta}{n}\left(\sum_{i:y_{i,1}=1, i\in\mathcal{D}_2}\left|l_{1,i,1}^{(\tau-1)}\right|\sigma'\left(\left\langle \mathbf{w}_{1,r^*}^{(\tau-1)}, \mathbf{v}_1\right\rangle\right)\right)$$

$$= (1 - \eta\lambda)\,\Psi_1^{(\tau-1)} + \frac{\eta}{n}\left(\sum_{i:y_{i,1}=1, i\in\mathcal{D}_2}\left|l_{1,i,1}^{(\tau-1)}\right|\sigma'\left(\Psi_1^{(\tau)}\right)\right)$$

where (i) holds since $\mathrm{sgn}\left(1\cdot y_{i,1}l_{1,i,1}^{(\tau-1)}\right) = 1$. Then $\max_{r\in[m]}\left\langle \mathbf{w}_{1,r}^{(\tau)}, \mathbf{v}_1\right\rangle \geq \left\langle \mathbf{w}_{1,r^*}^{(\tau)}, \mathbf{v}_1\right\rangle \geq 0$, which implies that $\Psi_1^{(\tau)} = \max_{r\in[m]}\left\langle \mathbf{w}_{1,r}^{(\tau)}, \mathbf{v}_1\right\rangle \geq 0$. So we have

$$\Psi_1^{(\tau)} \geq (1 - \eta\lambda)\,\Psi_1^{(\tau-1)} + \frac{\eta}{n}\left(\sum_{i:y_{i,1}=1, i\in\mathcal{D}_2}\left|l_{1,i,1}^{(t-1)}\right|\sigma'\left(\Psi_1^{(t-1)}\right)\right)$$

So we have the hypothesis and the inequality hold at $\tau$. Therefore, by induction, they holds for all $\tau \in [0, t]$.
Then we will prove the following inequality. For $\tau \in [1, t]$,

$$\Psi_1^{(\tau)} \leq \max\left\{(1 - \eta\lambda)\,\Psi_1^{(\tau-1)} + \frac{\eta}{n}\left(\sum_{i:y_{i,1}=1, i\in\mathcal{D}_2}\left|l_{1,i,1}^{(\tau-1)}\right|\sigma'\left(\Psi_1^{(\tau)}\right)\right), \widetilde{\Theta}\left(\sigma_0^{q-1}\right)\right\}.$$

Denote $r^* = \arg\max_{r\in[m]}\left\langle \mathbf{w}_{1,r}^{(\tau)}, \mathbf{v}_1\right\rangle$, then from the above results we can get that $\Psi_1^{(\tau)} = \left\langle \mathbf{w}_{1,r^*}^{(\tau)}, \mathbf{v}_1\right\rangle$ and $\Psi_1^{(\tau-1)} \geq \left\langle \mathbf{w}_{1,r^*}^{(\tau-1)}, \mathbf{v}_1\right\rangle$. From Lemma D.1 we can get that $\left\langle \mathbf{w}_{1,r^*}^{(\tau-1)}, -1\cdot\mathbf{v}_1\right\rangle \leq \widetilde{\Theta}\left(\sigma_0\right) \leq \rho$, so the activation function for $\left\langle \mathbf{w}_{1,r^*}^{(\tau-1)}, -1\cdot\mathbf{v}_1\right\rangle$ and $\widetilde{\Theta}\left(\sigma_0\right)$ is $\frac{z^q}{q\rho^{q-1}}$, which implies that $\sigma'\left(\widetilde{\Theta}\left(\sigma_0\right)\right) = \frac{(\widetilde{\Theta}(\sigma_0))^{q-1}}{\rho^{q-1}} = \widetilde{\Theta}\left(\sigma_0^{q-1}\right)$. Then from the gradient calculations on $\Psi_1^{(\tau)}$, we have

$$\Psi_1^{(\tau)} = \left\langle \mathbf{w}_{1,r^*}^{(\tau)}, \mathbf{v}_1\right\rangle = (1 - \eta\lambda)\cdot\left\langle \mathbf{w}_{1,r^*}^{(\tau-1)}, \mathbf{v}_1\right\rangle + \frac{\eta}{n}\cdot\left(\sum_{i\in\mathcal{D}_2}y_{i,1}l_{1,i,1}^{(\tau-1)}\sigma'\left(\left\langle \mathbf{w}_{1,r^*}^{(t-1)}, y_{i,1}\cdot\mathbf{v}_1\right\rangle\right)\right)$$

$$\overset{(i)}{=} (1 - \eta\lambda)\cdot\left\langle \mathbf{w}_{1,r^*}^{(\tau-1)}, \mathbf{v}_1\right\rangle + \frac{\eta}{n}\left(\sum_{i\in\mathcal{D}_2}\left|l_{1,i,1}^{(\tau-1)}\right|\sigma'\left(\left\langle \mathbf{w}_{1,r^*}^{(\tau-1)}, y_{i,1}\cdot\mathbf{v}_1\right\rangle\right)\right)$$

$$= (1 - \eta\lambda)\cdot\left\langle \mathbf{w}_{1,r^*}^{(\tau-1)}, \mathbf{v}_1\right\rangle + \frac{\eta}{n}\left(\sum_{i:y_{i,1}=1, i\in\mathcal{D}_2}\left|l_{1,i,1}^{(\tau-1)}\right|\sigma'\left(\left\langle \mathbf{w}_{1,r^*}^{(\tau-1)}, y_{i,1}\cdot\mathbf{v}_1\right\rangle\right)\right.$$

$$\left. + \sum_{i:y_{i,1}=-1, i\in\mathcal{D}_2}\left|l_{1,i,1}^{(\tau-1)}\right|\sigma'\left(\left\langle \mathbf{w}_{1,r^*}^{(\tau-1)}, y_{i,1}\cdot\mathbf{v}_1\right\rangle\right)\right)$$

$$= (1 - \eta\lambda)\cdot\left\langle \mathbf{w}_{1,r^*}^{(\tau-1)}, \mathbf{v}_1\right\rangle + \frac{\eta}{n}\left(\sum_{i:y_{i,1}=1, i\in\mathcal{D}_2}\left|l_{1,i,1}^{(\tau-1)}\right|\sigma'\left(\left\langle \mathbf{w}_{1,r^*}^{(\tau-1)}, \mathbf{v}_1\right\rangle\right)\right.$$

$$\left. + \sum_{i:y_{i,1}=-1, i\in\mathcal{D}_2}\left|l_{1,i,1}^{(\tau-1)}\right|\sigma'\left(\left\langle \mathbf{w}_{1,r^*}^{(\tau-1)}, -1\cdot\mathbf{v}_1\right\rangle\right)\right)$$

where (i) holds since $\mathrm{sgn}\left(1\cdot y_{i,1}l_{1,i,1}^{(t-1)}\right) = 1$. Then there are two situations:

1.    If $\left\langle \mathbf{w}_{1,r^*}^{(\tau-1)}, \mathbf{v}_1\right\rangle \geq 0$, then $\left\langle \mathbf{w}_{1,r^*}^{(\tau-1)}, -1\cdot\mathbf{v}_1\right\rangle \leq 0$, which implies that

$\sigma'\left(\left\langle \mathbf{w}_{1,r^*}^{(\tau-1)}, -1\cdot\mathbf{v}_1\right\rangle\right) = 0$, then we have

$$\Psi_1^{(\tau)} = (1-\eta\lambda)\cdot\left\langle \mathbf{w}_{1,r^*}^{(\tau-1)}, \mathbf{v}_1\right\rangle + \frac{\eta}{n}\left(\sum_{i:y_{i,1}=1,i\in\mathcal{D}_2}\left|l_{1,i,1}^{(\tau-1)}\right|\sigma'\left(\left\langle \mathbf{w}_{1,r^*}^{(\tau-1)}, \mathbf{v}_1\right\rangle\right)\right.$$

$$+ \sum_{i:y_{i,1}=-1,i\in\mathcal{D}_2}\left|l_{1,i,1}^{(\tau-1)}\right|\sigma'\left(\left\langle \mathbf{w}_{1,r^*}^{(\tau-1)}, -1\cdot\mathbf{v}_1\right\rangle\right)\right)$$

$$= (1-\eta\lambda)\cdot\left\langle \mathbf{w}_{1,r^*}^{(\tau-1)}, \mathbf{v}_1\right\rangle + \frac{\eta}{n}\left(\sum_{i:y_{i,1}=1,i\in\mathcal{D}_2}\left|l_{1,i,1}^{(\tau-1)}\right|\sigma'\left(\left\langle \mathbf{w}_{1,r^*}^{(\tau-1)}, \mathbf{v}_1\right\rangle\right)\right)$$

$$\leq (1-\eta\lambda)\Psi_1^{(\tau-1)} + \frac{\eta}{n}\left(\sum_{i:y_{i,1}=1,i\in\mathcal{D}_2}\left|l_{1,i,1}^{(\tau-1)}\right|\sigma'\left(\Psi_1^{(t-1)}\right)\right)$$

$$\leq \max\left\{(1-\eta\lambda)\Psi_1^{(\tau-1)} + \frac{\eta}{n}\left(\sum_{i:y_{i,1}=1,i\in\mathcal{D}_2}\left|l_{1,i,1}^{(\tau-1)}\right|\sigma'\left(\Psi_1^{(t-1)}\right)\right), \widetilde{\Theta}\left(\sigma_0^{q-1}\right)\right\}.$$

Therefore, by induction, this inequality holds for all $\tau\in[1,t]$.

2. If $\left\langle \mathbf{w}_{1,r^*}^{(\tau-1)}, \mathbf{v}_1\right\rangle < 0$, which implies that $\sigma'\left(\left\langle \mathbf{w}_{1,r^*}^{(\tau-1)}, \mathbf{v}_1\right\rangle\right) = 0$. From Lemma D.1 we can get that $\left\langle \mathbf{w}_{1,r^*}^{(\tau-1)}, -1\cdot\mathbf{v}_1\right\rangle \leq \widetilde{\Theta}(\sigma_0) \leq \rho$, so the activation function for $\left\langle \mathbf{w}_{1,r^*}^{(\tau-1)}, -1\cdot\mathbf{v}_1\right\rangle$ and $\widetilde{\Theta}(\sigma_0)$ is $\frac{z^q}{q\rho^{q-1}}$, which implies that $\sigma'\left(\widetilde{\Theta}(\sigma_0)\right) = \frac{(\widetilde{\Theta}(\sigma_0))^{q-1}}{\rho^{q-1}} = \widetilde{\Theta}\left(\sigma_0^{q-1}\right)$. Then we have

$$\Psi_1^{(\tau)} = (1-\eta\lambda)\cdot\left\langle \mathbf{w}_{1,r^*}^{(\tau-1)}, \mathbf{v}_1\right\rangle + \frac{\eta}{n}\left(\sum_{i:y_{i,1}=1,i\in\mathcal{D}_2}\left|l_{1,i,1}^{(\tau-1)}\right|\sigma'\left(\left\langle \mathbf{w}_{1,r^*}^{(\tau-1)}, \mathbf{v}_1\right\rangle\right)\right.$$

$$+ \sum_{i:y_{i,1}=-1,i\in\mathcal{D}_2}\left|l_{1,i,1}^{(\tau-1)}\right|\sigma'\left(\left\langle \mathbf{w}_{1,r^*}^{(\tau-1)}, -1\cdot\mathbf{v}_1\right\rangle\right)\right)$$

$$\leq \frac{\eta}{n}\sum_{i:y_{i,1}=-1,i\in\mathcal{D}_2}\left|l_{1,i,1}^{(\tau-1)}\right|\sigma'\left(\left\langle \mathbf{w}_{1,r^*}^{(\tau-1)}, -1\cdot\mathbf{v}_1\right\rangle\right)$$

$$\overset{(i)}{\leq} \frac{\eta}{n}\sum_{i:y_{i,1}=-1,i\in\mathcal{D}_2}\sigma'\left(\widetilde{\Theta}(\sigma_0)\right) = \widetilde{\Theta}\left(\sigma_0^{q-1}\right)$$

$$\leq \max\left\{(1-\eta\lambda)\Psi_1^{(\tau-1)} + \frac{\eta}{n}\left(\sum_{i:y_{i,1}=1,i\in\mathcal{D}_2}\left|l_{1,i,1}^{(\tau-1)}\right|\sigma'\left(\Psi_1^{(t-1)}\right)\right), \widetilde{\Theta}\left(\sigma_0^{q-1}\right)\right\}.$$

where (i) holds since $\left|l_{1,i,1}^{(\tau-1)}\right| \leq 1$. So this inequality holds for $\tau\in[1,t]$. Therefore, by induction, they holds for all $\tau\in[1,t]$.

(i) First we prove the lower bound. Since $t\geq P_1$, then according to our definition of $P_1$, $\Psi_1^{(P_1)} \geq \Theta(1/m)$, so we only need to consider $t > P_1$. If $\Psi_1^{(t-1)} \geq (\log(1/\sigma_0))/m$, then according to the above results and $\sigma'(z)\geq 0$, we have

$$\Psi_1^{(t)} \geq (1-\eta\lambda)\Psi_1^{(t-1)} + \frac{\eta}{n}\left(\sum_{i:y_{i,1}=1,i\in\mathcal{D}_2}\left|l_{1,i,1}^{(t-1)}\right|\sigma'\left(\Psi_1^{(t-1)}\right)\right)$$

$$\geq (1-\eta\lambda)\Psi_1^{(t-1)} \geq \frac{\Psi_1^{(t-1)}}{2} \geq (\log(1/\sigma_0))/2m$$

then the lower bound holds. Now we consider the case that $\Psi_1^{(t-1)} \leq (\log(1/\sigma_0))/m$, so without loss of generality, we can assume that $\tau_0$ is the smallest index in $[P_1, t-1]$ when the inequality

$\Psi_1^{(\tau)} \leq \left(\log\left(1/\sigma_0\right)\right)/m$ holds for all $\tau \in [\tau_0, t-1]$. From our definition of $P_1$, we can get that $\Psi_1^{(P_1)} \geq \Theta\left(1/m\right)$, so if $\tau_0 = P_1$, then we have $\Psi_1^{(\tau_0)} = \Psi_1^{(P_1)} \geq \Theta\left(1/m\right)$. If $\tau_0 > P_1$, then since $\tau_0$ is the smallest index in $[P_1, t-1]$ when the inequality $\Psi_1^{(\tau)} \leq \left(\log\left(1/\sigma_0\right)\right)/m$ holds for all $\tau \in [\tau_0, t-1]$, so the inequality does not hold at $\tau_0 - 1$, which implies that $\Psi_1^{(\tau_0-1)} \geq \left(\log\left(1/\sigma_0\right)\right)/m$. According to the above results and $\sigma'\left(z\right) \geq 0$, we have

$$\Psi_1^{(\tau_0)} \geq \left(1 - \eta\lambda\right)\Psi_1^{(\tau_0-1)} + \frac{\eta}{n}\left(\sum_{i:y_{i,1}=1, i\in\mathcal{D}_2}\left|l_{1,i,1}^{(\tau_0-1)}\right|\sigma'\left(\Psi_1^{(\tau_0-1)}\right)\right)$$

$$\geq \left(1 - \eta\lambda\right)\Psi_1^{(\tau_0-1)} \geq \frac{\Psi_1^{(\tau_0-1)}}{2} \geq \left(\log\left(1/\sigma_0\right)\right)/2m$$

So $\Psi_1^{(\tau_0)} \geq \min\left\{\Theta\left(1/m\right), \left(\log\left(1/\sigma_0\right)\right)/2m\right\} \geq \widetilde{\Theta}\left(1\right)$. Now we consider any $\tau \in [\tau_0, t-1]$ such that $\Psi_1^{(\tau)} \leq \left(\log\left(1/\sigma_0\right)\right)/m$. From Hypothesis 7 and Hypothesis 9, we can get that for $\tau \geq P_1$ and $l \in \{-1, 1\}$, $\left[\left\langle\mathbf{w}_{l,r}^{(\tau)}, \boldsymbol{\xi}_{i,1,h'}\right\rangle\right]_+ \leq \left[\left\langle\mathbf{w}_{l,r}^{(\tau)}, \boldsymbol{\zeta}_{i,1,h'}\right\rangle\right]_+ + \beta_h\left[\left\langle\mathbf{w}_{l,r}^{(\tau)}, -y_{i,1}\mathbf{v}\right\rangle\right]_+ \leq \widetilde{\Theta}\left(\sqrt{s}\sigma_p\sigma_0\right) + \beta_{h'}\widetilde{\Theta}\left(\sigma_0^{\frac{1}{3}}\right) \leq \rho$, which implies that the activation function for $\left\langle\mathbf{w}_{l,r}^{(\tau)}, \boldsymbol{\xi}_{i,1,h'}\right\rangle$ is $\frac{z^q}{q\rho^{q-1}}$, so $\sigma\left(\left\langle\mathbf{w}_{l,r}^{(\tau)}, \boldsymbol{\xi}_{i,1,h'}\right\rangle\right) \leq \frac{\rho^q}{q\rho^{q-1}} = \frac{\rho}{q}$, then we have

$$\sum_{r=1}^{m}\sum_{h'=2}^{H}\sigma\left(\left\langle\mathbf{w}_{l,r}^{(\tau)}, \boldsymbol{\xi}_{i,1,h'}\right\rangle\right) \leq \frac{mH\rho}{q} \leq 1.$$

From Lemma D.1, we can get that for $\tau \geq P_1$, $\left[\left\langle\mathbf{w}_{-1,r}^{(\tau)}, \mathbf{v}_1\right\rangle\right]_+ \leq \widetilde{\Theta}\left(\sigma_0\right) \leq \rho$, which implies that the activation function for $\left\langle\mathbf{w}_{-1,r}^{(\tau)}, \mathbf{v}_1\right\rangle$ is $\frac{z^q}{q\rho^{q-1}}$, so $\sigma\left(\left\langle\mathbf{w}_{-1,r}^{(\tau)}, \mathbf{v}_1\right\rangle\right) \leq \frac{\rho^q}{q\rho^{q-1}} = \frac{\rho}{q}$, then we have

$$\sum_{r=1}^{m}\sigma\left(\left\langle\mathbf{w}_{-1,r}^{(\tau)}, \mathbf{v}_1\right\rangle\right) \leq \frac{m\rho}{q} \leq 1.$$

Apparently $\left[\left\langle\mathbf{w}_{1,r}^{(\tau)}, \mathbf{v}_1\right\rangle\right]_+ \leq \Psi_1^{(\tau)}$. Combining all the results above, through calculations on $l_{1,i,1}^{(\tau)}$, we have that for $y_{i,1} = 1$,

$$\left|l_{1,i,1}^{(\tau)}\right| = \frac{\exp\left(\sum_{r=1}^{m}\left[\sigma\left(\left\langle\mathbf{w}_{-1,r}^{(\tau)}, y_{i,1}\cdot\mathbf{v}_1\right\rangle\right) + \sum_{h'=2}^{H}\sigma\left(\left\langle\mathbf{w}_{-1,r}^{(\tau)}, \boldsymbol{\xi}_{i,1,h'}\right\rangle\right)\right]\right)}{\sum_{l\in\{-1,1\}}\exp\left(\sum_{r=1}^{m}\left[\sigma\left(\left\langle\mathbf{w}_{l,r}^{(\tau)}, y_{i,1}\cdot\mathbf{v}_1\right\rangle\right) + \sum_{h'=2}^{H}\sigma\left(\left\langle\mathbf{w}_{l,r}^{(\tau)}, \boldsymbol{\xi}_{i,1,h'}\right\rangle\right)\right]\right)}$$

$$\overset{(i)}{\geq} \frac{1}{\sum_{l\in\{-1,1\}}\exp\left(\sum_{r=1}^{m}\left[\sigma\left(\left\langle\mathbf{w}_{l,r}^{(\tau)}, \mathbf{v}_1\right\rangle\right) + \sum_{h'=2}^{H}\sigma\left(\left\langle\mathbf{w}_{l,r}^{(\tau)}, \boldsymbol{\xi}_{i,1,h'}\right\rangle\right)\right]\right)}$$

$$\geq \frac{1}{\sum_{l\in\{-1,1\}}\exp\left(\sum_{r=1}^{m}\sigma\left(\left\langle\mathbf{w}_{l,r}^{(\tau)}, \mathbf{v}_1\right\rangle\right) + 1\right)}$$

$$\geq \Theta\left(\frac{1}{\sum_{l\in\{-1,1\}}\exp\left(\sum_{r=1}^{m}\sigma\left(\left\langle\mathbf{w}_{l,r}^{(\tau)}, \mathbf{v}_1\right\rangle\right)\right)}\right)$$

$$= \Theta\left(\frac{1}{\exp\left(\sum_{r=1}^{m}\sigma\left(\left\langle\mathbf{w}_{1,r}^{(\tau)}, \mathbf{v}_1\right\rangle\right)\right) + \exp\left(\sum_{r=1}^{m}\sigma\left(\left\langle\mathbf{w}_{-1,r}^{(\tau)}, \mathbf{v}_1\right\rangle\right)\right)}\right)$$

$$\geq \Theta\left(\frac{1}{\exp\left(m\sigma\left(\Psi_1^{(\tau)}\right)\right) + e}\right)$$

$$\overset{(ii)}{\geq} \Theta\left(\frac{1}{\exp\left(m\sigma\left(\Psi_1^{(\tau)}\right)\right) + e\exp\left(m\sigma\left(\Psi_1^{(\tau)}\right)\right)}\right) = \Theta\left(\exp\left(-m\sigma\left(\Psi_1^{(\tau)}\right)\right)\right)$$

where (i) holds since $\sigma\left(\cdot\right) \geq 0$, and (ii) holds since $\exp\left(\sigma\left(\cdot\right)\right) \geq e^0 \geq 1$. Then if $\Psi_1^{(\tau)} \leq \rho$ which means that the activation function for $\Psi_1^{(\tau)}$ is $\frac{z^q}{q\rho^{q-1}}$, then $\sigma\left(\Psi_j^\tau\right) = \frac{\left(\Psi_j^{(\tau)}\right)^q}{q\rho^{q-1}}$, $\sigma'\left(\Psi_1^{(\tau)}\right) = \frac{\left(\Psi_1^{(\tau)}\right)^{q-1}}{\rho^{q-1}}$, so we have

$$\left|l_{1,i,1}^{(\tau)}\right| \frac{\sigma'\left(\Psi_1^{(\tau)}\right)}{\Psi_1^{(\tau)}} = \left|l_{1,i,1}^{(\tau)}\right| \frac{\left(\Psi_1^{(\tau)}\right)^{q-2}}{\rho^{q-1}} \geq \Theta\left(\exp\left(-m\sigma\left(\Psi_1^{(\tau)}\right)\right)\right) \frac{\left(\Psi_1^{(\tau)}\right)^{q-2}}{\rho^{q-1}}$$

$$\geq \Theta\left(\exp\left(-\frac{m}{q\rho^{q-1}}\left(\Psi_1^{(t)}\right)^q\right)\right) \frac{\left(\Psi_1^{(\tau)}\right)^{q-2}}{\rho^{q-1}}$$

$$\geq \Theta\left(\exp\left(-\frac{m\Psi_1^{(\tau)}}{q}\right)\right) \frac{\left(\Psi_1^{(\tau)}\right)^{q-2}}{\rho^{q-1}}$$

$$\geq \widetilde{\Theta}\left(\exp\left(-\log\left(1/\sigma_0\right)\right)\right) = \widetilde{\Theta}\left(\sigma_0\right).$$

if $\Psi_1^{(\tau)} \geq \rho$ which means that the activation function for $\Psi_1^{(\tau)}$ is $z - \left(1 - \frac{1}{q}\right)\rho$, then $\sigma\left(\Psi_1^{(\tau)}\right) \leq \Psi_1^{(\tau)}$, $\sigma'\left(\Psi_1^{(\tau)}\right) = 1$, so we have

$$\left|l_{1,i,1}^{(\tau)}\right| \frac{\sigma'\left(\Psi_1^{(\tau)}\right)}{\Psi_1^{(\tau)}} = \left|l_{1,i,1}^{(\tau)}\right| \left(\Psi_1^{(\tau)}\right)^{-1} \geq \Theta\left(\exp\left(-m\sigma\left(\Psi_1^{(\tau)}\right)\right)\right)\left(\Psi_1^{(\tau)}\right)^{-1}$$

$$\geq \Theta\left(\exp\left(-m\Psi_1^{(\tau)}\right)\right)\left(\Psi_1^{(\tau)}\right)^{-1}$$

$$\geq \widetilde{\Theta}\left(\exp\left(-\log\left(1/\sigma_0\right)\right)\right) = \widetilde{\Theta}\left(\sigma_0\right)$$

So combining both parts, we have that for $i \in \mathcal{D}_2$,

$$\left|l_{1,i,1}^{(\tau)}\right| \frac{\sigma'\left(\Psi_1^{(\tau)}\right)}{\Psi_1^{(\tau)}} \geq \widetilde{\Theta}\left(\sigma_0\right)$$

Applying the above inequalities, and notice that $\lambda \leq \widetilde{\Theta}\left(\sigma_0\right)$, so we further can get that

$$\Psi_1^{(\tau+1)} \geq \left(1 - \eta\lambda\right)\Psi_1^{(\tau)} + \frac{\eta}{n}\left(\sum_{i:y_{i,1}=1,i\in\mathcal{D}_2}\left|l_{1,i,1}^{(\tau)}\right|\sigma'\left(\Psi_1^{(\tau)}\right)\right)$$

$$= \left(1 - \eta\lambda\right)\Psi_1^{(\tau)} + \frac{\eta\Psi_1^{(\tau)}}{n}\left(\sum_{i:y_{i,1}=1,i\in\mathcal{D}_2}\left|l_{1,i,1}^{(\tau)}\right|\frac{\sigma'\left(\Psi_1^{(\tau)}\right)}{\Psi_1^{(\tau)}}\right)$$

$$\overset{(i)}{\geq} \left(1 - \eta\lambda\right)\Psi_1^{(\tau)} + \frac{\eta\Psi_1^{(\tau)}}{n}\Theta\left(n\right)\widetilde{\Theta}\left(\sigma_0\right)$$

$$= \left(1 - \eta\lambda\right)\Psi_1^{(\tau)} + \eta\widetilde{\Theta}\left(\sigma_0\right)\Psi_1^{(\tau)}$$

$$= \Psi_1^{(\tau)} + \eta\left(\widetilde{\Theta}\left(\sigma_0\right) - \lambda\right)\Psi_1^{(\tau)} \geq \Psi_1^{(\tau)}.$$

in which (i) holds since $y_{i,1} = 1$ with probability $\frac{1}{2}$ according to our data generation method. This implies that $\Psi_1^{(\tau)}$ will keep increasing in the case, so we have that

$$\Psi_1^{(t)} \geq \Psi_1^{(\tau_0)} \geq \widetilde{\Theta}\left(1\right)$$

which completes the proof of the first part.

(ii) Then we will prove the upper bound. Since $t \geq P_1$, then according to our definition of $P_1$, $\Psi_1^{(P_1-1)} \leq \Theta\left(1/m\right)$, then according to the above results, we have

$$\Psi_1^{(P_1)} \leq \max\left\{\left(1 - \eta\lambda\right)\Psi_1^{(P_1-1)} + \frac{\eta}{n}\left(\sum_{i:y_{i,1}=1,i\in\mathcal{D}_2}\left|l_{1,i,1}^{(P_1-1)}\right|\sigma'\left(\Psi_1^{(P_1-1)}\right)\right), \widetilde{\Theta}\left(\sigma_0^{q-1}\right)\right\}$$

$$\overset{(i)}{\le} \Psi_1^{(P_1-1)} + \frac{\eta}{n} \sum_{i \in \mathcal{D}_2} \left( \Theta \left( \left| l_{1,i,1}^{(P_1-1)} \right| \right) + \widetilde{\Theta} \left( \sigma_0^{q-1} \right) \right)$$

$$\le \Theta(1/m) + \eta\Theta(1) + \eta\widetilde{\Theta}\left(\sigma_0^{q-1}\right) \le \Theta\left(\log\left(1/\lambda^2\right)\right)$$

where (i) holds since $\sigma'(\cdot) \le 1$. So we only need to consider $t > P_1$. If $\Psi_1^{(t-1)} \le \Theta(\log(1/\lambda))$, then according to the above results, we have

$$\Psi_1^{(t)} \le \max\left\{ (1-\eta\lambda)\Psi_1^{(t-1)} + \frac{\eta}{n}\left( \sum_{i:y_{i,1}=1, i \in \mathcal{D}_2} \left| l_{1,i,1}^{(t-1)} \right| \sigma'\left(\Psi_1^{(t-1)}\right) \right), \widetilde{\Theta}\left(\sigma_0^{q-1}\right) \right\}$$

$$\overset{(i)}{\le} \Psi_1^{(t-1)} + \frac{\eta}{n} \sum_{i \in \mathcal{D}_2} \left( \Theta\left( \left| l_{1,i,1}^{(t-1)} \right| \right) + \widetilde{\Theta}\left(\sigma_0^{q-1}\right) \right)$$

$$\le \Theta(\log(1/\lambda)) + \eta\Theta(1) + \eta\widetilde{\Theta}\left(\sigma_0^{q-1}\right) \le \Theta\left(\log\left(1/\lambda^2\right)\right)$$

where (i) holds since $\sigma'(\cdot) \le 1$. Then the upper bound holds. Now we consider the case that $\Psi_1^{(t-1)} \ge \Theta(\log(1/\lambda)) \ge \rho$, so without loss of generality, we can assume that $\tau_1$ is the smallest index in $[P_1, t-1]$ when the inequality $\Psi_1^{(\tau)} \ge \Theta(\log(1/\lambda)) \ge \rho$ holds for all $\tau \in [\tau_1, t-1]$. From our definition of $P_1$, we can get that $\Psi_1^{(P_1-1)} \le \Theta(1/m)$, so if $\tau_1 = P_1$, then $\Psi_1^{(\tau_1-1)} = \Psi_1^{(P_1-1)} \le \Theta(1/m)$, then we have

$$\Psi_1^{(\tau_1)} \le \max\left\{ (1-\eta\lambda)\Psi_1^{(\tau_1-1)} + \frac{\eta}{n}\left( \sum_{i:y_{i,1}=1, i \in \mathcal{D}_2} \left| l_{1,i,1}^{(\tau_1-1)} \right| \sigma'\left(\Psi_1^{(\tau_1-1)}\right) \right) \widetilde{\Theta}\left(\sigma_0^{q-1}\right) \right\}$$

$$\le \Psi_1^{(\tau_1-1)} + \frac{\eta}{n} \sum_{i \in \mathcal{D}_2} \left( \Theta\left( \left| l_{1,i,1}^{(\tau_1-1)} \right| \right) + \widetilde{\Theta}\left(\sigma_0^{q-1}\right) \right)$$

$$\le \Theta(1/m) + \eta\Theta(1) + \eta\widetilde{\Theta}\left(\sigma_0^{q-1}\right) \le \widetilde{\Theta}(1)$$

If $\tau_1 > P_1$, then since $\tau_1$ is the smallest index in $[P_1, t-1]$ when the inequality $\Psi_1^{(\tau)} \ge \Theta(\log(1/\lambda)) \ge \rho$ holds for all $\tau \in [\tau_1, t-1]$, so the inequality does not hold at $\tau_1 - 1$, which implies that $\Psi_1^{(\tau_1-1)} \le \Theta(\log(1/\lambda))$. According to the above results and $\sigma'(z) \le 1$, we have that

$$\Psi_1^{(\tau_1)} \le \max\left\{ (1-\eta\lambda)\Psi_1^{(\tau_1-1)} + \frac{\eta}{n}\left( \sum_{i:y_{i,1}=1, i \in \mathcal{D}_2} \left| l_{1,i,1}^{(\tau_1-1)} \right| \sigma'\left(\Psi_1^{(\tau_1-1)}\right) \right), \widetilde{\Theta}\left(\sigma_0^{q-1}\right) \right\}$$

$$\le \Psi_1^{(\tau_1-1)} + \frac{\eta}{n} \sum_{i \in \mathcal{D}_2} \left( \Theta\left( \left| l_{1,i,1}^{(\tau_1-1)} \right| \right) + \widetilde{\Theta}\left(\sigma_0^{q-1}\right) \right)$$

$$\le \Theta(\log(1/\lambda)) + \eta\Theta(1) + \eta\widetilde{\Theta}\left(\sigma_0^{q-1}\right) \le \Theta\left(\log\left(1/\lambda^2\right)\right)$$

So $\Psi_1^{(\tau_1)} \le \max\left\{ \widetilde{\Theta}(1), \Theta\left(\log\left(1/\lambda^2\right)\right) \right\} \le \widetilde{\Theta}\left(\log\left(1/\lambda^2\right)\right)$. Now we consider any $\tau \in [\tau_1, t-1]$ such that $\Psi_1^{(\tau)} \ge \Theta(\log(1/\lambda))$. From Hypothesis 7 and Hypothesis 9, we can get that for $\tau \ge P_1$, $y_{i,1} = 1$ and $h' \in [2, H]$, $\left[ \left\langle \mathbf{w}_{-1,r}^{(\tau)}, \boldsymbol{\xi}_{i,1,h'} \right\rangle \right]_+ \le \left[ \left\langle \mathbf{w}_{1,r}^{(\tau)}, \boldsymbol{\zeta}_{i,1,h'} \right\rangle \right]_+ + \beta_h \left[ \left\langle \mathbf{w}_{1,r}^{(\tau)}, -y_{i,1}\mathbf{v} \right\rangle \right]_+ \le \widetilde{\Theta}(\sqrt{s}\sigma_p\sigma_0) + \beta_{h'}\widetilde{\Theta}\left(\sigma_0^{\frac{1}{3}}\right) \le \rho$ and from Lemma D.1 we can get that $\left[ \left\langle \mathbf{w}_{-1,r}^{(\tau)}, \mathbf{v}_1 \right\rangle \right]_+ \le \widetilde{\Theta}(\sigma_0) \le \rho$, which implies that the activation function for both of them is $\frac{z^q}{q\rho^{q-1}}$, so $\sigma\left( \left\langle \mathbf{w}_{-1,r}^{(\tau)}, \boldsymbol{\xi}_{i,1,h'} \right\rangle \right) \le \frac{\rho^q}{q\rho^{q-1}} = \frac{q}{\rho}$ and similarly $\sigma\left( \left\langle \mathbf{w}_{-1,r}^{(\tau)}, \mathbf{v}_1 \right\rangle \right) \le \frac{\rho}{q}$. Then we have

$$\sum_{r=1}^{m} \left[ \sigma\left( \left\langle \mathbf{w}_{-1,r}^{(\tau)}, \mathbf{v}_1 \right\rangle \right) + \sum_{h'=2}^{H} \sigma\left( \left\langle \mathbf{w}_{-1,r}^{(\tau)}, \boldsymbol{\xi}_{i,1,h'} \right\rangle \right) \right] \le \frac{mH\rho}{q} \le 1$$

From the above results we have that $\Psi_1^{(\tau)} = \max_{r \in [m]} \left\langle \mathbf{w}_{1,r}^{(\tau)}, \mathbf{v}_1 \right\rangle$. Combining all the results above, through calculations on $l_{1,i,1}^{(\tau)}$, we have that for $y_{i,1} = 1$,

$$
\begin{aligned}
\left| l_{1,i,1}^{(\tau)} \right| &= \frac{\exp\left(\sum_{r=1}^m \left[ \sigma\left(\left\langle \mathbf{w}_{-1,r}^{(\tau)}, y_{i,1} \cdot \mathbf{v}_1 \right\rangle\right) + \sum_{h'=2}^H \sigma\left(\left\langle \mathbf{w}_{-1,r}^{(\tau)}, \boldsymbol{\xi}_{i,1,h'} \right\rangle\right) \right]\right)}{\sum_{l \in \{-1,1\}} \exp\left(\sum_{r=1}^m \left[ \sigma\left(\left\langle \mathbf{w}_{l,r}^{(\tau)}, y_{i,1} \cdot \mathbf{v}_1 \right\rangle\right) + \sum_{h'=2}^H \sigma\left(\left\langle \mathbf{w}_{l,r}^{(\tau)}, \boldsymbol{\xi}_{i,1,h'} \right\rangle\right) \right]\right)} \\
&\overset{(i)}{\leq} \frac{\exp\left(\sum_{r=1}^m \left[ \sigma\left(\left\langle \mathbf{w}_{-1,r}^{(\tau)}, \mathbf{v}_1 \right\rangle\right) + \sum_{h'=2}^H \sigma\left(\left\langle \mathbf{w}_{-1,r}^{(\tau)}, \boldsymbol{\xi}_{i,1,h'} \right\rangle\right) \right]\right)}{\exp\left(\sum_{r=1}^m \left[ \sigma\left(\left\langle \mathbf{w}_{1,r}^{(\tau)}, \mathbf{v}_1 \right\rangle\right) + \sum_{h'=2}^H \sigma\left(\left\langle \mathbf{w}_{1,r}^{(\tau)}, \boldsymbol{\xi}_{i,1,h'} \right\rangle\right) \right]\right)} \\
&\overset{(ii)}{\leq} \frac{e}{\exp\left(\sum_{r=1}^m \sigma\left(\left\langle \mathbf{w}_{1,r}^{(\tau)}, \mathbf{v}_1 \right\rangle\right)\right)} \leq \frac{e}{\exp\left(\max_{r \in [m]} \sigma\left(\left\langle \mathbf{w}_{1,r}^{(\tau)}, \mathbf{v}_1 \right\rangle\right)\right)} \\
&\overset{(iii)}{\leq} \Theta\left(\exp\left(-\sigma\left(\Psi_1^{(\tau)}\right)\right)\right)
\end{aligned}
$$

where (i) and (ii) hold since $\sigma(\cdot) \geq 0$ and (iii) holds since $\sigma$ is monotonously increasing. Then since $\Psi_1^{(\tau)} \geq \Theta\left(\log\left(1/\lambda\right)\right) \geq \rho$ which means that the activation function for $\Psi_1^{(\tau)}$ is $z - \left(1 - \frac{1}{q}\right)\rho$, then $\sigma\left(\Psi_1^{(\tau)}\right) \geq \frac{\Psi_1^{(\tau)}}{q}$, $\sigma'\left(\Psi_1^{(\tau)}\right) = 1$, so we have

$$
\begin{aligned}
\left| l_{1,i,1}^{(\tau)} \right| \frac{\sigma'\left(\Psi_1^{(\tau)}\right)}{\Psi_1^{(\tau)}} = \left| l_{1,i,1}^{(\tau)} \right| \left(\Psi_1^{(\tau)}\right)^{-1} &\leq \Theta\left(\exp\left(-\sigma\left(\Psi_1^{(\tau)}\right)\right)\right) \left(\Psi_1^{(\tau)}\right)^{-1} \\
&\leq \Theta\left(\exp\left(-\Theta\left(\Psi_1^{(\tau)}\right)\right)\right) \left(\Psi_1^{(\tau)}\right)^{-1} \\
&\leq \widetilde{\Theta}\left(\exp\left(-\Theta\left(\log\left(1/\lambda\right)\right)\right)\right) = \widetilde{\Theta}\left(\text{poly}\left(\lambda\right)\right)
\end{aligned}
$$

So we have that for $i \in \mathcal{D}_2$,

$$
\left| l_{1,i,1}^{(\tau)} \right| \frac{\sigma'\left(\Psi_1^{(\tau)}\right)}{\Psi_1^{(\tau)}} \leq \widetilde{\Theta}\left(\text{poly}\left(\lambda\right)\right)
$$

Applying the above inequalities, and notice that $\frac{\lambda}{2} \geq \widetilde{\Theta}\left(\text{poly}\left(\lambda\right)\right)$,

$$
\begin{aligned}
\Psi_1^{(\tau+1)} &\leq \max\left\{ (1 - \eta\lambda)\Psi_1^{(\tau)} + \frac{\eta}{n}\left(\sum_{i: y_{i,1}=1, i \in \mathcal{D}_2} \left| l_{1,i,1}^{(\tau)} \right| \sigma'\left(\Psi_1^{(\tau)}\right)\right), \widetilde{\Theta}\left(\sigma_0^{q-1}\right) \right\} \\
&= \max\left\{ (1 - \eta\lambda)\Psi_1^{(\tau)} + \frac{\eta\Psi_1^{(\tau)}}{n}\left(\sum_{i: y_{i,1}=1, i \in \mathcal{D}_2} \left| l_{1,i,1}^{(\tau)} \right| \frac{\sigma'\left(\Psi_1^{(\tau)}\right)}{\Psi_1^{(\tau)}}\right), \widetilde{\Theta}\left(\sigma_0^{q-1}\right) \right\} \\
&\overset{(i)}{\leq} \max\left\{ (1 - \eta\lambda)\Psi_1^{(\tau)} + \frac{\eta\Psi_1^{(\tau)}}{n}\Theta\left(n\right)\widetilde{\Theta}\left(\text{poly}\left(\lambda\right)\right), \widetilde{\Theta}\left(\sigma_0^{q-1}\right) \right\} \\
&= \max\left\{ (1 - \eta\lambda)\Psi_1^{(t)} + \eta\widetilde{\Theta}\left(\text{poly}\left(\lambda\right)\right)\Psi_1^{(\tau)}, \widetilde{\Theta}\left(\sigma_0^{q-1}\right) \right\} \\
&\leq \max\left\{ \Psi_1^{(\tau)} - \eta\left(\lambda - \widetilde{\Theta}\left(\text{poly}\left(\lambda\right)\right)\right)\Psi_1^{(\tau)}, \widetilde{\Theta}\left(\sigma_0^{q-1}\right) \right\} \\
&\leq \max\left\{ \Psi_1^{(\tau)}, \widetilde{\Theta}\left(\sigma_0^{q-1}\right) \right\} \leq \Psi_1^{(\tau)}
\end{aligned}
$$

in which (i) holds since $y_{i,1} = 1$ with probability $\frac{1}{2}$ according to our data generation method. This implies that $\Psi_1^{(\tau)}$ will keep decreasing in the case, so we have that

$$
\Psi_1^{(t)} \leq \Psi_1^{(\tau_1)} \leq \widetilde{\Theta}\left(\log\left(1/\lambda^2\right)\right)
$$

which completes the proof of the second part.

Therefore, we can get an upper bound and a lower bound which are both $\widetilde{\Theta}(1)$, so we verify Hypothesis 5.

(vi) In terms of Hypothesis 6, there are two different stages:

1. If $t \leq T_i$, then according to our definition of $T_i$, $\Phi_{y_{i,1},i}^{((T_i))}$ is the first time in the sequence that reaches $\Theta(1/m)$, which implies that $\Phi_{y_{i,1},i}^{(t)} \leq \Theta(1/m) = \widetilde{\Theta}(1)$; 2. If $t \geq T_i$, now we want to prove the following inequality by induction. For $\tau \in [0, t]$,

$$\Phi_{y_{i,1},i}^{(\tau)} \geq (1 - \eta\lambda)\Phi_{y_{i,1},i}^{(\tau-1)} + \frac{\eta}{2\rho^{q-1}}\Theta\left(\frac{s\sigma_p^2}{n}\right)\left|l_{y_{i,1},i,1}^{(\tau-1)}\right|\sigma'\left(\Phi_{y_{i,1},i}^{(\tau-1)} - \widetilde{\Theta}(\sigma_0)\right),$$

$$\text{and } \Phi_{y_{i,1},i}^{(\tau)} = \max_{r\in[m]}\max_{h\in[2,H]}\left\langle \mathbf{w}_{y_{i,1},r}^{(\tau)}, \boldsymbol{\zeta}_{i,1,h}\right\rangle,$$

and we set $\Phi_{y_{i,1},i}^{(-1)} = 0$.

For $\tau = 0$, according to previous calculations, we have $\Phi_{y_{i,1},i}^{(0)} = \widetilde{\Theta}(\sqrt{s}\sigma_p\sigma_0)$. Since $\Phi_{y_{i,1},i}^{(-1)} = 0$, then the inequality holds at $\tau = 0$., so we verify the hypothesis at $\tau = 0$. By induction, we assume that for any $\tau_0 \in [0, \tau-1]$, $\Phi_{y_{i,1},i}^{(\tau_0)} = \max_{r\in[m]}\max_{h\in[2,H]}\left\langle \mathbf{w}_{y_{i,1},r}^{(\tau_0)}, \boldsymbol{\zeta}_{i,1,h}\right\rangle \geq 0$, then for $\tau$, denote $r_h^* = \arg\max_{r\in[m]}\left\langle \mathbf{w}_{y_{i,1},r}^{(\tau-1)}, \boldsymbol{\zeta}_{i,1,h}\right\rangle$, then we can apply Hypothesis 2 at time $\tau - 1$ and get that $\Phi_{y_{i,1},i}^{(\tau-1)} = \max_{h\in[2,H]}\max_{r\in[m]}\left\langle \mathbf{w}_{y_{i,1},r}^{(t-1)}, \boldsymbol{\zeta}_{i,1,h}\right\rangle = \max_{h\in[2,H]}\left\langle \mathbf{w}_{y_{i,1},r_h^*}^{(t-1)}, \boldsymbol{\zeta}_{i,1,h}\right\rangle \geq 0$, and apparently $\Phi_{y_{i,1},i,h}^{(\tau)} \geq \left\langle \mathbf{w}_{y_{i,1},r_h^*}^{(\tau)}, \boldsymbol{\zeta}_{i,1,h}\right\rangle$, then the gradient calculation implies that

$$\left\langle \mathbf{w}_{y_{i,1},r_h^*}^{(\tau)}, \boldsymbol{\zeta}_{i,1,h}\right\rangle = (1 - \eta\lambda)\cdot\left\langle \mathbf{w}_{y_{i,1},r_h^*}^{(\tau-1)}, \boldsymbol{\zeta}_{i,1,h}\right\rangle$$

$$+ \frac{\eta}{n}\sum_{g=2}^{H}\sum_{s=1}^{n}l_{y_{i,1},s,1}^{(\tau-1)}\sigma'\left(\left\langle \mathbf{w}_{y_{i,1},r_h^*}^{(\tau-1)}, \boldsymbol{\xi}_{s,1,g}\right\rangle\right)\left\langle \boldsymbol{\zeta}_{s,1,g}, \boldsymbol{\zeta}_{i,1,h}\right\rangle$$

According to Lemma B.3, with probability exceeding $1 - n^{-2}$, for all $(s, g) \neq (i, h)$, $\langle \boldsymbol{\zeta}_{s,1,g}, \boldsymbol{\zeta}_{i,1,h}\rangle = 0$. By using the calculations above, we can get that with probability exceeding $1 - 2n^{-2}$, $\left|\|\boldsymbol{\zeta}_{i,1,h}\|^2 - s\sigma_p^2\right| \leq \widetilde{\Theta}(\sqrt{s}\sigma_p^2)$, which implies that $\|\boldsymbol{\zeta}_{i,1,h}\| \geq \Theta(\sqrt{s}\sigma_p)$, then we have

$$\left\langle \mathbf{w}_{y_{i,1},r_h^*}^{(\tau)}, \boldsymbol{\zeta}_{i,1,h}\right\rangle \geq (1 - \eta\lambda)\cdot\left\langle \mathbf{w}_{y_{i,1},r_h^*}^{(\tau-1)}, \boldsymbol{\zeta}_{i,1,h}\right\rangle + \frac{\eta}{n}\Theta(s\sigma_p^2)l_{y_{i,1},i,1}^{(\tau-1)}\sigma'\left(\left\langle \mathbf{w}_{y_{i,1},r_h^*}^{(\tau-1)}, \boldsymbol{\xi}_{i,1,h}\right\rangle\right)$$

According to Hypothesis 2 at $[0, \tau-1]$, we have $\Phi_{y_{i,1},i}^{(\tau-1)} = \max_{h\in[2,H]}\left\langle \mathbf{w}_{y_{i,1},r_h^*}^{(\tau-1)}, \boldsymbol{\zeta}_{i,1,h}\right\rangle \geq 0$ and $\Phi_{y_{i,1},i}^{(\tau)}$ is monotonically non-decreasing for $\tau_0 \in [0, \tau-1]$, which implies that $\Phi_{y_{i,1},i}^{(\tau-1)} \geq \Phi_{y_{i,1},i}^{(0)} \geq \widetilde{\Theta}(\sqrt{s}\sigma_p\sigma_0) \geq \widetilde{\Theta}(\sigma_0)$. According to Hypothesis 8 at time $\tau - 1$, we have $\max_{r\in[m]}\left\langle \mathbf{w}_{y_{i,1},r}^{(\tau-1)}, y_{i,1}\mathbf{v}\right\rangle \leq \Lambda_{y_{i,1}}^{(t-1)} \leq \widetilde{\Theta}(\sigma_0)$. Then taking maximum according to $h \in [2, H]$ on the both side of the inequality, we further get that

$$\max_{h\in[2,H]}\left\langle \mathbf{w}_{y_{i,1},r_h^*}^{(\tau)}, \boldsymbol{\zeta}_{i,1,h}\right\rangle$$

$$\overset{(i)}{\geq} (1 - \eta\lambda)\cdot\max_{h\in[2,H]}\left\langle \mathbf{w}_{y_{i,1},r_h^*}^{(t-1)}, \boldsymbol{\zeta}_{i,1,h}\right\rangle + \frac{\eta}{n}\Theta(s\sigma_p^2)\left|l_{y_{i,1},i,1}^{(\tau-1)}\right|\max_{h\in[2,H]}\sigma'\left(\left\langle \mathbf{w}_{y_{i,1},r_h^*}^{(t-1)}, \boldsymbol{\xi}_{i,1,h}\right\rangle\right)$$

$$\overset{(ii)}{\geq} (1 - \eta\lambda)\cdot\max_{h\in[2,H]}\left\langle \mathbf{w}_{y_{i,1},r_h^*}^{(\tau-1)}, \boldsymbol{\zeta}_{i,1,h}\right\rangle + \frac{\eta}{n}\Theta(s\sigma_p^2)\left|l_{y_{i,1},i,1}^{(\tau-1)}\right|\sigma'\left(\max_{h\in[2,H]}\left\langle \mathbf{w}_{y_{i,1},r_h^*}^{(\tau-1)}, \boldsymbol{\xi}_{i,1,h}\right\rangle\right)$$

$$= (1 - \eta\lambda)\cdot\max_{h\in[2,H]}\left\langle \mathbf{w}_{y_{i,1},r_h^*}^{(\tau-1)}, \boldsymbol{\zeta}_{i,1,h}\right\rangle$$

$$+ \frac{\eta}{n}\Theta(s\sigma_p^2)\left|l_{y_{i,1},i,1}^{(\tau-1)}\right|\sigma'\left(\max_{h\in[2,H]}\left\langle \mathbf{w}_{y_{i,1},r_h^*}^{(\tau-1)}, \boldsymbol{\zeta}_{i,1,h} - \beta_h y_{i,1}\mathbf{v}\right\rangle\right)$$

$$\geq (1 - \eta\lambda)\cdot\max_{h\in[2,H]}\left\langle \mathbf{w}_{y_{i,1},r_h^*}^{(\tau-1)}, \boldsymbol{\zeta}_{i,1,h}\right\rangle$$

$$+ \frac{\eta}{n}\Theta\left(s\sigma_p^2\right)\left|l_{y_{i,1},i,1}^{(\tau-1)}\right|\sigma'\left(\max_{h\in[2,H]}\left\langle\mathbf{w}_{y_{i,1},r_h^*}^{(\tau-1)},\boldsymbol{\zeta}_{i,1,h}\right\rangle - \beta_h\max_{h\in[2,H]}\left\langle\mathbf{w}_{y_{i,1},r_h^*}^{(\tau-1)},y_{i,1}\mathbf{v}\right\rangle\right)$$

$$\geq (1-\eta\lambda)\cdot\max_{h\in[2,H]}\left\langle\mathbf{w}_{y_{i,1},r_h^*}^{(\tau-1)},\boldsymbol{\zeta}_{i,1,h}\right\rangle$$

$$+ \frac{\eta}{n}\Theta\left(s\sigma_p^2\right)\left|l_{y_{i,1},i,1}^{(\tau-1)}\right|\sigma'\left(\max_{h\in[2,H]}\left\langle\mathbf{w}_{y_{i,1},r_h^*}^{(\tau-1)},\boldsymbol{\zeta}_{i,1,h}\right\rangle - \max_{r\in[m]}\left\langle\mathbf{w}_{y_{i,1},r_h^*}^{(\tau-1)},y_{i,1}\mathbf{v}\right\rangle\right)$$

$$\geq (1-\eta\lambda)\cdot\max_{h\in[2,H]}\left\langle\mathbf{w}_{y_{i,1},r_h^*}^{(\tau-1)},\boldsymbol{\zeta}_{i,1,h}\right\rangle$$

$$+ \frac{\eta}{n}\Theta\left(s\sigma_p^2\right)\left|l_{y_{i,1},i,1}^{(\tau-1)}\right|\sigma'\left(\max_{h\in[2,H]}\left\langle\mathbf{w}_{y_{i,1},r_h^*}^{(t-1)},\boldsymbol{\zeta}_{i,1,h}\right\rangle - \widetilde{\Theta}(\sigma_0)\right)$$

$$= (1-\eta\lambda)\Phi_{y_{i,1},i}^{(\tau-1)} + \frac{\eta}{n}\Theta\left(s\sigma_p^2\right)\left|l_{y_{i,1},i,1}^{(\tau-1)}\right|\sigma'\left(\Phi_{y_{i,1},i}^{(\tau-1)} - \widetilde{\Theta}(\sigma_0)\right) \geq 0$$

in which (i) holds since $\mathrm{sgn}\left(l_{y_{i,1},i,1}^{(\tau-1)}\right) = 1$, (ii) holds since $\sigma'(\cdot)$ is a monotonic function, and (iii) holds since $0 \leq \Phi_{y_{i,1},i}^{(\tau-1)} - \widetilde{\Theta}(\sigma_0) \leq \Phi_{y_{i,1},i}^{(\tau-1)} \leq \rho$ according to our assumption, then the activation function for $\Phi_{y_{i,1},i}^{(\tau-1)}$ is $\frac{z^q}{q\rho^{q-1}}$, which implies that $\sigma'\left(\Phi_{y_{i,1},i}^{(\tau-1)} - \widetilde{\Theta}(\sigma_0)\right) = \frac{\left(\Phi_{y_{i,1},i}^{(\tau-1)} - \widetilde{\Theta}(\sigma_0)\right)^{q-1}}{\rho^{q-1}}$. So we have

$$\max_{h\in[2,H]}\left\langle\mathbf{w}_{y_{i,1},r_h^*}^{(\tau)},\boldsymbol{\zeta}_{i,1,h}\right\rangle = \max_{h\in[2,H]}\max_{r\in[m]}\left\langle\mathbf{w}_{y_{i,1},r}^{(t)},\boldsymbol{\zeta}_{i,1,h}\right\rangle \geq 0$$

which implies that $\Phi_{y_{i,1},i}^{(\tau)} = \max_{h\in[2,H]}\max_{r\in[m]}\left\langle\mathbf{w}_{y_{i,1},r}^{(t)},\boldsymbol{\zeta}_{i,1,h}\right\rangle \geq 0$. Then

$$\Phi_{y_{i,1},i}^{(\tau)} \geq (1-\eta\lambda)\Phi_{y_{i,1},i}^{(\tau-1)} + \frac{\eta}{\rho^{q-1}}\Theta\left(\frac{s\sigma_p^2}{n}\right)\left|l_{y_{i,1},i,1}^{(\tau-1)}\right|\sigma'\left(\Phi_{y_{i,1},i}^{(\tau-1)} - \widetilde{\Theta}(\sigma_0)\right).$$

Therefore, by induction, this inequality holds for all $\tau\in[0,t]$.
Then we will prove the following inequality. For all $\tau\in[1,t]$,

$$\Phi_{y_{i,1},i}^{(\tau)} \leq (1-\eta\lambda)\Phi_{y_{i,1},i}^{(\tau-1)} + \frac{\eta}{\rho^{q-1}}\Theta\left(\frac{s\sigma_p^2}{n}\right)\left|l_{y_{i,1},i,1}^{(\tau-1)}\right|\sigma'\left(\Phi_{y_{i,1},i}^{(\tau-1)} - \widetilde{\Theta}\left(\sigma_0^{\frac{2}{3}}\right)\right)$$

Denote $r_h^* = \arg\max_{r\in[m]}\left\langle\mathbf{w}_{y_{i,1},r}^{(\tau)},\boldsymbol{\zeta}_{i,1,h}\right\rangle$, then apparently $\Phi_{y_{i,1},i,h}^{(\tau)} = \left[\left\langle\mathbf{w}_{y_{i,1},r_h^*}^{(\tau)},\boldsymbol{\zeta}_{i,1,h}\right\rangle\right]_+$ and $\Phi_{y_{i,1},i,h}^{(\tau-1)} \geq \left[\left\langle\mathbf{w}_{y_{i,1},r_h^*}^{(\tau-1)},\boldsymbol{\zeta}_{i,1,h}\right\rangle\right]_+$. Then the gradient calculation implies that

$$\Phi_{y_{i,1},i,h}^{(\tau)} = \left[\left\langle\mathbf{w}_{y_{i,1},r_h^*}^{(\tau)},\boldsymbol{\zeta}_{i,1,h}\right\rangle\right]_+$$

$$= \left[(1-\eta\lambda)\cdot\left\langle\mathbf{w}_{y_{i,1},r_h^*}^{(\tau-1)},\boldsymbol{\zeta}_{i,1,h}\right\rangle + \frac{\eta}{n}\sum_{g=2}^{H}\sum_{s=1}^{n}l_{y_{i,1},s,1}^{(\tau-1)}\sigma'\left(\left\langle\mathbf{w}_{y_{i,1},r_h^*}^{(\tau-1)},\boldsymbol{\xi}_{s,1,g}\right\rangle\right)\left\langle\boldsymbol{\zeta}_{s,1,g},\boldsymbol{\zeta}_{i,1,h}\right\rangle\right]_+$$

According to Lemma B.3, with probability exceeding $1 - n^{-2}$, for all $(s,g) \neq (i,h)$, $\left\langle\boldsymbol{\zeta}_{s,1,g},\boldsymbol{\zeta}_{i,1,h}\right\rangle = 0$. By using the calculations above, we can get that with probability exceeding $1 - 2n^{-2}$, $\left|\|\boldsymbol{\zeta}_{i,1,h}\|^2 - s\sigma_p^2\right| \leq \widetilde{\Theta}\left(\sqrt{s}\sigma_p^2\right)$, which implies that $\|\boldsymbol{\zeta}_{i,1,h}\| \leq \Theta\left(\sqrt{s}\sigma_p\right)$, then we have

$$\Phi_{y_{i,1},i,h}^{(\tau)} = \left[(1-\eta\lambda)\cdot\left\langle\mathbf{w}_{y_{i,1},r_h^*}^{(\tau-1)},\boldsymbol{\zeta}_{i,1,h}\right\rangle + \frac{\eta}{n}l_{y_{i,1},i,1}^{(\tau-1)}\sigma'\left(\left\langle\mathbf{w}_{y_{i,1},r_h^*}^{(\tau-1)},\boldsymbol{\xi}_{i,1,h}\right\rangle\right)\|\boldsymbol{\zeta}_{i,1,h}\|^2\right]_+$$

$$\overset{(i)}{\leq} (1-\eta\lambda)\cdot\left[\left\langle\mathbf{w}_{y_{i,1},r_h^*}^{(\tau-1)},\boldsymbol{\zeta}_{i,1,h}\right\rangle\right]_+ + \frac{\eta}{n}\left|l_{y_{i,1},i,1}^{(t-1)}\right|\sigma'\left(\left\langle\mathbf{w}_{y_{i,1},r_h^*}^{(\tau-1)},\boldsymbol{\xi}_{i,1,h}\right\rangle\right)\|\boldsymbol{\zeta}_{i,1,h}\|^2$$

$$= (1-\eta\lambda)\left[\left\langle\mathbf{w}_{y_{i,1},r_h^*}^{(\tau-1)},\boldsymbol{\zeta}_{i,1,h}\right\rangle\right]_+ + \frac{\eta}{\rho^{q-1}}\Theta\left(\frac{s\sigma_p^2}{n}\right)\left|l_{y_{i,1},i}^{(\tau-1)}\right|\sigma'\left(\left[\left\langle\mathbf{w}_{y_{i,1},r_h^*}^{(\tau-1)},\boldsymbol{\xi}_{i,1,h}\right\rangle\right]_+\right)$$

$$\overset{(ii)}{\leq} (1-\eta\lambda)\left[\left\langle\mathbf{w}_{y_{i,1},r_h^*}^{(\tau-1)},\boldsymbol{\zeta}_{i,1,h}\right\rangle\right]_+$$

$$+ \frac{\eta}{\rho^{q-1}} \Theta\left(\frac{s\sigma_p^2}{n}\right) \left|l_{y_{i,1},i}^{(\tau-1)}\right| \sigma'\left(\left[\left\langle \mathbf{w}_{y_{i,1},r_h^*}^{(\tau-1)}, \boldsymbol{\zeta}_{i,1,h}\right\rangle\right]_+ + \beta_h \left[\left\langle \mathbf{w}_{y_{i,1},r_h^*}^{(\tau-1)}, -y_{i,1}\mathbf{v}\right\rangle\right]_+\right)$$

$$\leq (1-\eta\lambda)\Phi_{y_{i,1},i}^{(\tau-1)} + \frac{\eta}{\rho^{q-1}}\Theta\left(\frac{s\sigma_p^2}{n}\right)\left|l_{y_{i,1},i,1}^{(\tau-1)}\right|\sigma'\left(\Phi_{y_{i,1},i}^{(\tau-1)} + \beta_h\left\langle \mathbf{w}_{y_{i,1},r_h^*}^{(\tau-1)}, -y_{i,1}\mathbf{v}\right\rangle\right)$$

$$\overset{(iii)}{\leq} (1-\eta\lambda)\Phi_{y_{i,1},i}^{(\tau-1)} + \frac{\eta}{\rho^{q-1}}\Theta\left(\frac{s\sigma_p^2}{n}\right)\left|l_{y_{i,1},i,1}^{(\tau-1)}\right|\sigma'\left(\Phi_{y_{i,1},i}^{(\tau-1)} + \widetilde{\Theta}\left(\sigma_0^{\frac{2}{3}}\right)\right)$$

where (i) holds according to the definition of $[\cdot]_+$ and the triangle inequality, (ii) holds according to the triangle inequality, and (iii) holds since $\left\langle \mathbf{w}_{y_{i,1},r_h^*}^{(\tau-1)}, -y_{i,1}\mathbf{v}\right\rangle \leq \widetilde{\Theta}\left(\sigma_0^{\frac{1}{3}}\right)$ according to Hypothesis 9 at $t-1$. Taking maximum according to $h \in [2,H]$ on the left hand side of the inequality, we further get that

$$\Phi_{y_{i,1},i}^{(\tau)} \leq (1-\eta\lambda)\Phi_{y_{i,1},i}^{(\tau-1)} + \frac{\eta}{\rho^{q-1}}\Theta\left(\frac{s\sigma_p^2}{n}\right)\left|l_{y_{i,1},i,1}^{(\tau)}\right|\sigma'\left(\Phi_{y_{i,1},i}^{(\tau-1)} + \widetilde{\Theta}\left(\sigma_0^{\frac{2}{3}}\right)\right)$$

so the hypothesis holds at $\tau$. Therefore, by induction, these two inequalities hold for all $\tau \in [1,t]$.
(i) First we prove the lower bound. Since $t \geq T_i$, then according to our definition of $T_i$, $\Phi_{y_{i,1},i}^{(T_i)} \geq \Theta(1/m)$, so we only need to consider $t > T_i$. If $\Phi_{y_{i,1},i}^{(t-1)} \geq (\log(1/\sigma_0))/m$, then according to the above results and $\sigma'(z) \geq 0$, we have

$$\Phi_{y_{i,1},i}^{(\tau)} \geq (1-\eta\lambda)\Phi_{y_{i,1},i}^{(\tau-1)} + \frac{\eta}{2\rho^{q-1}}\Theta\left(\frac{s\sigma_p^2}{n}\right)\left|l_{y_{i,1},i,1}^{(\tau-1)}\right|\sigma'\left(\Phi_{y_{i,1},i}^{(\tau-1)} - \widetilde{\Theta}(\sigma_0)\right)$$

$$\geq (1-\eta\lambda)\Phi_{y_{i,1},i}^{(t-1)} \geq \frac{\Phi_{y_{i,1},i}^{(t-1)}}{2} \geq (\log(1/\sigma_0))/2m$$

then the lower bound holds. Now we consider the case that $\Phi_{y_{i,1},i}^{(t-1)} \leq (\log(1/\sigma_0))/m$, so without loss of generality, we can assume that $\tau_0$ is the smallest index in $[T_i, t-1]$ when the inequality $\Phi_{y_{i,1},i}^{(\tau)} \leq (\log(1/\sigma_0))/m$ holds for all $\tau \in [\tau_0, t-1]$. From our definition of $T_i$, we can get that $\Phi_{y_{i,1},i}^{(T_i)} \geq \Theta(1/m)$, so if $\tau_0 = T_i$, then we have $\Phi_{y_{i,1},i}^{(\tau_0)} = \Phi_{y_{i,1},i}^{(T_i)} \geq \Theta(1/m)$. If $\tau_0 > T_i$, then since $\tau_0$ is the smallest index in $[T_i, t-1]$ when the inequality $\Phi_{y_{i,1},i}^{(\tau)} \leq (\log(1/\sigma_0))/m$ holds for all $\tau \in [\tau_0, t-1]$, so the inequality does not hold at $\tau_0 - 1$, which implies that $\Phi_{y_{i,1},i}^{(\tau_0-1)} \geq (\log(1/\sigma_0))/m$. According to the above results and $\sigma'(z) \geq 0$, we have

$$\Phi_{y_{i,1},i}^{(\tau)} \geq (1-\eta\lambda)\Phi_{y_{i,1},i}^{(\tau-1)} + \frac{\eta}{2\rho^{q-1}}\Theta\left(\frac{s\sigma_p^2}{n}\right)\left|l_{y_{i,1},i,1}^{(\tau-1)}\right|\sigma'\left(\Phi_{y_{i,1},i}^{(\tau-1)} - \widetilde{\Theta}(\sigma_0)\right)$$

$$\geq (1-\eta\lambda)\Phi_{y_{i,1},i}^{(\tau_0-1)} \geq \frac{\Phi_{y_{i,1},i}^{(\tau_0-1)}}{2} \geq (\log(1/\sigma_0))/2m$$

So $\Phi_{y_{i,1},i}^{(\tau_0)} \geq \min\{\Theta(1/m), (\log(1/\sigma_0))/2m\} \geq \widetilde{\Theta}(1)$. Now we consider any $\tau \in [\tau_0, t-1]$ such that $\Phi_{y_{i,1},i}^{(\tau)} \leq (\log(1/\sigma_0))/m$. From Hypothesis 8 and Lemma D.2, we can get that for $\tau \geq T_i$, $h' \in [2,H]$, $\left[\left\langle \mathbf{w}_{-y_{i,1},r}^{(\tau)}, \boldsymbol{\xi}_{i,1,h'}\right\rangle\right]_+ \leq \left[\left\langle \mathbf{w}_{-y_{i,1},r}^{(\tau)}, \boldsymbol{\zeta}_{i,1,h'}\right\rangle\right]_+ + \beta_h \left[\left\langle \mathbf{w}_{-y_{i,1},r}^{(\tau)}, -y_{i,1}\mathbf{v}\right\rangle\right]_+ \leq \widetilde{\Theta}(\sqrt{s}\sigma_p\sigma_0) + \beta_{h'}\widetilde{\Theta}(\sigma_0) \leq \rho$, which implies that the activation function for $\left\langle \mathbf{w}_{-y_{i,1},r}^{(\tau)}, \boldsymbol{\xi}_{i,1,h'}\right\rangle$ is $\frac{z^q}{q\rho^{q-1}}$, so $\sigma\left(\left\langle \mathbf{w}_{-y_{i,1},r}^{(\tau)}, \boldsymbol{\xi}_{i,1,h'}\right\rangle\right) \leq \frac{\rho^q}{q\rho^{q-1}} = \frac{\rho}{q}$, then we have

$$\sum_{r=1}^{m}\sum_{h'=2}^{H}\sigma\left(\left\langle \mathbf{w}_{-y_{i,1},r}^{(\tau)}, \boldsymbol{\xi}_{i,1,h'}\right\rangle\right) \leq \frac{mH\rho}{q} \leq 1.$$

From Hypothesis 9, we can get that for $\tau \geq T_i$, $\left[\left\langle \mathbf{w}_{-y_{i,1},r}^{(\tau)}, y_{i,1} \cdot \mathbf{v} \right\rangle\right]_+ \leq \widetilde{\Theta}\left(\sigma_0^{\frac{1}{3}}\right) \leq \rho$, which implies that the activation function for $\left\langle \mathbf{w}_{-y_{i,1},r}^{(\tau)}, y_{i,1} \cdot \mathbf{v} \right\rangle$ is $\frac{z^q}{q\rho^{q-1}}$, so $\sigma\left(\left\langle \mathbf{w}_{-y_{i,1},r}^{(\tau)}, y_{i,1} \cdot \mathbf{v} \right\rangle\right) \leq \frac{\rho^q}{q\rho^{q-1}} = \frac{\rho}{q}$, then we have

$$\sum_{r=1}^m \sigma\left(\left\langle \mathbf{w}_{-y_{i,1},r}^{(\tau)}, y_{i,1} \cdot \mathbf{v} \right\rangle\right) \leq \frac{m\rho}{q} \leq 1.$$

From Hypothesis 8, we can get that for $\tau \geq T_i$, $\left[\left\langle \mathbf{w}_{y_{i,1},r}^{(\tau)}, y_{i,1} \cdot \mathbf{v} \right\rangle\right]_+ \leq \widetilde{\Theta}(\sigma_0) \leq \rho$, which implies that the activation function for $\left\langle \mathbf{w}_{y_{i,1},r}^{(\tau)}, y_{i,1} \cdot \mathbf{v} \right\rangle$ is $\frac{z^q}{q\rho^{q-1}}$, so $\sigma\left(\left\langle \mathbf{w}_{y_{i,1},r}^{(\tau)}, y_{i,1} \cdot \mathbf{v} \right\rangle\right) \leq \frac{\rho^q}{q\rho^{q-1}} = \frac{\rho}{q}$, then we have

$$\sum_{r=1}^m \sigma\left(\left\langle \mathbf{w}_{y_{i,1},r}^{(\tau)}, y_{i,1} \cdot \mathbf{v} \right\rangle\right) \leq \frac{m\rho}{q} \leq 1.$$

Apparently from Hypothesis 6 at $\tau$, $\Phi_{y_{i,1},i}^{(\tau)} \geq \widetilde{\Theta}(1) \geq \widetilde{\Theta}\left(\sigma_0^{\frac{1}{3}}\right)$, then we have $\left[\left\langle \mathbf{w}_{y_{i,1},r}^{(\tau)}, \boldsymbol{\xi}_{i,1,h} \right\rangle\right]_+ \leq \left[\left\langle \mathbf{w}_{y_{i,1},r}^{(\tau)}, \boldsymbol{\zeta}_{i,1,h} \right\rangle\right]_+ + \beta_h \left[\left\langle \mathbf{w}_{y_{i,1},r}^{(\tau)}, -y_{i,1}\mathbf{v} \right\rangle\right]_+ \leq \Phi_{y_{i,1},i}^{(\tau)} + \widetilde{\Theta}\left(\sigma_0^{\frac{1}{3}}\right) \leq 2\Phi_{y_{i,1},i}^{(\tau)}$. Combining all the results above, through calculations on $l_{y_{i,1},i,1}^{(\tau)}$, we have that

$$\left|l_{y_{i,1},i,1}^{(\tau)}\right| = \frac{\exp\left(\sum_{r=1}^m \left[\sigma\left(\left\langle \mathbf{w}_{-y_{i,1},r}^{(\tau)}, y_{i,1} \cdot \mathbf{v} \right\rangle\right) + \sum_{h'=2}^H \sigma\left(\left\langle \mathbf{w}_{-y_{i,1},r}^{(\tau)}, \boldsymbol{\xi}_{i,1,h'} \right\rangle\right)\right]\right)}{\sum_{l\in\{-1,1\}} \exp\left(\sum_{r=1}^m \left[\sigma\left(\left\langle \mathbf{w}_{l,r}^{(\tau)}, y_{i,1} \cdot \mathbf{v} \right\rangle\right) + \sum_{h'=2}^H \sigma\left(\left\langle \mathbf{w}_{l,r}^{(\tau)}, \boldsymbol{\xi}_{i,1,h'} \right\rangle\right)\right]\right)}$$

$$\overset{(i)}{\geq} \frac{1}{\sum_{l\in\{-1,1\}} \exp\left(\sum_{r=1}^m \left[\sigma\left(\left\langle \mathbf{w}_{l,r}^{(\tau)}, \mathbf{v} \right\rangle\right) + \sum_{h'=2}^H \sigma\left(\left\langle \mathbf{w}_{l,r}^{(\tau)}, \boldsymbol{\xi}_{i,1,h'} \right\rangle\right)\right]\right)}$$

$$\geq \Theta\left(\frac{1}{\sum_{l\in\{-1,1\}} \exp\left(\sum_{r=1}^m \sum_{h=2}^H \sigma\left(\left\langle \mathbf{w}_{y_{i,1},r}^{(\tau)}, \boldsymbol{\xi}_{i,1,h} \right\rangle\right) + 1\right)}\right)$$

$$\geq \Theta\left(\frac{1}{\sum_{l\in\{-1,1\}} \exp\left(\sum_{r=1}^m \sum_{h=2}^H \sigma\left(\left\langle \mathbf{w}_{y_{i,1},r}^{(\tau)}, \boldsymbol{\xi}_{i,1,h} \right\rangle\right)\right)}\right)$$

$$\geq \Theta\left(\exp\left(-2mH\sigma\left(\Phi_{y_{i,1},i}^{(\tau)}\right)\right)\right)$$

where (i) holds since $\sigma(\cdot) \geq 0$, Then if $\Phi_{y_{i,1},i}^{(\tau)} \leq \rho$ which means that the activation function for $\Phi_{y_{i,1},i}^{(\tau)}$ is $\frac{z^q}{q\rho^{q-1}}$, then $\sigma\left(\Phi_{y_{i,1},i}^{(\tau)}\right) = \frac{\left(\Phi_{y_{i,1},i}^{(\tau)}\right)^q}{q\rho^{q-1}}$, $\sigma'\left(\Psi_1^{(\tau)}\right) = \frac{\left(\Phi_{y_{i,1},i}^{(\tau)}\right)^{q-1}}{\rho^{q-1}}$, and $\Phi_{y_{i,1},i}^{(\tau)} - \widetilde{\Theta}(\sigma_0) \geq \frac{\Phi_{y_{i,1},i}^{(\tau)}}{2}$, so we have

$$\left|l_{1,i,1}^{(\tau)}\right| \frac{\sigma'\left(\Phi_{y_{i,1},i}^{(\tau)}\right)}{\Phi_{y_{i,1},i}^{(\tau)}} \geq \left|l_{y_{i,1},i,1}^{(\tau)}\right| \Theta\left(\frac{\sigma'\left(\Phi_{y_{i,1}}^{(\tau)}\right)}{\Phi_{y_{i,1},i}^{(\tau)}}\right) = \left|l_{y_{i,1},i,1}^{(\tau)}\right| \Theta\left(\frac{\left(\Phi_{y_{i,1},i}^{(\tau)}\right)^{q-2}}{\rho^{q-1}}\right)$$

$$\geq \Theta\left(\exp\left(-2m\sigma\left(\Phi_{y_{i,1},i}^{(\tau)}\right)\right)\right) \cdot \Theta\left(\frac{\left(\Phi_{y_{i,1},i}^{(\tau)}\right)^{q-2}}{\rho^{q-1}}\right)$$

$$\geq \Theta\left(\exp\left(-\frac{2m}{q\rho^{q-1}}\left(\Phi_{y_{i,1},i}^{(t)}\right)^q\right)\right) \frac{\left(\Phi_{y_{i,1},i}^{(\tau)}\right)^{q-2}}{\rho^{q-1}}$$

$$\geq \Theta\left(\exp\left(-\frac{2m\Phi_{y_{i,1},i}^{(\tau)}}{q}\right)\right) \frac{\left(\Phi_{y_{i,1},i}^{(\tau)}\right)^{q-2}}{\rho^{q-1}}$$

$$\geq \widetilde{\Theta}\left(\exp\left(-\log\left(1/\sigma_0\right)\right)\right) = \widetilde{\Theta}\left(\sigma_0\right).$$

if $\Phi_{y_{i,1},i}^{(\tau)} \geq \rho$ which means that the activation function for $\Phi_{y_{i,1},i}^{(\tau)}$ is $z - \left(1 - \frac{1}{q}\right)\rho$, then $\sigma\left(\Phi_{y_{i,1},i}^{(\tau)} - \widetilde{\Theta}\left(\sigma_0\right)\right) \leq \Phi_{y_{i,1},i}^{(\tau)} - \widetilde{\Theta}\left(\sigma_0\right), \sigma'\left(\Phi_{y_{i,1},1}^{(\tau)} - \widetilde{\Theta}\left(\sigma_0\right)\right) = 1$, so we have

$$\left|l_{y_{i,1},i,1}^{(\tau)}\right| \frac{\sigma'\left(\Phi_{y_{i,1}}^{(\tau)} - \widetilde{\Theta}\left(\sigma_0\right)\right)}{\Phi_{y_{i,1},i}^{(\tau)}} \geq \left|l_{y_{i,1},i,1}^{(\tau)}\right| \Theta\left(\frac{\sigma'\left(\Phi_{y_{i,1}}^{(\tau)}\right)}{\Phi_{y_{i,1},i}^{(\tau)}}\right)$$

$$= \left|l_{1,i,1}^{(\tau)}\right| \Theta\left(\left(\Phi_{y_{i,1},i}^{(\tau)}\right)^{-1}\right)$$

$$\geq \Theta\left(\exp\left(-2m\sigma\left(\Phi_{y_{i,1},i}^{(\tau)}\right)\right)\right)\Theta\left(\left(\Phi_{y_{i,1},i}^{(\tau)}\right)^{-1}\right)$$

$$\geq \Theta\left(\exp\left(-2m\Phi_{y_{i,1},i}^{(\tau)}\right)\right)\Theta\left(\left(\Phi_{y_{i,1},i}^{(\tau)}\right)^{-1}\right)$$

$$\geq \widetilde{\Theta}\left(\exp\left(-\log\left(1/\sigma_0\right)\right)\right) = \widetilde{\Theta}\left(\sigma_0\right)$$

So combining both parts, we have that for $i \in \mathcal{D}_1$,

$$\left|l_{y_{i,1},i,1}^{(\tau)}\right| \frac{\sigma'\left(\Phi_{y_{i,1},i}^{(\tau)} - \widetilde{\Theta}\left(\sigma_0\right)\right)}{\Phi_{y_{i,1},i}^{(\tau)}} \geq \widetilde{\Theta}\left(\sigma_0\right)$$

Applying the above inequalities, and notice that $\lambda \leq \widetilde{\Theta}\left(\sigma_0\right)$, so we further can get that

$$\Phi_{y_{i,1},i}^{(\tau)} \geq (1 - \eta\lambda)\Phi_{y_{i,1},i}^{(\tau-1)} + \frac{\eta}{\rho^{q-1}}\Theta\left(\frac{s\sigma_p^2}{n}\right)\left|l_{y_{i,1},i,1}^{(\tau-1)}\right|\sigma'\left(\Phi_{y_{i,1},i}^{(\tau-1)} - \widetilde{\Theta}\left(\sigma_0\right)\right)$$

$$= (1 - \eta\lambda)\Phi_{y_{i,1},i}^{(\tau)} + \Theta\left(\frac{\eta s\sigma_p^2\Phi_{y_{i,1},i}^{(\tau)}}{n\rho^{q-1}}\right)\left|l_{y_{i,1},i,1}^{(\tau)}\right|\frac{\sigma'\left(\Phi_{y_{i,1},i}^{(\tau)} - \widetilde{\Theta}\left(\sigma_0\right)\right)}{\Phi_{y_{i,1},i}^{(\tau)}}$$

$$\geq (1 - \eta\lambda)\Phi_{y_{i,1},i}^{(\tau)} + \Theta\left(\frac{\eta s\sigma_p^2\Phi_{y_{i,1},i}^{(\tau)}}{n\rho^{q-1}}\right)\widetilde{\Theta}\left(\sigma_0\right)$$

$$= (1 - \eta\lambda)\Phi_{y_{i,1},i}^{(\tau)} + \eta\widetilde{\Theta}\left(\frac{s\sigma_0\sigma_p^2}{n\rho^{q-1}}\right)\Phi_{y_{i,1},i}^{(\tau)}$$

$$= \Phi_{y_{i,1},i}^{(\tau)} + \eta\left(\widetilde{\Theta}\left(\frac{s\sigma_0\sigma_p^2}{n\rho^{q-1}}\right) - \lambda\right)\Phi_{y_{i,1},i}^{(\tau)} \geq \Phi_{y_{i,1},i}^{(\tau)}.$$

This implies that $Phi_{y_{i,1},i}^{(\tau)}$ will keep increasing in the case, so we have that

$$\Phi_{y_{i,1},i}^{(t)} \geq \Psi_1^{(\tau_0)} \geq \widetilde{\Theta}(1)$$

which completes the proof of the first part.

(ii) Then we will prove the upper bound. Since $t \geq T_i$, then according to our definition of $T_i$, $\Phi_{y_{i,1},i}^{(T_i-1)} \leq \Theta\left(1/m\right)$, then according to the above results, we have

$$\Phi_{y_{i,1}}^{(P_1)} \leq \Phi_{y_{i,1},i}^{(T_i)} \leq (1 - \eta\lambda)\Phi_{y_{i,1},i}^{(\tau-1)} + \frac{\eta}{\rho^{q-1}}\Theta\left(\frac{s\sigma_p^2}{n}\right)\left|l_{y_{i,1},i,1}^{(\tau)}\right|\sigma'\left(\Phi_{y_{i,1},i}^{(\tau-1)} + \widetilde{\Theta}\left(\sigma_0^{\frac{2}{3}}\right)\right)$$

$$\overset{\text{(i)}}{\leq} \Phi_{y_{i,1},i}^{(T_i-1)} + \Theta\left(\frac{\eta s\sigma_p^2}{n\rho^{q-1}}\right)\Theta\left(\left|l_{y_{i,1},i,1}^{(T_i-1)}\right|\right)$$

$$\leq \Theta\left(1/m\right) + \Theta\left(\frac{\eta s\sigma_p^2}{n\rho^{q-1}}\right) \leq \Theta\left(\log\left(1/\lambda^2\right)\right)$$

where (i) holds since $\sigma'(\cdot) \leq 1$. So we only need to consider $t > T_i$. If $\Phi_{y_{i,1},i}^{(t-1)} \leq \Theta(\log(1/\lambda))$, then according to the above results, we have

$$
\Phi_{y_{i,1},i}^{(\tau)} \leq (1 - \eta\lambda)\Phi_{y_{i,1},i}^{(\tau-1)} + \frac{\eta}{\rho^{q-1}}\Theta\left(\frac{s\sigma_p^2}{n}\right)\left|l_{y_{i,1},i,1}^{(\tau)}\right|\sigma'\left(\Phi_{y_{i,1},i}^{(\tau-1)} + \widetilde{\Theta}\left(\sigma_0^{\frac{2}{3}}\right)\right)
$$

$$
\overset{(i)}{\leq} \Phi_{y_{i,1},i}^{(T_i-1)} + \Theta\left(\frac{\eta s\sigma_p^2}{n\rho^{q-1}}\right)\Theta\left(\left|l_{y_{i,1},i,1}^{(T_i-1)}\right|\right)
$$

$$
\leq \Theta(1/m) + \Theta\left(\frac{\eta s\sigma_p^2}{n\rho^{q-1}}\right) \leq \Theta\left(\log(1/\lambda^2)\right)
$$

where (i) holds since $\sigma'(\cdot) \leq 1$. Then the upper bound holds. Now we consider the case that $\Phi_{y_{i,1},i}^{(t-1)} \geq \Theta(\log(1/\lambda)) \geq \rho$, so without loss of generality, we can assume that $\tau_1$ is the smallest index in $[T_i, t-1]$ when the inequality $\Phi_{y_{i,1},i}^{(\tau)} \geq \Theta(\log(1/\lambda)) \geq \rho$ holds for all $\tau \in [\tau_1, t-1]$. From our definition of $T_i$, we can get that $\Phi_{y_{i,1},i}^{(T_i-1)} \leq \Theta(1/m)$, so if $\tau_1 = T_i$, then $\Phi_{y_{i,1},i}^{(\tau_1-1)} = \Phi_{y_{i,1},i}^{(T_i-1)} \leq \Theta(1/m)$, then we have

$$
\Phi_{y_{i,1},i}^{(\tau_1)} \leq (1 - \eta\lambda)\Phi_{y_{i,1},i}^{(\tau-1)} + \frac{\eta}{\rho^{q-1}}\Theta\left(\frac{s\sigma_p^2}{n}\right)\left|l_{y_{i,1},i,1}^{(\tau)}\right|\sigma'\left(\Phi_{y_{i,1},i}^{(\tau-1)} + \widetilde{\Theta}\left(\sigma_0^{\frac{2}{3}}\right)\right)
$$

$$
\leq \Phi_{y_{i,1},i}^{(T_i-1)} + \Theta\left(\frac{\eta s\sigma_p^2}{n\rho^{q-1}}\right)\Theta\left(\left|l_{y_{i,1},i,1}^{(T_i-1)}\right|\right)
$$

$$
\leq \Theta(1/m) + \Theta\left(\frac{\eta s\sigma_p^2}{n\rho^{q-1}}\right) \leq \Theta\left(\log(1/\lambda^2)\right)
$$

If $\tau_1 > T_i$, then since $\tau_1$ is the smallest index in $[T_i, t-1]$ when the inequality $\Phi_{y_{i,1},i}^{(\tau)} \geq \Theta(\log(1/\lambda)) \geq \rho$ holds for all $\tau \in [\tau_1, t-1]$, so the inequality does not hold at $\tau_1 - 1$, which implies that $\Phi_{y_{i,1},i}^{(\tau_1-1)} \leq \Theta(\log(1/\lambda))$. According to the above results and $\sigma'(z) \leq 1$, we have that

$$
\Phi_{y_{i,1},i}^{(\tau_1)} \leq (1 - \eta\lambda)\Phi_{y_{i,1},i}^{(\tau_1-1)} + \frac{\eta}{\rho^{q-1}}\Theta\left(\frac{s\sigma_p^2}{n}\right)\left|l_{y_{i,1},i,1}^{(\tau_1-1)}\right|\sigma'\left(\Phi_{y_{i,1},i}^{(\tau-1)} + \widetilde{\Theta}\left(\sigma_0^{\frac{2}{3}}\right)\right)
$$

$$
\leq \Phi_{y_{i,1},i}^{(\tau_1)} + \Theta\left(\frac{\eta s\sigma_p^2}{n\rho^{q-1}}\right)\left|l_{y_{i,1},i,1}^{(\tau_1-1)}\right|
$$

$$
\leq \Theta(\log(1/\lambda)) + \Theta\left(\frac{\eta s\sigma_p^2}{n\rho^{q-1}}\right) \leq \Theta\left(\log(1/\lambda^2)\right)
$$

So $\Phi_{y_{i,1},i}^{(\tau_1)} \leq \max\left\{\widetilde{\Theta}(1), \Theta\left(\log(1/\lambda^2)\right)\right\} \leq \widetilde{\Theta}\left(\log(1/\lambda^2)\right)$. Now we consider any $\tau \in [\tau_1, t-1]$ such that $\Phi_{y_{i,1},i}^{(\tau)} \geq \Theta(\log(1/\lambda))$. From Hypothesis 8 and Lemma D.2, we can get that for $\tau \geq T_i$, and $h' \in [2, H]$, $\left[\left\langle \mathbf{w}_{-y_{i,1},r}^{(\tau)}, \boldsymbol{\xi}_{i,1,h'}\right\rangle\right]_+ \leq \left[\left\langle \mathbf{w}_{-y_{i,1},r}^{(\tau)}, \boldsymbol{\zeta}_{i,1,h'}\right\rangle\right]_+ + \beta_h\left[\left\langle \mathbf{w}_{-y_{i,1},r}^{(\tau)}, -y_{i,1}\mathbf{v}\right\rangle\right]_+ \leq \widetilde{\Theta}(\sqrt{s}\sigma_p\sigma_0) + \beta_{h'}\widetilde{\Theta}(\sigma_0) \leq \rho$, which implies that the activation function for $\left\langle \mathbf{w}_{-y_{i,1},r}^{(\tau)}, \boldsymbol{\xi}_{i,1,h'}\right\rangle$ is $\frac{z^q}{q\rho^{q-1}}$, so $\sigma\left(\left\langle \mathbf{w}_{-y_{i,1},r}^{(\tau)}, \boldsymbol{\xi}_{i,1,h'}\right\rangle\right) \leq \frac{\rho^q}{q\rho^{q-1}} = \frac{\rho}{q}$, then we have

$$
\sum_{r=1}^{m}\sum_{h'=2}^{H}\sigma\left(\left\langle \mathbf{w}_{-y_{i,1},r}^{(\tau)}, \boldsymbol{\xi}_{i,1,h'}\right\rangle\right) \leq \frac{mH\rho}{q} \leq 1.
$$

From Hypothesis 9, we can get that for $\tau \geq T_i$, $\left[\left\langle \mathbf{w}_{-y_{i,1},r}^{(\tau)}, y_{i,1}\cdot\mathbf{v}\right\rangle\right]_+ \leq \widetilde{\Theta}\left(\sigma_0^{\frac{1}{3}}\right) \leq \rho$, which implies that the activation function for $\left\langle \mathbf{w}_{-y_{i,1},r}^{(\tau)}, y_{i,1}\cdot\mathbf{v}\right\rangle$ is $\frac{z^q}{q\rho^{q-1}}$, so $\sigma\left(\left\langle \mathbf{w}_{-y_{i,1},r}^{(\tau)}, y_{i,1}\cdot\mathbf{v}\right\rangle\right) \leq$

$\frac{\rho^q}{q\rho^{q-1}} = \frac{\rho}{q}$, then we have

$$\sum_{r=1}^{m} \sigma\left(\left\langle \mathbf{w}_{-y_{i,1},r}^{(\tau)}, y_{i,1} \cdot \mathbf{v} \right\rangle\right) \leq \frac{m\rho}{q} \leq 1.$$

From Hypothesis 8, we can get that for $\tau \geq T_i$, $\left[\left\langle \mathbf{w}_{y_{i,1},r}^{(\tau)}, y_{i,1} \cdot \mathbf{v} \right\rangle\right]_{+} \leq \widetilde{\Theta}(\sigma_0) \leq \rho$, which implies that the activation function for $\left\langle \mathbf{w}_{y_{i,1},r}^{(\tau)}, y_{i,1} \cdot \mathbf{v} \right\rangle$ is $\frac{z^q}{q\rho^{q-1}}$, so $\sigma\left(\left\langle \mathbf{w}_{y_{i,1},r}^{(\tau)}, y_{i,1} \cdot \mathbf{v} \right\rangle\right) \leq \frac{\rho^q}{q\rho^{q-1}} = \frac{\rho}{q}$, then we have

$$\sum_{r=1}^{m} \sigma\left(\left\langle \mathbf{w}_{y_{i,1},r}^{(\tau)}, y_{i,1} \cdot \mathbf{v} \right\rangle\right) \leq \frac{m\rho}{q} \leq 1.$$

Apparently from Hypothesis 6 at $\tau$, $\Phi_{y_{i,1},i}^{(\tau)} \geq \widetilde{\Theta}(1) \geq \widetilde{\Theta}(\sigma_0)$, then we have $\Phi_{y_{i,1},i}^{(\tau)} - \widetilde{\Theta}(\sigma_0) \geq \frac{\Phi_{y_{i,1},i}^{(\tau)}}{2}$. Combining all the results above, through calculations on $l_{y_{i,1},i,1}^{(\tau)}$, we have that

$$\left| l_{y_{i,1},i,1}^{(\tau)} \right| = \frac{\exp\left(\sum_{r=1}^{m}\left[\sigma\left(\left\langle \mathbf{w}_{-y_{i,1},r}^{(\tau)}, y_{i,1} \cdot \mathbf{v} \right\rangle\right) + \sum_{h'=2}^{H} \sigma\left(\left\langle \mathbf{w}_{-y_{i,1},r}^{(\tau)}, \boldsymbol{\xi}_{i,1,h'} \right\rangle\right)\right]\right)}{\sum_{l\in\{-1,1\}} \exp\left(\sum_{r=1}^{m}\left[\sigma\left(\left\langle \mathbf{w}_{l,r}^{(\tau)}, y_{i,1} \cdot \mathbf{v} \right\rangle\right) + \sum_{h'=2}^{H} \sigma\left(\left\langle \mathbf{w}_{l,r}^{(\tau)}, \boldsymbol{\xi}_{i,1,h'} \right\rangle\right)\right]\right)}$$

$$\overset{(i)}{\leq} \frac{\exp\left(\sum_{r=1}^{m}\left[\sigma\left(\left\langle \mathbf{w}_{-y_{i,1},r}^{(\tau)}, y_{i,1} \cdot \mathbf{v} \right\rangle\right) + \sum_{h'=2}^{H} \sigma\left(\left\langle \mathbf{w}_{-y_{i,1},r}^{(\tau)}, \boldsymbol{\xi}_{i,1,h'} \right\rangle\right)\right]\right)}{\exp\left(\sum_{r=1}^{m}\left[\sigma\left(\left\langle \mathbf{w}_{y_{i,1},r}^{(\tau)}, y_{i,1} \cdot \mathbf{v} \right\rangle\right) + \sum_{h'=2}^{H} \sigma\left(\left\langle \mathbf{w}_{y_{i,1},r}^{(\tau)}, \boldsymbol{\xi}_{i,1,h'} \right\rangle\right)\right]\right)}$$

$$\overset{(ii)}{\leq} \frac{e}{\exp\left(\sum_{r=1}^{m} \sum_{h'=2}^{H} \sigma\left(\left\langle \mathbf{w}_{y_{i,1},r}^{(\tau)}, \boldsymbol{\xi}_{i,1,h'} \right\rangle\right)\right)}$$

$$\leq \frac{e}{\exp\left(\sigma\left(\max_{r\in[m]} \max_{h'\in[2,H]} \sigma\left(\left\langle \mathbf{w}_{y_{i,1},r}^{(\tau)}, \boldsymbol{\zeta}_{i,1,h'} \right\rangle - \beta_h \left\langle \mathbf{w}_{y_{i,1},r}^{(\tau)}, y_{i,1}\mathbf{v} \right\rangle\right)\right)\right)}$$

$$\leq \frac{e}{\exp\left(\Phi_{y_{i,1},i}^{(\tau)} - \beta_h \widetilde{\Theta}(\sigma_0)\right)} \leq \frac{e}{\exp\left(\sigma\left(\frac{\Phi_{y_{i,1},i}^{(\tau)}}{2}\right)\right)} = \Theta\left(\exp\left(-\sigma\left(\frac{\Psi_1^{(\tau)}}{2}\right)\right)\right)$$

where (i) and (ii) hold since $\sigma(\cdot) \geq 0$. Then since $\Phi_{y_{i,1},i}^{(\tau)} \geq \Theta(\log(1/\lambda)) \geq \rho$ which means that the activation function for $\Phi_{y_{i,1},i}^{(\tau)}$ and $\Phi_{y_{i,1},i}^{(\tau)} + \widetilde{\Theta}\left(\sigma_0^{\frac{2}{3}}\right)$ is $z - \left(1 - \frac{1}{q}\right)\rho$, then $\sigma\left(\Phi_{y_{i,1},i}^{(\tau)}\right) \geq \frac{\Phi_{y_{i,1},i}^{(\tau)}}{q}$, $\sigma'\left(\Phi_{y_{i,1},i}^{(\tau)} + \widetilde{\Theta}\left(\sigma_0^{\frac{2}{3}}\right)\right) = 1$, so we have

$$\left| l_{y_{i,1},i,1}^{(\tau)} \right| \frac{\sigma'\left(\Phi_{y_{i,1},i}^{(\tau)} + \widetilde{\Theta}\left(\sigma_0^{\frac{2}{3}}\right)\right)}{\Phi_{y_{i,1},i}^{(\tau)}} = \left| l_{y_{i,1},i,1}^{(\tau)} \right| \left(\Phi_{y_{i,1},i}^{(\tau)}\right)^{-1}$$

$$\leq \Theta\left(\exp\left(-\sigma\left(\frac{\Phi_{y_{i,1},i}^{(\tau)}}{2}\right)\right)\right) \left(\Phi_{y_{i,1},i}^{(\tau)}\right)^{-1}$$

$$\leq \Theta\left(\exp\left(-\Theta\left(\frac{\Phi_{y_{i,1},i}^{(\tau)}}{2}\right)\right)\right) \left(\Phi_{y_{i,1},i}^{(\tau)}\right)^{-1}$$

$$\leq \widetilde{\Theta}\left(\exp\left(-\Theta\left(\log(1/\lambda)\right)\right)\right) = \widetilde{\Theta}(\text{poly}(\lambda))$$

So we have that for $i \in \mathcal{D}_1$,

$$\left| l_{y_{i,1},i,1}^{(\tau)} \right| \frac{\sigma'\left(\Phi_{y_{i,1},i}^{(\tau)} + \widetilde{\Theta}\left(\sigma_0^{\frac{2}{3}}\right)\right)}{\Phi_{y_{i,1},i}^{(\tau)}} \leq \widetilde{\Theta}(\text{poly}(\lambda))$$

Applying the above inequalities, and notice that $\frac{\lambda}{2} \geq \widetilde{\Theta}\left(\text{poly}\left(\lambda\right)\right)$,

$$
\begin{aligned}
\Phi_{y_{i,1},i}^{(\tau+1)} &\leq (1-\eta\lambda)\,\Phi_{y_{i,1},i}^{(\tau)} + \frac{\eta}{\rho^{q-1}}\Theta\left(\frac{s\sigma_p^2}{n}\right)\left|l_{y_{i,1},i,1}^{(\tau)}\right|\sigma'\left(\Phi_{y_{i,1},i}^{(\tau)} + \widetilde{\Theta}\left(\sigma_0^{\frac{2}{3}}\right)\right) \\
&\leq (1-\eta\lambda)\,\Phi_{y_{i,1},i}^{(\tau)} + \Theta\left(\frac{\eta s\sigma_p^2\Phi_{y_{i,1},i}^{(\tau)}}{n\rho^{q-1}}\right)\left|l_{y_{i,1},i,1}^{(\tau)}\right|\frac{\sigma'\left(\Phi_{y_{i,1},i}^{(\tau)} + \widetilde{\Theta}\left(\sigma_0^{\frac{2}{3}}\right)\right)}{\Phi_{y_{i,1},i}^{(\tau)}} \\
&\leq (1-\eta\lambda)\,\Phi_{y_{i,1},i}^{(\tau)} + \Theta\left(\frac{\eta s\sigma_p^2\Phi_{y_{i,1},i}^{(\tau)}}{n\rho^{q-1}}\right)\widetilde{\Theta}\left(\text{poly}\left(\lambda\right)\right) \\
&= (1-\eta\lambda)\,\Phi_{y_{i,1},i}^{(\tau)} + \eta\widetilde{\Theta}\left(\frac{s\sigma_p^2\text{poly}\left(\lambda\right)}{n\rho^{q-1}}\right)\Phi_{y_{i,1},i}^{(\tau)} \\
&= \Phi_{y_{i,1},i}^{(\tau)} - \eta\left(\lambda - \widetilde{\Theta}\left(\frac{s\sigma_p^2\text{poly}\left(\lambda\right)}{n\rho^{q-1}}\right)\right)\Phi_{y_{i,1},i}^{(\tau)} \leq \Phi_{y_{i,1},i}^{(\tau)}
\end{aligned}
$$

This implies that $\Phi_{y_{i,1},i}^{(\tau)}$ will keep decreasing in the case, so we have that

$$
\Phi_{y_{i,1},i}^{(t)} \leq \Phi_{y_{i,1},i}^{(\tau_1)} \leq \widetilde{\Theta}\left(\log\left(1/\lambda^2\right)\right)
$$

which completes the proof of the second part.

Therefore, we can get an upper bound and a lower bound which are both $\widetilde{\Theta}\left(1\right)$, so we verify Hypothesis 6.

(vii) In terms of Hypothesis 7, there are two stages:

1. If $t \leq P_0$, there are two situations:

(i). If $y_{i,1} = -j$, then according to Lemma D.2,

$$
\Phi_{j,i}^{(t)} = \Phi_{-y_{i,1},i}^{(t)} = \max_{h\in[2,H]}\max_{r\in[m]}\left\langle \mathbf{w}_{-y_{i,1},r}^{(t)}, \zeta_{i,1,h}\right\rangle \leq \widetilde{\Theta}\left(\sqrt{s}\sigma_p\sigma_0\right)
$$

(ii)). If $y_{i,1} = j$, then from Hypothesis 3 at $\tau \in [0,t]$, we can get that for any $\tau \in [0,t]$,

$$
\Phi_{j,i}^{(\tau)} = \Phi_{y_{i,1},i}^{(\tau)} \leq \Phi_{y_{i,1},i}^{(\tau-1)} + \frac{\eta}{\rho^{q-1}}\widetilde{\Theta}\left(\frac{s\sigma_p^2}{n}\right)\Theta\left(\left(\Phi_{y_{i,1},i}^{(\tau-1)} + \widetilde{\Theta}\left(\sigma_0^{\frac{2}{3}}\right)\right)^{q-1}\right)
$$

Denote $\widehat{\Phi}_{y_{i,1},i}^{(\tau)} = \Phi_{y_{i,1},i}^{(\tau)} + \widetilde{\Theta}\left(\sigma_0^{\frac{2}{3}}\right)$, then we have

$$
\widehat{\Phi}_{y_{i,1},i}^{(\tau)} \leq \widehat{\Phi}_{y_{i,1},i}^{(\tau)} + \frac{\eta}{\rho^{q-1}}\Theta\left(\frac{s\sigma_p^2}{n}\right)\Theta\left(\left(\widehat{\Phi}_{y_{i,1},i}^{(\tau-1)}\right)^{q-1}\right)
$$

From Hypothesis 1 at $\tau \in [0,t]$, we can get that for any $\tau \in [0,t]$ and any $j' \in \{-1,1\}$

$$
\Psi_{j'}^{(\tau)} \geq \Psi_{j'}^{(\tau-1)} + \frac{\eta}{2\rho^{q-1}}\Theta\left(\left(\Psi_{j'}^{(\tau-1)}\right)^{q-1}\right)
$$

Due to our assumptions, we have

$$
\eta\cdot\Theta\left(\frac{1}{2\rho^{q-1}}\right)\cdot\widetilde{\Theta}\left(\sigma_0^{q-2}\right) \geq \eta\cdot\Theta\left(\frac{s\sigma_p^2}{n\rho^{q-1}}\right)\widetilde{\Theta}\left(\left(\sqrt{s}\sigma_p\sigma_0\right)^{q-2}\right)
$$

So using Lemma 5.1 at $[0,t]$ with our initialization, we can conclude that for each $j' \in \{-1,1\}$ and $t \leq P_{j'}$, $\Phi_{y_{i,1},i}^{(t)} \leq \widehat{\Phi}_{y_{i,1},i}^{(t)} \leq \Theta\left(\widehat{\Phi}_{j'}^{(0)}\right) = \Theta\left(\Phi_{j'}^{(0)} + \widetilde{\Theta}\left(\sigma_0^{\frac{2}{3}}\right)\right) = \widetilde{\Theta}\left(\sqrt{s}\sigma_p\sigma_0\right)$. So $\Phi_{j,i}^{(t)} = \Phi_{y_{i,1},i}^{(t)} \leq \widetilde{\Theta}\left(\sqrt{s}\sigma_p\sigma_0\right)$ for $t \leq \max_{j'\in\{-1,1\}} P_{j'} = P_0$.

Combining these two parts, we have that the hypothesis holds at $t$.

2. If $t \geq P_0$, then from Hypothesis 5, we can get that for $\tau \in [P_0, t]$, $\Psi_j^{(t)} = \widetilde{\Theta}(1)$. Using the same inequality in proving Hypothesis 5, we have for $\tau \in [P_0, t]$,

$$\Psi_j^{(\tau)} \geq (1 - \eta\lambda) \Psi_j^{(\tau-1)} + \frac{\eta}{n} \left( \sum_{i:y_{i,1}=j, i \in \mathcal{D}_2} \left| l_{j,i,1}^{(\tau-1)} \right| \sigma' \left( \Psi_j^{(\tau-1)} \right) \right)$$

$$= (1 - \eta\lambda) \Psi_j^{(\tau-1)} + \widetilde{\Theta}\left( \frac{\eta}{n} \right) \left( \sum_{i:y_{i,1}=j, i \in \mathcal{D}_2} \left| l_{j,i,1}^{(\tau-1)} \right| \right)$$

Recursively applying this inequality for $\tau \in [P_0 + 1, t]$ gives that

$$\Psi_j^{(t)} \geq (1 - \eta\lambda)^{t-P_0} \Psi_j^{(P_0)} + \frac{\eta}{n} \sum_{\tau=0}^{t-P_0-1} (1 - \eta\lambda)^\tau \left( \sum_{i:y_{i,1}=j, i \in \mathcal{D}_2} \left| l_{j,i,1}^{(t-1-\tau)} \right| \right)$$

Taking summation over $j \in \{-1, 1\}$, we have

$$\sum_{j \in \{-1,1\}} \Psi_j^{(t)} \geq (1 - \eta\lambda)^{t-P_0} \sum_{j \in \{-1,1\}} \Psi_j^{(P_0)}$$

$$+ \widetilde{\Theta}\left( \frac{\eta}{n} \right) \sum_{\tau=0}^{t-P_0-1} (1 - \eta\lambda)^\tau \sum_{j \in \{-1,1\}} \sum_{i:y_{i,1}, i \in \mathcal{D}_2} \left| l_{j,i,1}^{(t-1-\tau)} \right|$$

Again by $\Psi_j^{(t)} = \widetilde{\Theta}(1)$ and $\Psi_j^{(P_0)} = \widetilde{\Theta}(1)$, we have

$$\widetilde{\Theta}\left( \frac{\eta}{n} \right) \sum_{\tau=0}^{t-P_0-1} (1 - \eta\lambda)^\tau \sum_{j \in \{-1,1\}} \sum_{i:y_{i,1}=j, i \in \mathcal{D}_2} \left| l_{j,i,1}^{(t-1-\tau)} \right|$$

$$\leq \sum_{j \in \{-1,1\}} \Psi_j^{(t)} - (1 - \eta\lambda)^{t-P_0} \sum_{j \in \{-1,1\}} \Psi_j^{(P_0)}$$

$$\leq \widetilde{\Theta}(1).$$

Since $y_{i,1} = 1$ or $y_{i,1} = -1$ always happens, and $\left| l_{j,i,1}^{(\tau)} \right| = \left| l_{-j,i,1}^{(\tau)} \right|$, so we have

$$\widetilde{\Theta}\left( \frac{\eta}{n} \right) \sum_{\tau=0}^{t-P_0-1} (1 - \eta\lambda)^\tau \sum_{i \in \mathcal{D}_2} \left| l_{j,i,1}^{(t-1-\tau)} \right| \leq \widetilde{\Theta}(1).$$

According to Hypothesis 7, $\Phi_{j,i}^{(\tau-1)} \leq \widetilde{\Theta}(\sqrt{s}\sigma_p\sigma_0)$, which implies that the activation function for $\Phi_{j,i}^{(\tau-1)} + \widetilde{\Theta}\left( \sigma_0^{\frac{2}{3}} \right)$ is $\frac{z^q}{q\rho^{q-1}}$, then $\sigma'\left( \Phi_{j,i}^{(\tau-1)} + \widetilde{\Theta}\left( \sigma_0^{\frac{2}{3}} \right) \right) = \frac{\left( \Phi_{j,i}^{(\tau-1)} + \widetilde{\Theta}\left( \sigma_0^{\frac{2}{3}} \right) \right)^{q-1}}{\rho^{q-1}} \leq \widetilde{\Theta}\left( \sigma_0^{\frac{2(q-1)}{3}} \right)$.

Using the same inequality as verifying Hypothesis 6, we have that

$$\Phi_{j,i}^{(\tau)} \leq (1 - \eta\lambda) \Phi_{j,i}^{(\tau-1)} + \Theta\left( \frac{\eta s \sigma_p^2}{n\rho^{q-1}} \right) \left| l_{j,i,1}^{(\tau)} \right| \sigma'\left( \Phi_{j,i}^{(\tau-1)} + \widetilde{\Theta}\left( \sigma_0^{\frac{2}{3}} \right) \right)$$

$$\leq (1 - \eta\lambda) \Phi_{j,i}^{(\tau-1)} + \widetilde{\Theta}\left( \frac{\eta s \sigma_p^2 \sigma_0^{\frac{2(q-1)}{3}}}{n\rho^{q-1}} \right) \left| l_{j,i,1}^{(\tau)} \right|$$

Recursively applying this inequality for $\tau \in [P_0 + 1, t]$, we have

$$\Phi_{j,i}^{(t)} \leq (1 - \eta\lambda)^{t-P_0} \Phi_{j,i}^{(P_0)} + \Theta\left( \frac{\eta s \sigma_p^2}{n\rho^{q-1}} \right) \sum_{\tau=0}^{t-P_0-1} (1 - \eta\lambda)^\tau \left| l_{y_{i,1},i,1}^{(t-1-\tau)} \right|$$

$$\leq (1 - \eta\lambda)^{t-P_0} \widetilde{\Theta}\left( \sqrt{s}\sigma_p\sigma_0 \right) + \widetilde{\Theta}\left( \frac{\eta s \sigma_p^2 \sigma_0^{\frac{2(q-1)}{3}}}{n\rho^{q-1}} \right) \cdot \widetilde{\Theta}\left( \frac{n\sqrt{s}\sigma_p\sigma_0}{\eta} \right) \leq \widetilde{\Theta}\left( \sqrt{s}\sigma_p\sigma_0 \right)$$

so we verify Hypothesis 7 at $t$.

(viii) In terms of Hypothesis 8, there are two stages:

1. If $t \leq T_0$, from Hypothesis 4 at $t$, we can get that

$$\Lambda_j^{(t)} \leq \max \left\{ \Lambda_j^{(t-1)} + \frac{\eta \alpha^q}{\rho^{q-1}} \Theta \left( \left( \Lambda_j^{(t-1)} \right)^{q-1} \right), \widetilde{\Theta} \left( \alpha^q \sigma_0^{\frac{2}{3}} \right) \right\}$$

then there are two situations:

1. If $\Lambda_j^{(t)} \leq \widetilde{\Theta} \left( \alpha^q \sigma_0^{\frac{2}{3}} \right) \leq \widetilde{\Theta} (\sigma_0)$ holds, then the hypothesis holds at $t$.

2. If $\Lambda_j^{(t)} \leq \widetilde{\Theta} \left( \alpha^q \sigma_0^{\frac{2}{3}} \right)$ does not hold, then from Hypothesis 4 at $\tau \in [0, t]$, we can get that for any $\tau \in [0, t]$,

$$\Lambda_j^{(\tau)} \leq \max \left\{ \Lambda_j^{(t-1)} + \frac{\eta \alpha^q}{\rho^{q-1}} \Theta \left( \left( \Lambda_j^{(\tau-1)} \right)^{q-1} \right), \widetilde{\Theta} \left( \alpha^q \sigma_0^{\frac{2}{3}} \right) \right\}$$

Assume $\tau_0$ is the smallest index in $[0, t]$ such that $\Lambda_j^{(\tau)} \leq \widetilde{\Theta} \left( \alpha^q \sigma_0^{\frac{2}{3}} \right)$ does not hold for any $\tau \in [\tau_0, t]$, then we have that

$$\Lambda_j^{(\tau)} \leq \Lambda_j^{(\tau-1)} + \frac{\eta \alpha^q}{\rho^{q-1}} \Theta \left( \left( \Lambda_j^{(\tau-1)} \right)^{q-1} \right)$$

Specifically, if $\tau_0 = 0$, then $\Lambda_j^{(\tau_0)} \leq \widetilde{\Theta} (\sigma_0)$; if $\tau_0 > 0$, then according to the definition of $\tau_0$, we must have $\Lambda_j^{(\tau_0-1)} \leq \widetilde{\Theta} \left( \alpha^q \sigma_0^{\frac{2}{3}} \right)$, then we can get that

$$\Lambda_j^{(\tau_0)} \leq \Lambda_j^{(\tau_0-1)} + \frac{\eta \alpha^q}{\rho^{q-1}} \Theta \left( \left( \Lambda_j^{(\tau-1)} \right)^{q-1} \right) \leq \widetilde{\Theta} \left( \alpha^q \sigma_0^{\frac{2}{3}} \right) + \frac{\eta \alpha^q}{\rho^{q-1}} \left( \widetilde{\Theta} \left( \alpha^q \sigma_0^{\frac{2}{3}} \right) \right)^{q-1} \leq \widetilde{\Theta} (\sigma_0)$$

so $\Lambda_j^{(\tau_0)} \leq \widetilde{\Theta} (\sigma_0)$ always holds. Again from Hypothesis 1 at $\tau \in [0, t]$, we can get that for any $\tau \in [0, t]$ and any $j' \in \{-1, 1\}$,

$$\Psi_{j'}^{(\tau)} \geq \Psi_{j'}^{(\tau-1)} + \frac{\eta}{2\rho^{q-1}} \Theta \left( \left( \Psi_{j'}^{(\tau-1)} \right)^{q-1} \right)$$

which also implies that $\Psi_{j'}^{(\tau)}$ is increasing, specifically, $\Psi_{j'}^{(\tau_0)} \geq \Psi_{j'}^{(0)} = \widetilde{\Theta} (\sigma_0)$. Due to our assumptions,

$$\eta \cdot \Theta \left( \frac{1}{2\rho^{q-1}} \right) \cdot \widetilde{\Theta} \left( \left( \Psi_{j'}^{(\tau_0)} \right)^{q-2} \right) \geq \eta \cdot \Theta \left( \frac{1}{2\rho^{q-1}} \right) \cdot \widetilde{\Theta} \left( \sigma_0^{q-2} \right)$$

$$\geq \eta \cdot \frac{\alpha^q}{\rho^{q-1}} \cdot \widetilde{\Theta} \left( \sigma_0^{q-2} \right) \geq \eta \cdot \frac{\alpha^q}{\rho^{q-1}} \cdot \widetilde{\Theta} \left( \left( \Lambda_j^{(\tau_0)} \right)^{q-2} \right)$$

So using Lemma 5.1 at $[\tau_0, t]$ with our initialization, we can conclude that for each $j' \in \{-1, 1\}$ and $t \leq P_{j'}$, $\Lambda_j^{(t)} \leq \Theta \left( \Lambda_j^{(\tau_0)} \right) \leq \widetilde{\Theta} (\sigma_0)$. So $\Lambda_j^{(t)} \leq \widetilde{\Theta} (\sigma_0)$ for $t \leq \max_{j' \in \{-1,1\}} P_{j'} = P_0$.

Similarly, from Hypothesis 2 at $\tau \in [0, t]$, we can get that for any $\tau \in [0, t]$ any $i \in \mathcal{D}_1$,

$$\Phi_{y_{i,1},i}^{(\tau)} \geq \Phi_{y_{i,1},i}^{(\tau-1)} + \frac{\eta}{2\rho^{q-1}} \Theta \left( \frac{s\sigma_p^2}{n} \right) \Theta \left( \left( \Phi_{y_{i,1},i}^{(\tau-1)} - \widetilde{\Theta} (\sigma_0) \right)^{q-1} \right)$$

Denote $\widetilde{\Phi}_{y_{i,1},i}^{(\tau)} = \Phi_{y_{i,1},i}^{(t)} - \widetilde{\Theta} (\sigma_0)$, then we have

$$\widetilde{\Phi}_{y_{i,1},i}^{(\tau)} \geq \widetilde{\Phi}_{y_{i,1},i}^{(\tau-1)} + \frac{\eta}{2\rho^{q-1}} \Theta \left( \frac{s\sigma_p^2}{n} \right) \Theta \left( \left( \widetilde{\Phi}_{y_{i,1},i}^{(\tau-1)} \right)^{q-1} \right)$$

and $\Phi_{y_{i,1},i}^{(\tau)}$ is non-decreasing, specifically, $\Phi_{y_{i,1},i}^{(\tau_0)} \geq \Phi_{y_{i,1},i}^{(0)} = \widetilde{\Theta} (\sqrt{s}\sigma_p\sigma_0)$. Due to our assumptions, we have

$$\eta \cdot \frac{1}{2\rho^{q-1}} \Theta \left( \frac{s\sigma_p^2}{n} \right) \cdot \widetilde{\Theta} \left( \left( \Phi_{y_{i,1},i}^{\tau_0} \right)^{q-2} \right) \geq \eta \cdot \frac{1}{2\rho^{q-1}} \Theta \left( \frac{s\sigma_p^2}{n} \right) \cdot \widetilde{\Theta} \left( \left( \sqrt{s}\sigma_p\sigma_0 \right)^{q-2} \right)$$

$$\geq \eta \cdot \frac{\alpha^q}{\rho^{q-1}} \cdot \widetilde{\Theta}\left(\sigma_0^{q-2}\right) \geq \eta \cdot \frac{\alpha^q}{\rho^{q-1}} \cdot \widetilde{\Theta}\left(\left(\Lambda_j^{(\tau_0)}\right)^{q-2}\right)$$

So using Lemma 5.1 at $[\tau_0, t]$ with our initialization, we can conclude that for each $i \in \mathcal{D}_1$ and $t \leq T_i$, $\Lambda_j^{(t)} \leq \Theta\left(\Lambda_j^{(0)}\right) = \widetilde{\Theta}\left(\sigma_0\right)$. So $\Lambda_j^{(t)} \leq \widetilde{\Theta}\left(\sigma_0\right)$ for $t \leq \max_{i \in \mathcal{D}_1} T_i = T_0$.

Combining these two situations, we can get that $\Lambda_j^{(t)} \leq \widetilde{\Theta}\left(\sigma_0\right)$ holds at $t \leq T_0$. So the hypothesis holds at $t$.

2. If $t \geq T_0$, then from Hypothesis 6, we can get that for $\tau \in [T_0, t]$, $\Phi_{y_{i,1},i}^{(t)} = \widetilde{\Theta}\left(1\right)$. Using the same inequality in proving Hypothesis 6, we have for $\tau \in [T_0, t]$,

$$\Phi_{y_{i,1},i}^{(\tau)} \geq (1 - \eta\lambda)\Phi_{y_{i,1},i}^{(\tau-1)} + \frac{\eta}{\rho^{q-1}}\Theta\left(\frac{s\sigma_p^2}{n}\right)\left|l_{y_{i,1},i,1}^{(\tau-1)}\right|\sigma'\left(\Phi_{y_{i,1},i}^{(\tau-1)} - \widetilde{\Theta}\left(\sigma_0\right)\right)$$

$$= (1 - \eta\lambda)\Phi_{y_{i,1},i}^{(\tau-1)} + \frac{\eta}{\rho^{q-1}}\Theta\left(\frac{s\sigma_p^2}{n}\right)\left|l_{y_{i,1},i,1}^{(\tau-1)}\right|$$

Recursively applying this inequality for $\tau \in [T_0 + 1, t]$ gives that

$$\Phi_{y_{i,1},i}^{(t)} \geq (1 - \eta\lambda)^{t-P_0}\Phi_{y_{i,1},i}^{(T_0)} + \Theta\left(\frac{\eta s\sigma_p^2}{n\rho^{q-1}}\right)\left|l_{y_{i,1},i,1}^{(t-1-\tau)}\right|$$

Taking summation over $i \in \mathcal{D}_1$, we have

$$\sum_{i \in \mathcal{D}_1}\Phi_{y_{i,1},i}^{(t)} \geq (1 - \eta\lambda)^{t-T_0}\sum_{i \in \mathcal{D}_1}\Phi_{y_{i,1},i}^{(Y_0)} + \Theta\left(\frac{\eta s\sigma_p^2}{n\rho^{q-1}}\right)\sum_{\tau=0}^{t-T_0-1}(1 - \eta\lambda)^\tau\sum_{i \in \mathcal{D}_1}\left|l_{y_{i,1},i}^{(t-1-\tau)}\right|$$

Again by $\Phi_{y_{i,1},i}^{(t)} = \widetilde{\Theta}\left(1\right)$ and $\Phi_{y_{i,1},i}^{(T_0)} = \widetilde{\Theta}\left(1\right)$, we have

$$\Theta\left(\frac{\eta s\sigma_p^2}{n\rho^{q-1}}\right)\sum_{\tau=0}^{t-T_0-1}(1 - \eta\lambda)^\tau\sum_{i \in \mathcal{D}_1}\left|l_{y_{i,1},i}^{(t-1-\tau)}\right| \leq \sum_{i \in \mathcal{D}_1}\Phi_{y_{i,1},i}^{(t)} - (1 - \eta\lambda)^{t-T_0}\sum_{i \in \mathcal{D}_1}\Phi_{y_{i,1},i}^{(T_0)}$$

$$\leq \widetilde{\Theta}\left(n\right)$$

Since $y_{i,1} = 1$ or $y_{i,1} = -1$ always happens, and $\left|l_{j,i,1}^{(\tau)}\right| = \left|l_{-j,i,1}^{(\tau)}\right|$, so we have

$$\Theta\left(\frac{\eta s\sigma_p^2}{n\rho^{q-1}}\right)\sum_{\tau=0}^{t-T_0-1}(1 - \eta\lambda)^\tau\sum_{i \in \mathcal{D}_1}\left|l_{j,i,1}^{(t-1-\tau)}\right| \leq \widetilde{\Theta}\left(n\right)$$

According to Hypothesis 8, $\Lambda_j^{(\tau-1)} \leq \widetilde{\Theta}\left(\sigma_0\right)$, which implies that the activation function for $\Lambda_j^{(\tau-1)}$ is $\frac{z^q}{q\rho^{q-1}}$, then $\sigma'\left(\Lambda_j^{(t)}\right) = \frac{\left(\Lambda_j^{(\tau-1)}\right)^{q-1}}{\rho^{q-1}} \leq \widetilde{\Theta}\left(\sigma_0^{q-1}\right)$. Using similar calculations, we can get that

$$\Lambda_j^{(\tau)} \leq \max\left\{(1 - \eta\lambda)\Lambda_j^{(\tau-1)} + \frac{\eta\alpha^q}{n}\sum_{i:y_{i,1}=j,i \in \mathcal{D}_1}\left|l_{j,i,1}^{(\tau-1)}\right|\sigma'\left(\alpha\Lambda_j^{(\tau-1)}\right), \widetilde{\Theta}\left(\alpha^q\sigma_0^{\frac{2}{3}}\right)\right\}$$

$$\leq \max\left\{(1 - \eta\lambda)\Lambda_j^{(\tau-1)} + \widetilde{\Theta}\left(\frac{\eta\alpha^q\sigma_0^{q-1}}{n}\right)\sum_{i:y_{i,1}=j,i \in \mathcal{D}_1}\left|l_{j,i,1}^{(\tau-1)}\right|, \widetilde{\Theta}\left(\alpha^q\sigma_0^{\frac{2}{3}}\right)\right\}$$

then assume $\tau_1$ is the smallest index that $\Lambda_j^{(\tau')} \geq \widetilde{\Theta}\left(\alpha^q\sigma_0^{\frac{2}{3}}\right)$ holds for all $\tau' \in [\tau_1, t]$, then recursively applying this inequality for $\tau \in [\tau_1, t]$, we have

$$\Lambda_j^{(t)} \leq (1 - \eta\lambda)^{t-\tau_1+1}\Lambda_j^{\tau_1-1} + \widetilde{\Theta}\left(\frac{\eta\alpha^q\sigma_0^{q-1}}{n}\right)\sum_{\tau=0}^{(t-\tau_1-1)}(1 - \eta\lambda)^\tau\sum_{i:y_{i,1}=j,i \in \mathcal{D}_1}\left|l_{j,i,1}^{(t-1\tau)}\right|$$

$$\leq \Lambda_j^{\tau_1-1} + \widetilde{\Theta}\left(\frac{\eta\alpha^q\sigma_0^{q-1}}{n}\right) \sum_{\tau=0}^{(t-T_0-1)} (1-\eta\lambda)^\tau \sum_{i:i\in\mathcal{D}_1} \left|l_{j,i,1}^{(t-1\tau)}\right|$$

$$\leq \widetilde{\Theta}\left(\alpha^q\sigma_0^{\frac{2}{3}}\right) + \widetilde{\Theta}\left(\frac{\eta\alpha^q\sigma_0^{q-1}}{n}\right)\cdot\widetilde{\Theta}(n)\cdot\widetilde{\Theta}\left(\frac{n}{\eta s\sigma_p^2}\right) \leq \widetilde{\Theta}(\sigma_0).$$

(vii) In terms of Hypothesis 9, according to our calculation of update, we have

$$\left[\left\langle \mathbf{w}_{-j,r}^{(t)}, j\cdot\mathbf{v}\right\rangle\right]_+ = \Bigg[(1-\eta\lambda)\left\langle \mathbf{w}_{-j,r}^{(t-1)}, j\cdot\mathbf{v}\right\rangle$$

$$+ \frac{\eta}{n}\cdot j\cdot\left(\sum_{s\in\mathcal{D}_1}\alpha y_{s,1}l_{-j,s,1}^{(t-1)}\sigma'\left(\left\langle\mathbf{w}_{-j,r}^{(t-1)},\alpha y_{s,1}\cdot\mathbf{v}\right\rangle\right)\right.$$

$$\left.- \sum_{h=2}^{H}\sum_{s=1}^{n}\beta_h y_{s,1}l_{-j,s,1}^{(t-1)}\sigma'\left(\left\langle\mathbf{w}_{-j,r}^{(t-1)},\boldsymbol{\xi}_{s,1,h}\right\rangle\right)\right)\Bigg]_+$$

$$\overset{(i)}{=} \Bigg[(1-\eta\lambda)\cdot\left\langle\mathbf{w}_{-j,r}^{(t-1)}, j\cdot\mathbf{v}\right\rangle - \frac{\eta}{n}\sum_{s\in\mathcal{D}_1}\alpha\left|l_{-j,s,1}^{(t-1)}\right|\sigma'\left(\left\langle\mathbf{w}_{-j,r}^{(t-1)},\alpha y_{s,1}\cdot\mathbf{v}\right\rangle\right)$$

$$+ \frac{\eta}{n}\sum_{h=2}^{H}\sum_{s=1}^{n}\beta_h\left|l_{-j,s,1}^{(t-1)}\right|\sigma'\left(\left\langle\mathbf{w}_{-j,r}^{(t-1)},\boldsymbol{\xi}_{s,1,h}\right\rangle\right)\Bigg]_+$$

$$\overset{(ii)}{\leq} \Bigg[(1-\eta\lambda)\left\langle\mathbf{w}_{-j,r}^{(t-1)}, j\cdot\mathbf{v}\right\rangle + \frac{\eta}{n}\sum_{h=2}^{H}\sum_{s=1}^{n}\beta_h\left|l_{-j,s,1}^{(t-1)}\right|\sigma'\left(\left\langle\mathbf{w}_{-j,r}^{(t-1)},\boldsymbol{\xi}_{s,1,h}\right\rangle\right)\Bigg]_+$$

$$\overset{(iii)}{\leq} (1-\eta\lambda)\left[\left\langle\mathbf{w}_{-j,r}^{(t-1)}, j\cdot\mathbf{v}\right\rangle\right]_+ + \frac{\eta}{n}\sum_{h=2}^{H}\sum_{s=1}^{n}\beta_h\left|l_{-j,s,1}^{(t-1)}\right|\sigma'\left(\left\langle\mathbf{w}_{-j,r}^{(t-1)},\boldsymbol{\xi}_{s,1,h}\right\rangle\right)$$

$$= (1-\eta\lambda)\left[\left\langle\mathbf{w}_{-j,r}^{(t-1)}, j\cdot\mathbf{v}\right\rangle\right]_+$$

$$+ \frac{\eta}{n}\sum_{h=2}^{H}\sum_{s:y_{s,1}=j}\beta_h\left|l_{-j,s,1}^{(t-1)}\right|\sigma'\left(\left\langle\mathbf{w}_{-j,r}^{(t-1)},\boldsymbol{\xi}_{s,1,h}\right\rangle\right)$$

$$+ \frac{\eta}{n}\sum_{h=2}^{H}\sum_{s:s\in\mathcal{D}_2,y_{s,1}=-j}\beta_h\left|l_{-j,s,1}^{(t-1)}\right|\sigma'\left(\left\langle\mathbf{w}_{-j,r}^{(t-1)},\boldsymbol{\xi}_{s,1,h}\right\rangle\right)$$

$$+ \frac{\eta}{n}\sum_{h=2}^{H}\sum_{s:s\in\mathcal{D}_1,y_{s,1}=-j}\beta_h\left|l_{-j,s,1}^{(t-1)}\right|\sigma'\left(\left\langle\mathbf{w}_{-j,r}^{(t-1)},\boldsymbol{\xi}_{s,1,h}\right\rangle\right)$$

where (i) holds since $\text{sgn}\left(y_{i,1}y_{s,1}l_{-y_{i,1},s,1}^{(t-1)}\right) = -1$, (ii) holds since $[\cdot]_+$ is a monotone function, and (iii) holds according to the definition of $[\cdot]_+$ and the triangle inequality. Recursively applying this inequality from $0$ to $t$ gives that

$$\left[\left\langle\mathbf{w}_{-j,r}^{(t)}, j\cdot\mathbf{v}\right\rangle\right]_+ \leq (1-\eta\lambda)^t\left[\left\langle\mathbf{w}_{-j,r}^{(0)}, j\cdot\mathbf{v}\right\rangle\right]_+$$

$$+ \frac{\eta}{n}\sum_{\tau=0}^{t-1}(1-\eta\lambda)^{t-\tau-1}\sum_{h=2}^{H}\sum_{s:y_{s,1}=j}\beta_h\left|l_{-j,s,1}^{(\tau)}\right|\sigma'\left(\left\langle\mathbf{w}_{-j,r}^{(\tau)},\boldsymbol{\xi}_{s,1,h}\right\rangle\right)$$

$$+ \frac{\eta}{n}\sum_{\tau=0}^{t-1}(1-\eta\lambda)^{t-\tau-1}\sum_{h=2}^{H}\sum_{s:s\in\mathcal{D}_2,y_{s,1}=-j}\beta_h\left|l_{-j,s,1}^{(\tau)}\right|\sigma'\left(\left\langle\mathbf{w}_{-j,r}^{(\tau)},\boldsymbol{\xi}_{s,1,h}\right\rangle\right)$$

$$+ \frac{\eta}{n}\sum_{\tau=0}^{t-1}(1-\eta\lambda)^{t-\tau-1}\sum_{h=2}^{H}\sum_{s:s\in\mathcal{D}_1,y_{s,1}=-j}\beta_h\left|l_{-j,s,1}^{(\tau)}\right|\sigma'\left(\left\langle\mathbf{w}_{-j,r}^{(\tau)},\boldsymbol{\xi}_{s,1,h}\right\rangle\right)$$

For the first term, we can get from the initialization from Lemma D.1 that $\left|\left\langle \mathbf{w}^{(0)}_{-j,r}, \mathbf{v} \right\rangle\right| \leq \widetilde{\Theta}(\sigma_0)$, so we have

$$(1 - \eta\lambda)^t \left[\left\langle \mathbf{w}^{(0)}_{-j,r}, j \cdot \mathbf{v} \right\rangle\right]_+ \leq \left[\left\langle \mathbf{w}^{(0)}_{-j,r}, j \cdot \mathbf{v} \right\rangle\right]_+ \leq \left|\left\langle \mathbf{w}^{(0)}_{-j,r}, \mathbf{v} \right\rangle\right| \leq \widetilde{\Theta}(\sigma_0)$$

For the second term, from Lemma D.2 at $\tau \in [0, t-1]$ we can get that

$$\left\langle \mathbf{w}^{(\tau)}_{-j,r}, \boldsymbol{\zeta}_{s,1,h} \right\rangle = \left\langle \mathbf{w}^{(\tau)}_{-y_{s,1},r}, \boldsymbol{\zeta}_{s,1,h} \right\rangle \leq \widetilde{\Theta}\left(\sqrt{s}\sigma_p\sigma_0\right)$$

From Hypothesis 8 at $\tau \in [0, t-1]$ we can get that

$$\left\langle \mathbf{w}^{(\tau)}_{-j,r}, -y_{s,1}\mathbf{v} \right\rangle = \left\langle \mathbf{w}^{(\tau)}_{-y_{s,1},r}, -y_{s,1}\mathbf{v} \right\rangle \leq \Lambda^{(\tau)}_{-y_{s,1}} \leq \widetilde{\Theta}(\sigma_0)$$

so we have

$$\left\langle \mathbf{w}^{(\tau)}_{-j,r}, \boldsymbol{\xi}_{s,1,h} \right\rangle = \left\langle \mathbf{w}^{(\tau)}_{-j,r}, \boldsymbol{\zeta}_{s,1,h} - \beta_h y_{s,1}\mathbf{v} \right\rangle = \left\langle \mathbf{w}^{(\tau)}_{-j,r}, \boldsymbol{\zeta}_{s,1,h} \right\rangle + \beta_h \left\langle \mathbf{w}^{(\tau)}_{-j,r}, -y_{s,1}\mathbf{v} \right\rangle$$
$$\leq \widetilde{\Theta}\left(\sqrt{s}\sigma_p\sigma_0\right) + \beta_h\widetilde{\Theta}(\sigma_0) \leq \widetilde{\Theta}\left(\sqrt{s}\sigma_p\sigma_0\right)$$

then we can get that

$$\frac{\eta}{n} \sum_{\tau=0}^{t-1} (1-\eta\lambda)^{t-\tau-1} \sum_{h=2}^{H} \sum_{s:y_{s,1}=j} \beta_h \left|l^{(\tau)}_{-j,s,1}\right| \sigma'\left(\left\langle \mathbf{w}^{(\tau)}_{-j,r}, \boldsymbol{\xi}_{s,1,h} \right\rangle\right)$$
$$\leq \frac{\eta}{n} \sum_{\tau=0}^{t-1} (1-\eta\lambda)^{t-\tau-1} \sum_{h=2}^{H} \sum_{s:y_{s,1}=j} \beta_h \left|l^{(\tau)}_{-j,s,1}\right| \sigma'\left(\widetilde{\Theta}\left(\sqrt{s}\sigma_p\sigma_0\right)\right)$$
$$\leq \frac{\eta}{n} \sum_{\tau=0}^{t-1} (1-\eta\lambda)^{t-\tau-1} \sum_{h=2}^{H} \sum_{s:y_{s,1}=j} \beta_h \left|l^{(\tau)}_{-j,s,1}\right| \widetilde{\Theta}\left(\left(\sqrt{s}\sigma_p\sigma_0\right)^{q-1}\right)$$

where we use the fact that $\widetilde{\Theta}\left(\sqrt{s}\sigma_p\sigma_0\right) \leq \rho$ so that the activation function for $\widetilde{\Theta}\left(\sqrt{s}\sigma_p\sigma_0\right)$ is $\frac{z^q}{q\rho^{q-1}}$, which implies that $\sigma'\left(\widetilde{\Theta}\left(\sqrt{s}\sigma_p\sigma_0\right)\right) = \frac{\left(\widetilde{\Theta}\left(\sqrt{s}\sigma_p\sigma_0\right)\right)^{q-1}}{\rho^{q-1}} = \widetilde{\Theta}\left(\left(\sqrt{s}\sigma_p\sigma_0\right)^{q-1}\right)$.

For the third term, from Hypothesis 7 at $\tau \in [0, t-1]$ we can get that

$$\left\langle \mathbf{w}^{(\tau)}_{-j,r}, \boldsymbol{\zeta}_{s,1,h} \right\rangle = \left\langle \mathbf{w}^{(\tau)}_{y_{s,1},r}, \boldsymbol{\zeta}_{s,1,h} \right\rangle \leq \Phi^{(\tau)}_{y_{s,1},s} \leq \widetilde{\Theta}\left(\sqrt{s}\sigma_p\sigma_0\right)$$

from Hypothesis 9 at $\tau \in [0, t-1]$ we can get that

$$\left\langle \mathbf{w}^{(\tau)}_{-j,r}, -y_{s,1}\mathbf{v} \right\rangle = \left\langle \mathbf{w}^{(\tau)}_{y_{s,1},r}, -y_{s,1}\mathbf{v} \right\rangle \leq \widetilde{\Theta}\left(\sigma_0^{\frac{1}{3}}\right)$$

so we have

$$\left\langle \mathbf{w}^{(\tau)}_{-j,r}, \boldsymbol{\xi}_{i,1,h} \right\rangle = \left\langle \mathbf{w}^{(\tau)}_{-j,r}, \boldsymbol{\zeta}_{s,1,h} - \beta_h y_{s,1}\mathbf{v} \right\rangle = \left\langle \mathbf{w}^{(\tau)}_{-j,r}, \boldsymbol{\zeta}_{s,1,h} \right\rangle + \beta_h \left\langle \mathbf{w}^{(\tau)}_{-j,r}, -y_{s,1}\mathbf{v} \right\rangle$$
$$\leq \widetilde{\Theta}\left(\sqrt{s}\sigma_p\sigma_0\right) + \sigma_0^{\frac{1}{3}} \cdot \widetilde{\Theta}\left(\sigma_0^{\frac{1}{3}}\right) \leq \widetilde{\Theta}\left(\sqrt{s}\sigma_p\sigma_0\right)$$

then we can get that

$$\frac{\eta}{n} \sum_{\tau=0}^{t-1} (1-\eta\lambda)^{t-\tau-1} \sum_{h=2}^{H} \sum_{s:s\in\mathcal{D}_2, y_{s,1}=-j} \beta_h \left|l^{(\tau)}_{-j,s,1}\right| \sigma'\left(\left\langle \mathbf{w}^{(\tau)}_{-j,r}, \boldsymbol{\xi}_{s,1,h} \right\rangle\right)$$
$$\leq \frac{\eta}{n} \sum_{\tau=0}^{t-1} (1-\eta\lambda)^{t-\tau-1} \sum_{h=2}^{H} \sum_{s:s\in\mathcal{D}_2, y_{s,1}=-j} \beta_h \left|l^{(\tau)}_{-j,s,1}\right| \sigma'\left(\widetilde{\Theta}\left(\sqrt{s}\sigma_p\sigma_0\right)\right)$$
$$\leq \frac{\eta}{n} \sum_{\tau=0}^{t-1} (1-\eta\lambda)^{t-\tau-1} \sum_{h=2}^{H} \sum_{s:s\in\mathcal{D}_2, y_{s,1}=-j} \beta_h \left|l^{(\tau)}_{-j,s,1}\right| \widetilde{\Theta}\left(\left(\sqrt{s}\sigma_p\sigma_0\right)^{q-1}\right)$$

where we use the fact that $\widetilde{\Theta}\left(\sqrt{s}\sigma_p\sigma_0\right) \le \rho$ so that the activation function for $\widetilde{\Theta}\left(\sqrt{s}\sigma_p\sigma_0\right)$ is $\frac{z^q}{q\rho^{q-1}}$, which implies that $\sigma'\left(\widetilde{\Theta}\left(\sqrt{s}\sigma_p\sigma_0\right)\right) = \frac{\left(\widetilde{\Theta}\left(\sqrt{s}\sigma_p\sigma_0\right)\right)^{q-1}}{\rho^{q-1}} = \widetilde{\Theta}\left(\left(\sqrt{s}\sigma_p\sigma_0\right)^{q-1}\right)$.

Using the same results as verifying Hypothesis 7 and Hypothesis 8, we have that

$$\widetilde{\Theta}\left(\frac{\eta}{n}\right)\sum_{\tau=0}^{t-P_0-1}(1-\eta\lambda)^\tau\sum_{i\in\mathcal{D}_2}\left|l_{j,i,1}^{(t-1-\tau)}\right| \le \widetilde{\Theta}(1)$$

$$\Theta\left(\frac{\eta s\sigma_p^2}{n\rho^{q-1}}\right)\sum_{\tau=0}^{t-T_0-1}(1-\eta\lambda)^\tau\sum_{i\in\mathcal{D}_1}\left|l_{j,i,1}^{(t-1-\tau)}\right| \le \widetilde{\Theta}(n)$$

Then the summation of the second term and third term will not exceed

$$\frac{\eta}{n}\sum_{\tau=0}^{t-1}(1-\eta\lambda)^{t-\tau-1}\sum_{h=2}^{H}\sum_{s:y_{s,1}=j}\beta_h\left|l_{-j,s,1}^{(\tau)}\right|\widetilde{\Theta}\left(\left(\sqrt{s}\sigma_p\sigma_0\right)^{q-1}\right)$$

$$+\frac{\eta}{n}\sum_{\tau=0}^{t-1}(1-\eta\lambda)^{t-\tau-1}\sum_{h=2}^{H}\sum_{s:s\in\mathcal{D}_2,y_{s,1}=-j}\beta_h\left|l_{-j,s,1}^{(\tau)}\right|\widetilde{\Theta}\left(\left(\sqrt{s}\sigma_p\sigma_0\right)^{q-1}\right)$$

$$=\frac{2\eta}{n}\sum_{\tau=0}^{t-1}(1-\eta\lambda)^{t-\tau-1}\sum_{h=2}^{H}\sum_{s:s\in\mathcal{D}_2,y_{s,1}=-j}\beta_h\left|l_{-j,s,1}^{(\tau)}\right|\widetilde{\Theta}\left(\left(\sqrt{s}\sigma_p\sigma_0\right)^{q-1}\right)$$

$$+\frac{\eta}{n}\sum_{\tau=0}^{t-1}(1-\eta\lambda)^{t-\tau-1}\sum_{h=2}^{H}\sum_{s:s\in\mathcal{D}_1,y_{s,1}=-j}\beta_h\left|l_{-j,s,1}^{(\tau)}\right|\widetilde{\Theta}\left(\left(\sqrt{s}\sigma_p\sigma_0\right)^{q-1}\right)$$

$$\le\frac{2\eta\beta_h}{n}\cdot\widetilde{\Theta}\left(\frac{\eta}{n}\right)\cdot\widetilde{\Theta}\left(\left(\sqrt{s}\sigma_p\sigma_0\right)^{q-1}\right)+\frac{\eta\beta_h}{n}\cdot\widetilde{\Theta}\left(\frac{n^2\rho^{q-1}}{\eta s\sigma_p^2}\right)\cdot\widetilde{\Theta}\left(\left(\sqrt{s}\sigma_p\sigma_0\right)^{q-1}\right)\le\widetilde{\Theta}\left(\sigma_0^{\frac{1}{3}}\right)$$

For the fourth term, the gradient calculation implies that for each $s$ which satisfies that $s \in \mathcal{D}_1$ and $y_{s,1}=-j$, and $h\in[2,H]$,

$$\left\langle\mathbf{w}_{y_{s,1},r}^{(t)},\boldsymbol{\zeta}_{s,1,h}\right\rangle \ge (1-\eta\lambda)\cdot\left\langle\mathbf{w}_{y_{s,1},r}^{(t-1)},\boldsymbol{\zeta}_{s,1,h}\right\rangle$$

$$+\frac{\eta}{n}\sum_{g=2}^{H}\sum_{i=1}^{n}l_{y_{s,1},i,1}\sigma'\left(\left\langle\mathbf{w}_{y_{s,1},r}^{(t-1)},\boldsymbol{\xi}_{i,1,g}\right\rangle\right)\langle\boldsymbol{\zeta}_{i,1,g},\boldsymbol{\zeta}_{s,1,h}\rangle$$

According to Lemma B.3, with probability exceeding $1-n^{-2}$, for all $(i,g)\ne(s,h)$, $\langle\boldsymbol{\zeta}_{i,1,g},\boldsymbol{\zeta}_{s,1,h}\rangle=0$. By using the calculations above, we can get that with probability exceeding $1-2n^{-2}$, $\left|\|\boldsymbol{\zeta}_{s,1,h}\|^2-s\sigma_p^2\right|\le\widetilde{\Theta}\left(\sqrt{s}\sigma_p^2\right)$, which implies that $\|\boldsymbol{\zeta}_{s,1,h}\|\ge\Theta\left(\sqrt{s}\sigma_p\right)$. Since $\text{sgn}\left(l_{y_{s,1},s,1}^{(t-1)}\right)=1$, then we have

$$\left\langle\mathbf{w}_{y_{s,1},r}^{(t)},\boldsymbol{\zeta}_{s,1,h}\right\rangle \ge (1-\eta\lambda)\cdot\left\langle\mathbf{w}_{y_{s,1},r}^{(t-1)},\boldsymbol{\zeta}_{s,1,h}\right\rangle$$

$$+\frac{\eta}{n}\Theta\left(s\sigma_p^2\right)\left|l_{y_{s,1},s,1}^{(t-1)}\right|\sigma'\left(\left\langle\mathbf{w}_{y_{s,1},r}^{(t-1)},\boldsymbol{\xi}_{s,1,h}\right\rangle\right)$$

Recursively applying this inequality from 0 to $t$ gives that

$$\left\langle\mathbf{w}_{y_{s,1},r}^{(t)},\boldsymbol{\zeta}_{s,1,h}\right\rangle \ge (1-\eta\lambda)^t\left\langle\mathbf{w}_{y_{s,1},r}^{(0)},\boldsymbol{\zeta}_{s,1,h}\right\rangle$$

$$+\frac{\eta}{n}\Theta\left(s\sigma_p^2\right)\sum_{\tau=0}^{t-1}(1-\eta\lambda)^{t-\tau-1}\left|l_{y_{s,1},s,1}^{(\tau)}\right|\sigma'\left(\left\langle\mathbf{w}_{y_{s,1},r}^{(\tau)},\boldsymbol{\xi}_{s,1,h}\right\rangle\right)$$

Since $t\le\min\{P_0,T_0\}\le\min_{s\in\mathcal{D}_1}T_s$, then we can get that $\Phi_{y_{s,1},s}^{(t)}\le\widetilde{\Theta}(1)$. From the initialization in Lemma D.2, we can get that $\left|\left\langle\mathbf{w}_{y_{s,1},r}^{(0)},\boldsymbol{\zeta}_{s,1,h}\right\rangle\right|\le\widetilde{\Theta}\left(\sqrt{s}\sigma_p\sigma_0\right)$. So we have

$$\frac{\eta}{n}\sum_{\tau=0}^{t-1}(1-\eta\lambda)^{t-\tau-1}\left|l_{y_{s,1},s,1}^{(\tau)}\right|\sigma'\left(\left\langle\mathbf{w}_{y_{s,1},r}^{(\tau)},\boldsymbol{\xi}_{s,1,h}\right\rangle\right)$$

$$\leq \frac{1}{\Theta\left(s\sigma_p^2\right)} \left( \left\langle \mathbf{w}_{y_{s,1},r}^{(t)}, \boldsymbol{\zeta}_{s,1,h} \right\rangle + (1-\eta\lambda)^t \left\langle \mathbf{w}_{y_{s,1},r}^{(0)}, \boldsymbol{\zeta}_{s,1,h} \right\rangle \right)$$

$$\leq \frac{1}{\Theta\left(s\sigma_p^2\right)} \left( \Phi_{y_{s,1},s}^{(t)} + \left| \left\langle \mathbf{w}_{y_{s,1},r}^{(0)}, \boldsymbol{\zeta}_{s,1,h} \right\rangle \right| \right)$$

$$\leq \frac{1}{\Theta\left(s\sigma_p^2\right)} \left( \widetilde{\Theta}(1) + \widetilde{\Theta}\left(\sqrt{s}\sigma_p\sigma_0\right) \right) \leq \frac{\widetilde{\Theta}(1)}{\Theta\left(s\sigma_p^2\right)}$$

Adding up all the above inequalities for those $s$ which satisfies $s \in \mathcal{D}_1$ and $y_{s,1} = -j$, and $h \in [2, H]$, we can get that

$$\frac{\eta}{n} \sum_{\tau=0}^{t-1} (1-\eta\lambda)^{t-\tau-1} \sum_{h=2}^{H} \sum_{s:s\in\mathcal{D}_1,y_{s,1}=-j} \beta_h \left| l_{-j,s,1}^{(\tau)} \right| \sigma'\left( \left\langle \mathbf{w}_{-j,r}^{(\tau)}, \boldsymbol{\xi}_{s,1,h} \right\rangle \right)$$

$$= \beta_h \frac{\eta}{n} \sum_{\tau=0}^{t-1} (1-\eta\lambda)^{t-\tau-1} \sum_{h=2}^{H} \sum_{s:s\in\mathcal{D}_1,y_{s,1}=-j} \left| l_{y_{s,1},s,1}^{(\tau)} \right| \sigma'\left( \left\langle \mathbf{w}_{y_{s,1},r}^{(\tau)}, \boldsymbol{\xi}_{s,1,h} \right\rangle \right)$$

$$\leq \beta_h \cdot n \cdot (H-1) \cdot \widetilde{\Theta}(1) \leq \widetilde{\Theta}\left(\sigma_0^{\frac{1}{3}}\right)$$

Combining these inequalities together, we have that

$$\left[ \left\langle \mathbf{w}_{-j,r}^{(t)}, j \cdot \mathbf{v} \right\rangle \right]_+ \leq (1-\eta\lambda)^t \left[ \left\langle \mathbf{w}_{-j,r}^{(0)}, j \cdot \mathbf{v} \right\rangle \right]_+$$

$$+ \frac{\eta}{n} \sum_{\tau=0}^{t-1} (1-\eta\lambda)^{t-\tau-1} \sum_{h=2}^{H} \sum_{s:y_{s,1}=j} \beta_h \left| l_{-j,s,1}^{(\tau)} \right| \sigma'\left( \left\langle \mathbf{w}_{-j,r}^{(\tau)}, \boldsymbol{\xi}_{s,1,h} \right\rangle \right)$$

$$+ \frac{\eta}{n} \sum_{\tau=0}^{t-1} (1-\eta\lambda)^{t-\tau-1} \sum_{h=2}^{H} \sum_{s:s\in\mathcal{D}_2,y_{s,1}=-j} \beta_h \left| l_{-j,s,1}^{(\tau)} \right| \sigma'\left( \left\langle \mathbf{w}_{-j,r}^{(\tau)}, \boldsymbol{\xi}_{s,1,h} \right\rangle \right)$$

$$+ \frac{\eta}{n} \sum_{\tau=0}^{t-1} (1-\eta\lambda)^{t-\tau-1} \sum_{h=2}^{H} \sum_{s:s\in\mathcal{D}_1,y_{s,1}=-j} \beta_h \left| l_{-j,s,1}^{(\tau)} \right| \sigma'\left( \left\langle \mathbf{w}_{-j,r}^{(\tau)}, \boldsymbol{\xi}_{s,1,h} \right\rangle \right)$$

$$\leq \widetilde{\Theta}(\sigma_0) + \widetilde{\Theta}\left(\sigma_0^{\frac{1}{3}}\right) + \widetilde{\Theta}\left(\sigma_0^{\frac{1}{3}}\right) \leq \widetilde{\Theta}\left(\sigma_0^{\frac{1}{3}}\right)$$

so the hypothesis holds at $t$.

Therefore, by induction, we finish the proof of all hypotheses. □

-

## F  MULTI-TASK LEARNING

Next, we will focus on the proof of multi-task part. The proof techniques and structures are basically the same as the proofs in Section C. In order to prove Theorem 4.2, we need the following technical lemmas.

**Lemma F.1** (Convergence Guarantee of GD). *If the step size satisfies $\eta \leq O(\sigma_0)$, then for any $t \geq P_0$, it holds that*

$$L(\mathbf{W}^{(t+1)}) - L(\mathbf{W}^{(t)}) \leq -\frac{\eta}{2}\|\nabla L(\mathbf{W}^{(t)})\|_{\mathrm{F}}^2.$$

**Lemma F.2** (Generalization Performance of GD). *Let*

$$\mathbf{W}^* = \arg\min_{\left\{ \mathbf{W}^{(1)},...,\mathbf{W}^{(T)} \right\}} \left\| \nabla L\left(\mathbf{W}^{(t)}\right) \right\|_{\mathrm{F}}.$$

*Then by selecting $T = \frac{\mathrm{poly}(n)}{\eta}$, for all training data, we have*

$$\frac{1}{nK} \sum_{i=1}^{n} \sum_{k=1}^{K} \mathbf{1}\left[ F_{y_{i,k}}\left(\mathbf{W}^*, \mathbf{x}_{i,k}\right) \leq F_{-y_{i,k}}\left(\mathbf{W}^*, \mathbf{x}_{i,k}\right) \right] = 0.$$

*Moreover, in terms of the test data $(\mathbf{x}, y) \sim \mathcal{D}$, we have*

$$\mathbb{P}_{(\mathbf{x},y)\sim\mathcal{D}}\left[ F_y\left(\mathbf{W}^*, \mathbf{x}\right) \leq F_{-y}\left(\mathbf{W}^*, \mathbf{x}\right) \right] \leq \mathrm{poly}\left(n^{-1}\right).$$

## G  PROOF OF LEMMAS IN APPENDIX F

In order to prove Lemma F.1 and Lemma F.2, we need the following technical lemmas.

**Lemma G.1** (Off-diagonal Correlations for Task-specific Feature). *For any $j \in \{-1,1\}$, $k \in [K]$, and any $t$, it holds that $[\langle \mathbf{w}_{-j,r}^{(t)}, j \cdot \mathbf{v}_k \rangle]_+ \leq \widetilde{\Theta}(\sigma_0)$.*

**Lemma G.2** (Off-diagonal correlations for Random Noises). *For any data $(\mathbf{x}_{i,k}, y_{i,k})$, any $h \in [2, H]$ and any $t$, it holds that $\left[\left\langle \mathbf{w}_{-y_{i,k},r}^{(t)}, \boldsymbol{\zeta}_{i,k,h} \right\rangle\right]_+ \leq \widetilde{\Theta}\left(\sqrt{s}\sigma_p\sigma_0\right)$.*

**Lemma G.3.** *Suppose the training data is generated according to Definition 3.1 and Definition 3.2. Let $\Lambda_j^{(t)} = \max_{r \in [m]} \left[\left\langle \mathbf{w}_{j,r}^{(t)}, j \cdot \mathbf{v} \right\rangle\right]_+$, $\Psi_{j,k}^{(t)} = \max_{r \in [m]} \left[\left\langle \mathbf{w}_{j,r}^{(t)}, j \cdot \mathbf{v}_k \right\rangle\right]_+$, $\Phi_{j,i,h}^{(t)} = \max_{r \in [m]} \max_{k \in [K]} \left[\left\langle \mathbf{w}_{j,r}^{(t)}, \boldsymbol{\xi}_{i,k,h} \right\rangle\right]_+$, and $\Phi_{j,i}^{(t)} = \max_h \Phi_{j,i,h}^{(t)}$ and $\Phi_j^{(t)} = \max_{i \in [n]} \Phi_{j,i}^{(t)}$ Then let $P_{j,k}$ be the iteration number that $\Psi_{j,k}^{(t)}$ reaches $\Theta(1/m)$ for $j \in \{-1,1\}$, $Q_{j,k}$ be the iteration number that $\Lambda_j^{(t)}$ reaches $\Theta(1/m)$ for $j \in \{-1,1\}$, we have $P_{j,k} \leq \widetilde{\Theta}\left(\sigma_0^{2-q}/\eta\right)$ for all $j \in \{-1,1\}$ and $k \in [K]$, and $Q_j = \widetilde{\Theta}\left(\sigma_0^{2-q}/\eta\right)$ for all $j \in \{-1,1\}$. Moreover, let $P_0 = \max_{j \in \{-1,1\}, k \in [K]} P_j$ and $Q_0 = \max_{j \in \{-1,1\}} Q_j$. For all $t \geq 0$, it holds that $\Phi_{j,i}^{(t)} \leq \widetilde{\Theta}\left(\sqrt{s}\sigma_p\sigma_0\right)$ for all $j \in \{-1,1\}$ and $i \in [n]$, and $\left[\left\langle \mathbf{w}_{-j,r}^{(t)}, j \cdot \mathbf{v} \right\rangle\right]_+ \leq \widetilde{\Theta}\left(\sigma_0^{\frac{1}{3}}\right)$ for all $j \in \{-1,1\}$.*

Given these three useful lemmas, we are ready to prove Lemma F.1 and Lemma F.2.

### G.1  PROOF OF LEMMA F.1

*Proof of Lemma F.1.* The proof of this lemma is basically relying on the smoothness property of the loss function $L(\mathbf{W})$ given certain constraints on the inner products with each patch.

Let $\Delta F_{j,i,k} = F_j\left(\mathbf{W}^{(t+1)}, \mathbf{x}_{i,k}\right) - F_j\left(\mathbf{W}^{(t)}, \mathbf{x}_{i,k}\right)$, we can get the following Taylor expansion on the loss function $L_{i,k}\left(\mathbf{W}^{(t+1)}\right)$,

$$L_{i,k}\left(\mathbf{W}^{(t+1)}\right) - L_{i,k}\left(\mathbf{W}^{(t)}\right) \leq \sum_j \frac{\partial L_{i,k}\left(\mathbf{W}^{(t)}\right)}{\partial F_j\left(\mathbf{W}^{(t)}, \mathbf{x}_{i,k}\right)} \cdot \Delta F_{j,i,k} + \sum_j \left(\Delta F_{j,i,k}\right)^2.$$

In particular, by Lemma G.1 to Lemma G.3, we know that $\left\langle \mathbf{w}_{j,r}^{(t)}, \alpha y_{i,k} \cdot \mathbf{v} \right\rangle \leq \widetilde{\Theta}(1)$, $\left\langle \mathbf{w}_{j,r}^{(t)}, y_{i,k} \cdot \mathbf{v}_k \right\rangle \leq \widetilde{\Theta}(1)$ and $\left\langle \mathbf{w}_{j,r}^{(t)}, \xi_{i,k} \right\rangle \leq \widetilde{\Theta}(1)$. Then we can apply first-order Taylor expansion to $F_j\left(\mathbf{W}^{(t+1)}, \mathbf{x}_{i,k}\right)$, which requires to characterize the second-order error of the Taylor expansions on $\sigma\left(\left\langle \mathbf{w}_{j,r}^{(t+1)}, \alpha y_{i,k}\mathbf{v} \right\rangle\right)$, $\sigma\left(\left\langle \mathbf{w}_{j,r}^{(t+1)}, y_{i,k}\mathbf{v}_k \right\rangle\right)$ and $\sigma\left(\left\langle \mathbf{w}_{j,r}^{(t+1)}, \xi_{i,k} \right\rangle\right)$,

$$\left| \sigma\left(\left\langle \mathbf{w}_{j,r}^{(t+1)}, \alpha y_{i,k}\mathbf{v} \right\rangle\right) - \sigma\left(\left\langle \mathbf{w}_{j,r}^{(t)}, \alpha y_{i,k}\mathbf{v} \right\rangle\right) - \left\langle \nabla_{\mathbf{w}_{j,r}}\sigma\left(\left\langle \mathbf{w}_{j,r}^{(t+1)}, \alpha y_{i,k}\mathbf{v} \right\rangle\right), \mathbf{w}_{j,r}^{(t+1)} - \mathbf{w}_{j,r}^{(t)} \right\rangle \right|$$
$$\leq \widetilde{\Theta}\left(\left\| \mathbf{w}_{j,r}^{(t+1)} - \mathbf{w}_{j,r}^{(t)} \right\|_2^2\right) = \widetilde{\Theta}\left(\eta^2 \left\| \nabla_{\mathbf{w}_{j,r}} L\left(\mathbf{W}^{(t)}\right) \right\|_2^2\right),$$
$$\left| \sigma\left(\left\langle \mathbf{w}_{j,r}^{(t+1)}, y_{i,k}\mathbf{v}_k \right\rangle\right) - \sigma\left(\left\langle \mathbf{w}_{j,r}^{(t)}, y_{i,k}\mathbf{v}_k \right\rangle\right) - \left\langle \nabla_{\mathbf{w}_{j,r}}\sigma\left(\left\langle \mathbf{w}_{j,r}^{(t+1)}, y_{i,k}\mathbf{v}_k \right\rangle\right), \mathbf{w}_{j,r}^{(t+1)} - \mathbf{w}_{j,r}^{(t)} \right\rangle \right|$$
$$\leq \widetilde{\Theta}\left(\left\| \mathbf{w}_{j,r}^{(t+1)} - \mathbf{w}_{j,r}^{(t)} \right\|_2^2\right) = \widetilde{\Theta}\left(\eta^2 \left\| \nabla_{\mathbf{w}_{j,r}} L\left(\mathbf{W}^{(t)}\right) \right\|_2^2\right),$$
$$\left| \sigma\left(\left\langle \mathbf{w}_{j,r}^{(t+1)}, \xi_{i,k} \right\rangle\right) - \sigma\left(\left\langle \mathbf{w}_{j,r}^{(t)}, \xi_{i,k} \right\rangle\right) - \left\langle \nabla_{\mathbf{w}_{j,r}}\sigma\left(\left\langle \mathbf{w}_{j,r}^{(t+1)}, \xi_{i,k} \right\rangle\right), \mathbf{w}_{j,r}^{(t+1)} - \mathbf{w}_{j,r}^{(t)} \right\rangle \right|$$
$$\leq \widetilde{\Theta}\left(\left\| \mathbf{w}_{j,r}^{(t+1)} - \mathbf{w}_{j,r}^{(t)} \right\|_2^2\right) = \widetilde{\Theta}\left(\eta^2 \left\| \nabla_{\mathbf{w}_{j,r}} L\left(\mathbf{W}^{(t)}\right) \right\|_2^2\right),$$

Then combining the above bounds for every $r \in [m]$, we can get the following bound for $\Delta F_{j,i,k}$

$$\left| \Delta F_{j,i,k} - \left\langle \nabla_{\mathbf{W}} F_j \left( \mathbf{W}^{(t)}, \mathbf{x}_{i,k} \right), \mathbf{W}^{(t+1)} - \mathbf{W}^{(t)} \right\rangle \right| \leq \widetilde{\Theta} \left( \eta^2 \sum_{r \in [m]} \left\| \nabla_{\mathbf{w}_{j,r}} L \left( \mathbf{W}^{(t)} \right) \right\|_2^2 \right)$$

$$= \widetilde{\Theta} \left( \eta^2 \left\| \nabla L \left( \mathbf{W}^{(t)} \right) \right\|_{\mathrm{F}}^2 \right).$$

Moreover, since $\langle \mathbf{w}_{j,r}^{(t)}, \alpha y_{i,k} \cdot \mathbf{v} \rangle \leq \widetilde{\Theta}(1)$, $\langle \mathbf{w}_{j,r}^{(t)}, y_{i,k} \cdot \mathbf{v}_k \rangle \leq \widetilde{\Theta}(1)$, $\langle \mathbf{w}_{j,r}^{(t)}, \xi_{i,k} \rangle \leq \widetilde{\Theta}(1)$ and $\sigma(\cdot)$ is convex, then we have

$$\left| \sigma \left( \left\langle \mathbf{w}_{j,r}^{(t+1)}, \alpha y_{i,k} \mathbf{v} \right\rangle \right) - \sigma \left( \left\langle \mathbf{w}_{j,r}^{(t+1)}, \alpha y_{i,k} \mathbf{v} \right\rangle \right) \right|$$

$$\leq \max \left\{ \left| \sigma' \left( \left\langle \mathbf{w}_{j,r}^{(t+1)}, \alpha y_{i,k} \mathbf{v} \right\rangle \right) \right|, \left| \sigma' \left( \left\langle \mathbf{w}_{j,r}^{(t)}, \alpha y_{i,k} \mathbf{v} \right\rangle \right) \right| \right\} \cdot \left| \left\langle \mathbf{v}, \mathbf{w}_{j,r}^{(t+1)} - \mathbf{w}_{j,r}^{(t)} \right\rangle \right|$$

$$\leq \widetilde{\Theta} \left( \left\| \mathbf{w}_{j,r}^{(t+1)} - \mathbf{w}_{j,r}^{(t)} \right\|_2 \right).$$

Similarly we also have

$$\left| \sigma \left( \left\langle \mathbf{w}_{j,r}^{(t+1)}, y_{i,k} \mathbf{v}_k \right\rangle \right) - \sigma \left( \left\langle \mathbf{w}_{j,r}^{(t+1)}, y_{i,k} \mathbf{v}_k \right\rangle \right) \right| \leq \widetilde{\Theta} \left( \left\| \mathbf{w}_{j,r}^{(t+1)} - \mathbf{w}_{j,r}^{(t)} \right\|_2 \right).$$

and

$$\left| \sigma \left( \left\langle \mathbf{w}_{j,r}^{(t+1)}, \xi_{i,k} \right\rangle \right) - \sigma \left( \left\langle \mathbf{w}_{j,r}^{(t+1)}, \xi_{i,k} \right\rangle \right) \right| \leq \widetilde{\Theta} \left( \left\| \mathbf{w}_{j,r}^{(t+1)} - \mathbf{w}_{j,r}^{(t)} \right\|_2 \right).$$

Combining the above inequalities for every $r \in [m]$, we have

$$|\Delta F_{j,i,k}|^2 \leq \widetilde{\Theta} \left( \left[ \sum_{r \in [m]} \left\| \mathbf{w}_{j,r}^{(t+1)} - \mathbf{w}_{j,r}^{(t)} \right\|_2 \right]^2 \right)$$

$$\leq \widetilde{\Theta} \left( m \eta^2 \left\| \nabla L \left( \mathbf{W}^{(t)} \right) \right\|_{\mathrm{F}}^2 \right)$$

$$= \widetilde{\Theta} \left( \eta^2 \left\| \nabla L \left( \mathbf{W}^{(t)} \right) \right\|_{\mathrm{F}}^2 \right).$$

Now we combine all the above inequalities, which gives

$$L_{i,k}(\mathbf{W}^{(t+1)}) - L_{i,k} \left( \mathbf{W}^{(t)} \right) \leq \sum_j \frac{\partial L_{i,k} \left( \mathbf{W}^{(t)} \right)}{\partial F_j \left( \mathbf{W}^{(t)}, \mathbf{x}_{i,k} \right)} \cdot \Delta F_{j,i,k} + \sum_j \left( \Delta F_{j,i,k} \right)^2$$

$$= \left\langle \nabla L_{i,k} \left( \mathbf{W}^{(t)} \right), \mathbf{W}^{(t+1)} - \mathbf{W}^{(t)} \right\rangle$$

$$+ \widetilde{\Theta} \left( \eta^2 \left\| \nabla L \left( \mathbf{W}^{(t)} \right) \right\|_{\mathrm{F}}^2 \right).$$

Taking sum over $i \in [n]$ and $k \in [K]$ and applying the smoothness property of the regularization function $\lambda \|\mathbf{W}\|_{\mathrm{F}}^2$, we can get

$$L \left( \mathbf{W}^{(t+1)} \right) - L \left( \mathbf{W}^{(t)} \right)$$

$$= \frac{1}{nK} \sum_{i=1}^n \sum_{k=1}^K \left\{ \left[ L_{i,k} \left( \mathbf{W}^{(t+1)} \right) - L_{i,k} \left( \mathbf{W}^{(t)} \right) \right] + \lambda \left( \left\| \mathbf{W}^{(t+1)} \right\|_{\mathrm{F}}^2 - \left\| \mathbf{W}^{(t)} \right\|_{\mathrm{F}}^2 \right) \right\}$$

$$\leq \left\langle \nabla L \left( \mathbf{W}^{(t)} \right), \mathbf{W}^{(t+1)} - \mathbf{W}^{(t)} \right\rangle + \widetilde{\Theta} \left( \eta^2 \left\| \nabla L \left( \mathbf{W}^{(t)} \right) \right\|_{\mathrm{F}}^2 \right)$$

$$= - \left( \eta - \widetilde{\Theta} \left( \eta^2 \right) \right) \cdot \left\| \nabla L \left( \mathbf{W}^{(t)} \right) \right\|_{\mathrm{F}}^2$$

$$\leq - \frac{\eta}{2} \left\| \nabla L \left( \mathbf{W}^{(t)} \right) \right\|_{\mathrm{F}}^2,$$

where the last inequality is due to our choice of step size $\eta = o(1)$. This completes the proof. $\square$

## G.2 PROOF OF LEMMA F.2

*Proof of Lemma F.2.* From Lemma G.2 and Lemma G.3, we can get that

$$F_{y_{i,k}} \left( \mathbf{W}^*, \mathbf{x}_{i,k} \right) = \sum_{r=1}^{m} \left[ \sigma \left( \left\langle \mathbf{w}^*_{y_{i,k},r}, \alpha y_{i,k} \mathbf{v} \right\rangle \right) + \sum_{h=2}^{H} \sigma \left( \left\langle \mathbf{w}^*_{y_{i,k},r}, \boldsymbol{\xi}_{i,k,h} \right\rangle \right) \right]$$

$$\geq \max_{r \in [m]} \sigma \left( \left\langle \mathbf{w}^*_{y_{i,k},r}, \alpha y_{i,k} \mathbf{v} \right\rangle \right) = \widetilde{\Theta} \left( \alpha^q \right)$$

$$F_{-y_{i,k}} \left( \mathbf{W}^*, \mathbf{x}_{i,k} \right) = \sum_{r=1}^{m} \left[ \sigma \left( \left\langle \mathbf{w}^*_{-y_{i,k},r}, \alpha y_{i,k} \mathbf{v} \right\rangle \right) + \sum_{h=2}^{H} \sigma \left( \left\langle \mathbf{w}^*_{-y_{i,k},r}, \boldsymbol{\xi}_{i,k,h} \right\rangle \right) \right]$$

$$\leq m \max_{r \in [m]} \sigma \left( \left\langle \mathbf{w}^*_{-y_{i,k},r}, \alpha y_{i,k} \mathbf{v} \right\rangle \right)$$

$$+ m \left( H - 1 \right) \sigma \left( \max_{r \in [m]} \max_{h \in [2,H]} \left\langle \mathbf{w}^*_{-y_{i,k},r}, \boldsymbol{\zeta}_{i,k,h} \right\rangle + \max_{h \in [2,H]} \beta_h \max_{r \in [m]} \left\langle \mathbf{w}^*_{-y_{i,k},r}, -y_{i,k} \mathbf{v} \right\rangle \right)$$

$$\leq m \widetilde{\Theta} \left( \alpha^q \sigma_0^{\frac{q}{3}} \right) + m \left( H - 1 \right) \widetilde{\Theta} \left( \left( \max_{h \in [2,H]} \beta_h \right)^q \right),$$

so $F_{y_{i,k}} \left( \mathbf{W}^*, \mathbf{x}_{i,k} \right) \geq F_{-y_{i,k}} \left( \mathbf{W}^*, \mathbf{x}_{i,k} \right)$ holds for $i \in \mathcal{D}_1$. Similarly, from Lemma G.1 to Lemma G.3, we also have

$$F_{y_{i,k}} \left( \mathbf{W}^*, \mathbf{x}_{i,k} \right) = \sum_{r=1}^{m} \left[ \sigma \left( \left\langle \mathbf{w}^*_{y_{i,k},r}, y_{i,k} \mathbf{v}_k \right\rangle \right) + \sum_{h=2}^{H} \sigma \left( \left\langle \mathbf{w}^*_{y_{i,k},r}, \boldsymbol{\xi}_{i,k,h} \right\rangle \right) \right]$$

$$\geq \max_{r \in [m]} \sigma \left( \left\langle \mathbf{w}^*_{y_{i,k},r}, y_{i,k} \mathbf{v}_k \right\rangle \right) = \widetilde{\Theta} \left( 1 \right)$$

$$F_{-y_{i,k}} \left( \mathbf{W}^*, \mathbf{x}_{i,k} \right) = \sum_{r=1}^{m} \left[ \sigma \left( \left\langle \mathbf{w}^*_{-y_{i,k},r}, y_{i,k} \mathbf{v}_k \right\rangle \right) + \sum_{h=2}^{H} \sigma \left( \left\langle \mathbf{w}^*_{-y_{i,k},r}, \boldsymbol{\xi}_{i,k,h} \right\rangle \right) \right]$$

$$\leq m \max_{r \in [m]} \sigma \left( \left\langle \mathbf{w}^*_{-y_{i,k},r}, y_{i,k} \mathbf{v}_k \right\rangle \right)$$

$$+ m \left( H - 1 \right) \sigma \left( \max_{r \in [m]} \max_{h \in [2,H]} \left\langle \mathbf{w}^*_{-y_{i,k},r}, \boldsymbol{\zeta}_{i,k,h} \right\rangle + \max_{h \in [2,H]} \beta_h \max_{r \in [m]} \left\langle \mathbf{w}^*_{-y_{i,k},r}, -y_{i,k} \mathbf{v} \right\rangle \right)$$

$$\leq m \widetilde{\Theta} \left( \sigma_0^q \right) + m \left( H - 1 \right) \widetilde{\Theta} \left( \left( \max_{h \in [2,H]} \beta_h \right)^q \right) = o \left( 1 \right).$$

so $F_{y_{i,k}} \left( \mathbf{W}^*, \mathbf{x}_{i,k} \right) \geq F_{-y_{i,k}} \left( \mathbf{W}^*, \mathbf{x}_{i,k} \right)$ holds for $i \in \mathcal{D}_2$. Combining these two parts, we have that for $i \in [n]$, $F_{y_{i,k}} \left( \mathbf{W}^*, \mathbf{x}_{i,k} \right) \geq F_{-y_{i,k}} \left( \mathbf{W}^*, \mathbf{x}_{i,k} \right)$ holds, which directly implies that

$$\frac{1}{n} \sum_{i=1}^{n} \mathbf{1} \left[ F_{-y_{i,k}} \left( \mathbf{W}^*, \mathbf{x}_{i,k} \right) \leq F_{y_{i,k}} \left( \mathbf{W}^*, \mathbf{x}_{i,k} \right) \right] = 0$$

Therefore, $\mathbf{W}^*$ can correctly classify all training data and thus achieve zero training error.
In terms of the test data $(\mathbf{x}, y)$ which is generated according to our assumptions, then with probability $p$, it will have the patch of task-shared feature and the patches of noise, like the training data for $i \in \mathcal{D}_1$, then $\mathbf{x} = [\alpha y \mathbf{v}, \boldsymbol{\xi}_2, ..., \boldsymbol{\xi}_H]$. Similar to Lemma B.3, the support set of the random noise of this data will have no interpolation with the support sets of random noises in training samples with probability larger than $1 - n^2$, which implies that $\left\langle \mathbf{w}^*_{y,r}, \boldsymbol{\zeta}_{\mathbf{x},h} \right\rangle = 0$. For this kind of data, we have

$$F_y \left( \mathbf{W}^*, \mathbf{x} \right) = \sum_{r=1}^{m} \left[ \sigma \left( \left\langle \mathbf{w}^*_{y,r}, \alpha y \mathbf{v} \right\rangle \right) + \sum_{h=2}^{H} \left( \left\langle \mathbf{w}^*_{y,r}, \boldsymbol{\zeta}_{\mathbf{x},h} - \beta_h \mathbf{v} \right\rangle \right) \right]$$

$$= \sum_{r=1}^{m} \left[ \sigma \left( \left\langle \mathbf{w}^*_{y,r}, \alpha y \mathbf{v} \right\rangle \right) + \sum_{h=2}^{H} \left( \left\langle \mathbf{w}^*_{y,r}, -\beta_h \mathbf{v} \right\rangle \right) \right]$$

$$\geq \max_{r\in[m]} \sigma\left(\left\langle \mathbf{w}^*_{y,r}, \alpha y\mathbf{v}\right\rangle\right)$$

$$\geq \widetilde{\Theta}\left(\alpha^q\right)$$

and

$$
\begin{aligned}
F_{-y}\left(\mathbf{W}^*, \mathbf{x}\right) &= \sum_{r=1}^{m}\left[\sigma\left(\left\langle \mathbf{w}^*_{-y,r}, \alpha y\mathbf{v}\right\rangle\right) + \sum_{h=2}^{H}\sigma\left(\left\langle \mathbf{w}^*_{-y,r}, \boldsymbol{\zeta}_{\mathbf{x},h} - \beta_h y\mathbf{v}\right\rangle\right)\right] \\
&= \sum_{r=1}^{m}\left[\sigma\left(\left\langle \mathbf{w}^*_{-y,r}, \alpha y\mathbf{v}\right\rangle\right) + \sum_{h=2}^{H}\sigma\left(\left\langle \mathbf{w}^*_{-y,r}, -\beta_h y\mathbf{v}\right\rangle\right)\right] \\
&\leq m\left[\sigma\left(\max_{r\in[m]}\left\langle \mathbf{w}^*_{-y,r}, \alpha y\mathbf{v}\right\rangle\right) + (H-1)\sigma\left(\max_{h\in[2,H]}\beta_h\max_{r\in[m]}\left\langle \mathbf{w}^*_{-y,r}, -y\mathbf{v}\right\rangle\right)\right] \\
&\leq \widetilde{\Theta}\left(mH\left(\max_{h\in[2,H]}\beta_h\right)^q\right)
\end{aligned}
$$

then $F_y\left(\mathbf{W}^*, \mathbf{x}\right) > F_{-y}\left(\mathbf{W}^*, \mathbf{x}\right)$ holds for this kind of data. With probability $1-p$, the data will have the patch of task-specific feature and the patches of noise, like the training data for $i\in\mathcal{D}_2$, assume it belongs to the $k$-th task, then $\mathbf{x} = [y\mathbf{v}_k, \boldsymbol{\xi}_2, ..., \boldsymbol{\xi}_H]$. Similar to Lemma B.3, the support set of the random noise of this data will have no interpolation with the support sets of random noises in training samples with probability larger than $1-n^2$, which implies that $\langle \mathbf{w}_{y,r}, \boldsymbol{\zeta}_{\mathbf{x},h}\rangle = 0$. For this kind of data, we have

$$
\begin{aligned}
F_y\left(\mathbf{W}^*, \mathbf{x}\right) &= \sum_{r=1}^{m}\left[\sigma\left(\left\langle \mathbf{w}^*_{y,r}, y\mathbf{v}_k\right\rangle\right) + \sum_{h=2}^{H}\left(\left\langle \mathbf{w}^*_{y,r}, \boldsymbol{\zeta}_{\mathbf{x},h} - \beta_h\mathbf{v}\right\rangle\right)\right] \\
&= \sum_{r=1}^{m}\left[\sigma\left(\left\langle \mathbf{w}^*_{y,r}, y\mathbf{v}_k\right\rangle\right) + \sum_{h=2}^{H}\left(\left\langle \mathbf{w}^*_{y,r}, -\beta_h\mathbf{v}\right\rangle\right)\right] \\
&\geq \max_{r\in[m]}\sigma\left(\left\langle \mathbf{w}^*_{y,r}, y\mathbf{v}_k\right\rangle\right) \\
&\geq \widetilde{\Theta}\left(1\right)
\end{aligned}
$$

and

$$
\begin{aligned}
F_{-y}\left(\mathbf{W}^*, \mathbf{x}\right) &= \sum_{r=1}^{m}\left[\sigma\left(\left\langle \mathbf{w}^*_{-y,r}, y\mathbf{v}_k\right\rangle\right) + \sum_{h=2}^{H}\sigma\left(\left\langle \mathbf{w}^*_{-y,r}, \boldsymbol{\zeta}_{\mathbf{x},h} - \beta_h y\mathbf{v}\right\rangle\right)\right] \\
&= \sum_{r=1}^{m}\left[\sigma\left(\left\langle \mathbf{w}^*_{-y,r}, y\mathbf{v}_k\right\rangle\right) + \sum_{h=2}^{H}\sigma\left(\left\langle \mathbf{w}^*_{-y,r}, -\beta_h y\mathbf{v}\right\rangle\right)\right] \\
&\leq m\left[\sigma\left(\max_{r\in[m]}\left\langle \mathbf{w}^*_{-y,r}, y\mathbf{v}_k\right\rangle\right) + (H-1)\sigma\left(\max_{h\in[2,H]}\beta_h\max_{r\in[m]}\left\langle \mathbf{w}^*_{-y,r}, -y\mathbf{v}\right\rangle\right)\right] \\
&\leq mH\widetilde{\Theta}\left(\left(\max_{h\in[2,H]}\beta_h\right)^q\right)
\end{aligned}
$$

then $F_y\left(\mathbf{W}^*, \mathbf{x}\right) > F_{-y}\left(\mathbf{W}^*, \mathbf{x}\right)$ holds for this kind of data. Combining two parts, we can get that

$$\mathbb{P}_{(\mathbf{x},y)\sim\mathcal{D}}\left[F_y\left(\mathbf{W}^*, \mathbf{x}\right) \leq F_{-y}\left(\mathbf{W}^*, \mathbf{x}\right)\right] = o\left(1\right).$$

Then we complete the proof. $\qquad\square$

## H    Proof of Lemmas used in Appendix G

### H.1    Proof of Lemma G.1

*Proof of Lemma G.1.* Note that at the initialization, since $\mathbf{w}^{(0)}_{j,r} \sim \mathcal{N}\left(0, \sigma_0^2\mathbf{I}_d\right)$, then denote $\mathbf{w}^{(0)u}_{j,r}$ the $u$-th coordinate of $\mathbf{w}^{(0)}_{j,r}$, denote $\mathbf{v}^u_k$ the $u$-th coordinate of $\mathbf{v}$, then using Hoeffding's inequality,

we can get that

$$\mathbb{P}\left\{\left|\left\langle \mathbf{w}_{j,r}^{(0)}, \mathbf{v}_k \right\rangle\right| \geq a\right\} = \mathbb{P}\left\{\left|\sum_{u=1}^{d} \mathbf{w}_{j,r}^{(0)u} \mathbf{v}_k^u\right| \geq t\right\} \leq 2\exp\left(-\frac{ca^2}{\sigma_0^2 \|\mathbf{v}_k\|^2}\right) = 2\exp\left(-\frac{ca^2}{\sigma_0^2}\right)$$

so one can conclude that with probability exceeding $1 - 2n^{-2}$,

$$\left|\left\langle \mathbf{w}_{j,r}^{(0)}, \mathbf{v}_k \right\rangle\right| = \left|\sum_{u=1}^{d} \mathbf{w}_{j,r}^{(0)u} \mathbf{v}_k^u\right| \leq \widetilde{\Theta}(\sigma_0)$$

then with probability exceeding $1 - 4mn^{-2} \geq 1 - 4n^{-1}$, $\left|\left\langle \mathbf{w}_{j,r}^{(0)}, \mathbf{v}_k \right\rangle\right| \leq \widetilde{\Theta}(\sigma_0)$ holds for all $r \in [m]$ and all $j \in \{-1, 1\}$. Similarly, we can get that with probability exceeding $1 - 4n^{-1}$, $\left|\left\langle \mathbf{w}_{j,r}^{(0)}, \mathbf{v} \right\rangle\right| \leq \widetilde{\Theta}(\sigma_0)$ holds for all $r \in [m]$ and $j \in \{-1, 1\}$. Then we have

$$\left[\left\langle \mathbf{w}_{-j,r}^{(0)}, j \cdot \mathbf{v}_k \right\rangle\right]_+ \leq \left|\left\langle \mathbf{w}_{-j,r}^{(0)}, \mathbf{v}_k \right\rangle\right| \leq \widetilde{\Theta}(\sigma_0)$$

According to our calculations of update, we can get that

$$\left[\left\langle \mathbf{w}_{-j,r}^{(t+1)}, j \cdot \mathbf{v}_k \right\rangle\right]_+$$

$$= \left[(1 - \eta\lambda) \cdot \left\langle \mathbf{w}_{-j,r}^{(t)}, j \cdot \mathbf{v}_k \right\rangle + \frac{\eta}{nK} \cdot j \cdot \left(\sum_{s \in \mathcal{D}_2} y_{s,k} l_{-j,s,k}^{(t)} \sigma'\left(\left\langle \mathbf{w}_{-j,r}^{(t)}, y_{s,k} \cdot \mathbf{v}_1 \right\rangle\right)\right)\right]_+$$

$$\overset{(i)}{=} \left[(1 - \eta\lambda) \cdot \left\langle \mathbf{w}_{-j,r}^{(t)}, j \cdot \mathbf{v}_k \right\rangle - \frac{\eta}{n} \sum_{s \in \mathcal{D}_2} \left|l_{-j,s,k}^{(t)}\right| \sigma'\left(\left\langle \mathbf{w}_{-j,r}^{(t)}, y_{s,k} \cdot \mathbf{v}_k \right\rangle\right)\right]_+$$

$$\overset{(ii)}{\leq} (1 - \eta\lambda)\left[\left\langle \mathbf{w}_{-j,r}^{(t)}, j \cdot \mathbf{v}_k \right\rangle\right]_+ \leq \left[\left\langle \mathbf{w}_{-j,r}^{(t)}, j \cdot \mathbf{v}_k \right\rangle\right]_+$$

where (i) holds since $\text{sgn}\left(jy_{s,k} l_{-j,s,k}^{(t)}\right) = -1$, (ii) holds since $[\cdot]_+$ is a monotone function. By applying the above inequality recursively, we can get that

$$\left[\left\langle \mathbf{w}_{-j,r}^{(t+1)}, j \cdot \mathbf{v}_k \right\rangle\right]_+ \leq \left[\left\langle \mathbf{w}_{-j,r}^{(t)}, j \cdot \mathbf{v}_k \right\rangle\right]_+ \leq \left[\left\langle \mathbf{w}_{-j,r}^{(0)}, j \cdot \mathbf{v}_k \right\rangle\right]_+ \leq \widetilde{\Theta}(\sigma_0)$$

$\square$

## H.2 PROOF OF LEMMA G.2

*Proof of Lemma G.2.* Note that at the initialization, according to the definition, $\boldsymbol{\zeta}_{i,k,h}$ is s random vector which selects $s$ coordinates from a random vector which follows $\mathcal{N}\left(0, \sigma_p^2 \mathbf{I}_d\right)$, then denote $\mathcal{B}_{i,1,h} = \text{supp}\left(\boldsymbol{\zeta}_{i,k,h}\right)$ be the support of $\boldsymbol{\zeta}_{i,k,h}$, then $|\boldsymbol{\zeta}_{i,k,h}| = s$. Denote $\boldsymbol{\zeta}_{i,k,h}^u$ the $u$-th coordinate of $\boldsymbol{\zeta}_{i,k,h}$. Since $\mathbf{w}_{j,r}^{(0)} \sim \mathcal{N}\left(0, \sigma_0^2 \mathbf{I}_s\right)$, then denote $\mathbf{w}_{j,r}^{(0)u}$ the $u$-th coordinate of $\mathbf{w}_{j,r}^{(0)}$, and using Bernstein's inequality, we can get that

$$\mathbb{P}\left\{\left|\left\langle \mathbf{w}_{j,r}^{(0)}, \boldsymbol{\zeta}_{i,k,h} \right\rangle\right| \geq a\right\} = \mathbb{P}\left\{\left|\sum_{u=1}^{d} \mathbf{w}_{j,r}^{(0)u} \boldsymbol{\zeta}_{i,k,h}^u\right| \geq a\right\}$$

$$= \mathbb{P}\left\{\left|\sum_{u \in \mathcal{B}_{i,k,h}} \mathbf{w}_{j,r}^{(0)u} \boldsymbol{\zeta}_{i,k,h}^u\right| \geq a\right\} \leq 2\exp\left(-c\min\left(\frac{a^2}{s\sigma_p^2\sigma_0^2}, \frac{t}{\sigma_p\sigma_0}\right)\right)$$

so one can conclude that with probability exceeding $1 - 2n^{-2}$,

$$\left|\left\langle \mathbf{w}_{j,r}^{(0)}, \boldsymbol{\zeta}_{i,k,h} \right\rangle\right| = \left|\sum_{u=1}^{d} \mathbf{w}_{j,r}^{(0)u} \boldsymbol{\zeta}_{i,1,h}^u\right| \leq \widetilde{\Theta}\left(\sqrt{s}\sigma_p\sigma_0\right)$$

then with probability exceeding $1 - 4mHn^{-2} \geq 1 - 4n^{-1}$, $\left|\left\langle \mathbf{w}_{j,r}^{(0)}, \boldsymbol{\zeta}_{i,k,h} \right\rangle\right| \leq \widetilde{\Theta}\left(\sqrt{s}\sigma_p\sigma_0\right)$ holds for all $r \in [m]$, $h \in [2, H]$ and $j \in \{-1, 1\}$ Then we have

$$\left[\left\langle \mathbf{w}_{-y_{i,1},r}^{(0)}, \boldsymbol{\zeta}_{i,k,h} \right\rangle\right]_+ \leq \left|\left\langle \mathbf{w}_{-y_{i,k},r}^{(0)}, \boldsymbol{\zeta}_{i,1,h} \right\rangle\right| \leq \widetilde{\Theta}\left(\sqrt{s}\sigma_p\sigma_0\right)$$

According to our calculations of update, we can get that

$$\left[\left\langle \mathbf{w}_{-y_{i,k},r}^{(t+1)}, \boldsymbol{\zeta}_{i,k,h} \right\rangle\right]_+$$
$$= \left[(1-\eta\lambda) \cdot \left\langle \mathbf{w}_{y_{i,k},r}^{(t)}, \boldsymbol{\zeta}_{i,k,h} \right\rangle + \frac{\eta}{nK} \sum_{l=1}^{K} \sum_{g=2}^{H} \sum_{s=1}^{n} l_{-y_{i,k},s,l}^{(t)} \sigma'\left(\left\langle \mathbf{w}_{-y_{i,k},r}^{(t)}, \boldsymbol{\xi}_{s,l,g} \right\rangle\right) \langle \boldsymbol{\zeta}_{s,l,g}, \boldsymbol{\zeta}_{i,k,h} \rangle \right]_+$$

According to Lemma B.3, with probability exceeding $1 - n^{-2}$, for all $(s, l, g) \neq (i, k, h)$, $\langle \boldsymbol{\zeta}_{s,l,g}, \boldsymbol{\zeta}_{i,k,h} \rangle = 0$, then we have

$$\left[\left\langle \mathbf{w}_{-y_{i,k},r}^{(t+1)}, \boldsymbol{\zeta}_{i,k,h} \right\rangle\right]_+ = \left[(1-\eta\lambda) \cdot \left\langle \mathbf{w}_{y_{i,k},r}^{(t)}, \boldsymbol{\zeta}_{i,k,h} \right\rangle + \frac{\eta}{n} l_{-y_{i,k},i,k}^{(t)} \sigma'\left(\left\langle \mathbf{w}_{-y_{i,k},r}^{(t)}, \boldsymbol{\xi}_{i,k,h} \right\rangle\right) \langle \boldsymbol{\zeta}_{i,k,h}, \boldsymbol{\zeta}_{i,k,h} \rangle \right]_+$$

$$\overset{(i)}{=} \left[(1-\eta\lambda) \cdot \left\langle \mathbf{w}_{-y_{i,k},r}^{(t)}, \boldsymbol{\zeta}_{i,k,h} \right\rangle - \frac{\eta}{n} \left|l_{-y_{i,k},i,k}^{(t)}\right| \sigma'\left(\left\langle \mathbf{w}_{-y_{i,k},r}^{(t)}, \boldsymbol{\xi}_{i,k,h} \right\rangle\right) \|\boldsymbol{\zeta}_{i,k,h}\|^2 \right]_+$$

$$\overset{(ii)}{\leq} (1-\eta\lambda) \left[\left\langle \mathbf{w}_{-y_{i,k},r}^{(t)}, \boldsymbol{\zeta}_{i,k,h} \right\rangle\right]_+ \leq \left[\left\langle \mathbf{w}_{-y_{i,k},r}^{(t)}, \boldsymbol{\zeta}_{i,k,h} \right\rangle\right]_+$$

where (i) holds since $\mathrm{sgn}\left(l_{-y_{i,k},i,k}^{(t)}\right) = -1$, (ii) holds since $[\cdot]_+$ is a monotone functions. By applying the above inequality recursively, we can get that

$$\left[\left\langle \mathbf{w}_{-y_{i,k},r}^{(t+1)}, \boldsymbol{\zeta}_{i,k,h} \right\rangle\right]_+ \leq \left[\left\langle \mathbf{w}_{-y_{i,k},r}^{(t)}, \boldsymbol{\zeta}_{i,k,h} \right\rangle\right]_+ \leq \left[\left\langle \mathbf{w}_{-y_{i,k},r}^{(0)}, \boldsymbol{\zeta}_{i,k,h} \right\rangle\right]_+ \leq \widetilde{\Theta}\left(\sqrt{s}\sigma_p\sigma_0\right)$$

$\square$

## H.3 PROOF OF LEMMA G.3

*Proof of Lemma G.3.* From the calculations of initialization in Lemma G.1 and Lemma G.2, we have that with probability exceeding $1 - 12n^{-1}$, $\left|\left\langle \mathbf{w}_{j,r}^{(0)}, \mathbf{v}_k \right\rangle\right| \leq \widetilde{\Theta}(\sigma_0)$, $\left|\left\langle \mathbf{w}_{j,r}^{(0)}, \mathbf{v} \right\rangle\right| \leq \widetilde{\Theta}(\sigma_0)$, and $\left|\left\langle \mathbf{w}_{j,r}^{(0)}, \boldsymbol{\zeta}_{i,k,h} \right\rangle\right| \leq \widetilde{\Theta}\left(\sqrt{s}\sigma_p\sigma_0\right)$ holds simultaneously for all $j \in \{-1, 1\}$, $r \in [m]$ and $h \in [2, H]$. Therefore,

$$\Psi_{j,k}^{(0)} = \max_{r \in [m]} \left[\left\langle \mathbf{w}_{j,r}^{(0)}, j \cdot \mathbf{v}_k \right\rangle\right]_+ \leq \max_{r \in [m]} \left|\left\langle \mathbf{w}_{j,r}^{(0)}, j \cdot \mathbf{v}_k \right\rangle\right| = \max_{r \in [m]} \left|\left\langle \mathbf{w}_{j,r}^{(0)}, \mathbf{v}_k \right\rangle\right| \leq \widetilde{\Theta}(\sigma_0) \leq \rho$$

$$\Lambda_j^{(0)} = \max_{r \in [m]} \left[\left\langle \mathbf{w}_{j,r}^{(0)}, j \cdot \mathbf{v} \right\rangle\right]_+ \leq \max_{r \in [m]} \left|\left\langle \mathbf{w}_{j,r}^{(0)}, j \cdot \mathbf{v} \right\rangle\right| = \max_{r \in [m]} \left|\left\langle \mathbf{w}_{j,r}^{(0)}, \mathbf{v} \right\rangle\right| \leq \widetilde{\Theta}(\sigma_0) \leq \rho$$

$$\Phi_{j,i,h}^{(0)} \leq \Phi_{j,i}^{(0)} \leq \Phi_j^{(0)} = \max_{k \in [K]} \max_{i \in [n]} \max_{h \in [2,H]} \max_{r \in [m]} \left[\left\langle \mathbf{w}_{j,r}^{(0)}, \boldsymbol{\zeta}_{i,1,h} \right\rangle\right]_+$$

$$\leq \max_{k \in [K]} \max_{i \in [n]} \max_{h \in [2,H]} \max_{r \in [m]} \left|\left\langle \mathbf{w}_{j,r}^{(0)}, \boldsymbol{\zeta}_{i,k,h} \right\rangle\right| \leq \widetilde{\Theta}\left(\sqrt{s}\sigma_p\sigma_0\right) \leq \rho$$

holds simultaneously, which also implies that

$$\max_{k \in [K]} \max_{i \in [n]} \max_{h \in [2,H]} \max_{r \in [m]} \left[\left\langle \mathbf{w}_{j,r}^{(0)}, \boldsymbol{\xi}_{i,k,h} \right\rangle\right]_+$$
$$= \max_{k \in [K]} \max_{i \in [n]} \max_{h \in [2,H]} \max_{r \in [m]} \left[\left\langle \mathbf{w}_{j,r}^{(0)}, \boldsymbol{\zeta}_{i,k,h} - \beta_h y_{i,k} \mathbf{v} \right\rangle\right]_+$$
$$\leq \max_{k \in [K]} \max_{i \in [n]} \max_{r \in [2,H]} \max_{r \in [m]} \left[\left\langle \mathbf{w}_{j,r}^{(0)}, \boldsymbol{\zeta}_{i,k,h} \right\rangle\right]_+ + \max_{r \in [m]} \left|\left\langle \mathbf{w}_{j,r}^{(0)}, \mathbf{v} \right\rangle\right|$$
$$\leq \widetilde{\Theta}\left(\sqrt{s}\sigma_p\sigma_0\right) + \widetilde{\Theta}(\sigma_0) \leq \rho$$

so that the activation function for all of them are $\frac{z^q}{q\rho^{q-1}}$.

Besides, for any $r \in [m]$, since $\mathbf{w}_{j,r}^{(0)} \sim \mathcal{N}\left(0, \sigma_0^2 \mathbf{I}_d\right)$, we can get that $\left\langle \mathbf{w}_{j,r}^{(0)}, j \cdot \mathbf{v}_k \right\rangle \sim$

$\mathcal{N}\left(0, \sigma_0^2 \|\mathbf{v}_k\|^2\right) = \mathcal{N}\left(0, \sigma_0^2\right)$, so $\mathbb{P}\left(\left\langle \mathbf{w}_{j,r}^{(0)}, j \cdot \mathbf{v}_k \right\rangle < \frac{\sigma_0}{2}\right)$ is an absolute constant, then we can get that

$$\mathbb{P}\left(\max_{r \in [m]} \left\langle \mathbf{w}_{j,r}^{(0)}, j \cdot \mathbf{v}_k \right\rangle \geq \frac{\sigma_0}{2}\right) = 1 - \mathbb{P}\left(\max_{r \in [m]} \left\langle \mathbf{w}_{j,r}^{(0)}, j \cdot \mathbf{v}_k \right\rangle < \frac{\sigma_0}{2}\right)$$

$$= 1 - \mathbb{P}\left(\left\langle \mathbf{w}_{j,r}^{(0)}, j \cdot \mathbf{v}_k \right\rangle < \frac{\sigma_0}{2}, \forall r \in [m]\right)$$

$$= 1 - \mathbb{P}\left(\left\langle \mathbf{w}_{j,r}^{(0)}, j \cdot \mathbf{v}_k \right\rangle < \frac{\sigma_0}{2}\right)^m$$

$$\geq 1 - n^{-1}$$

so with probability exceeding $1 - n^{-1}$, $\Psi_{j,k}^{(0)} = \max_{r \in [m]} \left[\left\langle \mathbf{w}_{j,r}^{(0)}, j \cdot \mathbf{v}_k \right\rangle\right]_+ = \max_{r \in [m]} \left\langle \mathbf{w}_{j,r}^{(0)}, j \cdot \mathbf{v}_k \right\rangle \geq \frac{\sigma_0}{2}$. Similarly, we can get that with probability exceeding $1 - n^{-1}$, $\Lambda_j^{(0)} = \max_{r \in [m]} \left[\left\langle \mathbf{w}_{j,r}^{(0)}, j \cdot \mathbf{v} \right\rangle\right]_+ = \max_{r \in [m]} \left\langle \mathbf{w}_{j,r}^{(0)}, j \cdot \mathbf{v} \right\rangle \geq \frac{\sigma_0}{2}$. And conditioning on $\boldsymbol{\zeta}_{i,k,h}$, we can get that $\left\langle \mathbf{w}_{j,r}^{(0)}, \boldsymbol{\zeta}_{i,k,h} \right\rangle \sim \mathcal{N}\left(0, \sigma_0^2 \|\boldsymbol{\zeta}_{i,k,h}\|^2\right)$, so $\mathbb{P}\left(\left\langle \mathbf{w}_{j,r}^{(0)}, \boldsymbol{\zeta}_{i,k,h} \right\rangle < \frac{\sigma_0 \|\boldsymbol{\zeta}_{i,k,h}\|}{2}\right)$ is an absolute constant, then we can get that

$$\mathbb{P}\left(\max_{r \in [m]} \left\langle \mathbf{w}_{j,r}^{(0)}, \boldsymbol{\zeta}_{i,k,h} \right\rangle \geq \frac{\sigma_0 \|\boldsymbol{\zeta}_{i,k,h}\|}{2}\right) = 1 - \mathbb{P}\left(\max_{r \in [m]} \left\langle \mathbf{w}_{j,r}^{(0)}, \boldsymbol{\zeta}_{i,k,h} \right\rangle < \frac{\sigma_0 \|\boldsymbol{\zeta}_{i,k,h}\|}{2}\right)$$

$$= 1 - \mathbb{P}\left(\left\langle \mathbf{w}_{j,r}^{(0)}, \boldsymbol{\zeta}_{i,k,h} \right\rangle < \frac{\sigma_0 \|\boldsymbol{\zeta}_{i,k,h}\|}{2}, \forall r \in [m]\right)$$

$$= 1 - \mathbb{P}\left(\left\langle \mathbf{w}_{j,r}^{(0)}, \boldsymbol{\zeta}_{i,k,h} \right\rangle < \frac{\sigma_0 \|\boldsymbol{\zeta}_{i,k,h}\|}{2}\right)^m$$

$$\geq 1 - n^{-1}$$

so conditioning on $\boldsymbol{\zeta}_{i,k,h}$, with probability exceeding $1 - n^{-1}$, $\max_{r \in [m]} \left\langle \mathbf{w}_{j,r}^{(0)}, \boldsymbol{\zeta}_{i,k,h} \right\rangle \geq \frac{\sigma_0 \|\boldsymbol{\zeta}_{i,1,h}\|}{2}$. According to the definition, $\boldsymbol{\zeta}_{i,k,h}$ is s random vector which selects $s$ coordinates from a random vector which follows $\mathcal{N}\left(0, \sigma_p^2 \mathbf{I}_d\right)$, then denote $\mathcal{B}_{i,k,h} = \text{supp}\left(\boldsymbol{\zeta}_{i,k,h}\right)$ be the support of $\boldsymbol{\zeta}_{i,k,h}$, then $|\boldsymbol{\zeta}_{i,k,h}| = s$, $\|\boldsymbol{\zeta}_{i,k,h}\|^2 = \sum_{u \in \mathcal{B}_{i,k,h}} {\zeta_{i,k,h}^u}^2$ and for each $u \in \mathcal{B}_{i,k,h}$, $\zeta_{i,k,h}^u \sim \mathcal{N}\left(0, \sigma_p^2\right)$. Then we have that

$$\mathbb{E}\|\boldsymbol{\zeta}_{i,k,h}\|^2 = \mathbb{E}\sum_{u \in \mathcal{B}_{i,k,h}} {\zeta_{i,k,h}^u}^2 = \sum_{u \in \mathcal{B}_{i,k,h}} \mathbb{E}{\zeta_{i,k,h}^u}^2 = s\sigma_p^2$$

By using Bernstein's inequality, we can get that

$$\mathbb{P}\left\{\left|\|\boldsymbol{\zeta}_{i,k,h}\|^2 - \mathbb{E}\|\boldsymbol{\zeta}_{i,k,h}\|^2\right| \geq a\right\} = \mathbb{P}\left\{\left|\sum_{u \in \mathcal{B}_{i,k,h}} {\zeta_{i,k,h}^u}^2 - \mathbb{E}\sum_{u \in \mathcal{B}_{i,k,h}} {\zeta_{i,k,h}^u}^2\right| \geq a\right\}$$

$$= \mathbb{P}\left\{\left|\sum_{u \in \mathcal{B}_{i,k,h}} \zeta_{i,k,h}^2 - s\sigma_p^2\right| \geq a\right\}$$

$$\leq 2\exp\left(-c\min\left(\frac{a^2}{s\sigma_p^2}, \frac{t}{\sigma_p}\right)\right)$$

so one conclude that with probability exceeding $1 - 2n^{-2}$,

$$\left|\|\boldsymbol{\zeta}_{i,k,h}\|^2 - s\sigma_p^2\right| \leq \widetilde{\Theta}\left(\sqrt{s}\sigma_p^2\right)$$

which implies that which probability exceeding $1 - 2n^{-2}$,

$$\|\boldsymbol{\zeta}_{i,k,h}\| \geq O\left(\sqrt{s}\sigma_p\right)$$

Combining these two parts, we have that with probability exceeding $1 - n^{-1} - 2n^{-2} \geq 1 - 2n^{-1}$,

$$\max_{r \in [m]} \left\langle \mathbf{w}_{j,r}^{(0)}, \boldsymbol{\zeta}_{i,k,h} \right\rangle \geq \frac{\sigma_0 \left\| \boldsymbol{\zeta}_{i,k,h} \right\|}{2} \geq O\left( \sqrt{s} \sigma_p \sigma_0 \right)$$

So we have that with probability exceeding $1 - 2n^{-1}$,

$$\Phi_j^{(0)} \geq \Phi_{j,i}^{(0)} \geq \Phi_{j,i,h}^{(0)} = \max_{k \in [K]} \max_{k \in [K]} \max_{r \in [m]} \left[ \left\langle \mathbf{w}_{j,r}^{(0)}, \boldsymbol{\zeta}_{i,k,h} \right\rangle \right]_+$$

$$\geq \max_{r \in [m]} \left[ \left\langle \mathbf{w}_{j,r}^{(0)}, \boldsymbol{\zeta}_{i,1,h} \right\rangle \right]_+ = \max_{r \in [m]} \left\langle \mathbf{w}_{j,r}^{(0)}, \boldsymbol{\zeta}_{i,k,h} \right\rangle \geq O\left( \sqrt{s} \sigma_p \sigma_0 \right)$$

Therefore, with probability exceeding $1 - 8n^{-1}$,

$$\Psi_{j,k}^{(0)} = \max_{r \in [m]} \left[ \left\langle \mathbf{w}_{j,r}^{(0)}, j \cdot \mathbf{v}_k \right\rangle \right]_+ \geq \frac{\sigma_0}{2}$$

$$\Lambda_j^{(0)} = \max_{r \in [m]} \left[ \left\langle \mathbf{w}_{j,r}^{(0)}, j \cdot \mathbf{v} \right\rangle \right]_+ \geq \frac{\sigma_0}{2}$$

$$\Phi_j^{(0)} \geq \Phi_{j,i}^{(0)} \geq \Phi_{j,i,h}^{(0)} = \max_{k \in [K]} \max_{r \in [m]} \left[ \left\langle \mathbf{w}_{j,r}^{(0)}, \boldsymbol{\zeta}_{i,k,h} \right\rangle \right]_+ \geq O\left( \sqrt{s} \sigma_p \sigma_0 \right)$$

holds simultaneously for all $j \in \{-1, 1\}$. Combining these two parts, we have that with probability exceeding $1 - 20n^{-1}$,

$$\Psi_{j,k}^{(0)} = \widetilde{\Theta}\left( \sigma_0 \right), \Lambda_j^{(0)} = \widetilde{\Theta}\left( \sigma_0 \right), \Phi_j^{(0)} = \widetilde{\Theta}\left( \sqrt{s} \sigma_p \sigma_0 \right), \Phi_{j,i}^{(0)} = \widetilde{\Theta}\left( \sqrt{s} \sigma_p \sigma_0 \right), \Phi_{j,i,h}^{(0)} = \widetilde{\Theta}\left( \sqrt{s} \sigma_p \sigma_0 \right)$$

Then according to our definition, it can be shown that for $i \in \mathcal{D}_1$,

$$F_j \left( \mathbf{W}^{(0)}, \mathbf{x}_{i,k} \right) = \sum_{r=1}^{m} \left[ \sigma \left( \left\langle \mathbf{w}_{j,r}^{(0)}, \alpha y_{i,k} \cdot \mathbf{v} \right\rangle \right) + \sum_{h=2}^{H} \sigma \left( \left\langle \mathbf{w}_{j,r}^{(0)}, \boldsymbol{\xi}_{i,k,h} \right\rangle \right) \right]$$

$$= \sum_{r=1}^{m} \left[ \frac{\left[ \left\langle \mathbf{w}_{j,r}^{(0)}, \alpha y_{i,k} \cdot \mathbf{v} \right\rangle \right]_+^q}{q \rho^{q-1}} + \sum_{h=2}^{H} \frac{\left[ \left\langle \mathbf{w}_{j,r}^{(0)}, \boldsymbol{\xi}_{i,k,h} \right\rangle \right]_+^q}{q \rho^{q-1}} \right]$$

$$\leq m \left( \frac{\rho^q}{q \rho^{q-1}} + \sum_{h=2}^{H} \frac{\rho^q}{q \rho^{q-1}} \right) = \frac{m H \rho}{q} = o(1)$$

for all $j \in \{-1, 1\}$, and similarly for $i \in \mathcal{D}_2$,

$$F_j \left( \mathbf{W}^{(0)}, \mathbf{x}_{i,1} \right) = \sum_{r=1}^{m} \left[ \sigma \left( \left\langle \mathbf{w}_{j,r}^{(0)}, y_{i,k} \cdot \mathbf{v}_k \right\rangle \right) + \sum_{h=2}^{H} \sigma \left( \left\langle \mathbf{w}_{j,r}^{(0)}, \boldsymbol{\xi}_{i,k,h} \right\rangle \right) \right]$$

$$= \sum_{r=1}^{m} \left[ \frac{\left[ \left\langle \mathbf{w}_{j,r}^{(0)}, y_{i,k} \cdot \mathbf{v}_k \right\rangle \right]_+^q}{q \rho^{q-1}} + \sum_{h=2}^{H} \frac{\left[ \left\langle \mathbf{w}_{j,r}^{(0)}, \boldsymbol{\xi}_{i,k,h} \right\rangle \right]_+^q}{q \rho^{q-1}} \right]$$

$$\leq m \left( \frac{\rho^q}{q \rho^{q-1}} + \sum_{h=2}^{H} \frac{\rho^q}{q \rho^{q-1}} \right) = \frac{m H \rho}{q} = o(1)$$

for all $j \in \{-1, 1\}$. Then we have

$$\frac{e^{F_j \left( \mathbf{W}^{(0)}, \mathbf{x}_{i,k} \right)}}{\sum_s e^{F_s \left( \mathbf{W}^{(0)}, \mathbf{x}_{i,k} \right)}} = \Theta(1).$$

so that

$$\left| l_{j,i,k}^{(0)} \right| = \Theta(1).$$

According to the definition of $l_{j,i,k}^{(t)}$, we have $\operatorname{sgn}\left( l_{j,i,k}^{(t)} \right) = j y_{i,k}$, so $\operatorname{sgn}\left( j y_{i,k} l_{j,i,k}^{(t)} \right) = 1$. We will prove the desired argument based on the following induction hypothesis:

$$\Psi_{j,k}^{(t)} \geq \Psi_{j,k}^{(t-1)} + \frac{\eta}{2K \rho^{q-1}} \Theta\left( \left( \Psi_{j,k}^{(t-1)} \right)^{q-1} \right),$$

$$\Psi_{j,k}^{(t)} = \max_{r \in [m]} \left\langle \mathbf{w}_{j,r}^{(t)}, j \cdot \mathbf{v}_k \right\rangle,$$

$\Psi_{j,k}^{(t)}$ is monotonically non-decreasing, $\forall j \in \{-1, 1\}, k \in [K], t \leq P_j$ $\qquad$ (1)

$$\Lambda_j^{(t)} \geq \Lambda_j^{(t-1)} + \frac{\eta \alpha^q}{\rho^{q-1}} \Theta \left( \left( \Lambda_j^{(t-1)} \right)^{q-1} \right) - \widetilde{\Theta} \left( \left( \sqrt{s} \sigma_p \sigma_0 \right)^{q-1} \right),$$

$$\Lambda_j^{(t)} = \max_{r \in [m]} \left\langle \mathbf{w}_{j,r}^{(t)}, j \cdot \mathbf{v} \right\rangle,$$

$\Lambda_j^{(t)}$ is monotonically non-decreasing, $\forall j \in \{-1, 1\}, t \leq Q_j$ $\qquad$ (2)

$$\Phi_{y_{i,k},i}^{(t)} \leq \Phi_{y_{i,k},i}^{(t-1)} + \frac{\eta}{K\rho^{q-1}} \Theta \left( \frac{s\sigma_p^2}{n} \right) \Theta \left( \left( \Phi_{y_{i,k},i}^{(t-1)} + \widetilde{\Theta} \left( \sigma_0^{\frac{2}{3}} \right) \right)^{q-1} \right), \forall i \in \mathcal{D}_2, t \leq P_0$$
$\qquad$ (3)

$$\Phi_{y_{i,k},i}^{(t)} \leq \Phi_{y_{i,k},i}^{(t-1)} + \frac{\eta}{K\rho^{q-1}} \Theta \left( \frac{s\sigma_p^2}{n} \right) \Theta \left( \left( \Phi_{y_{i,k},i}^{(t-1)} + \widetilde{\Theta} \left( \sigma_0^{\frac{2}{3}} \right) \right)^{q-1} \right), \forall i \in \mathcal{D}_1, t \leq Q_0$$
$\qquad$ (4)

$$\Psi_{j,k}^{(t)} \leq \widetilde{\Theta}(1), \forall j \in \{-1, 1\}, k \in [K], t \leq P_{j,k},$$

$$\Psi_{j,k}^{(t)} = \widetilde{\Theta}(1), \Psi_{j,k}^{(t)} = \max_{r \in [m]} \left\langle \mathbf{w}_{j,r}^{(t)}, j \cdot \mathbf{v}_k \right\rangle, \forall j \in \{-1, 1\}, k \in [K], P_{j,k} \leq t \leq T \quad (5)$$

$$\Lambda_j^{(t)} \leq \widetilde{\Theta}(1), \forall j \in \{\pm 1\}, t \leq Q_j,$$

$$\Lambda_j^{(t)} = \widetilde{\Theta}(1), \Lambda_j^{(t)} = \max_{r \in [m]} \left\langle \mathbf{w}_{j,r}^{(t)}, j \cdot \mathbf{v} \right\rangle, \forall j \in \{-1, 1\}, Q_j \leq t \leq T \qquad (6)$$

$$\Phi_{j,i}^{(t)} \leq \widetilde{\Theta} \left( \sqrt{s} \sigma_p \sigma_0 \right), \forall j \in \{-1, 1\}, \forall i \in \mathcal{D}_2, t \leq T \qquad (7)$$

$$\Phi_{j,i}^{(t)} \leq \widetilde{\Theta} \left( \sqrt{s} \sigma_p \sigma_0 \right), \forall j \in \{-1, 1\}, \forall i \in \mathcal{D}_2, t \leq T \qquad (8)$$

$$\left\langle \mathbf{w}_{-j,r}^{(t)}, j \cdot \mathbf{v} \right\rangle \leq \widetilde{\Theta} \left( \sigma_0^{\frac{1}{3}} \right), \forall j \in \{-1, 1\}, t \leq T \qquad (9)$$

and for $t = -1$, we set $\Psi_{j,k}^{(-1)} = 0$ for all $j \in \{-1, 1\}$ and $k \in [K]$, $\Lambda_j^{(-1)} = 0$ for all $j \in \{-1, 1\}$, and $\Phi_{j,i}^{(-1)} = 1$ for all $j \in \{-1, 1\}$ and $i \in [n]$. Now we consider the situation at $t = 0$.

(i) In terms of Hypothesis 1 and 5, since the distribution of $\mathbf{w}_{j,r}^{(0)}$ is symmetric, then for each $r \in [m]$ and $j \in \{-1, 1\}$, with probability $\frac{1}{2}$, $\left\langle \mathbf{w}_{j,r}^{(0)}, j \cdot \mathbf{v}_k \right\rangle < 0$. So with probability $\frac{1}{2^m}$, $\left\langle \mathbf{w}_{j,r}^{(0)}, j \cdot \mathbf{v}_k \right\rangle < 0$ holds for all $r \in [m]$. Denote $r^* = \arg\max_{r \in [m]} \left\langle \mathbf{w}_{j,r}^{(0)}, j \cdot \mathbf{v}_k \right\rangle$, then

$$\mathbb{P} \left\{ \left\langle \mathbf{w}_{j,r^*}^{(0)}, j \cdot \mathbf{v}_k \right\rangle \geq 0, \forall j \in \{-1, 1\} \right\}$$

$$\geq 1 - \mathbb{P} \left\{ \left\langle \mathbf{w}_{1,r^*}^{(0)}, 1 \cdot \mathbf{v}_k \right\rangle < 0 \right\} - \mathbb{P} \left\{ \left\langle \mathbf{w}_{-1,r^*}^{(0)}, -1 \cdot \mathbf{v}_k \right\rangle < 0 \right\}$$

$$= 1 - \mathbb{P} \left\{ \left\langle \mathbf{w}_{1,r}^{(0)}, 1 \cdot \mathbf{v}_k \right\rangle < 0, \forall r \in [m] \right\} - \mathbb{P} \left\{ \left\langle \mathbf{w}_{-1,r}^{(0)}, -1 \cdot \mathbf{v}_k \right\rangle < 0, \forall r \in [m] \right\}$$

$$= 1 - \frac{1}{2^{m-1}} \geq 1 - n^{-1}$$

so with probability exceeding $1 - n^{-1}$, $\Psi_{j,k}^{(0)} = \left\langle \mathbf{w}_{j,r^*}^{(0)}, j \cdot \mathbf{v}_k \right\rangle \geq 0$ holds for all $j \in \{-1, 1\}$ and $j \in [K]$. Since $\Psi_{j,k}^{(-1)} = 0$, then the inequality

$$\Psi_{j,k}^{(0)} \geq \Psi_{j,k}^{(-1)} + \frac{\eta}{2\rho^{q-1}} \Theta \left( \left( \Psi_{j,k}^{(-1)} \right)^{q-1} \right)$$

holds, so we verify Hypothesis 1 at $t = 0$. According to previous calculations, we have that $\Psi_{j,k}^{(0)} = \widetilde{\Theta}(\sigma_0) \leq \widetilde{\Theta}(1)$ for all $j \in \{-1, 1\}$, so we verify Hypothesis 5 at $t = 0$.

(ii) In terms of Hypothesis 2 and 6, since the distribution of $\mathbf{w}_{j,r}^{(0)}$ is symmetric, then for each $r \in [m]$

and $j \in \{-1, 1\}$, with probability $\frac{1}{2}$, $\left\langle \mathbf{w}_{j,r}^{(0)}, j \cdot \mathbf{v} \right\rangle < 0$. So with probability $\frac{1}{2^m}$, $\left\langle \mathbf{w}_{j,r}^{(0)}, j \cdot \mathbf{v} \right\rangle < 0$ holds for all $r \in [m]$. Denote $r^* = \arg\max_{r \in [m]} \left\langle \mathbf{w}_{j,r}^{(0)}, j \cdot \mathbf{v} \right\rangle$, then

$$
\begin{aligned}
\mathbb{P} &\left\{ \left\langle \mathbf{w}_{j,r^*}^{(0)}, j \cdot \mathbf{v} \right\rangle \geq 0, \forall j \in \{-1, 1\} \right\} \\
&\geq 1 - \mathbb{P} \left\{ \left\langle \mathbf{w}_{1,r^*}^{(0)}, 1 \cdot \mathbf{v} \right\rangle < 0 \right\} - \mathbb{P} \left\{ \left\langle \mathbf{w}_{-1,r^*}^{(0)}, -1 \cdot \mathbf{v} \right\rangle < 0 \right\} \\
&= 1 - \mathbb{P} \left\{ \left\langle \mathbf{w}_{1,r}^{(0)}, 1 \cdot \mathbf{v} \right\rangle < 0, \forall r \in [m] \right\} - \mathbb{P} \left\{ \left\langle \mathbf{w}_{-1,r}^{(0)}, -1 \cdot \mathbf{v} \right\rangle < 0, \forall r \in [m] \right\} \\
&= 1 - \frac{1}{2^{m-1}} \geq 1 - n^{-1}
\end{aligned}
$$

so with probability exceeding $1 - n^{-1}$, $\Lambda_j^{(0)} = \left\langle \mathbf{w}_{j,r^*}^{(0)}, j \cdot \mathbf{v} \right\rangle \geq 0$ holds for both $j \in \{-1, 1\}$. Since $\Lambda_j^{(-1)} = 0$, then the inequality

$$
\Lambda_j^{(t)} \geq \Lambda_j^{(t-1)} + \frac{\eta \alpha^q}{2\rho^{q-1}} \Theta \left( \left( \Lambda_j^{(t-1)} \right)^{q-1} \right) - \widetilde{\Theta} \left( \left( \sqrt{s} \sigma_p \sigma_0 \right)^{q-1} \right)
$$

holds, so we verify Hypothesis 2 at $t = 0$. According to previous calculations, we have that $\Lambda_j^{(0)} = \widetilde{\Theta}(\sigma_0) \leq \widetilde{\Theta}(1)$ for all $j \in \{-1, 1\}$, so we verify Hypothesis 6 at $t = 0$.

(iii) In terms of Hypothesis 3, 4, 7, and 8, from the calculations of initialization in Lemma G.2, we have that for any $i \in [n]$, $\Phi_{y_{i,k},i}^{(0)} \leq \Phi_{y_{i,k}}^{(0)} \leq \widetilde{\Theta}(\sqrt{s}\sigma_p \sigma_0)$. Since $\Phi_{y_{i,k},i}^{(-1)} = 1$, then

$$
\Phi_{y_{i,k},i}^{(0)} \leq \widetilde{\Theta}\left(\sqrt{s}\sigma_p\sigma_0\right) \leq 1 \leq \Phi_{y_{i,k},i}^{(-1)} + \frac{\eta}{\rho^{q-1}} \widetilde{\Theta}\left(\frac{d\sigma_p^2}{n}\right) \Theta\left( \left( \Phi_{y_{i,k},i}^{(-1)} + \widetilde{\Theta}\left(\sigma_0^{\frac{2}{3}}\right) \right)^{q-1} \right)
$$

so we verify the hypothesis at $t = 0$.

(iv) In terms of Hypothesis 9, from the calculations of initialization in Lemma G.1, we have that for any $j \in \{-1, 1\}$, $\left\langle \mathbf{w}_{-j,r}^{(0)}, j \cdot \mathbf{v} \right\rangle \leq \left| \left\langle \mathbf{w}_{-j,r}^{(0)}, \mathbf{v} \right\rangle \right| \leq \widetilde{\Theta}(\sigma_0) \leq \widetilde{\Theta}\left(\sigma_0^{\frac{1}{3}}\right)$, so we verify the hypothesis at $t = 0$.

For the induction part, we will use similar steps as in Lemma D.3, so we will omit the proof of this part here. $\square$

