# OpenReview forum: "On the Power of Multitask Representation Learning with Gradient Descent"
_ICLR.cc/2024/Conference — Submitted to ICLR 2024_

### Official Review · Reviewer_vmJ4 · 2023-10-31

**Soundness:** 2 fair
**Presentation:** 2 fair
**Contribution:** 2 fair
**Rating:** 3
**Confidence:** 3

**Summary:**

This work studies the feature learning of a kind of mutitask representation learning where multiple classification tasks share the same output space (same y space). Through some theoretical and experimental studies, this paper concludes that this kind of multipletask learning helps learn shared features and thus helps generalization.

**Strengths:**

- The idea of investigating shared feature and task-specified feature in neural network representation learning is interesting.
- The author provide a rich liturature review of Multi-task Representation Learning.

**Weaknesses:**

- The mutipletask setting in this work, various single-tasks contain different noise in input X but share the same output categories Y, is almost exactly data-augmentation. The shared feature and task-specified feature conclusion in this paper driven by mutipletask is also close to the invariant feature and spurious feature view in data-augmentation.  From data-augmentation's view, one can easily get these conclusions.

- In the theory part, the paper mentioned that each single-task contain $n$ examples, but didn't give the number of examples in mutipletask. In the experiment settings, the paper considers the **union** of single-tasks as a mutipletask: *"Therefore, we consider learning on each specific corruption as the single-task setting and learning on a union of corruptions as the multi-task setting. "*.  That means mutipletask in this work contains much more examples than each single-task.  From the large training set's view, one can again easily get this paper's conclusion.

**Questions:**

- Does the mutipletask contain the union of single-tasks? In other words, does the mutipletask contain more training examples?

---

> ### Author Response · Authors · 2023-11-23
> **Response to Reviewer vmJ4 [Part I]**
>
> Thank you for your feedback!
>
> **Question 1**:
>
> The multiple task setting in this work, various single-tasks containing different noise in input X but sharing the same output categories Y, is almost exactly data-augmentation. The shared feature and task-specified feature conclusion in this paper driven by mutipletask is also close to the invariant feature and spurious feature view in data-augmentation. From data-augmentation's view, one can easily get these conclusions.
>
> **Answer 1**:
>
> Our setting is quite different from data augmentation. Data augmentation usually applies random and mild transformations to the data during training, such as cropping, flipping, rotation, and adding mild noise. The goal of data augmentation is to improve the model's generalization within the same task. An important property of data augmentation is that the actual feature of the data remains very strong as compared to introduced noises. To our understanding, our problem setup is very different from data augmentation in the following ways.
>
> First, according to our Definition 3.1, our multi-task setting involves task-shared feature, task-specific features and noises. Moreover, our task-shared feature is weakened by $\alpha<1$, so the strength of task-shared feature is weaker than the strength of task-specific features. This can also be seen from the fact that in single-task learning, the task-shared feature even has less impact than noises, such that the neural network prefers to memorize noises instead of learning the task-shared feature. Therefore, our model differs greatly from the property of data augmentation.
>
> Second, the CIFAR10-C and MNIST-C datasets used in our paper are datasets commonly used by other works on multi-task learning [1,2]. From the experiments, we can also verify that the task-shared feature in our setting is much weaker than the strength of features in data augmentation. For example, the model trained on Bright/Contrast corruption only has a 17.00/13.35 test accuracy on Impulse corruption, which is only slightly higher than random guess, so the task-shared feature among them does not have a very large strength.
>
> Therefore, we believe our multi-task learning setting is very different from data augmentation, so our results are also different and can not be deduced from results in data augmentation.

---

> ### Author Response · Authors · 2023-11-23
> **Response to Reviewer vmJ4 [Part II]**
>
> **Question 2**:
>
> Does the mutipletask contain the union of single-tasks? In other words, does the mutipletask contain more training examples?
>
> **Answer 2**:
>
> Yes, according to our Problem Setup in Section 3, there are $n$ training samples for each of the $K$ tasks, so in single-task learning, there are $n$ training samples, and in multi-task learning, there are $n*K$ training samples.
>
> **Question 3**:
>
> That means multi-task learning in this work contains much more examples than each single-task learning. From the large training set's view, one can again easily get this paper's conclusion.
>
> **Answer 3**:
>
> The observation that multi-task learning involves more training samples than single-task learning does not necessarily translate to improved test errors. Typically, a larger sample size only straightforwardly results in better test errors when the samples originate from the same task/distribution.
>
> According to our Definition 3.1 and Definition 3.2, the task-shared feature, the task-specific features and random noises are orthogonal to each other. Then for those data points whose feature patches are equal to the task-specific features, the learning process of each task-specific feature is only influenced by data within the corresponding task according to gradient calculations in Lemma B.2, so the learning results of each task-specific feature does not benefit from a larger training set. Therefore, the only possible data points that can benefit from more training data points are the data points whose feature patches are equal to the task-shared feature. But the problem is that the task-shared feature also appears in feature noises of every data point, so the learning process of the task-shared feature will also be influenced by other data points whose feature patches are equal to the task-specific features as shown in the gradient calculations in Lemma B.2. This is obviously different from the setting of simply adding more data points from the same distribution.
>
> Therefore, our conclusion does not stem from a straightforward outcome of having a larger training set.

---

### Official Review · Reviewer_yKNZ · 2023-11-01

**Soundness:** 3 good
**Presentation:** 3 good
**Contribution:** 3 good
**Rating:** 6
**Confidence:** 3

**Summary:**

The submission uses a simple mathematical model of a convolutional neural network and data that obeys certain properties analogous to single-task and specific multi-task settings, in order to develop an analysis of learning dynamics and theoretical bounds that illustrate the benefits of multi-task learning under such modelling assumptions. The mathematical tools are based on tensor-power methods developed in the literature. Some experiments are performed to confirm the benefits of multi-task learning.

**Strengths:**

* Originality: To my knowledge, the analysis carried out in this submission is quite original, both in terms of the choice of mathematical tools utilized, as well as the analysis about learning dynamics in single and multi-task settings.

* Quality: The submitted work seems to generally be of high quality, in terms of the theoretical contributions.

* Clarity: With the caveat that I’m no mathematician and didn’t verify any of the proofs, I found the main body of the paper overall very readable. I particularly appreciated the intuitive descriptions along with the formal statements.

* Significance: The model and mathematical tooling adopted in the paper seems well-suited to analyzing neural networks, and the demonstration for how it can be used to explain the empirical observations about the usefulness of multi-task learning is potentially inspirational for similar analyses applied elsewhere.

**Weaknesses:**

* The experiments could be in greater correspondence with the developed theory. Currently, the empirical results only demonstrate that multi-task learning is an improvement, something that is fairly well-acknowledged and doesn’t really need further demonstration. What would be far more interesting is if the learning dynamics posited in the submission were shown to be reflected in real-life learning dynamics (i..e the stage-wise learning processes described in Section 5).

* The submission seems to model specifically one variant of a multi-task learning setup, i.e. different data sources for a common task. There are other variants that might be more commonly associated with multi-task learning, in particular that of the same data source/domain but different objectives. Perhaps the setup that has been tackled in the manuscript is better thought of as a model for analyzing the benefits of data augmentation? Or perhaps it is possible to draw stronger links between the data and task model discussed with other variants of multi-task learning setups.

**Questions:**

1. If I followed right, the theory only focuses on binary classification problems. Do the authors anticipate any difficulties with extensions into multi-label categorization or regression problems?

2. What is c in Other Requirements on Page 4? An arbitrary constant?

3. What is p on the first line on page 5? The probability of the feature patch being the task-shared feature?  How would one intuitively interpret this bound, in terms of how p and n interact to raise or lower the error rate bound?

4. Just to confirm, In Theorem 4.2, the test error is equal to poly(1/n), or should it be poly(1/nK)?

---

> ### Author Response · Authors · 2023-11-23
> **Response to Reviewer yKNZ [Part I]**
>
> Thank you for your positive feedback!
>
> **Question 1**:
>
> The experiments could be in greater correspondence with the developed theory. What would be far more interesting is if the learning dynamics posited in the submission were shown to be reflected in real-life learning dynamics (i.e., the stage-wise learning processes described in Section 5).
>
> **Answer 1**:
>
> We have also done further experiments regarding the performance on both synthetic data and CIFAR10-C in Appendix A.1 and Appendix A.2 to show how they back up our theoretical results, especially the different capabilities of learning the task-shared feature between single-task learning and multi-task learning, as well as different performances between two learning stages on both task-shared feature and task-specific feature.
>
> For synthetic data, we have plotted the inner products between the model weight vectors and both the key feature vectors and noises identified in our study. We consider Setting 1 for these experiments, and take the average of all noise vectors in training data to show the overall trend of learning noise. In the case of single-task learning, our observations confirm that the model primarily learns the task-specific feature, while the learning of noise surpasses the learning of the task-shared feature. Conversely, multi-task learning demonstrates successful learning of the task-shared feature, surpassing the learning of noise.
>
> Furthermore, the learning trends observed in our synthetic experiments are consistent with our two-stage training analysis. In single-task learning, the learning processes of the task-specific feature and noise are going through the pattern learning stage before around 50th and 350th epoch respectively, in which the inner products are growing rapidly; and they both keep steady and enter the regularization stage after the turning points. The inner product with the task-shared feature remains near initialization. In multi-task learning, the inner products with all features are increasing fast in the pattern learning stage before around 250th epoch, until each of them reaches a certain level and keeps the same magnitude in the regularization stage. The learning level of noise stays at a low level.
>
> Since in the CIFAR10-C dataset, one cannot explicitly separate task-shared features and task-specific features, we try to use the test errors under various training and testing situations to reflect the learning process and the learning results of task-shared feature and task-specific features.
>
> In Figure 5, we present a comparison of test accuracy for models trained on a single task (brightness) and multiple tasks (encompassing the 9 datasets discussed previously). The test data includes impulse noise, an element not present in the training data for either the single-task or multi-task models. The performance on impulse noise serves as an indicator of the model's capability in learning the task-shared feature. Additionally, we examine test data related to brightness and corruption encountered in both single-task and multi-task learning. This assessment is used to indicate the model's capability in learning task-specific features.
>
> As shown in Figure 5, multi-task learning successfully learns the task-shared feature while single-task learning fails to do so. Moreover, the trend in the figure aligns with our two-stage analysis. For single-task learning, before around 80th epoch, the task-specific feature is learned relatively rapidly, which corresponds to our pattern learning stage; after around 80th epoch, it basically remains at the same level, which corresponds to our regularization stage. The learning level of the task-shared feature stays low. For multi-task learning, before around 50th epoch, both the task-specific features and the task-shared feature are learned rapidly by the neural network, which matches our pattern learning stage; after around 50th epoch, they both keep at a constant level, which matches our regularization stage.

---

> ### Author Response · Authors · 2023-11-23
> **Response to Reviewer yKNZ [Part II]**
>
> **Question 2**:
>
> The submission seems to model specifically one variant of a multi-task learning setup, i.e. different data sources for a common task. There are other variants that might be more commonly associated with multi-task learning, in particular that of the same data source/domain but different objectives. Perhaps the setup that has been tackled in the manuscript is better thought of as a model for analyzing the benefits of data augmentation? Or perhaps it is possible to draw stronger links between the data and task model discussed with other variants of multi-task learning setups.
>
> **Answer 2**:
>
> Our setup for multi-task learning is not on different data sources for a common task. As in our Definition 3.1, we have $K$ different classification tasks, and for each task there are $n$ training samples. The task-specific features for different tasks and the task-shared feature are orthogonal to each other, which implies that these classification tasks are different from each other.
>
> Besides, our setting is also quite different from data augmentation. Data augmentation usually applies random and mild transformations to the data during training, such as cropping, flipping, rotation, and adding mild noise. The goal of data augmentation is to improve the model's generalization within the same task. An important property of data augmentation is that the actual feature of the data remains very strong as compared to introduced noises. To our understanding, our problem setup is very different from data augmentation in the following ways.
>
> First, according to our Definition 3.1, our multi-task setting involves task-shared feature, task-specific features and noises. Moreover, our task-shared feature is weakened by $\alpha<1$, so the strength of task-shared feature is weaker than the strength of task-specific features. This can also be seen from the fact that in single-task learning, the task-shared feature even has less impact than noises, such that the neural network prefers to memorize noises instead of learning the task-shared feature. Therefore, our model differs greatly from the property of data augmentation.
>
> Second, the CIFAR10-C and MNIST-C datasets used in our paper are datasets commonly used by other works on multi-task learning [1,2]. From the experiments, we can also verify that the task-shared feature in our setting is much weaker than the strength of features in data augmentation. For example, the model trained on Bright/Contrast corruption only has a 17.00/13.35 test accuracy on Impulse corruption, which is only slightly higher than random guess, so the task-shared feature among them does not have a very large strength.
>
> Therefore, we believe our multi-task learning setting is very different from data augmentation, so our results are also different and can not be deduced from results in data augmentation.
>
> **Question 3**:
>
> The theory only focuses on binary classification problems. Do the authors anticipate any difficulties with extensions into multi-label categorization or regression problems?
>
> **Answer 3**:
>
> Yes, in our current paper, we are only focusing on binary classification problems. The number of classes does not play a significant role in our analysis, so we believe we can directly extend into multi-label categorization with cross entropy loss. We can use the data model (Definition 3.1) in [3], and the expected results are a test error lower bounded by a constant in single-task learning and a test error close to $0$ in multi-task learning.
>
> Regarding the extension to regression problems, we need to consider square loss and we are not sure if the extension is straightforward.

---

> ### Author Response · Authors · 2023-11-23
> **Response to Reviewer yKNZ [Part III]**
>
> **Question 4**:
>
> What is $c$ in Other Requirements on Page 4? An arbitrary constant?
>
> **Answer 4**:
>
> Yes, $c$ in the parameter requirements is an arbitrary constant.
>
> **Question 5**:
>
> What is $p$ on the first line on page 5? The probability of the feature patch being the task-shared feature?
>
> **Answer 5**:
>
> $p$ is defined in our Definition 3.1, as the probability of the feature patch being the task-shared feature.
>
> **Question 6**:
>
> How would one intuitively interpret this bound, in terms of how $p$ and $n$ interact to raise or lower the error rate bound?
>
> **Answer 6**:
>
> We can intuitively interpret this bound in terms of $p$ and $n$ in the following way:
>
> 1. First, we will explain intuitively on the impact of $p$. In the Theorem 4.1, which is under the single-task setting, the task-shared feature is weakened by $\alpha$. This will result in that for some data points with the feature patch equal to the task-shared feature, noises will have a stronger impact than the task-shared feature, causing the neural network fail to learn the task-shared feature, which means that these data points cannot be correctly classified. Therefore, at least some of the misclassified data points come from a relatively fixed proportion of the data where the feature patch is equal to the task-shared feature, noticing that $p$ is approximately the proportion of data where the feature patch is equal to the task-shared feature, so the lower bound will increase when $p$ increases.
>
> 2. Second, according to our proof, the quantity that is directly related to $n$ in the test error, $\text{poly}(n^{-1})$, comes from the fact that by concentration inequalities, with probability $1-\text{poly}(n^{-1})$, we can get a good enough initialization on the weight vectors in the neural network and the noise patch. Throughout the entire proof, we only consider those situations with this good initialization. Conditioned on good initialization, the probability of making an incorrect classification is lower bounded by $p/4$. Since p is a constant, so the lower bound of misclassification without conditioning should be $p/4(1-\text{poly}(n^{-1}))=p/4-\text{poly}(n^{-1})$.
>
> **Question 7**:
>
> In Theorem 4.2, the test error is equal to $\text{poly}(1/n)$, or should it be $\text{poly}(1/nK)$?
>
> **Answer 7**:
>
> According to our assumptions, $K$ is lower bounded by a polynomial of $n$, which implies that the polynomial of $1/(nK)$ should be upper bounded by another polynomial of $1/n$. Since it is an upper bound on the test error, we choose to write as $\text{poly}(1/n)$.
>
> [1] Dan Hendrycks and Thomas Dietterich. Benchmarking neural network robustness to common corruptions and perturbations. In International Conference on Learning Representations, 2018.
>
> [2] Norman Mu and Justin Gilmer. Mnist-c: A robustness benchmark for computer vision. arXiv preprint arXiv:1906.02337, 2019.
>
> [3] Zeyuan Allen-Zhu and Yuanzhi Li. Towards understanding ensemble, knowledge distillation and self-distillation in deep learning. arXiv preprint arXiv:2012.09816, 2020.

---

### Official Review · Reviewer_rtpX · 2023-11-01

**Soundness:** 3 good
**Presentation:** 3 good
**Contribution:** 2 fair
**Rating:** 6
**Confidence:** 2

**Summary:**

The paper derives generalization bounds for multi-task learning and single-task learning, under a data generating process with task-specific and task-shared features. Under this assumption, both single and multi-task learning achieve zero training error but only multi-task learning achieve asymptotically 0 generalization error. The authors derive these results for a 2-layer neural network trained with gradient descent. Finally, the authors present a few small experiments on multi-task learning on CIFAR-10 and a synthetic dataset.

**Strengths:**

**Novelty and Quality.** The authors consider a setting, that has not been previously tackled theoretically.  Most prior works have dealt with the existence of a single shared representation This work instead assumes that some input features are shared across all the tasks and task-specific features have some noise. Previous works primarily conduct a learning theoretic analysis for multi-task learning while this work conducts an analysis using the gradient descent update.

**Presentation.** The authors present thorough analysis of two-layer neural networks with this specific data generating process. The manuscript details fairly involved calculations to derive these results. The authors also do a great job at presenting some of these results; the notation in section 5 is fairly involved and was hard to parse but I appreciate the effort from the authors.

**Weaknesses:**

**Why is the noise present only in the task-specific features?**
From my understanding, the noise is only present for the task-specific features. This seems to be the reason for the lower-bound on single-task learning. But why is this a reasonable assumption reasonable? Why is the noise not present in the shared features? For example, one can imagine that "dogs" may have task-specific features that may not be important for classifying "birds" and I believe this scenario is far more common in multi-task learning.

**How does this work differentiate itself from work from Baxter et al., Maurer et al., Tripuraneni et al.**
Prior works have deal with the scenario where there exists a single representation shared across all the tasks. The assumption in this work is more restrictive:  it assumes the existence of shared input features. Is it possible to derive similar conclusions using results from prior work? In particular, prior work also predicts that the error decreases asymptotically as $\sqrt{n^{-1}}$ where $n$ is the number of tasks.

**Do the synthetic and real-world experiments show two stages?**
Are the predictions made by theory representative of what occurs in practice? Do we see different observations if there is no weight decay for the multi-task models?


**It isn't clear how the CIFAR10-C experiments complement the theory.** The results of the experiment are unsurprising: multi-task learning works on a dataset like CIFAR10-C. Is the data generating process, accurate for even simple datasets like CIFAR10-C? The experiments do not seem to complement any of the theoretical findings in the paper (two stages of learning process, rate of convergence, lower bound of single-task learning, etc..).

**Questions:**

1. **Importance of these techniques towards understanding multi-task learning.**  In practice, neural networks are deep and over-parameterized. Can these proof techniques be scaled beyond two layers to understand multi-task learning? Currently, they do not seem to provide a complete theory for even a highly idealized two layer neural network.

2. **Is the data generating process reflective of real world datasets?** Multi-task learning, works for a broad range of settings outside of having a shared input patch across all the tasks. My concern is that this assumption is too narrow and maybe weaker than prior works that have assumed shared representations.

3. **Verifying two stages of learning using experiments**. It would be helpful to verify that two stages of learning even on synthetic data. What happens if the regularization is set to 0?

**Edit:**
Thank you for addressing many of my concerns and for clarifying some misconceptions. I have my raised my score to a 6. I still have concerns regarding assumptions on the data generating process---I think they do not capture many multi-task learning problems that we typically consider in benchmarks. Nevertheless, I think this is interesting work and has new insights.

---

> ### Author Response · Authors · 2023-11-23
> **Response to Reviewer rtpX [Part I]**
>
> Thank you for your helpful feedback!
>
> **Question 1:**
>
> Why is the noise present only in the task-specific features? From my understanding, the noise is only present for the task-specific features. This seems to be the reason for the lower bound on single-task learning. But why is this a reasonable assumption? Why is the noise not present in the shared features?
>
> **Answer 1:**
>
> We believe there might be a misunderstanding in our data model. According to our Definition 3.1, for each data point, there is one feature patch and $H-1$ noise patches. With probability $p$, the feature patch is taken as the task-shared feature $\alpha y\cdot \boldsymbol{v}$ and with probability $1-p$, the feature patch is taken as the task-specific feature $y\cdot \boldsymbol{v}_{k}$. So each data point will have either the task-shared feature or the task-specific feature, but there is always at least ($H-1\geq 1$) one noise patch in each data point.
>
> **Question 2:**
>
> How does this work differentiate itself from works from Baxter et al., Maurer et al., Tripuraneni et al.? Is it possible to derive similar conclusions using results from prior work? Prior works have dealt with the scenario where there exists a single representation shared across all the tasks. The assumption in this work is more restrictive: it assumes the existence of shared input features.
>
> **Answer 2:**
>
> We believe that the problem in our paper is different from those considered in previous works. In those papers on multi-task learning mentioned above, including [1][2][3] in our references, they only work on bounding the excess risk of some empirical risk minimizer (ERM) on multi-task learning, while we try to depict the entire learning process of single-task learning and multi-task learning optimized by gradient descent, and the test error bounds are only parts of our conclusions. So our results are algorithm-dependent and can not be derived from prior work. Since we need to go over the entire optimization dynamics, it is reasonable to make slightly stronger assumptions than previous works. We believe it is possible to relax our assumptions such that there exists a single representation shared across all the tasks.

---

> ### Author Response · Authors · 2023-11-23
> **Response to Reviewer rtpX [Part II]**
>
> **Question 3:**
>
> Do the synthetic and real-world experiments show two stages? Are the predictions made by theory representative of what occurs in practice? How do the CIFAR10-C experiments complement the theory?
>
> **Answer 3:**
>
> Our experiments in Table 3 demonstrate results in many folds. First, it indicates the existence of shared features across the tasks we defined, as models trained on any single task invariably outperform random guessing (10%) on other tasks. Second, it shows that single-task learning can not learn shared features well, because many single-task learning results in poor accuracy of less than 20% (e.g., training on brightness and testing on impulse noise only gets 17%). Lastly, the results explain why multi-task learning can outperform single-task learning in Table 2, by showing that multi-task learning on the 9 tasks resulted in better results on the other 10 unseen tasks, which is attributed to better learning of the shared feature.
>
> We have also done further experiments regarding the performance on both synthetic data and CIFAR10-C in Appendix A.1 and Appendix A.2 to show how they backup our theoretical results, especially the different capabilities of learning the task-shared feature between single-task learning and multi-task learning, as well as different performances between two learning stages on both task-shared feature and task-specific feature.
>
> For synthetic data, we have plotted the inner products between the model weight vectors and both the key feature vectors and noises identified in our study. We consider Setting 1 for these experiments, and take the average of all noise vectors in training data to show the overall trend of learning noise. In the case of single-task learning, our observations confirm that the model primarily learns the task-specific feature, while the learning of noise surpasses the learning of the task-shared feature. Conversely, multi-task learning demonstrates successful learning of the task-shared feature, surpassing the learning of noise.
>
> Furthermore, the learning trends observed in our synthetic experiments are consistent with our two-stage training analysis. In single-task learning, the learning processes of the task-specific feature and noise are going through the pattern learning stage before around 50th and 350th epoch respectively, in which the inner products are growing rapidly; and they both keep steady and enter the regularization stage after the turning points. The inner product with the task-shared feature remains near initialization. In multi-task learning, the inner products with all features are increasing fast in the pattern learning stage before around 250th epoch, until each of them reach a certain level and keep the same magnitude in the regularization stage. The learning level of noise stays at a low level.
>
> Since in the CIFAR10-C dataset, one cannot explicitly separate task-shared feature and task-specific feature, we try to use the test errors under various training and testing situations to reflect the learning process and the learning results of task-shared feature and task-specific features.
>
> In Figure 5, we present a comparison of test accuracy for models trained on a single task (brightness) and multiple tasks (encompassing the 9 datasets discussed previously). The test data includes impulse noise, an element not present in the training data for either the single-task or multi-task models. The performance on impulse noise serves as an indicator of the model's capability in learning task-shared feature. Additionally, we examine test data related to brightness and corruption encountered in both single-task and multi-task learning. This assessment is used to indicate the model's capability in learning task-specific features.
>
> As shown in Figure 5, multi-task learning successfully learns the task-shared feature while single-task learning fails to do so. Moreover, the trend in the figure aligns with our two-stage analysis. For single-task learning, before around 80th epoch, the task-specific feature is learned relatively rapidly, which corresponds to our pattern learning stage; after around 80th epoch, it basically remains at the same level, which corresponds to our regularization stage. The learning level of the task-shared feature stays low. For multi-task learning, before around 50th epoch, both the task-specific features and the task-shared feature are learned rapidly by the neural network, which matches our pattern learning stage; after around 50th epoch, they both keep at a constant level, which matches our regularization stage.

---

> ### Author Response · Authors · 2023-11-23
> **Response to Reviewer rtpX [Part III]**
>
> **Question 4:**
>
> Do we see different observations if there is no weight decay for the multi-task models?
>
> **Answer 4:**
>
> During the rebuttal, we have done further experiments without weight decay on the synthetic data and make a comparison between the results with and without weight decay in Appendix A.1. The results are illustrated in Figure 4. We can see that although there are turning points in the dynamics of inner products between weight vectors and features both with and without weight decay, the curves after the turning points are obviously different between these two settings.
>
> With weight decay, which is the setting considered in our analysis, the learning processes of features enter the regularization stage after the turning points, and the inner products with features basically stay still, and even decrease a little. Without weight decay, however, the inner products with features keep increasing at a lower speed, so there are not very significant regularization stages in this case.
>
> Since it is difficult to explicitly express the different performance with and without weight decay, we do not show experiment results and observations on real datasets.
>
> **Question 5:**
>
> Is the data generating process reflective of real world datasets? Especially, is the data generating process accurate for even simple datasets like CIFAR10-C?
>
> **Answer 5:**
>
> The data-generating process with a common feature is close to the CIFAR10-C dataset. According to our experiments, for each of the situations listed in Table 3 where the neural network is trained on one corruption, it has an accuracy higher than a random guess on another corruption. This implies that these corruptions must have a common feature which is learned during training. More generally, many current pretrained models such as BERT, CLIP and GPT, are used in a large amount of downstream tasks, which are not even in the same categories as the pre-training tasks. However, the same pre-trained model performs well in fine-tuning/zero-shot learning on many tasks, and this is also due to the shared features among these tasks.
>
> **Question 6:**
>
> In practice, neural networks are deep and over-parameterized. Can these proof techniques be scaled beyond two layers to understand multi-task learning? Currently, they do not seem to provide a complete theory for even a highly idealized two layer neural network.
>
> **Answer 6:**
>
> We agree with the reviewer's observation about the simplicity of the problem setup in our current paper. Nevertheless, even with this simple data and neural network model, there is no existing research that has demonstrated similar results in this particular direction. We believe that analyzing multi-task learning with two-layer CNNs and the simple data model serves as a valuable initial step in comprehending multi-task learning with more complicated neural networks and more general data models. Without a deep understanding of the potential outcomes in this simplified setting, it becomes challenging to provide satisfactory explanations for the remarkable success of multi-task learning.  Currently, we have not figured out how to extend our results to multi-layer networks, and we will leave it for future work.
>
> [1] Nilesh Tripuraneni, Michael Jordan, and Chi Jin. On the theory of transfer learning: The importance of task diversity. Advances in neural information processing systems, 33:7852–7862, 2020.
>
> [2] Andreas Maurer, Massimiliano Pontil, and Bernardino Romera-Paredes. The benefit of multitask representation learning. Journal of Machine Learning Research, 17(81):1–32, 2016.
>
> [3] Jonathan Baxter. A model of inductive bias learning. Journal of artificial intelligence research, 12:149–198, 2000.

---

### Official Review · Reviewer_tYKP · 2023-11-04

**Soundness:** 3 good
**Presentation:** 3 good
**Contribution:** 3 good
**Rating:** 6
**Confidence:** 4

**Summary:**

The paper provides a theoretical analysis to show why multi-task learning leads to models with better generalizability than single-task learning when training neural networks using gradient descent. The analysis was done with a much-simplified problem setup comparing to networks commonly used in current practice. In addition to the theoretical analysis, empirical results on synthetic datasets and real-world datasets were provided to show multi-task learning outperforms single-task learning.

**Strengths:**

+ In the era of full of papers of neural networks with only empirical results, the efforts in theoretical understanding of better generalizability of multi-task learning than that of single-task learning in the context of gradient descent is commendable.

**Weaknesses:**

- The considered problem setup is an oversimplification of networks commonly used in the practice. The implication of their results on complex networks is not clear and likely limited. On the other hand, though, it may seed future theoretical studies on more complicated cases.

- The included experiments provide little to no value. The reported results are just typical observations one would expect to see when comparing multi-task and single task learning in the cases where tasks are related.

**Questions:**

- The parameter \alpha in their set up (Definition 3.1) is expected to have high impact. Since it approaches to 1, there should be no difference between shared and task-specific features. However, I did not see \alpha involved in any of the derived quantities.

- In the synthetic experiments, there are both shared and task-specific features. According to Definition 3.1 and 3.2, there should be more than two patches when including noise patches. Then, how come H can be 2?

- A discussion on how realistic those requirements listed at the end of section 3 are would be helpful.

- There are mathematic symbols without proper introduction, for example, n and polylog(n) in Definition 3.3. Is this n the total number of examples in the training set? If that is the case, the test error bound in Theorem 4.1 is counter intuitive. Because according to the bound, larger training set size leads to larger test/generalization error.

---

> ### Author Response · Authors · 2023-11-23
> **Response to Reviewer tYKP [Part I]**
>
> Thank you for your positive feedback!
>
> **Question 1**:
>
> The considered problem setup is an oversimplification of networks commonly used in the practice. The implication of their results on complex networks is not clear and likely limited.
>
> **Answer 1**:
>
> We agree with the reviewer's observation about the simplicity of the problem setup in our current paper. Nevertheless, even with this simple data and neural network model, there is no existing research that has demonstrated similar results in this particular direction. We believe that analyzing multi-task learning with two-layer CNNs and the simple data model serves as a valuable initial step in comprehending multi-task learning with more complicated neural networks and more general data models. Without a deep understanding of the potential outcomes in this simplified setting, it becomes challenging to provide satisfactory explanations for the remarkable success of multi-task learning.
>
> **Question 2**:
>
> The included experiments provide little to no value. The reported results are just typical observations one would expect to see when comparing multi-task and single-task learning in the cases where tasks are related.
>
> **Answer 2**:
>
> We would like to highlight the value of our experiments. Our experiments in Table 3 demonstrate results in many folds. First, it indicates the existence of shared features across the tasks we defined, as models trained on any single task invariably outperform random guessing (10%) on other tasks. Second, it shows that single-task learning cannot learn shared features well, because many single-task learning results in poor accuracy of less than 20% (e.g., training on brightness and testing on impulse noise only gets 17%). Lastly, the results explain why multi-task learning can outperform single-task learning in Table 2, by showing that multi-task learning on the 9 tasks resulted in better results on the other 10 unseen tasks, which is attributed to better learning of the shared feature. During the rebuttal, we have added additional experiments in Appendices A.1 and A.2 to corroborate our theoretical analysis.
>
> **Question 3**:
>
> The parameter $\alpha$ in their setup (Definition 3.1) is expected to have high impact. Since it approaches to $1$, there should be no difference between shared and task-specific features.
>
> **Answer 3**:
>
> We believe there is a misunderstanding by the reviewer. In our parameter setting at the end of Section 3, $\alpha$ is an important parameter of our data model with order $\Theta(n^{-c/10q^2})$.  $\alpha$ is always smaller than $1$, and when sample size $n$ is getting larger and larger, $\alpha$ will approach $0$ instead of $1$. Notice that the strength of the task-specific feature is $1$. The strength of the task-shared feature is always smaller than the task-specific features and their difference will get larger as $n$ gets larger.
>
> **Question 4**:
>
> I did not see $\alpha$ involved in any of the derived quantities.
>
> **Answer 4**:
>
> $\alpha$ doesn't show up in our Theorems 4.1 and 4.2, since $\alpha = \Theta(n^{-c/10q^2})$ is the condition of our theorems and only gets involved in the learning level of features and noises.  $\alpha$ represents the strength of the task-shared feature. In Theorem 4.1, $\alpha = \Theta(n^{-c/10q^2})$ is small enough. So the noises have a larger impact than the task-shared feature, and the neural network memorizes noises instead of learning the task-shared feature, which leads to a large test error. In Theorem 4.2, conversely,  $\alpha$ is large enough, the task-shared feature has a larger impact than noises, so the neural network is able to learn the task-shared feature, which leads to a small test error close to $0$. Specifically, $\alpha$ is getting involved in many quantities in Section 5.  In the pattern learning stage, it appears in updating inequalities of the task-shared feature (e.g., Lemma 5), which shows the learning speed of the task-shared feature in this stage; in the regularization stage, $\alpha$ influences the choices of parameters and the learning level of the task-shared feature implicitly and explicitly in the proof. Therefore, $\alpha$ plays an important role during learning and the output after learning.

---

> ### Author Response · Authors · 2023-11-23
> **Response to Reviewer tYKP [Part II]**
>
> **Question 5**:
>
> In the synthetic experiments, there are both shared and task-specific features. According to Definition 3.1 and 3.2, there should be more than two patches when including noise patches. Then, how come $H$ can be $2$?
>
> **Answer 5**:
>
> According to our Definition 3.1 and Definition 3.2, among all $H$ patches, there is one feature patch, and $H-1$ noise patch(es). So when $H = 2$, there will be one feature patch and one noise patch. In synthetic experiments, we set up our data model strictly according to Definitions 3.1 and  3.2. Note that for each data point from each task, it will either have a task-specific feature or a task-shared feature.
>
> **Question 6**:
>
> A discussion on how realistic those requirements listed at the end of section 3 are would be helpful.
>
> **Answer 6**:
>
> We set all the parameters based on n, the sample size for one task. According to our parameter settings, the dimension of each patch is set as $d=\Omega(n^{2c+4}K)$, so our problem is in the high dimensional setting, which aligns with most real datasets. Besides, the number of tasks is $K=\Omega(n^{\frac{c}{8q}})$, which is a constant degree polynomial of $n$, and can be chosen to be small or large based on the choice of constant $c$. In our synthetic dataset, we apply our parameter settings under a relatively small $c$ and get good results.
>
> **Question 7**:
>
> There are mathematical symbols without proper introduction, for example, $n$ and $\text{polylog}(n)$ in Definition 3.3. Is this $n$ the total number of examples in the training set?
>
> **Answer 7**:
>
> Thank you for your valuable suggestion. We have added the notation paragraph in our paper. Yes, $n$ is the sample size for one task which is defined in our Problem Setup (Section 3). $\text{polylog}(n)$ denotes a constant degree polynomial of $\log(n)$.
>
> **Question 8**:
>
> If that is the case, the test error bound in Theorem 4.1 is counterintuitive. Because according to the bound, larger training set size leads to larger test/generalization errors.
>
> **Answer 8**:
>
> The test error bound in Theorem 4.1 is reasonable for the following reasons.
>
> First, this is a lower bound of the test error, so it might not be able to accurately show the trend of the test error changing with $n$;
>
> Second, according to our data model, basically all parameters are related to n, including $\sigma_{0}$ and $\sigma_{p}$, the initialization of the weight vectors in the neural network, the noise patch, $\alpha$, the strength of the task-shared feature. When $n$ is getting larger, these quantities will get smaller, and they all have influences on the test error as well. So changing $n$ does not only affect the sample size, and we can not directly discern the trend of the test error only from sample size.
>
> Third, according to our proof, the quantity that is directly related to $n$ in the test error, $\text{poly}(n^{-1})$, comes from the fact that by concentration inequalities, with probability $1-\text{poly}(n^{-1})$, we can get a good enough initialization on the weight vectors in the neural network and the noise patches. Throughout the entire proof, we only consider those situations with this good initialization. Conditioned on good initialization, the probability of making an incorrect classification is lower bounded by $p/4$. Since $p$ is a constant, so the lower bound of misclassification without conditioning should be $p/4(1-\text{poly}(n^{-1}))=p/4-\text{poly}(n^{-1})$.
>
> Above all, the lower bound on the test error in Theorem 4.1 is reasonable.

---

### Meta-Review · Area_Chair_7LGu · 2023-12-08

**Metareview:**

This paper presents a comprehensive analysis of multi-task learning in neural networks, illustrating its superiority in generalizability compared to single-task learning when trained using gradient descent. The authors provide theoretical generalization bounds for multi-task learning, highlighting its capability to achieve asymptotically zero generalization error under certain conditions, unlike single-task learning. The analysis is conducted using a simplified model of a two-layer neural network and data with task-specific and task-shared features, employing tensor-power methods for mathematical modeling. Empirical evidence supporting these findings is demonstrated through experiments on both synthetic datasets and real-world datasets, including CIFAR-10. The work underscores the effectiveness of multi-task learning in learning shared features, thereby enhancing generalization in multiple classification tasks sharing the same output space.

This AC read all the reviews and the rebuttals as well as the post-discussion with the reviewers.
Three reviewers leaned toward positive about the theoretical contribution of this work, while one reviewer was against this decision due to the uncommon multi-task setting and the unfair advantage of leveraging more training data.
During the discussion, another reviewer stated: "I share the concerns with other reviewers that the assumptions are too specific (perhaps even too specific to call it generally 'multi-task learning'), and that it is a little disingenuous to absorb the number of tasks K into the O(N) dependency."
This AC concurs with the same opinion as the above.

Despite the positive comments and its potential acknowledged by multiple reviewers, overall the reviewers seem not to champion this work and there is a violation of the page limit after the revision.

**Justification For Why Not Higher Score:**

The multi-task setting in this work does not follow common settings, which appears unfair in comparison to the single-task settings.
The authors need to put more effort into balancing the number of samples to be fair, at least in the theorems.
This may introduce the gap between the theory and empirical results.

It would have been more interesting if the authors derived the theorem from the common practical multi-task setup, where each sample has all the labels for every task.

There is a violation of the page limit after the revision, which should be corrected.

**Justification For Why Not Lower Score:**

Although the setting is quite artificial, the analysis is interesting, and the analysis could encourage applying similar analysis to other related problems.

---

### Decision · Program_Chairs · 2024-01-16

Reject